

# The crystallographic spin point groups and their representations

Hana Schiff[1*], Alberto Corticelli[2], Afonso Guerreiro[3],
Judit Romhányi[1] and Paul McClarty[2,4]

**1** Department of Physics and Astronomy, University of California,
Irvine, California 92697, USA
**2** Max Planck Institute for the Physics of Complex Systems,
Nöthnitzer Str. 38, 01187 Dresden, Germany
**3** CeFEMA, Instituto Superior Técnico, Universidade de Lisboa,
Av. Rovisco Pais, 1049-001 Lisboa, Portugal
**4** Laboratoire Léon Brillouin, CEA, CNRS, Université Paris-Saclay,
CEA-Saclay, 91191 Gif-sur-Yvette, France

* hschiff@uci.edu

## Abstract

The spin point groups are finite groups whose elements act on both real space and spin space. Among these groups are the magnetic point groups in the case where the real and spin space operations are locked to one another. The magnetic point groups are central to magnetic crystallography for strong spin-orbit coupled systems and the spin point groups generalize these to the intermediate and weak spin-orbit coupled cases. The spin point groups were introduced in the 1960's in the context of condensed matter physics and enumerated shortly thereafter. In this paper, we complete the theory ofcrystallographic spin point groups by presenting an account of these groups and their representation theory. Our main findings are that the so-called nontrivial spin point groups (numbering 598 groups) have co-irreps corresponding exactly to the (co-)-irreps of regular or black and white groups and we tabulate this correspondence for each nontrivial group. However a total spin group, comprising the product of a nontrivial group and a spin-only group, has new co-irreps in cases where there is continuous rotational freedom. We provide explicit co-irrep tables for all these instances. We also discuss new forms of spin-only group extending the Litvin-Opechowski classes. To exhibit the usefulness of these groups to physically relevant problems we discuss a number of examples from electronic band structures of altermagnets to magnons.

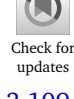

# 1 Introduction

Much of condensed matter physics is concerned with phenomena in crystalline solids. With discrete translation symmetry, physics happens within a periodic volume of crystal momentum space whose residual symmetries place constraints on band structures [1] and their topology [2–4] as well as static and dynamic correlation functions across a wide variety of systems including electronic, photonic, phononic and magnonic degrees of freedom to name a few out of many possibilities. The group theory of crystals is the foundation for understanding these and other aspects of solid state physics [1,5–8].

The history of group theory in relation to solid state physics goes back around 200 years with the realization from Hessel that 32 point groups are relevant to periodic crystals [9]. The space groups — that include discrete translations and that classify all periodic crystal structures — were classified towards the end of the 19th century [10].

Starting in the 20th century, magnetic crystals and their symmetries came under examination. Heesch [11], Tavger and Zaitsev [12] are associated with the discovery of the 122 magnetic point groups and Shubnikov [13] and Zamorzaev [14] with the magnetic space groups. In these studies, spin-orbit coupling is considered to be strong so that crystallographic operations such as rotations are locked to transformations of localized magnetic moments, meaning that operations performed on the crystal lattice simultaneously transform the magnetic moments according to the transformation properties of axial vectors. It was not until the 1960's that the weak spin-orbit coupling case was considered with the introduction of spin groups notably by Kitz [15], Brinkman and Elliott [16,17] later systematized by Litvin and Opechowski [18]. These groups have the feature of having elements that act on both real space and spin space but without locking the transformations to be the same in both. This work has been the subject of renewed interest with the recognition of its role in band structure topology in magnons [19] and electronic band structures [20–24] — the latter including anisotropic spin split Fermi surfaces enforced by spin group symmetries.

In this paper, we pick up the systematic exploration of these groups that has languished for about a half century by laying out the representation theory of the spin point groups first enumerated by Litvin [25] and more recently developed by Liu *et al.* [26]. While we focus on the finite set of crystallographic spin point groups as these are relevant to condensed matter physics, the same techniques discussed here are applicable to find the co-representations of all spin point groups.

The organization of the paper is as follows. In an effort to be as self-contained as is practicable, we give a complete discussion of what the spin point groups are, introducing notions of nontrivial spin point groups and spin-only groups and how to enumerate both (described in the first three parts of Section 2). Once we have both nontrivial spin point groups and the spin-only groups we may classify the total spin point groups (Section 2.4). A complete table of the nontrivial spin point groups is given in Appendix B. This table distinguishes the collinear and coplanar spin groups. These sections review material that can be found elsewhere in the literature [6,8,15–18,25–28]. In particular, the nontrivial spin point groups were enumerated by Litvin [25], the spin line groups by Lazić, Milivojević and Damnjanović [27] and the pairing of spin groups with the nontrivial spin groups was investigated by Liu *et al.* [26]. We then turn to the representation theory of these groups which had not previously been worked out. Making use of the main isomorphism theorem of Litvin and Opechowski [18], we demonstrate that the 598 nontrivial spin point groups have co-irreps corresponding to the regular or black and white point groups (Section 3.1). In Sections 3.2, we describe the effect on the representations of including the spin-only group to form the total spin group. We show that the coplanar spin groups are isomorphic to paramagnetic spin groups. Of particular interest are the spin groups corresponding to collinear magnetic structures with continuous spin rotation symmetry. These have new co-irreps that we compute and tabulate. The computation method is described in Section 3.2.4 with various technical results relegated to appendices. Complete tables of the co-irreps of the collinear spin groups are listed in Appendix F. In Section 4, we give some examples of how to put information about the representation theory to use in applications from band theory. In doing so, we remark on physically relevant extensions to the Litvin-Opechowski spin-only groups. Finally, we conclude with a broader perspective on the spin group representation theory including general results that may be inferred from the co-irrep tables. As a guide to the reader who may not be familiar with the (co-)representation theory of magnetic groups we review the relevant material in Appendix A.

## 2 Introduction to the spin point groups

### 2.1 Definitions

We imagine a situation where localized moments $M(\mathbf{R})$ are placed onto a finite number of sites in real space. As mentioned above the spin point groups are finite groups with elements $[S\|R]$ where $R$ is an ordinary point group element acting in real space and $S$ acts on spin space:[1]

$$[S\|R]M^{\alpha}(\mathbf{r}) = S_{\alpha\beta}M^{\beta}(R^{-1}\mathbf{r}). \tag{1}$$

A set of elements form a group if they leave the spin configuration on the real space sites invariant, i.e. $[S\|R]M(\mathbf{r}) = M(\mathbf{r})$ for all $[S\|R]$ in the set. We are interested in classifying spin point groups that can be relevant in crystallography. Therefore we consider only the 32 point groups relevant to crystals as candidates for the spatial point group elements, $R$.

Following Litvin and Opechowski [18], we distinguish between the spin point groups and so-called *nontrivial* spin point groups. The former may be written in general as a direct product $\mathbf{b} \otimes \mathbf{S}$ where $\mathbf{b}$, known as the *spin-only* group, acts only on spin space (it contains only elements of the form $[S\|E]$, where E is the identity element) while $\mathbf{S}$, the non-trivial spin point group, acts on both real space and spin space (and does not contain elements of the form $[S\|E]$ except for the trivial element, $[E\|E]$).

To enumerate the spin point groups we may proceed as follows [18, 25]. We first focus our attention on the nontrivial spin point groups defined above. Later we address the ways in which these can be decorated with pure spin groups $\mathbf{b}$. We choose one of the 32 point groups − hereafter referred to as spatial parent groups $\mathbf{G}$. Litvin and Opechowski explained [18, 25] that one first decomposes the group into a coset decomposition of a normal subgroup $\mathbf{g}$

$$\mathbf{G} = \mathbf{g} + G_2\mathbf{g} + \ldots + G_n\mathbf{g}. \tag{2}$$

We may find a group $\mathbf{B}$, referred to as the spin-space parent group, that is isomorphic to the quotient group $\mathbf{G}/\mathbf{g}$. A spin group is then formed from a pairing of these elements

$$\mathbf{S} = ([E\|E] + [B_2\|G_2] + \ldots [B_n\|G_n])[E\|\mathbf{g}]. \tag{3}$$

By finding all normal subgroups of $\mathbf{G}$ and all isomorphisms between $\mathbf{G}/\mathbf{g}$ and point groups $\mathbf{B}$ one may enumerate all nontrivial spin point groups (see Appendix C). Pairings leading to spin groups conjugate in $\mathbf{O(3)} \otimes \mathbf{O(3)}$ are considered equivalent. This program was accomplished by Litvin [25]. See also discussions in Refs. [6, 8].

### 2.2 Enumeration of the nontrivial spin point groups

Here we check the enumeration proceeding in a different but ultimately equivalent way [27]. The isomorphism between $\mathbf{G}/\mathbf{g}$ and $\mathbf{B}$ forms a homomorphism between $\mathbf{G}$ and $\mathbf{B} < \mathbf{O(3)}$ with kernel $\mathbf{g}$. Thus, all three dimensional real representations of $\mathbf{G}$ with kernel $\mathbf{g}$ exhaust the possible pairings between $\mathbf{G}/\mathbf{g}$ and $\mathbf{B}$. For a choice of $\mathbf{G}$ we have access to its irreducible representations, and build all possible three dimensional representations from them.

First take the one-dimensional real irreps of $\mathbf{G}$, $\Gamma_1^{(n)}(G_\alpha)$, with characters $\chi(\Gamma_1^{(n)}(G_\alpha))$ for element $G_\alpha$ where $n$ labels the irrep and the subscript 1 is the dimension of the irrep. We may construct a real three-dimensional irrep by taking three copies of the same irrep

$$\chi(\Gamma_1^{(n)}(G_\alpha)) \oplus \chi(\Gamma_1^{(n)}(G_\alpha)) \oplus \chi(\Gamma_1^{(n)}(G_\alpha)), \tag{4}$$

---

[1]These spin groups are not to be confused with the *continuous* groups of the same name Spin(N) defined as the double cover of $SO(N)$.

by taking

$$\chi(\Gamma_1^{(m)}(G_\alpha)) \oplus \chi(\Gamma_1^{(n)}(G_\alpha)) \oplus \chi(\Gamma_1^{(n)}(G_\alpha)), \tag{5}$$

or, in case there are three or more 1D irreps, we may take

$$\chi(\Gamma_1^{(m)}(G_\alpha)) \oplus \chi(\Gamma_1^{(n)}(G_\alpha)) \oplus \chi(\Gamma_1^{(p)}(G_\alpha)), \tag{6}$$

where $m \neq n \neq p$.

Given pseudoreal or complex 1D irreps we are forced to pair them with their complex conjugate perhaps performing a similarity transformation to ensure the reality of the resulting irrep.

$$W\left[\chi(\Gamma_1^{(m)}(G_\alpha)) \oplus \chi^*(\Gamma_1^{(m)}(G_\alpha))\right]W^{-1} \oplus \chi(\Gamma_1^{(n)}(G_\alpha)). \tag{7}$$

We may also pair a 2D irrep with a real 1D irrep

$$\Gamma_2^{(m)}(G_\alpha) \oplus \chi(\Gamma_1^{(n)}(G_\alpha)), \tag{8}$$

and, finally, if there is a 3D irrep of **G**, that is real, this itself suffices to supply a representation for the spins:

$$\Gamma_3^{(m)}(G_\alpha). \tag{9}$$

It is straightforward to construct all such combinations for all elements of all parent groups. With the associated spin elements for each real space element we identify the nature of the element. The reader may well have noticed that this process leads to improper elements which are forbidden for magnetic moments as they are axial vectors. We interpret these elements as antiunitary elements. For example, a pure inversion is identified with the time reversal operation. All improper elements are products of inversion with a proper element and are therefore identified with a proper element times time reversal.

This algorithm overcounts the possible groups. Fortunately all instances of overcounting come from permutations of the Cartesian axes. Removing all of these cases one ends up with 598 crystallographic spin point groups, confirming previous results in detail including the number of groups for each parent group and the nature of the spin and space generators [25].

## 2.3 The spin-only group

We now ask what constraints should be placed on the spin-only groups, **b**. To set the stage, we could, in principle, consider any crystallographic subgroup of the paramagnetic group consisting of $\mathbf{SO(3)} \times \{E, \tau\}$. If the nontrivial spin point group is compatible with paramagnetism, then **b** is precisely this group as any element leaves the zero net moments invariant. This is the non-magnetic or paramagnetic case.

Now we consider the cases where there is a net magnetic moment. Following Litvin and Opechowski we first distinguish three cases: spin textures that are (1) collinear, (2) coplanar and (3) non-coplanar.

In the non-coplanar case there is no global spin-only transformation that will leave the spin texture invariant. Therefore, in this case the only choice for **b** is the trivial group. In the coplanar case, the spin texture can be left invariant by doing nothing or by rotating all the moments about an axis perpendicular to the plane of the moments by $\pi$ and then carrying out a time reversal operation — then $\mathbf{b} \equiv \mathbf{b}^{\mathbb{Z}_2} = \{E, C_{2\perp}T\}$. Finally, in the collinear case (ferromagnetic, antiferromagnetic or ferrimagnetic), in general one can rotate by any angle $\phi$ around the global moment direction. One can also rotate through $\pi$ about any axis perpendicular to the axis containing the moments and then carry out time reversal. So the spin-only group in this case is $\mathbf{b} \equiv \mathbf{b}^\infty = \mathbf{SO(2)} \rtimes \{E, C_{2\perp}\tau\}$.

## 2.4 Pairing with a spin-only group

As noted above the total spin group takes the form of a direct product of a spin-only part and one of the nontrivial spin point groups. Here we describe what spin-only groups are possible and the constraints on their pairing with the nontrivial spin point groups. By the end of this section we will have a complete picture of the classification of the crystallographic spin point groups. The observations in this section first appeared in Ref. [26].

The nature of the nontrivial spin point group contains information about the invariant magnetic structure. As described in the previous section there is a discrete set of possibilities for the spin-only group. For any given nontrivial group there are constraints on the possible pairings. In Section 2.1, we outlined an enumeration procedure for the nontrivial spin point groups that involved pairing spin-only elements from group **B** with elements of an isomorphic quotient group acting on real space. Taking all the nontrivial point groups together, one finds that **B** runs over all 32 crystallographic point groups with improper elements being a proxy for antiunitary elements. From a group theoretic perspective, the constraints on the spin-only group imposed by the nontrivial spin point group arise from the structure of a direct product **b** × **S**. For direct product groups, both factors in the product must be normal subgroups of the total group. Further constraints arise from eliminating redundancies where two different nontrivial groups **S** give rise to the same total group **X**.

In the non-magnetic case, the spin-only group is $\mathbf{b}^{\mathrm{NM}} \equiv \mathbf{SO(3)} \rtimes \{E, \tau\}$. Consider a nontrivial spin point group **S** with spin-space parent group **B**. Then, the total spin point group $\mathbf{X} = \mathbf{b}^{\mathrm{NM}} \times \mathbf{S}$ contains exactly the same elements as $\mathbf{b}^{\mathrm{NM}} \times \mathbf{G}$. We can think of **X** containing cosets resembling those of equation 3, except with the coset representatives multiplying the group $[\mathbf{b}^{\mathrm{NM}}\|E] \times [E\|\mathbf{g}]$ (see Appendix D). Then, the coset $[B_j\|G_j]\big([\mathbf{b}^{\mathrm{NM}}\|E] \times [E\|\mathbf{g}]\big)$ is equal to the coset $[E\|G_j]\big([\mathbf{b}^{\mathrm{NM}}\|E] \times [E\|\mathbf{g}]\big)$, since $\mathbf{b}^{\mathrm{NM}}$ contains $B_j^{-1}$ by virtue of containing all proper and improper rotations. As a result, all nontrivial **S** with the same spatial parent group **G** will give rise to the same total group **X**. To prevent redundancies then, it is sufficient to consider the paramagnetic groups formed from the direct product of $\mathbf{b}^{\mathrm{NM}}$ with each of the 32 crystallographic point groups.

Now suppose **B** is one of the cubic groups $T$, $T_d$, $T_h$, $O$ or $O_h$. This means that there are rotations that forbid collinear and coplanar structures. As a result, wherever the nontrivial spin group is built from one such **B** group, the spin-only part of the group must be trivial, $\mathbf{b}^{\mathrm{triv}} = \{[E\|E]\}$. We note that these groups all coincide exactly with magnetic point groups (see table in Appendix B). In other words, the irreducibly non-coplanar groups are not true spin groups but lie in the older magnetic point group classification.

Suppose we act with the elements of a nontrivial spin point group on a single site. Then, if **B** is the trivial group, the elements of the group do nothing to the spin directions meaning that the resulting spin structure is collinear and ferromagnetic. In this case we expect to pair the group with $\mathbf{b}^{\infty}$. For group $\mathbf{B} = C_i$ — containing identity and time reversal — the structure is collinear and antiferromagnetic leading again to $\mathbf{b}^{\infty}$ as the allowed spin-only group. Conversely, we may ask which of the nontrivial spin groups may be paired with $\mathbf{b}^{\infty}$. The answer is generally that only the $C_1$ and $C_i$ are allowed. Suppose we take for **B** another group that is consistent with a collinear magnetic structure such as 222. This group is consistent with an antiferromagnetic collinear structure for certain conditions on the coordinates. However, the requirement of pairing with $\mathbf{b}^{\infty}$ makes the $C_2$ around the moment direction redundant since any rotation angle is now allowed. The remaining $C_2$ rotations are also contained in the antiunitary coset of $\mathbf{b}^{\infty}$. This argument generalizes reducing the allowed **B** to $C_1$ and $C_i$. There are 32 of the former total spin groups — one for each of the parent point groups — and 58 of the latter which are non-unitary at the level of the nontrivial point group. We refer to these groups as *naturally collinear* as they are compatible with collinear magnetic structures.

However, one could obtain a magnetic structure symmetric under these groups starting from inequivalent sites with non-collinear structures and applying group elements. These groups, as we shall see, turn out to be of particular interest from the point of view of their representation theory.

Now we consider coplanar spin configurations, giving rise to 252 coplanar spin groups. At first it might appear that any of the 27 non-cubic groups $\mathbf{B}$ are compatible with a coplanar structure, but this is not the case. Let us begin with groups $C_n$ for $n > 2$. These naturally lead to coplanar structures whenever a moment is perpendicular to the rotation axis. If a moment is parallel or antiparallel to the axis the elements generate a collinear configuration. Otherwise a non-coplanar configuration will arise. Now suppose we add horizontal mirrors to get $C_{nh}$. The combined constraints that the moments be coplanar and the $\mathbf{b} = \mathbf{b}^{\mathbb{Z}_2} = \{E, C_{2\perp}\tau\}$ spin-only group "mod out" the horizontal mirrors so that the group is reduced to $C_n$. Suppose instead we add vertical mirrors taking the group to $C_{nv}$. In this case, the coplanarity condition takes the group to $D_n$, since the composition of $C_{2\perp}$ and a two-fold rotation, $C_{2\parallel}$, whose axis lies in the spin plane is another $C'_{2\parallel}$ parallel to the spin plane, perpendicular to the original $C_{2\parallel}$. Similarly groups $S_n$, with roto-reflections, reduce to $C_n$ for coplanar configurations. We must now consider $D_{nh}$ and $D_{nd}$. $D_{nh}$ is obtained by including horizontal mirrors in $D_n$. As the moments are assumed to be coplanar, horizontal mirrors in $D_{nh}$ are essentially "modded out" by the presence of $C_{2\perp}\tau$ as the spin-only group as therefore they cannot arise as separate elements in their own right in the nontrivial part of the spin group. The groups $D_{nd}$ are obtained from $D_n$ by including diagonal reflections and, as a consequence, they also contain roto-reflections. But for coplanar moments, the roto-reflections amount to simple rotations so imposing $D_{nd}$ and then coplanarity reduces the effective symmetry to groups with pure rotations. The summary of the coplanar case is that only the groups $C_n$ and $D_n$ need be considered. All other groups have the non-$C_n$ and $D_n$ elements rendered redundant by the action of the spin-only part.

We now work out what groups arise by taking the direct product of coplanar nontrivial spin groups with the spin-only part $\{E, C_{2\perp}\tau\}$. We first dispense with simple cases. These belong to $D_n$ with $n > 2$ and $C_n$. Each of these groups has a principal rotation axis that generates the plane of the moments. There is then one single choice of $C_{2\perp}\tau$. For $C_2$ this is perpendicular to the $C_2$ axis and for the other groups it is parallel to the rotation axis. This leaves us with $D_2$ as the only subtle case. This group has three perpendicular $C_2$ axes. These may be paired with various real space group operations. Suppose the real space part is also $D_2$. Then the resulting coplanar structures in three perpendicular planes are related by a coordinate redefinition so they are equivalent. Although there are three distinct $C_{2\perp}\tau$ axes these are therefore all equivalent. A similar argument can be made for the $D_{2h}(mmm)$ group.

There are in total 26 groups with $\mathbf{B} = D_2$ and we have dealt with two of them. For the rest one must inspect the way that the 222 spin elements are paired with the real space elements. We describe one case explicitly as it serves to illustrate how to handle all remaining cases. We refer to the elements of group $^{2_y}m^{2_x}m^1m$ listed in Table 3. We observe that the $C_{2x}$ and $C_{2y}$ spin elements are both paired with one real space mirror and one two-fold rotation whereas $C_{2z}$ is paired with an inversion and a rotation. Now we make a choice of the spin-only group assuming coplanarity of the moments. If the perpendicular axis to the plane of the moments is $z$ then the resulting group is distinct from the cases where the axis perpendicular to the plane is $x$ or $y$. For the latter cases, the groups are equivalent after a permutation of real space and spin space axes. It follows that for spin group $^{2_y}m^{2_x}m^1m$ with the condition of coplanarity, there are two inequivalent groups after pairing with the spin-only group.

There is therefore a mechanism to increase the number of groups by pairing with the spin-only group. In Tables 1 and 2 we list all spin groups with 222 spin elements and the inequivalent groups after pairing with the spin-only group.

Table 1: Explicitly listing coplanar spin groups where the parent spin group was $\mathbf{B} = 222$.

| Parent Group | Litvin Number | Litvin Symbol | Inequivalent $C_{2\perp}$ axis choices |
|---|---|---|---|
| **2/m** | 24 | $^{2}2/^{2}m$ | x |
| | | | y |
| | | | z |
| **mm2** | 41 | $^{2_x}m^{2_y}m^{2_z}2$ | x,y |
| | | | z |
| **222** | 51 | $^{2_x}2^{2_y}2^{2_z}2$ | x,y,z |
| **mmm** | 64 | $^{2_z}m^{2_z}m^{2_x}m$ | x |
| | | | y |
| | | | z |
| **mmm** | 74 | $^{2_x}m^{2_y}m^{1}m$ | x,y |
| | | | z |
| **mmm** | 80 | $^{2_x}m^{2_y}m^{2_z}m$ | x,y,z |
| **4/m** | 116 | $^{2_z}4/^{2_x}m$ | x |
| | | | y |
| | | | z |
| **422** | 137 | $^{2_x}4^{2_z}2^{2_y}2$ | x |
| | | | y, z |
| **4mm** | 153 | $^{2_z}4^{2_y}m^{2_x}m$ | x |
| | | | y, z |
| **$\overline{4}$2m** | 172 | $^{2_x}4^{2_z}2^{2_y}m$ | x |
| | | | y |
| | | | z |
| **4/mmm** | 202 | $^{2_x}4/^{2_x}m^{2_z}m^{2_y}m$ | x |
| | | | y, z |
| **4/mmm** | 208 | $^{1}4/^{2}m^{2}m^{2}m$ | x |
| | | | y |
| | | | z |
| **4/mmm** | 218 | $^{2_z}4/^{1}m^{2_x}m^{2_y}m$ | x,y |
| | | | z |
| **4/mmm** | 224 | $^{2_z}4/^{2_x}m^{1}m^{2_z}m$ | x |
| | | | y |
| | | | z |
| **4/mmm** | 234 | $^{2_z}4/^{2_x}m^{2_x}m^{2_y}m$ | y |
| | | | y |
| | | | z |
| **$\overline{3}$m** | 304 | $^{2_x}\overline{3}^{2_z}m$ | x |
| | | | y |
| | | | z |
| **622** | 345 | $^{2_x}6^{2_y}2^{2_z}2$ | x |
| | | | y,z |
| **6/m** | 368 | $^{2_z}6/^{2_x}m$ | x |
| | | | y |
| | | | z |
| **6mm** | 400 | $^{2_x}6^{2_y}m^{2_z}m$ | x |
| | | | y, z |
| **$\overline{6}$m2** | 422 | $^{2_z}\overline{6}^{2_x}m^{2_y}2$ | x |
| | | | y |
| | | | z |

Table 2: Explicitly listing coplanar spin groups where the parent spin group was
$\mathbf{B} = 222$.

| Parent Group | Litvin Number | Litvin Symbol | Inequivalent $C_{2\perp}$ axis choices |
|:---:|:---:|:---:|:---:|
| **6/mmm** | 456 | $^{2_z}6/^{2_z}m^{2_x}m^{2_y}m$ | x,y |
| | | | z |
| **6/mmm** | 462 | $^{2_z}6/^{1}m^{2_x}m^{2_y}m$ | x,y |
| | | | z |
| **6/mmm** | 468 | $^{1}6/^{2_z}m^{2_x}m^{2_x}m$ | x |
| | | | y |
| | | | z |
| **6/mmm** | 478 | $^{2_z}6/^{2_x}m^{1}m^{2_z}m$ | x |
| | | | y |
| | | | z |
| **6/mmm** | 488 | $^{2_z}6/^{2_x}m^{2_y}m^{2_z}m$ | x |
| | | | y |
| | | | z |
| **m$\bar{3}$m** | 577 | $^{2_x}4/^{2_y}m^{2_y}\bar{3}^{2_x}2/^{2_z}m$ | x |
| | | | y |
| | | | z |

Table 3: Table of elements in $^{2_y}m^{2_x}m^{1}m$ with real space elements in the top row
and spin elements in the bottom row. The axis labels $e$ means $1/\sqrt{2}(0,1,1)$ and $f$ is
$1/\sqrt{2}(0,1,-1)$.

| Real space | $E$ | $C_{2f}$ | $C_{2e}$ | $C_{2x}$ | $I$ | $IC_{2x}$ | $IC_{2e}$ | $IC_{2f}$ |
|:---:|:---:|:---:|:---:|:---:|:---:|:---:|:---:|:---:|
| Spin space | $E$ | $C_{2z}$ | $C_{2y}$ | $C_{2x}$ | $C_{2z}$ | $C_{2y}$ | $C_{2x}$ | $E$ |

Finally, we remark on all the remaining groups — that make up the majority of nontrivial
spin groups. Since, of the four classes, we have found all cases that can be collinear and co-
planar and since the nontrivial spin groups are magnetic (none is of the form $\mathbf{G} + \tau\mathbf{G}$) all the
remaining groups are non-coplanar groups even though this is not manifest from the $\mathbf{B}$ group
as it is for the cubic groups discussed above.

## 2.5 Guide to the tables of the nontrivial spin point groups

In Appendix B we give a table with all 598 nontrivial spin point groups (Table 7). This table
is the original enumeration of Litvin using his notation to decorate the real space generators
with spin space elements. For example, nontrivial group $^{\bar{1}}4/^{1}m^{1}m^{\bar{1}}m$ tells us that the parent
point group $4/mmm$ has the $C_{4z}$ and one out of the three mirrors paired with a time reversal
operation the other two mirrors being paired with a trivial operation on spin space. The table
is organized by parent crystallographic point group and contains various other useful facts.
First of all we highlight in boldface those groups that are exactly magnetic point groups by
which we mean those groups for which the real and spin space transformations are locked
together. As we show below the nontrivial point groups are all isomorphic to (but usually not
equal to) magnetic point groups. In the last column of each table we provide the matching
magnetic point group label. In addition, we color code the Litvin symbols to indicate the
constraints on the possible spin-only point groups coming from the nontrivial groups. We
color (in blue) those groups for which pairing with $\mathbf{b}^{\infty}$ is possible. These are the groups that
belong naturally to simple collinear spin textures. We color in orange the groups that can be
paired with the discrete $\mathbf{b}^{\mathbb{Z}_2}$ spin-only group. All other groups can neither be paired with $\mathbf{b}^{\infty}$

nor the $\mathbf{b}^{\mathbb{Z}_2}$ spin-only groups. We remind the reader that there are nontrivial spin groups that necessarily belong to non-coplanar spin configurations and that, therefore, cannot be paired with a nontrivial spin-only group. We do not distinguish these in the table as all the non-highlighted groups are naturally non-coplanar once the collinear and coplanar constraints are taken into account as explained in Section 2.4.

# 3 Representations of the spin point groups

In this section, we lay out the representation theory for the spin point groups. The representation theory for the nontrivial spin point groups turns out to be simple: their (co-)irreps can be matched with one of the ordinary or black and white point group (co-)irreps. The non-magnetic and coplanar total spin groups also have (co-)irreps corresponding to some magnetic point group, as we will show. However, collinear spin point groups have new co-irreps. In the case of $\mathbf{B} = C_1$, the co-irreps can be easily found, whereas for $\mathbf{B} = C_i$, an additional step of induction is required. As such, the co-irreps in these cases are calculated explicitly and included in Appendix F. In this section, $\tau$ refers to the time-reversal element, and the ◁ symbol denotes that the group on the left is a normal subgroup in the group on the right. It should be noted that this work does not address the double spin point groups, which take point group elements from $\mathbf{SU(2)}$ and therefore allow half-integral angular momentum irreps.

## 3.1 Nontrivial spin point groups

Recall that the nontrivial spin point do not contain *any* spin-only elements (aside from the trivial element, $[E\|E]$). That the nontrivial spin point groups are isomorphic to their parent group $\mathbf{G}$ follows directly from the theorem Litvin & Opechowski used to construct the spin groups [18] and which is explained in Appendix C. The relevant argument for our current purposes is reproduced here. For spin group $\mathbf{S}$ with spatial parent group $\mathbf{G}$ and spin-space parent group $\mathbf{B}$ and defining $\mathbf{g} \equiv \mathbf{G} \cap \mathbf{S}$ and $\mathbf{b} \equiv \mathbf{B} \cap \mathbf{S}$, the central result, for our purposes, is that $\mathbf{S}/\mathbf{b} \cong \mathbf{G}$.

The spin group $\mathbf{S}$ is a subgroup of $\mathbf{B} \times \mathbf{G}$, i.e. $\mathbf{S} \leq \mathbf{B} \times \mathbf{G}$. Furthermore $\mathbf{B}, \mathbf{G} \triangleleft \mathbf{B} \times \mathbf{G}$. We now invoke Noether's 2nd isomorphism theorem that we state without proof.

**Theorem (Noether):** Let $\mathbf{H}$ be a group, with $\mathbf{J} < \mathbf{H}$ and $\mathbf{K} \triangleleft \mathbf{H}$. Then, $\mathbf{K} \cap \mathbf{J} \triangleleft \mathbf{J}$, and $\mathbf{J}/(\mathbf{K} \cap \mathbf{J}) \cong \mathbf{KJ}/\mathbf{K}$, where $\mathbf{JK} \equiv \{jk | j \in \mathbf{J}, k \in \mathbf{K}\}$.

This result allows us to conclude that $\mathbf{b}, \mathbf{g} \triangleleft \mathbf{S}$ as well as $\mathbf{S}/\mathbf{b} \cong \mathbf{BS}/\mathbf{B}$ and $\mathbf{S}/\mathbf{g} \cong \mathbf{GS}/\mathbf{G}$.

Next, given that, for example, $\mathbf{g} \triangleleft \mathbf{S}$, we can express $\mathbf{BS}$ as

$$\mathbf{BS} = \mathbf{B}[E\|\mathbf{g}] + \mathbf{B}[B_2\|G_2\mathbf{g}] + \ldots \mathbf{B}[B_n\|G_n\mathbf{g}] \cong \mathbf{B}[E\|\mathbf{g}] + \mathbf{B}[E\|G_2\mathbf{g}] + \ldots \mathbf{B}[E\|G_n\mathbf{g}],$$

since $\mathbf{B}B_i = \mathbf{B}$. So $\mathbf{BS}/\mathbf{B} \cong [E\|\mathbf{g}] + [E\|G_2\mathbf{g}] + \ldots [E\|G_n\mathbf{g}] \cong \mathbf{G}$.

We have shown that

$$\mathbf{S}/\mathbf{b} \cong \mathbf{BS}/\mathbf{B} \cong \mathbf{G}, \tag{10}$$

as claimed above. Evidently, when the spin group is nontrivial, $\mathbf{b}$ is trivial so $\mathbf{S} \cong \mathbf{G}$ demonstrating that nontrivial spin groups are isomorphic to the spatial parent group.

### 3.1.1 Unitary nontrivial spin point groups

When the nontrivial spin (point) group $\mathbf{S}$ is unitary, an immediate result of the isomorphism equation 10 is that the irreducible representations of $\mathbf{S}$ are the same as for $\mathbf{G}$. However, this will not be the case for the non-unitary nontrivial spin point groups, as we shall see in the following section.

### 3.1.2 Non-unitary nontrivial spin point groups

When the nontrivial spin point group **S** is such that **B** contains improper elements, we say that **S** is non-unitary because inversion in spin-space is interpreted as time-reversal. For every nontrivial spin point group, at the level of abstract groups, it still holds that $\mathbf{S} \cong \mathbf{G}$. However, the fact that half of the elements in **S** contain time reversal means that half of the group elements will be represented by antiunitary operators. As discussed in Appendix A, the representation theory of non-unitary groups must be modified to accommodate antiunitary operators. As a consequence, it does *not* follow that non-unitary nontrivial **S** will have the same irreps as **G**. However, a related result holds: these groups have irreducible co-representations equivalent to those of some black & white point group.

For convenience, we will call group elements with improper spin parts antiunitary as a reminder of the implications for their representation theory; similarly, group elements with proper spin parts will be called unitary. Recall that every non-unitary group can be expressed as the union of a coset of unitary elements (the "unitary coset" or "unitary halving subgroup") and a coset of antiunitary elements.

Let our nontrivial non-unitary spin point group be expressed as $\mathbf{S} = \mathbf{S}^* + a\mathbf{S}^*$, with unitary halving spin point group $\mathbf{S}^*$ and antiunitary coset representative $a = [\tau u_s \| u_o]$ where $u = [u_s \| u_o]$ is unitary (i.e. $u_s$ is proper). Because $[\tau \| E]$ commutes with every element of **S**, the group structure of $\tilde{\mathbf{S}} = \mathbf{S}^* + u\mathbf{S}^*$ is the same as that of **S**. As a result, it holds that $\mathbf{S} \cong \tilde{\mathbf{S}}$, and we will choose an explicit isomorphism $\tilde{\beta} : \mathbf{S} \to \mathbf{S}^*$ for later use, such that $\tilde{\beta}(\mathbf{S}^*) = \mathbf{S}^*$ and $\tilde{\beta}(a) = u$. However, we have also seen in the previous section(s) that $\mathbf{S} \cong \mathbf{G}$, so we have three group isomorphisms

$$\mathbf{S} \cong \tilde{\mathbf{S}} \cong \mathbf{G}.$$

Next, note that $\mathbf{S}^*$ is a unitary nontrivial spin point group. Using the result of the previous section, it follows that $\mathbf{S}^*$ is isomorphic to *its* spatial parent group $\mathbf{G}^*$ (the collection of elements on the right-hand side of the $[\star \| \star]$ in the elements of $\mathbf{S}^*$), which must be a halving subgroup of **G**. As a result, we can conclude that $\mathbf{S}^* \cong \mathbf{G}^*$ both have the same irreducible representations, and $\mathbf{G} = \mathbf{G}^* + u_o\mathbf{G}^*$.

A black and white point group, $\mathbf{G}^{\mathrm{BW}}$, can be formed from **G** and $\mathbf{G}^*$ via the group isomorphism $\beta : \mathbf{G} \to \mathbf{G}^{\mathrm{BW}}$ such that $\beta(\mathbf{G}^*) = \mathbf{G}^*$ and $\beta(u_o) = \tau u_o$, giving $\mathbf{G}^{\mathrm{BW}} = \mathbf{G}^* + \tau u_o\mathbf{G}^*$. The choice of **G** and $\mathbf{G}^*$ completely determines the black & white point group [28] and its co-irreps, since they enforce a set of possible of coset representatives $\tau u_o$, all of which give rise to the same Dimmock indicators in equation A.20.

We would like to use the previous isomorphisms and construction algorithm for the black and white point groups to demonstrate that **S** is not only isomorphic to the black and white point group $\mathbf{G}^{\mathrm{BW}}$ on the level of abstract groups, but that **S** and $\mathbf{G}^{\mathrm{BW}}$ must have equivalent co-irreps.

Since $\tilde{\mathbf{S}} \cong \mathbf{G}$ and $\mathbf{S}^* \cong \mathbf{G}^*$, there exists a group isomorphism $\varphi : \tilde{\mathbf{S}} \to \mathbf{G}$ such that $\varphi(\mathbf{S}^*) = \mathbf{G}^*$ and $\varphi(u) = u_o$. Then, we can compose the isomorphisms $\tilde{\beta}$, $\varphi$ and $\beta$ to obtain an isomorphism $\beta \circ \varphi \circ \tilde{\beta} : \mathbf{S} \to \mathbf{G}^{\mathrm{BW}}$, as shown in Figure 1. This particular isomorphism will not only ensure that **S** is isomorphic to $\mathbf{G}^{\mathrm{BW}}$, but also that their co-irreps are equivalent based on their Dimmock indicators.

We will now demonstrate that the co-irreps of **S** are equivalent to those of $\mathbf{G}^{\mathrm{BW}}$. To begin, recall that the Dimmock indicators of equation A.20 use the squares of group elements in the antiunitary coset. For every antiunitary group element, its square is a unitary group element from $\mathbf{S}^*$ or $\mathbf{G}^*$, respectively. Further, the Dimmock indicator will use the characters $\chi$ from irreps of the unitary halving subgroups, $d^{(\mu)}(\mathbf{S}^*) \sim d^{(\mu)}(\mathbf{G}^*)$. Because $\varphi$ is an isomorphism that maps $\mathbf{S}^*$ to $\mathbf{G}^*$, it will map conjugacy classes of $\mathbf{S}^*$ to conjugacy classes of $\mathbf{G}^*$. Because characters are functions on the conjugacy classes of a group, this also means that $\varphi$ induces

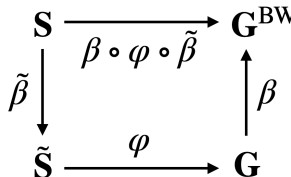

Figure 1: The isomorphisms between $\mathbf{S}$, $\tilde{\mathbf{S}}$, $\mathbf{G}$ and $\mathbf{G}^{\mathrm{BW}}$ that ensure the construction of equivalent co-irreps.

a mapping on the characters $\chi^{(\mu)}(\mathbf{S}^*)$ and $\chi^{(\mu)}(\mathbf{G}^*)$, and in particular, we are free to choose $\varphi$ to be any isomorphism that maps elements in $\mathbf{S}^*$ to those in $\mathbf{G}^*$ with the same character. Then, it is guaranteed that the characters $\{\chi^{(\mu)}(aS)^2 | S \in \mathbf{S}^*\}$ are equal to the characters $\{\chi^{(\mu)}(\tau u_o G)^2 | G \in \mathbf{G}^*\}$, implying that the Dimmock indicators for $\mathbf{S}$ will be the same as for $\mathbf{G}^{\mathrm{BW}}$.

This procedure also provides a scheme for identifying which black and white point group $\mathbf{G}^{\mathrm{BW}}$ has the co-irreps corresponding to our non-unitary nontrivial spin point group $\mathbf{S}$: $\mathbf{G}^{\mathrm{BW}}$ is the black and white point group whose unitary halving subgroup $\mathbf{G}^*$ is the spatial parent group for the unitary halving subgroup $\mathbf{S}^*$ of $\mathbf{S}$. It turns out that, conveniently, this process is on a practical level the same as replacing the improper spin elements in the Herman-Maugin notation of the spin point group with primes to obtain the name of the black and white point group.

We can demonstrate this principle using as an example the non-unitary nontrivial spin point groups isomorphic to spatial parent group $4mm$ ($C_{4v}$).[2] Table 4 lists all such spin point groups. Groups #160 and #161 exactly correspond to the black and white point groups $4m'm'$ and $4'mm'$, respectively. The unitary halving subgroups $\mathbf{S}^*$ of all the $\mathbf{S}$ in table 4 are isomorphic to either 4 ($C_4$) or $mm2$ ($C_{2v}$), the unitary halving subgroups $\mathbf{G}^*$ of the corresponding black and white point groups $\mathbf{G}^{\mathrm{BW}}$.

## 3.2 Effect of the spin-only group

In Section 2.3 we described the possible spin-only groups that can be paired with a nontrivial space group in general following [18]:

- Non-magnetic: $\mathbf{b}^{\mathrm{NM}} = \mathbf{SO(3)} \times \{[\, E \,\|\, E \,], [\, \tau \,\|\, E \,]\}$, (which is isomorphic to $\mathbf{O(3)}$).

- Collinear: $\mathbf{b}^{\infty} = \mathbf{SO(2)} \rtimes \{[\, E \,\|\, E \,], [\, \tau 2_{\perp \mathbf{n}} \,\|\, E \,]\}$, where $\mathbf{n}$ is the axis to which the spins are aligned.

- Coplanar: $\mathbf{b}^{\mathbb{Z}_2} = \{[\, E \,|\, E \,], [\, \tau 2_{\mathbf{n}} \,\|\, E \,]\}$, where the plane in which the spins lie is denoted by $\mathbf{n}$.

- Non-coplanar: $\mathbf{b}^{\mathrm{triv}} = \{[\, E \,\|\, E \,]\}$.

In Section 2.4 we presented constraints on the possible pairings of these spin-only groups with nontrivial spin point groups $\mathbf{S}$, confirming the results of [26]. Section 3.1 we demonstrated that the (co-)irreps of $\mathbf{S}$ correspond to those of $\mathbf{G}$ (if $\mathbf{S}$ is unitary) or a black and white point group derived from $\mathbf{G}$ (if $\mathbf{S}$ is non-unitary). In this section, we study the effect of the spin-only group $\mathbf{b}$ on the (co-)representation theory of the spin point groups, in order to systematically obtain the full (co-)representation theory of total spin point groups. We remind the reader

---

[2]Here we will use Hermann-Mauguin (HM) notation, as it is better suited for magnetic and spin point group naming conventions.

Table 4: Non-unitary, nontrivial spin point groups isomorphic to spatial parent group $4mm$ with Litvin number and adapted HM notation in the first and second columns. The third column expresses $\mathbf{S}$ in terms of its unitary halving spin point group $\mathbf{S}^*$ and a choice of antiunitary coset representative $a$ (in correspondence with the HM notation). In the final column, the black & white point group $\mathbf{G}^{\mathrm{BW}}$ to which the non-unitary spin point group corresponds. In this case, there are only two choices of black and white point group: $4m'm'$ and $4'mm'$. The unitary halving subgroups $\mathbf{G}^*$ in these cases are 4 and $mm2$, respectively. In each row, $\mathbf{S}^*$ is isomorphic to the appropriate $\mathbf{G}^*$.

| # | $\mathbf{S}$ | $\mathbf{S}^* + a\mathbf{S}^*$ | $\mathbf{G}^{\mathrm{BW}}$ |
|---|---|---|---|
| 148 | ${}^1 4^m m^m m$ | ${}^1 4 + [m\|m_x]{}^1 4$ | $4m'm'$ |
| 149 | ${}^1 4^{\overline{1}} m^{\overline{1}} m$ | ${}^1 4 + [\overline{1}\|m_x]{}^1 4$ | $4m'm'$ |
| 151 | ${}^m 4^1 m^m m$ | ${}^1 m^1 m^1 2 + [m\|4_z]{}^1 m^1 m^1 2$ | $4'mm'$ |
| 152 | ${}^{\overline{1}} 4^1 m^{\overline{1}} m$ | ${}^1 m^1 m^1 2 + [\overline{1}\|4_z]{}^1 m^1 m^1 2$ | $4'mm'$ |
| 154 | ${}^{2_z} 4^{m_x} m^{m_y} m$ | ${}^2 4 + [m_x\|m_x]{}^2 4$ | $4m'm'$ |
| 155 | ${}^{m_x} 4^{2_z} m^{m_y} m$ | ${}^2 m^2 m^1 2 + [m_x\|4_z]{}^2 m^2 m^1 2$ | $4'mm'$ |
| 156 | ${}^2 4^{\overline{1}} m^m m$ | ${}^2 4 + [\overline{1}\|m_x]{}^2 4$ | $4m'm'$ |
| 157 | ${}^{\overline{1}} 4^2 m^m m$ | ${}^2 m^2 m^1 2 + [\overline{1}\|4_z]{}^2 m^2 m^1 2$ | $4'mm'$ |
| 158 | ${}^m 4^2 m^{\overline{1}} m$ | ${}^2 m^2 m^1 2 + [m\|4_z]{}^2 m^2 m^1 2$ | $4'mm'$ |
| 160 | ${}^{4_z} 4^{m_x} m^{m_{xy}} m$ | ${}^4 4 + [m_x\|m_x]{}^4 4$ | $4m'm'$ |
| 161 | ${}^{\overline{4_z}} 4^{m_x} m^{2_{xy}} m$ | ${}^{2_{xy}} m^{2_{\overline{xy}}} m^{2_z} 2 + [\overline{4_z}\|4_z]{}^{2_{xy}} m^{2_{\overline{xy}}} m^{2_z} 2$ | $4'mm'$ |

that supporting material reviewing the representation theory for non-unitary groups may be found in Appendix A.

Both the spin-only group $\mathbf{b}$ and nontrivial spin point group $\mathbf{S}$ can be expressed as the union of a unitary and an antiunitary coset. We would like to express the total group $\mathbf{X} = \mathbf{b} \times \mathbf{S}$ in this form as well in order to investigate and derive the co-irreps of $\mathbf{X}$, because this form makes the unitary halving subgroup of $\mathbf{X}$ explicit. In general, this can be achieved using the procedure outlined in Appendix D. In the subsections that follow, we say that $\mathbf{S}$ is a spin group of spatial parent group $\mathbf{G}$ and spin-space parent group $\mathbf{B}$.

In the non-coplanar, non-magnetic, and coplanar cases, the co-irreps obtained for the total group $\mathbf{X}$ correspond trivially to existing magnetic group co-irreps. However, for collinear arrangements new co-irreps can be obtained due to the interplay between $\mathbf{b}^\infty$ and the nontrivial spin point group $\mathbf{S}$ (which is a consequence of the semi-direct product structure of $\mathbf{b}^\infty$). In this section, we demonstrate these results for each of the four cases of spin arrangements, depending on whether the nontrivial spin point group $\mathbf{S}$ is unitary or antiunitary.

### 3.2.1 Non-magnetic spin arrangements

In the non-magnetic case, the spin-only group is $\mathbf{b}^{\mathrm{NM}} = \mathbf{SO(3)} \times \{[E\|E], [\tau\|E]\}$. Since in this case $\mathbf{S}$ is always one of the 32 crystallographic point groups (see Section 2.4),

$$\mathbf{X} = \mathbf{b}^{\mathrm{NM}} \times \mathbf{G} = \mathbf{SO(3)} \times \mathbf{G} + \tau\big(\mathbf{SO(3)} \times \mathbf{G}\big).$$

Because the antiunitary coset representative is simply time-reversal, the Dimmock test in equation A.20 corresponds exactly to the Frobenius-Schur indicator for the reality of the irreps of

**SO(3) × G.** The irreps of $\mathbf{SO(3)} \times \mathbf{G}$ are the direct product of irreps $d^{(l)}(\mathbf{SO(3)})$ (for $l \in \mathbb{N}$) and $d^{(\nu)}(\mathbf{G})$, and the indicator will factor into an integral over the elements of $\mathbf{SO(3)}$ and a sum over the elements of $\mathbf{G}$:

$$X^{l,\nu} = \frac{1}{\Omega} \int d\mu \; \chi^{(l)}(R(\theta, \varphi, 2\psi)) \cdot \frac{1}{|\mathbf{G}|} \sum_{g \in \mathbf{G}} \chi^{(\nu)}(g^2),$$

where $\mu$ is the Haar measure for $\mathbf{SO(3)}$, $\Omega$ is an appropriate normalization constant, and $R(\theta, \varphi, \psi) \in \mathbf{SO(3)}$. In the irrep $d^{(l)}(\mathbf{SO(3)})$, the character of a rotation by angle $2\psi$ about any axis (determined by $\theta$ and $\varphi$) is given by $\chi^{(l)}(R(\theta, \varphi, 2\psi)) = 1 + \sum_{m=0}^{l} 2\cos(2m\psi)$. In the integral over $\psi$, the cosine terms will always yield zero, ensuring that the irreps of $\mathbf{SO(3)}$ are real. As a result, the Dimmock indicator for $\mathbf{X}$ will be the same as the Frobenius-Schur reality indicator of $\mathbf{G}$,

$$X^{l,\nu} = \frac{1}{|\mathbf{G}|} \sum_{g \in \mathbf{G}} \chi^{(\nu)}(g^2).$$

Therefore, the irreps of non-magnetic $\mathbf{X}$ will be constructed from the irreps $d^{(l)}(\mathbf{SO(3)}) \times d^{(\nu)}(\mathbf{G})$ based on the reality of the $\nu$ irrep of $\mathbf{G}$.

### 3.2.2 Spatial spin arrangements

For spatial spin arrangements, the spin-only group is $\mathbf{b}^{\mathrm{triv}} = \{[\,E\,\|\,E\,]\}$. Therefore, the total spin point group is equal to a nontrivial spin point group, and the result of Section 3.1 apply. We will reiterate the results here. If $\mathbf{S}$ is unitary, the irreps will simply correspond to the irreps of $\mathbf{G}$ by virtue of the isomorphism theorem in Section 3.1.

If $\mathbf{S} = \mathbf{S}^* + a\mathbf{S}^*$ is antiunitary, then it is a nontrivial non-unitary spin point group, and as such has co-irreps that correspond to a black and white point group. In particular, it corresponds to the black and white point group where in the HM symbol, improper spin elements are replaced by primes. This identification is explicitly tabulated in Table 7.

### 3.2.3 Coplanar spin arrangements

For coplanar spin arrangements, the spin-only group is $\mathbf{b}^{\mathbb{Z}_2} = \{[\,E\,\|\,E\,], [\,\tau 2_{\mathbf{n}}\,\|\,E\,]\}$, where $\mathbf{n}$ is the normal vector to the plane of the spins. If $\mathbf{S}$ is unitary, then

$$\mathbf{X} = \mathbf{b}^{\mathbb{Z}_2} \times \mathbf{S} = \{[\,E\,\|\,E\,], [\,\tau 2_{\mathbf{n}}\,\|\,E\,]\} \times \mathbf{S} = \mathbf{S} + [\,\tau 2_{\mathbf{n}}\,\|\,E\,]\mathbf{S}.$$

The co-irreps for $\mathbf{X}$ will be formed from the irreps of $\mathbf{S}$, which (as a unitary nontrivial SPG) are the same as for the spatial parent group $\mathbf{G}$. The Dimmock indicator (eq. A.20) for the $\mu$−th irrep of $\mathbf{S}$ (with elements given by $[\,s_s\,\|\,s_o\,]$) will be given by

$$X^{\mu} = \frac{1}{|\mathbf{S}|} \sum_{s \in \mathbf{S}} \chi^{(\mu)}\left(([\,\tau 2_{\mathbf{n}}\,\|\,E\,][\,s_s\,\|\,s_o\,])^2\right) = \frac{1}{|\mathbf{S}|} \sum_{s \in \mathbf{S}} \chi^{(\mu)}\left([\,2_{\mathbf{n}}s_s\,\|\,s_o\,]^2\right)$$

$$= \frac{1}{|\mathbf{S}|} \sum_{s \in \mathbf{S}} \chi^{(\mu)}\left([\,(2_{\mathbf{n}}s_s)^2\,\|\,s_o^2\,]\right).$$

Note that the axis of $s_s$ must be parallel or perpendicular to the plane of the spins, otherwise it will not map the spin plane back onto itself. If it is parallel to the spin plane then it is a $\pi$ rotation about some axis perpendicular to $\mathbf{n}$, and $2_{\mathbf{n}}2_{\perp \mathbf{n}} = 2_{\perp \mathbf{n}'}$ is a $\pi$ rotation about another axis perpendicular to $\mathbf{n}$, and so $(2_{\mathbf{n}}2_{\perp \mathbf{n}})^2 = E$. If $s_s$ is a rotation about $\mathbf{n}$, then $2_{\mathbf{n}}$ commutes with $s_s$ and $(2_{\mathbf{n}}s_s)^2 = s_s^2$. So, the indicator for the $\mu$−th irrep of $\mathbf{S}$ becomes

$$X^{\mu} = \begin{cases} \frac{1}{|\mathbf{S}|} \sum_{s \in \mathbf{S}} \chi^{(\mu)}\left([\,s_s\,\|\,s_o\,]^2\right), & s_s \text{ axis parallel to } \mathbf{n}, \\ \frac{1}{|\mathbf{S}|} \sum_{s \in \mathbf{S}} \chi^{(\mu)}\left([\,E\,\|\,s_o\,]^2\right), & s_s \text{ axis perpendicular to } \mathbf{n}. \end{cases}$$

Since $\mathbf{S} \cong \mathbf{G}$, we see explicitly that both of these cases correspond to the indicators of $\mathbf{G}$ for the grey group $\mathbf{G} + \tau\mathbf{G}$.

If $\mathbf{S} = \mathbf{S}^* + a\mathbf{S}^*$ is non-unitary and $a = [\tau u_s \| u_o]$, then (by Appendix D)

$$\mathbf{X} = \mathbf{b}^{\mathbb{Z}_2} \times \mathbf{S} = \{E, \tau 2_{\mathbf{n}}\} \times (\mathbf{S}^* + a\mathbf{S}^*) = \mathbf{S}^* + [\, 2_{\mathbf{n}} u_s \| u_o \,]\mathbf{S}^* + [\, \tau 2_{\mathbf{n}} \| E \,](\mathbf{S}^* + [\, 2_{\mathbf{n}} u_s \| u_o \,]\mathbf{S}^*)\,.$$

In this case, the unitary halving subgroup is also a nontrivial SPG of the family $\mathbf{G}$ (although its spin part may be smaller).[3] Using a similar analysis as for the unitary case, we find that the co-irreps of $\mathbf{X}$ will also correspond to the co-irreps of $\mathbf{G} + \tau\mathbf{G}$.

### 3.2.4 Collinear spin arrangements

For collinear spin arrangements with spins aligned along the $\mathbf{n}$ axis, the spin-only group is given by $\mathbf{b}^\infty = \mathbf{SO(2)} \rtimes \{E, \tau 2_{\perp\mathbf{n}}\}$. In Section 2.4, we demonstrated that for $\mathbf{X}$ to have the structure of a direct product and to avoid redundancy in identifying collinear spin point groups, the spin elements of $\mathbf{S}$ can only come from one of two groups: $\mathbf{B} = 1$ if $\mathbf{S}$ is unitary, and $\mathbf{B} = \bar{1}$ if $\mathbf{S}$ is nonunitary.

If $\mathbf{S}$ is unitary, then each element of $\mathbf{S}$ has the identity in the spin part, so $\mathbf{S}$ is essentially equal to the spatial parent group $\mathbf{G}$. The total spin group is given by

$$\begin{aligned} \mathbf{X} = \mathbf{b}^\infty \times \mathbf{S} &= \mathbf{SO(2)} \times \mathbf{S} + [\, \tau 2_{\perp\mathbf{n}} \| E \,](\mathbf{SO(2)} \times \mathbf{S}) \\ &= \mathbf{SO(2)} \times \mathbf{G} + [\, \tau 2_{\perp\mathbf{n}} \| E \,](\mathbf{SO(2)} \times \mathbf{G})\,. \end{aligned}$$

The co-irreps of $\mathbf{X}$ will be induced from the irreducible representations[4] of $\mathbf{SO(2)} \times \mathbf{G}$

$$d^{(\mu,\nu)}(\mathbf{SO(2)} \times \mathbf{G}) = \delta^{(\mu)}(\mathbf{SO(2)}) \otimes \Delta^{(\nu)}(\mathbf{G})\,,$$

where $\mu \in \mathbb{Z}$ and $\nu$ runs through the irreps of $\mathbf{G}$. These irreps have Dimmock indicators (see equation A.20) given by

$$\begin{aligned} X^{\mu,\nu} &= \frac{1}{2\pi} \int_0^{2\pi} d\varphi \, \frac{1}{|\mathbf{S}|} \sum_{s \in \mathbf{S}} \left( [\, \tau 2_{\perp\mathbf{n}} R_{\mathbf{n}}(\varphi) \| s_o \,]^2 \right) \\ &= \left( \frac{1}{2\pi} \int_0^{2\pi} d\varphi \, (\tau 2_{\perp\mathbf{n}} R_{\mathbf{n}}(\varphi))^2 \right) \left( \frac{1}{|\mathbf{G}|} \sum_{g \in \mathbf{G}} g^2 \right)\,, \end{aligned}$$

due to the direct product structure of $\mathbf{SO(2)} \times \mathbf{G}$, the spin elements $s \in \mathbf{S}$ being of the form $s = [E\|g]$ (and $g \in \mathbf{G}$), and denoting an arbitrary element of $\mathbf{SO(2)}$ as a rotation $R_{\mathbf{n}}(\varphi)$ where $\mathbf{n}$ is the axis of the spins. Notice that $2_{\perp\mathbf{n}} R_{\mathbf{n}}(\varphi)$ is an $\pi$−rotation about a different axis perpendicular to $\mathbf{n}$, i.e. $2_{\perp\mathbf{n}} R_{\mathbf{n}} = 2'_{\perp\mathbf{n}}$. Since this is a two-fold rotation its square is the identity element, and so the squared spin element in the argument of the Dimmock indicator is simply

$$(\tau 2_{\perp\mathbf{n}} R_{\mathbf{n}}(\varphi))^2 = (2'_{\perp\mathbf{n}})^2 = E\,.$$

As a result,[5] the Dimmock indicator becomes

$$X^{\mu,\nu} = \left( \frac{1}{2\pi} \int_0^{2\pi} d\varphi \, \chi^{(\mu)}(E) \right) \left( \frac{1}{|\mathbf{G}|} \sum_{g \in \mathbf{G}} \chi^{(\nu)}(g^2) \right) = \frac{1}{|\mathbf{G}|} \sum_{g \in \mathbf{G}} \chi^{(\nu)}(g^2)\,.$$

---

[3]If $a$ was any reflection, the spin-space parent group $\mathbf{B}$ will either remain the same, or swap to an isomorphic $\mathbf{B}'$. However, if $a$ was $S_{2n}$, for $n = 1$ or $n = 3$, $\mathbf{B}$ will have half as many elements, whereas $n = 2$ remains unchanged. This covers all possible cases of index-two subgroup coset representatives (see [28]).

[4]The irreducible representations of $\mathbf{SO(2)}$ are given by $\delta^{(\mu)}(R(\varphi)) = e^{i\mu\varphi}$, where $R(\varphi) \in \mathbf{SO(2)}$ is a rotation by angle $\varphi$, and $\mu \in \mathbb{Z}$.

[5]We also use the fact that the character of the identity element is the dimension of the representation, and that the irreducible representations of $\mathbf{SO(2)}$ are one dimensional so $\chi^{(\mu)}(E) = 1$.

Table 5: Summarizing which spin point groups have new (co-)irrep content. For those with existing (co-)irreps, the reader is referenced to the appropriate group's (co-)irreps. For the collinear groups, see Appendix F. Note that the collinear groups with unitary (non-unitary) $\mathbf{S}$ have Dimmock indicators corresponding to the grey point group $\mathbf{G}$ ($\mathbf{G}_{1/2}$).

| Spin Point Group Type | New (Co-)irreps? | Group with Corresponding (Co)-irreps |
|---|---|---|
| Nontrivial | No | Black & White group (see table 7) |
| Non-magnetic | No | Grey group $\mathbf{G} + \tau\mathbf{G}$ |
| Coplanar | No | Grey group $\mathbf{G} + \tau\mathbf{G}$ |
| Collinear | Yes | N/A (see Appendix F). |

This means that the irreps $d^{(\mu,\nu)}(\mathbf{SO(2)} \times \mathbf{G})$ have types that depend only on the irrep of the spatial parent group, $\Delta^{(\nu)}(\mathbf{G})$. Further, the type of $d^{(\mu,\nu)}(\mathbf{SO(2)} \times \mathbf{G})$ in $\mathbf{X}$ will be the same as for $\Delta^{(\nu)}(\mathbf{G})$ in the grey point group $\mathbf{G} + \tau\mathbf{G}$, and the tables of these co-irreps are given in Appendix F.

If $\mathbf{S}$ is nonunitary with unitary halving subgroup $\mathbf{S}^*$, then the spin elements of $\mathbf{S}$ are taken from the group $\mathbf{B} = \bar{1}$, the inversion group. Now, $\mathbf{S} = \mathbf{S}^* + [\tau\|s_o]\mathbf{S}^*$, and $\mathbf{S}^*$ is a unitary nontrivial spin point group. The total group $\mathbf{X}$ can be expressed as

$$\mathbf{X} = \mathbf{b}^\infty \times \mathbf{S} = (\mathbf{SO(2)} + \tau 2_{\perp\mathbf{n}}\mathbf{SO(2)}) \times (\mathbf{S}^* + [\tau\|s_o]\mathbf{S}^*)$$
$$= \mathbf{SO(2)} \times \mathbf{S}^* + [\,2_{\perp\mathbf{n}}\|s_o\,]\,\mathbf{SO(2)} \times \mathbf{S}^* + [\tau 2_{\perp\mathbf{n}}\|E]\Big(\mathbf{SO(2)} \times \mathbf{S}^* + [\,2_{\perp\mathbf{n}}\|s_o\,]\,\mathbf{SO(2)} \times \mathbf{S}^*\Big),$$

where we have used the result in Appendix D for the direct product of antiunitary groups.

The co-irreps of $\mathbf{X}$ must be induced from the irreps of

$$\mathbf{X}_{1/2} = \mathbf{SO(2)} \times \mathbf{S}^* + [\,2_{\perp\mathbf{n}}\|s_o\,]\,\mathbf{SO(2)} \times \mathbf{S}^*,$$

but the irreps of $\mathbf{X}_{1/2}$ are not known a priori. Therefore, we must first find the irreps of $\mathbf{X}_{1/2}$, and we do so using the irreps of $\mathbf{SO(2)} \times \mathbf{S}^*$. Since $\mathbf{S}^*$ is a unitary nontrivial spin point group, it is isomorphic to its parent spatial group; this spatial parent group will also be a halving subgroup of $\mathbf{G}$ (the parent group of $\mathbf{S}$) and so we call it $\mathbf{G}_{1/2}$. Then, the irreps of $\mathbf{SO(2)} \times \mathbf{S}^*$ will be equal to those of $\mathbf{SO(2)} \times \mathbf{G}_{1/2}$. From here, to obtain there irreps of $\mathbf{X}_{1/2}$, we use the standard algorithm for induction from a subgroup of index two [8, 29, 30]. An explanation of this procedure as well as an example are provided in Appendix E. With the irreps of $\mathbf{X}_{1/2}$, we derive the co-irreps of $\mathbf{X}$ using the procedure outlined in Appendix A and the resulting tables are provided in Appendix F.

We comment on the Dimmock indicators in this case. The squared elements appearing in the Dimmock indicator are $[E\|g^2]$ with $g \in \mathbf{G}_{1/2}$ for the first coset of $\mathbf{SO(2)} \times \mathbf{S}^*$, and $[R_{\mathbf{n}}(2\varphi)\|(s_o g)^2]$ for the second coset. Using results from Appendix E.1 one can see that the latter contribute zero to the Dimmock indicator. This is because these group elements have characters with a factor of $e^{2i\mu\varphi}$, which when integrated from 0 to $2\pi$ gives zero. The former is the same as the indicator for the grey point group $\mathbf{G}_{1/2} + \tau\mathbf{G}_{1/2}$.

In contrast to the coplanar, spatial, and non-magnetic spin point groups, the co-irreps of collinear spin point groups to *not* correspond to any of the well-known and tabulated magnetic (point) group irreps. The maximal irrep dimension was found to be six. The conclusions regarding existing versus new (co-)irreps are summarized in Table 5.

As an example of these nontrivially new co-irreps and the procedure used to induce them, we take the non-unitary parent group $\mathbf{S} = {}^{\bar{1}}4/{}^1m{}^1m{}^{\bar{1}}m$. In this case, the total spin point group

is given by $\mathbf{X} = \mathbf{b}^\infty \times \overline{1}4/^1m^1m^1m$. In this case, $\mathbf{X}$ can be expressed as

$$\mathbf{X} = \mathbf{X}_{1/2} + [\tau 2_{\perp \mathbf{n}} \| E]\mathbf{X}_{1/2},$$

where

$$\mathbf{X}_{1/2} = \mathbf{SO(2)} \times {}^1m^1m^1m + [2_{\perp \mathbf{n}} \| 4_z]\mathbf{SO(2)} \times {}^1m^1m^1m,$$

by Appendix D, since

$$\mathbf{S} = \mathbf{S}^* + [\tau \| s_o]\mathbf{S}^*$$

can be written as $\overline{1}4/^1m^1m^1m = {}^1m^1m^1m + [\overline{1}\|4_z]^1m^1m^1m$. To find the co-irreps of the total spin group, we must first find the irreducible represntations of $\mathbf{X}_{1/2}$, which is demonstrated in Appendix E.2. From here, we determine the types of the irreps of $\mathbf{X}_{1/2}$ by Dimmock's test (A.20). In this case, all irreps of $\mathbf{X}_{1/2}$ belong to case (a), and so we use equation A.16 to find the total co-irreps, to find the table on page 70.

# 4 Applications

## 4.1 Rutile altermagnetism

Altermagnetism in metals is the appearance of spin split electronic bands that are anisotropic in momentum space in the absence of spin-orbit coupling [20, 21, 31]. A characteristic feature of these systems as currently understood is the presence of collinear antiferromagnetic order on a pair of sublattices with neither inversion nor translation connecting the sublattices. One example of such magnetism is in rutiles with chemical formula $MX_2$ for example the metallic antiferromagnet $RuO_2$ in which band structure calculations reveal a d wave pattern of spin splitting greatly in excess of the splitting arising from spin-orbit coupling [31] which is consistent with subsequent experimental findings [32–35]. As has been observed in the literature on these systems, the absence of spin-orbit coupling calls for an understanding based on spin groups. Therefore, in this section, we give a toy model of rutile altermagnetism and provide an explanation of the salient features of the band structure from the point of view of symmetry.

We consider the lattice structure in Fig. 2. The magnetic ions live on a body-centred tetragonal lattice. The space group of this structure is $P4_2/mnm$ #136 with magnetic ions M on Wyckoff positions $2a$ and X ions on $2f$. The two sublattices are connected by $C_{4z}$ and a translation $(1/2, 1/2, 1/2)$ and, notably, not by a pure inversion or translation. The lattice symmetries are such that there are two inequivalent third neighbour hoppings and that on the different magnetic sublattices the 11 and $\overline{1}1$ hoppings are interchanged. These features are indicated in Fig. 2.

We consider a model of non-interacting fermions with a Hund coupling to fixed classical localized moments $\boldsymbol{h}$ in a Néel structure:

$$H = -t_1 \sum_{\langle i,j \rangle} c_{i\sigma}^\dagger c_{j\sigma} - \sum_{a=1,2} t_3^a \sum_{\langle\langle\langle i,j \rangle\rangle\rangle_a} c_{i\sigma}^\dagger c_{j\sigma} - J \sum_i c_{i\alpha}^\dagger \boldsymbol{h}_i \cdot \boldsymbol{\sigma}_{\alpha\beta} c_{i\beta}, \tag{11}$$

where $\boldsymbol{h}_i = (0,0,1)$ on $(0,0,0)$ Wyckoff sites and $(0,0,-1)$ on the $(1/2,1/2,1/2)$ position. Re-writing this in crystal momentum space, the Hamiltonian is block diagonal in spin space with blocks

$$H_\uparrow(\mathbf{k}) = \begin{pmatrix} c_{\mathbf{k}A\uparrow}^\dagger & c_{\mathbf{k}B\uparrow}^\dagger \end{pmatrix} \begin{pmatrix} -J - t_3^1\gamma_{3,\mathbf{k}}^1 - t_3^2\gamma_{3,\mathbf{k}}^2 & -t_1\gamma_{1,\mathbf{k}} \\ -t_1\gamma_{1,\mathbf{k}} & J - t_3^1\gamma_{3,\mathbf{k}}^2 - t_3^2\gamma_{3,\mathbf{k}}^1 \end{pmatrix} \begin{pmatrix} c_{\mathbf{k}A\uparrow} \\ c_{\mathbf{k}B\uparrow} \end{pmatrix}, \tag{12}$$

$$H_\downarrow(\mathbf{k}) = \begin{pmatrix} c_{\mathbf{k}A\downarrow}^\dagger & c_{\mathbf{k}B\downarrow}^\dagger \end{pmatrix} \begin{pmatrix} J - t_3^1\gamma_{3,\mathbf{k}}^1 - t_3^2\gamma_{3,\mathbf{k}}^2 & -t_1\gamma_{1,\mathbf{k}} \\ -t_1\gamma_{1,\mathbf{k}} & -J - t_3^1\gamma_{3,\mathbf{k}}^2 - t_3^2\gamma_{3,\mathbf{k}}^1 \end{pmatrix} \begin{pmatrix} c_{\mathbf{k}A\downarrow} \\ c_{\mathbf{k}B\downarrow} \end{pmatrix}, \tag{13}$$

where

$$\gamma_{1,\mathbf{k}} = \sum_{\mu=1,4} \cos\left(\mathbf{k} \cdot \mathbf{r}_\mu\right), \tag{14}$$

$$\gamma_{3,\mathbf{k}}^1 = \cos(k_x + k_y), \tag{15}$$

$$\gamma_{3,\mathbf{k}}^2 = \cos(k_x - k_y), \tag{16}$$

and $\mathbf{r}_1 = (1/2)(a\hat{\mathbf{x}} + a\hat{\mathbf{y}} + c\hat{\mathbf{z}})$, $\mathbf{r}_2 = (1/2)(a\hat{\mathbf{x}} + a\hat{\mathbf{y}} - c\hat{\mathbf{z}})$, $\mathbf{r}_3 = (1/2)(a\hat{\mathbf{x}} - a\hat{\mathbf{y}} + c\hat{\mathbf{z}})$, $\mathbf{r}_4 = (1/2)(-a\hat{\mathbf{x}} + a\hat{\mathbf{y}} + c\hat{\mathbf{z}})$.

The combination of the pattern of hoppings and the time-reversal breaking from the anti-collinear moments leads to electronic bands with a characteristic d wave spin splitting as shown in Fig. 3. As the magnetic order breaks the full rotational symmetry down to **SO(2)** that keeps the spin projection as a good quantum number the spin up or down nature of the bands is present across the zone. The d wave pattern originates from crossing of bands with well-defined spin along the lines $(H,0,0)$ and $(0,H,0)$. Any symmetric hopping parameters preserve this feature. The problem that faces us is to understand the origin of the band crossing along these lines from the perspective of symmetry.

Let us focus on the zone centre where the total spin-space group reduces to a spin point group. The identity of the nontrivial spin point group is $^{\bar{1}}4/^1 m^1 m^{\bar{1}}m$. In the tables, this is group number 195. In short, the time-reversal operation is linked to real space operations that take one magnetic sublattice into another. The representation theory of this group has been worked out in Appendix E.2 with the full co-irrep tables given in Appendix F. The first part of the calculation of these tables is to work out the irreps of the unitary halving subgroup of $^{\bar{1}}4/^1 m^1 m^{\bar{1}}m$ times the collinear pure spin group. This is nontrivial as the direct product of antiunitary groups leads to unitary part $\mathbf{SO(2)} \otimes^1 m^1 m^1 m + [2_{\perp\mathbf{n}}\|4]\mathbf{SO(2)} \otimes^1 m^1 m^1 m$. The interpretation of this is that the $^1 m^1 m^1 m$ coset contains elements that preserve the sublattice and the spin while the second piece $[2_{\perp\mathbf{n}}\|4] \otimes^1 m^1 m^1 m$ reverses both spin and sublattice.

The eigenstates are organized with the first pair and last pair of components referring to opposite spin orientations. Then the components of each of these pairs is resolved by magnetic sublattice. Inspection of the eigenstates at the $\Gamma$ point reveals two doubly degenerate states of the form $\{(\alpha, \beta, 0, 0), (0, 0, \beta, \alpha)\}$ and $\{(-\beta, \alpha, 0, 0), (0, 0, \alpha, -\beta)\}$. The eigenstates within each doubly degenerate pair are therefore related by sublattice (and spin) swaps. Within each degenerate eigenspace, we therefore expect the halving subgroup elements that preserve the sublattice to be of the form

$$\begin{pmatrix} e^{i\phi} & 0 \\ 0 & e^{-i\phi} \end{pmatrix}, \tag{17}$$

where the diagonal nonzero elements indicate that the sublattice is preserved. The elements that swap the sublattice and spin are

$$\begin{pmatrix} 0 & e^{-i\phi} \\ e^{i\phi} & 0 \end{pmatrix}, \tag{18}$$

where the phases are fixed by the fact that the perpendicular spin rotation reverses the **SO(2)** rotation sense. The tables in Appendix F contain a 2D co-irrep of precisely this form: $\Gamma_{11}$ of $^{\bar{1}}4/^1 m^1 m^{\bar{1}}m$. In the list of elements, the only swap of the spins and sublattices arises from the element $[2_{\perp\mathbf{n}}\|4]$ and its partner after multiplication by $[\tau 2_{\perp\mathbf{n}}\|E]$. The element labelled $C_{2y}$ in the table does not swap the sublattices as the $y$ axis in the group convention corresponds to the $\hat{\mathbf{x}} - \hat{\mathbf{y}}$ direction in the lattice convention of this section.

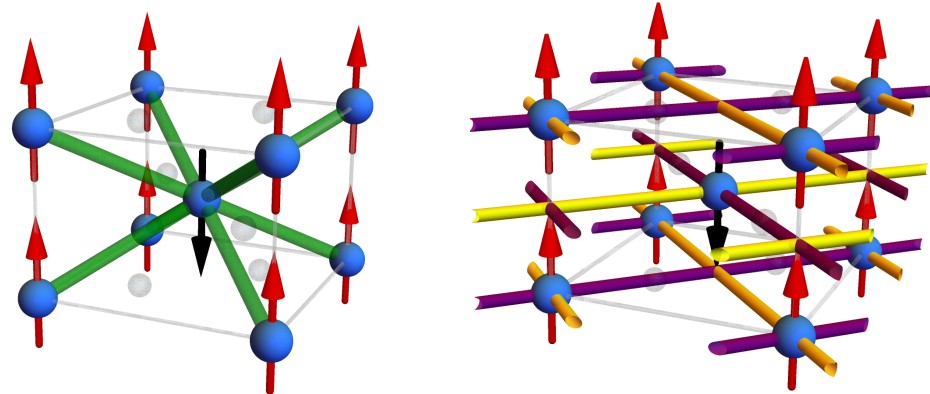

Figure 2: Crystal structure of rutile $MX_2$ where the magnetic M ions (blue) are on the tetragonal unit cell vertices and the X ions (translucent) crucially are arranged so that the local environments around the two magnetic sublattices are related by a $C_{4z}$ operation and a translation. The ↑ / ↓ sublattice structure is also shown, and the left figure indicates the $t_1$ hopping, while the right figure shows two inequivalent $t_3$ hoppings.

Now we consider the $(H, 0, 0)$ direction for points in the interior of the zone. We enumerate all group elements including those obtained by multiplication by $[\tau 2_{\perp \mathbf{n}} \| E]$ and ask which leave the momentum invariant. We find

$$\mathbf{SO(2)} \otimes \left([E\|E] + [\tau 2_{\perp \mathbf{n}} \| I]\right) \otimes \left([E\|E] + [E\|I2_z] + [\tau\|I2_x] + [\tau\|2_y]\right) \tag{19}$$

in the lattice convention of the toy model. This is isomorphic to the antiunitary collinear spin point group $\mathbf{b}^{\infty} \otimes^1 \mathbf{m}^{\bar{1}}\mathbf{m}^{\bar{1}}\mathbf{2}$ or Litvin number 40 with halving unitary subgroup $\mathbf{SO(2)} \otimes^1 m + [2_{\perp \mathbf{n}}\|2]\mathbf{SO(2)} \otimes^1 m$. The co-irreps include a 2D irrep, $\Gamma_5$, with matrices of the same form as for the $\Gamma$ point case albeit with fewer elements. These matrices concord with the transformation properties of the eigenstates along this line.

In contrast, for the $(H, H, 0)$ direction, the spin group that leaves the momentum invariant is

$$\mathbf{SO(2)} \otimes \left([E\|E] + [\tau 2_{\perp \mathbf{n}} \| I]\right) \otimes \left([E\|E] + [E\|I2_{110}] + [E\|I2_z] + [E\|2_{1\bar{1}0}]\right). \tag{20}$$

The nontrivial spin group is therefore unitary so there is no doubling of the irrep dimension beyond that of the unitary point group. As the point group is isomorphic to $mm2$ which has only 1D irreps so there is no symmetry enforced degeneracy along this line. This observation is consistent with the band structure from the toy model. These degeneracies remain in the presence of breaking down to the magnetic space group when the moment is pinned along the crystal $c$ axis.

It is interesting to note that the SPGs listed above strictly do *not* fall into Litvin's classification [25]. The spin-only groups in each of these cases is reduced to $\mathbf{SO(2)}$, as opposed to the collinear $\mathbf{b}^{\infty}$. What previously was a spin-only element, $[\tau 2_{\perp \mathbf{n}} \| E]$ is here paired with a real-space inversion. The group $\mathbf{SO(2)} \otimes \{[E\|E], [\tau 2_{\perp \mathbf{n}}\|I]\}$ is isomorphic to $\mathbf{b}^{\infty}$, but it is not conjugate to $\mathbf{b}^{\infty}$ in $\mathbf{O(3)} \otimes \mathbf{O(3)}$. Generically, the study of reciprocal space symmetries may give rise to SPGs beyond the Litvin classification.

## 4.2 MnTe

As another example, we consider the magnetic structure of MnTe which is also consistent with altermagnetism. The underlying crystal structure has space group $P6_3/mmc$ #194 with

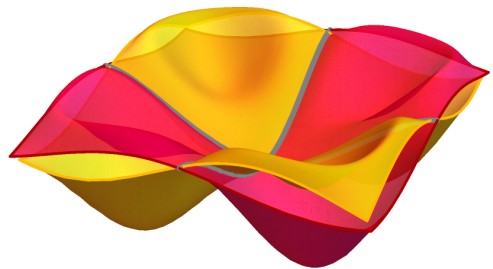

Figure 3: The two lowest energy bands of the toy model described in the main text. The colours indicate the spin state. The total spin is a good quantum number and it exhibits a d wave pattern in momentum space.

hexagonal crystal class. The magnetic ions live at Wyckoff position $2a$: $(0,0,0)$ and $(0,0,1/2)$ and the decorating Te ions live at Wyckoff position $2c$: $(1/3,2/3,1/4)$ and $(2/3,1/3,3/4)$. One may build a toy model for the band structure of this material in the same way as for the rutile case. Details may be found in Ref. [36]. One finds degeneracies along $(k_x, 0, k_z)$ and in the $(k_x, k_y, 0)$ plane.

The spin point group for the magnetic order in this system is $\mathbf{b}^\infty \otimes {}^{\bar{1}}6/{}^{\bar{1}}m{}^1m{}^{\bar{1}}m$ or Litvin number 443. We consider three planes in the Brillouin zone and determine the spin point group for each. The first is the $k_z = 0$ plane. We find

$$\mathbf{SO(2)} \otimes ([E\|E] + [\tau 2_{\perp\mathbf{n}}\|I]) \otimes ([E\|E] + [\tau\|2_z]) . \tag{21}$$

This is isomorphic to $\mathbf{b}^\infty \otimes^{\bar{1}} 2$ which has 2D co-irrep $\Gamma_3$ corresponding to the degenerate bands in the toy model. Along $(k_x, 0, k_z)$ the spin point group is

$$\mathbf{SO(2)} \otimes ([E\|E] + [\tau 2_{\perp\mathbf{n}}\|I]) \otimes ([E\|E] + [\tau\|2_{120}]) , \tag{22}$$

which again is isomorphic to $\mathbf{b}^\infty \otimes^{\bar{1}} 2$. Finally, along $(0, k_y, k_z)$ the spin point group is

$$\mathbf{SO(2)} \otimes ([E\|E] + [\tau 2_{\perp\mathbf{n}}\|I]) \otimes ([E\|E] + [E\|m_{100}]) , \tag{23}$$

or $\mathbf{b}^\infty \otimes^1 m$. As the nontrivial group is unitary with only 1D irreps there is no symmetry enforced degeneracy along this line.

It is interesting to note that the material MnTe has a magnetic anisotropy pinning the moment along six-fold axis in the $ab$ plane. The magnetic space group is a type III group 63.462. This has only 1D irreps along the directions considered above. Therefore the spin point group analysis reveals degeneracies at zero spin-orbit coupling that are not present in the anisotropic case. This may be useful to assess the proximity of altermagnets to the limit where the spin group is an exact description.

## 4.3 Magnons in Heisenberg-Kitaev model

Here we give an example of how spin point groups can be used to determine useful information on the spectrum of magnetic excitations following a model studied in Ref. [19]. Additionally, we discuss new forms of spin-only groups extending the Litvin-Opechowski classes when the Hamiltonian has anisotropies which break full rotational symmetry.

We consider the Kitaev-Heisenberg model on a hyperhoneycomb lattice, a tri-coordinated lattice in three-dimensions represented in Fig. 4. The space group of the structure is Fddd #70 with magnetic ions on Wyckoff positions $16g$. Such a lattice can be found for example

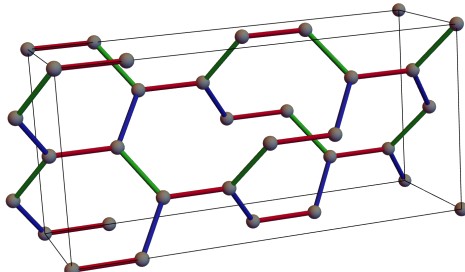

Figure 4: Plot of a unit cell of the hyperhoneycomb lattice with nearest neighbour bonds coloured according to the Kitaev coupling on each bond.

in the iridium $Ir^{4+}$ sublattice in $\beta$-$Li_2IrO_3$. In the material, the magnetic ions live in cages formed by edge-sharing octahedra of oxygen ions in such a way that the Ir-O-Ir bond angle is 90°. Such a configuration allows for a microscopic mechanism leading to Kitaev-Heisenberg couplings [37, 38].

The Hamiltonian reads

$$H = J \sum_{\langle ij \rangle} \boldsymbol{J}_i \cdot \boldsymbol{J}_j + K \sum_{\langle ij \rangle_\gamma} J_i^\gamma J_j^\gamma , \tag{24}$$

where $\boldsymbol{J}_i$ are the magnetic moments. In the paramagnetic state, the Heisenberg model alone has full spin rotational symmetry $\mathbf{SO(3)}$ plus time-reversal in addition to the space group symmetry. The presence of the Kitaev exchange coupling breaks $\mathbf{SO(3)}$ to three perpendicular $C_2$ rotations forming the discrete group $D_2 = 222$. The onset of magnetic order further reduces the symmetry to a subgroup mixing spin and space transformations. Magnon excitations propagating on top of a particular magnetic order inherit those symmetries.

In this model there are four phases in zero external field: the collinear ferromagnet, a Néel phase, and two further antiferromagnetic phases called skew-stripey and skew-zigzag [39, 40]. These collinear phases lie along one of the Cartesian axes ([001] direction chosen here). A large enough applied magnetic field can tune freely the direction of the ferromagnetic state, allowing us to explore further ordered directions. The summary of these phases and their relative spin point group at the zone center $\Gamma$ can be found in Table 6.

Let us start our discussion with the pure Heisenberg model, where two phases are possible: Heisenberg ferromagnetic and Néel magnetic order. The Heisenberg model is rotationally invariant so all spin ordering directions $[xyz]$ are equivalent. For both types of magnetic order, the spin-only group is composed of a free axial rotation around the spin direction $\mathbf{SO(2)}$ and a perpendicular $C_{2\perp}$ rotation coupled with time-reversal symmetry, giving the group indicated as $\mathbf{b}^\infty$.

The Heisenberg Néel state has symmetries captured by the total spin group $\mathbf{b}^\infty \times {}^1m^1m^{\bar{1}}m$, where the nontrivial spin point group is number 60 in the Litvin enumeration. The representation of the spin wave spectrum associated with this group can be obtained by the full co-irrep tables given in Appendix F and gives

$$\rho_{\text{Néel}}^{\Gamma} = \Gamma_9(2) + \Gamma_{10}(2) , \tag{25}$$

where the number in parentheses shows the dimensionality of the irreps. We expect therefore two doubly degenerate magnon modes. Considering a general position $GP$ in the zone the spin point group is isomorphic to $\mathbf{SO(2)} \times {}^{\bar{1}}\bar{1}$ with spin wave representation:

$$\rho_{\text{Néel}}^{GP} = 2\Gamma_3(2) . \tag{26}$$

Table 6: Different collinear phases in the hyperhoneycomb lattice Heisenberg-Kitaev model and relative spin point groups at Brillouin zone center Γ. The ordering direction $[xyz]$ is given in Kitaev coordinates and indicates a representative direction of the phase among other equivalent. The last column specifies when a double degeneracy is predicted in the spin wave spectrum by the spin point group representation theory.

| Phase | Spin Point Group | Degeneracy |
|---|---|---|
| Heisenberg FM $[x\,y\,z]$ | $\mathbf{b}^\infty \times {}^1m^1m^1m$ | |
| Heisenberg Néel $[x\,y\,z]$ | $\mathbf{b}^\infty \times {}^1m^1m^{\bar{1}}m$ | ✓ |
| Néel $[0\,0\,1]$ | $\mathbf{2'2'2} \times {}^mm^2m^1m$ | ✓ |
| Skew-Stripy $[0\,0\,1]$ | $\mathbf{2'2'2} \times {}^2m^2m^{\bar{1}}m$ | ✓ |
| Skew-ZigZag $[0\,0\,1]$ | $\mathbf{2'2'2} \times {}^2m^{\bar{1}}m^mm$ | ✓ |
| FM $[0\,0\,1]$ | $\mathbf{2'2'2} \times {}^1m^mm^mm$ | |
| FM $[1\text{-}1\,0]$ | $\mathbf{2'} \times {}^1m^2m^2m$ | |
| FM $[1\,1\,z]$ | ${}^1m^mm^mm$ | |
| FM $[1\,0\,0]$ | $\mathbf{2'2'2} \times {}^12/{}^1m$ | |
| FM $[x\,y\,0]$ | $\mathbf{2'} \times {}^12/{}^1m$ | |
| FM $[x\,y\,z]$ | ${}^12/{}^1m$ | |

Since both the zone centre, which is the highest symmetric position, and the general position, the lowest symmetric one, predict double degenerate modes we expect two double degenerate magnon bands everywhere inside the Brillouin zone. This consideration leaves out possible extra degeneracy on the zone boundary due to non-symmorphic spin space symmetry as discussed in [19].

In contrast, the ferromagnetic state at zone center is described by the total spin group $\mathbf{b}^\infty \times {}^1m^1m^1m$, where the nontrivial spin point group is Litvin number 54. The spin wave representation gives

$$\rho_{\text{FM}}^\Gamma = \Gamma_1(1) + \Gamma_2(1) + \Gamma_3(1) + \Gamma_4(1), \tag{27}$$

while the general position $GP$ in the zone has spin group $\mathbf{SO(2)} \times {}^1\bar{1}$ with representation

$$\rho_{\text{FM}}^{GP} = 2\Gamma_1(1) + 2\Gamma_2(1). \tag{28}$$

The ferromagnetic case therefore has four 1D bands everywhere inside the zone.

Following this example, we highlight a general mechanism that leads to degenerate volumes in the magnon spectrum. In particular, we consider Heisenberg models with antiferromagnetic (AFM) sublattices when there is a symmetry that spatially maps up-spin sublattices into down-spin sublattice together with a spin flip coming either from time-reversal or by a perpendicular $C_{2\perp}$ spin rotation. When this symmetry also maps the reciprocal vector $\mathbf{k}$ to itself then there is a double degeneracy everywhere in the Brillouin zone (general position GP). Here we are dealing with point group operations and we do not consider case of black and white translations which can satisfy this requirement. A point group symmetry which conserves $\mathbf{k}$ is PT symmetry, inversion coupled with time-reversal symmetry. This symmetry, if present alone without additional spin rotations, maps the AFM sublattices into one another leading to a degenerate spectrum.

From the full co-irrep tables given in Appendix F we can shed some new light on this mechanism. First, we see that the group $\mathbf{b}^\infty$ is essential. At the general position this group, as in the altermagnetic examples, contains also spatial inversion in order to leave the momentum

invariant. In the case of simple antiferromagnets, the dispersion on one sublattice is represented by $\nu$ and the other by $-\nu$ ($\sigma = \pm 1$ for magnons). Such irreps pair into a 2-dimensional co-irrep under PT symmetry as we see in the spin point group table $\mathbf{b}^\infty \times {}^{\bar{1}}\bar{1}$. In contrast the pure nontrivial spin group ${}^{\bar{1}}\bar{1}$ has only 1-D co-irreps. Similarly, the other collinear groups with non-unitary nontrivial group always have a symmetry which couples the $\sigma = \pm \nu$ **SO(2)** irreps into a 2-dimensional co-irrep. For example, in the Heisenberg Néel hyperhoneycomb the group at zone center $\mathbf{b}^\infty \times {}^1 m^1 m^{\bar{1}} m$ has the symmetry ${}^{\bar{1}} m$ which couples the two sublattices. Again, the pure nontrivial spin group ${}^1 m^1 m^{\bar{1}} m$ alone does not have 2-dimensional co-irreps.

Having considered magnon spectra for Heisenberg models, we now switch on the Kitaev terms and we consider again table 6. The situation here is much richer and we arrive to a total of 11 different spin point groups.

We take as an example the ferromagnetic $[11z]$ state. This phase is described at the zone center by the total spin group ${}^1 m^m m^m m$ (Litvin number 56). As we see from Table 7, this group is isomorphic to the magnetic point group $mm'm'$ and the spin wave representation in the Cracknell, Davies, Miller and Love notation is

$$\rho^\Gamma_{\text{FM}} = \Gamma_1^+(1) + \Gamma_1^-(1) + \Gamma_2^+(1) + \Gamma_2^-(1). \tag{29}$$

On the basis of the representation theory, we therefore account for the lack of enforced degeneracy at $\Gamma$ which has four symmetry distinct magnon modes.

### 4.3.1 Discrete spin-only groups

In Table 6 there are various spin groups of physical relevance that lie outside the Litvin-Opechowski classification. In particular, we find listed spin groups with discrete spin-only group $\mathbf{2'2'2}$ which is neither coplanar nor collinear. The Litvin-Opechowski classes categorize the invariances of magnetic structures in spin space and real space without reference to the Hamiltonian. Discrete spin-only groups may arise when there are anisotropies in the Hamiltonian originating from spin-orbit coupling.

It has been noted that spin point groups (and spin space groups in general) are relevant to certain cases where spin-orbit coupling is significant. These include Heisenberg-Kitaev exchange and Dzyaloshinskii-Moriya coupling when the **D** vector is common to all bonds. Beside exchange couplings, discrete anisotropies can also arise naturally from single-ion anisotropies. Just as spin groups are expected to be approximate symmetries in real materials (albeit to an excellent approximation in certain cases), so too the anisotropic terms mentioned above are generally expected to appear with additional couplings that break down the spin symmetry. In general, both the Hamiltonian symmetries and the symmetries of the magnetic structure are relevant to determine the full symmetry of the system under consideration. For the case of Kitaev-Heisenberg models, there are discrete anisotropies which limit the spin free rotation to the discrete group $D_2 = 222$ of three perpendicular $C_2$ axes.

As discussed above, magnetic order reduces the "paramagnetic" spin-only group to a subgroup. In the Heisenberg-Kitaev example there are three possible cases. Firstly, we have a collinear order along one of the Kitaev axis (e.g. $[001]$). This the most symmetric scenario since it preserves the spin-only group $\mathbf{2'2'2}$. Secondly, a collinear order perpendicular to a Kitaev axis (e.g. $[xy0]$) has the spin-only group $\mathbf{2'}$. Such a group, identified in the paper with the coplanar group $\mathbf{b}^{\mathbb{Z}_2}$, can appear also in a collinear phase in the presence of anisotropies as we see here. Finally, for a non-symmetric direction (e.g. $[xyz]$) there is no free spin rotation as we expect in a strong spin-orbit scenario, but there is still a nontrivial spin point group ${}^1 2/{}^1 m$ since the Kitaev axis is aligned to a crystallographic axis.

It is natural to ask what discrete spin-only groups are possible. In other words, what is the most general set of spin point groups? Since we limit ourselves to spin-orbit anisotropies which couple the spin with the lattice, such anisotropies must respect crystallographic constraints and therefore belong to one of the proper point groups.

For a paramagnetic phase, the relevant symmetries are those of the Hamiltonian. When the Hamiltonian is spin rotation symmetric the spin-only group is $\mathbf{b}^{\mathrm{NM}} \equiv \mathbf{SO(3)} \rtimes \{E, \tau\}$. However, when the Hamiltonian contains anisotropies, the allowed spin-only transformations — those that leave the Hamiltonian invariant — is enlarged to 13 possible groups — all subgroups of $\mathbf{b}^{\mathrm{NM}}$:

$$\mathbf{b}_{\mathrm{paramagnetic}} = \{\mathbf{b}^{\mathrm{NM}}, \mathbf{b}^{\mathrm{XY}}, \mathbf{11}', \mathbf{21}', \mathbf{2221}', \mathbf{31}', \mathbf{321}', \mathbf{41}', \mathbf{4221}', \mathbf{61}', \mathbf{6221}', \mathbf{231}', \mathbf{4321}'\}. \tag{30}$$

The group $\mathbf{b}^{\mathrm{XY}} = \mathbf{SO(2)} \rtimes \{E, \tau\}$ can be obtained for example in the $XY$ model or with Dzyaloshinskii-Moriya interaction with collinear $\mathbf{D}$ vector.

For a magnetically ordered phase, the spin-only groups must have an axial symmetry if they are to respect the invariance of the magnetic structure. We therefore extend the classification giving 12 possible spin-only groups for magnetically ordered systems:

$$\mathbf{b}_{\mathrm{ordered}} = \{\mathbf{b}^{\infty}, \mathbf{SO(2)}, \mathbf{1}, \mathbf{2}, \mathbf{2}'(\equiv \mathbf{b}^{\mathbb{Z}_2}), \mathbf{22}'\mathbf{2}', \mathbf{3}, \mathbf{32}', \mathbf{4}, \mathbf{42}'\mathbf{2}', \mathbf{6}, \mathbf{62}'\mathbf{2}'\}, \tag{31}$$

where $\mathbf{b}^{\infty} = \mathbf{SO(2)} \rtimes \{E, C_{2\perp}\tau\}$ and $\mathbf{b}^{\mathbb{Z}_2} = \{E, C_{2\perp}\tau\}$ are the zero spin-orbit collinear and coplanar case considered by Litvin and Opechowski.

These discrete spin-only groups lead to new total spin point groups when paired to the 598 nontrivial spin point groups. We leave for the future a calculation of the compatibility conditions of the discrete spin-only groups with the nontrivial groups as well as a discussion of their representation theory.

## 5 Discussion

In the foregoing we have described the theory of spin point groups that generalize the magnetic point groups to the case where spin and space transformations are correlated but not locked. These groups arise naturally in a condensed matter context in the zero spin-orbit coupled limit of magnetic crystals and in some cases when spin-orbit coupling is significant such as collinear structures with Kitaev-Heisenberg exchange. The existence of these groups and, to some extent, their significance were recognized in the 1960's. The first systematic studies, by Litvin and Opechowski and later by Litvin, outlined a general theory for the spin point groups introducing the spin-only groups and the nontrivial spin groups and enumerating the latter. Much more recent work on the formal side has studied constraints on the inter-relation of the spin-only part with the nontrivial group and calculated irreducible representations of certain crystallographic spin groups. Other work has explored further connections to materials.

This paper completes the programme of Litvin and Opechowski by working out the representation theory of the spin point groups. We have provided a self-contained description of the groups, an independent enumeration of the nontrivial spin groups and a discussion of the types of spin-only group that can arise in principle. One of the main results of this work is the observation that nontrivial spin groups are isomorphic to magnetic point groups, as a consequence of the main isomorphism theorem of Litvin and Opechowski, and we have worked out all such isomorphisms. The precise magnetic point group isomorphic to a given spin point group is not *a priori* obvious and the tabulation therefore has practical utility. For example, one may work out the spin group of a given magnetic structure in a crystal and directly find the possible irreducible representations corresponding to spin wave modes by translating to the well-known magnetic point groups irreps.

We then consider the total spin groups including the nontrivial group and the spin-only part. We find the amusing result that coplanar spin groups correspond to grey groups. Grey groups are usually to be found in the context of paramagnets where time reversal is unbroken. Here, though, they arise as the natural symmetry groups for certain magnetically ordered systems.

Even so, the grey groups are well known from traditional magnetic group theory. The representation theory of spin groups is enriched, however, by the spin-only group when that group has residual spin rotation symmetry. There are both unitary and non-unitary groups of these types. We have identified these groups and we have calculated all the (co-)irreps for these groups. An extensive set of tables for these new (co-)irreps may be found in Appendix F. One of the main findings is the pairing of (co-)irreps coming from the spin rotation symmetry. These findings systematize the possible enhanced degeneracies that can arise in band structures in magnetic crystals with spin rotation symmetry (with and without time reversal symmetry). We have given some examples of how this information can be useful in problems of interest such as determining the electronic band degeneracies in d wave altermagnets and features of magnon modes.

It is worth examining how doubling in the (co-)irrep dimension can arise in the (co-)irreps of the total spin point groups. Generically, (co-)irreps double in size during the induction procedure. When dealing with the total spin point groups, there are at most two induction steps: one to obtain the irreducible representations of the unitary halving subgroup (as described in Section E.1), and one to obtain co-irreps for the total antiunitary group (as described in Appendix A). Except in the case of total collinear groups with non-unitary $\mathbf{S}$, the first induction step is unnecessary for the spin point groups, since the irreducible representations of the unitary halving group are already known (and they correspond to a regular or magnetic point group, or the direct product of the irreps of $\mathbf{SO(2)}$ with those of the parent point group). In these cases, doubling (beyond the dimension of the point groups' irreps) will only arise when the unitary halving subgroup has irreps belonging to cases (b) or (c), as determined by Dimmock's test (in equation A.20). As summarized in table 5, for the nontrivial, non-magnetic, and coplanar groups these types are already known, since the co-irreps for the magnetic point groups are known [6]; doubling in the collinear groups requires additional analysis.

For the collinear groups, there are no irreps belonging to case (b), meaning that for these groups doubling in the co-irrep induction step can only occur via case (c). In collinear groups with unitary $\mathbf{S}$ (i.e. $\mathbf{S}$ exactly corresponds to one of the regular point groups), there are ten such instances.[6] Co-irreps of dimension larger than one in the remaining 22 collinear groups with unitary $\mathbf{S}$ arise from the parent point group itself having larger irreps. Among the 32 collinear groups with unitary $\mathbf{S}$, the largest co-irrep dimension is 3, indicating that only point group irreps of dimension one get doubled in the co-irreps (otherwise, we would see co-irreps of dimensions four or six).

For the 58 collinear groups with non-unitary $\mathbf{S}$, we must check for doubling in *both* of the induction steps. In inducing the irreps of the unitary halving subgroups $\mathbf{X}_{1/2}$ (according to the procedure laid out in Appendix E), *every* collinear group with non-unitary $\mathbf{S}$ possesses doubled irreps, arising from the pairing of an irrep labelled by non-zero integer $\mu$ (arising from $\mathbf{SO(2)}$) with the irrep labelled by $-\mu$ (giving rise to a new irrep labelled by positive non-zero integers $\nu$ in the tables of Appendix F). This is in addition to occasional doubling due to the point group structure alone. In the co-irrep induction step (described in Appendix A), doubling can occur for (1) irreps that were *not* doubled in the previous induction step, and (2) irreps that *were* doubled in the previous induction step. There are thirteen groups in the former category,[7]

---

[6]Where $\mathbf{S}$ is one of: $^{1}3$, $^{1}4$, $^{1}6$, $^{1}\bar{6}$, $^{1}4/^{1}m$, $^{1}6/^{1}m$, $^{1}\bar{4}$, $^{1}\bar{3}$, $^{1}2^{1}3$, or $^{1}2/^{1}m^{1}\bar{3}$.

[7]Where $\mathbf{S}$ is one of: $^{\bar{1}}4$, $^{\bar{1}}\bar{4}$, $^{\bar{1}}4/^{1}m$, $^{\bar{1}}4/^{\bar{1}}m$, $^{1}4/^{\bar{1}}m$, $^{1}4/^{1}m^{\bar{1}}m^{\bar{1}}m$, $^{\bar{1}}\bar{3}$, $^{\bar{1}}6/^{\bar{1}}m$, $^{\bar{1}}6/^{1}m$, $^{1}6/^{\bar{1}}m$, $^{1}2/^{\bar{1}}m^{\bar{1}}\bar{3}$, $^{\bar{1}}\bar{4}^{1}3^{\bar{1}}m$, $^{\bar{1}}4^{1}3^{\bar{1}}2$.

and twenty groups in the latter category.[8] Among these collinear groups with non-unitary **S**, while double-doubling is possible, it only ever occurs starting from a one-dimensional irrep of $\mathbf{SO(2)} \times \mathbf{G}_{1/2}$, giving a co-irrep of dimension four. The maximal co-irrep dimension is six, occuring from the doubling of a three dimensional irrep in the first induction step.

That the total collinear groups are fundamentally different from standard groups, and that their co-irreps are not known *a priori* is made clear by looking at even the simplest examples. Let us compare the standard group

$$\mathbf{C}_{\infty v} = \mathbf{SO(2)} \rtimes \{E, I2_{\perp \mathbf{n}}\}$$

(where in this discussion we will assume that these elements act on both real and spin space, as is standard under the assumption of strong spin-orbit coupling), with the simplest total collinear spin group

$$\mathbf{b}^{\infty} \times {}^1 1 = (\mathbf{SO(2)} \rtimes \{[E\|E], [\tau 2_{\perp \mathbf{n}}\|E]\}) \times {}^1 1 .$$

From a geometric perspective, both of these groups include rotations about the axis **n**, as well as all vertical reflections (i.e. those mirror planes containing the axis **n**). However, because spins are axial vectors, the group $\mathbf{C}_{\infty v}$ acts effectively on spins as the group $\mathbf{C}_{\infty} = \mathbf{SO(2)}$, as improper elements derived from real-space cannot flip the orientation of an axial vector. Only by including time-reversal can one achieve the effect of improper elements acting on the spins, and in the weak spin-orbit coupling limit it is possible for the symmetry of the spins to extend beyond the symmetry of the atomic arrangement. Further, while these groups are isomorphic as abstract groups, due to the requirement that elements with time-reversal must be represented by antiunitary matrices, the (co-)representation theory for these two groups differs. This is explicitly clear when one recognizes that the group $\mathbf{C}_{\infty v}$ contains two-dimensional irreducible representations, while $\mathbf{b}^{\infty} \times {}^1 1$ only contains one-dimensional co-irreps. Similarly, one could compare the (co-)irrep tables between $\mathbf{D}_{\infty h}$ and the simplest total collinear group with non-unitary **S**, $\mathbf{b}^{\infty} \times {}^{\bar{1}} 1$. Again, one notices that despite these groups containing elements with similar geometric content and being isomorphic as abstract groups, they differ in their irrep content ($\mathbf{D}_{\infty h}$ has two additional one-dimensional irreps, for example). Finally, it is also worth recognizing that the co-irreps of the total spin groups are *not* the direct product of the irreps of $\mathbf{b}^{\infty} \cong \mathbf{b}^{\infty} \times {}^1 1$ (which has only one-dimensional irreps labelled by any integer $\sigma$) with those of **S**. For example, we may examine the case $\mathbf{S} = {}^{\bar{1}} 6 / {}^{\bar{1}} m$, which has the same co-irreps as the black and white point group $6'/m'$. This group has four co-irreps, two of dimension one and two of dimension two. However, the total collinear group $\mathbf{b}^{\infty} \times {}^{\bar{1}} 6 / {}^{\bar{1}} m$ has co-irreps of dimension four, and therefore these co-irreps could not have arisen from a direct product of the irreps of $\mathbf{b}^{\infty}$ and ${}^{\bar{1}} 6 / {}^{\bar{1}} m$.

These spin point groups are the foundations of magnetic crystallography. A natural future direction is to explore the space groups built from these point groups - the so-called spin-space groups. Our isomorphism result carries over to that case: spin-space groups with no pure spin elements are necessarily isomorphic to a magnetic space group. The central problems to be addressed therefore are to determine the pattern of isomorphisms, constraints on the possible translation groups and the co-irreps at high symmetry momenta especially those at Brillouin zone boundaries.

In addition to demonstrating by example the utility of the spin point groups from the program of Litvin and Opechowski [25], we point out several contexts in which new spin point groups arise. In the context of electron band degeneracies in reciprocal space, spin point groups isomorphic but not conjugate to the collinear Litvin groups arise (in $\mathbf{O}(3) \times \mathbf{O}(3)$).

---

[8]Where **S** is one of: ${}^{\bar{1}} 4 / {}^{\bar{1}} m$, ${}^1 4 / {}^{\bar{1}} m$, ${}^1 4 {}^{\bar{1}} 2 {}^{\bar{1}} 2$, ${}^1 4 {}^{\bar{1}} m {}^{\bar{1}} m$, ${}^1 \bar{4} {}^{\bar{1}} 2 {}^{\bar{1}} m$, ${}^{\bar{1}} 3$, ${}^1 3 {}^{\bar{1}} 2$, ${}^1 3 {}^{\bar{1}} m$, ${}^1 \bar{3} {}^{\bar{1}} m$, ${}^{\bar{1}} \bar{6}$, ${}^{\bar{1}} 6$, ${}^1 6 {}^{\bar{1}} 2 {}^{\bar{1}} 2$, ${}^{\bar{1}} 6 / {}^{\bar{1}} m$, ${}^{\bar{1}} 6 / {}^1 m$, ${}^1 6 / {}^{\bar{1}} m$, ${}^1 6 {}^{\bar{1}} m {}^{\bar{1}} m$, ${}^1 \bar{6} {}^{\bar{1}} m 2$, ${}^1 6 / {}^1 m {}^{\bar{1}} m {}^{\bar{1}} m$, ${}^1 2 / {}^{\bar{1}} m {}^{\bar{1}} \bar{3}$, ${}^{\bar{1}} 4 / {}^1 m {}^1 \bar{3} {}^{\bar{1}} 2 / {}^{\bar{1}} m$.

In studying magnon excitations, we have found entirely new discrete spin-only groups describing Hamiltonian symmetries, further extending the classification of Litvin and Opechowski. As discussed above there are consistent total spin groups with nontrivial discrete spin-only groups that are not $2_{\perp \mathbf{n}}$ times time reversal. Another open problem is to explore the representation theory of spin-only groups beyond the collinear, coplanar and trivial cases.

## End note

Upon completion of this work, three independent groups posted preprints to the arXiv preprint server that provide a discussion and enumeration of spin-space groups [41–43]. References [41,42] contain discussions of the representation theory of these groups that has some overlap with the results contained here.

## Acknowledgments

PM would like to thank Jeff Rau and Pedro Ribeiro for useful discussions. The authors acknowledge use of the GTPACK [44] Mathematica group theory package developed by W. Hergert and M. Geilhufe.

**Funding information** JR and HS were supported by the NSF through grant DMR-2142554.

## A Review of magnetic representation theory

The representation theory of finite unitary groups tends to be covered in most introductory courses on group theory. The theory of the (co-)representation theory of magnetic groups [6,8] is perhaps less widely known though it is central to our work as many of the spin point groups have antiunitary elements. Therefore, we include here a self-contained review of the relevant modifications to unitary representation theory when time reversal operations are included.

All magnetic groups including spin point groups may be written in the form

$$\mathbf{M} = \mathbf{G} + A\mathbf{G}, \tag{A.1}$$

where $\mathbf{G}$ is a group with only unitary elements while coset $A\mathbf{G}$ is obtained from the elements of $\mathbf{G}$ by multiplying by antiunitary element $A$.[9] Following the notation of Bradley and Cracknell [6] in this section, we denote unitary elements by $R, S$ and antiunitary elements by $B, C$.

We suppose that the irreducible representations of the unitary part $\mathbf{G}$ are known. This is reasonable because, for the magnetic point groups, $\mathbf{G}$ is one of the 32 crystallographic point groups and, for the spin point groups, a similar result is true as will be shown in the following section. This means that the central problem to tackle is to build the so-called corepresentations obtained by including the antiunitary part of $\mathbf{M}$.

Suppose we have $d$-dimensional basis functions $\psi_\alpha$ ($\alpha = 1, \ldots, d$) for the unitary part of the group such that element $R \in \mathbf{G}$ acts like

$$R\psi_\alpha = \Delta(R)_{\alpha\beta}\psi_\beta. \tag{A.2}$$

We then need a basis for the antiunitary elements and we may choose $A\psi_\alpha \equiv \omega_\alpha$. It is then straightforward to see that a unitary element acting on the antiunitary part of the basis gives

---

[9]We say that a group element is antiunitary if it must be represented by an antiunitary operator. This is the case for any group element containing time-reversal. Recall that an antiunitary operator $U$ acts like $\langle U\phi | U\psi \rangle = \langle \psi | \phi \rangle$.

$R\omega_\alpha = \Delta^*(A^{-1}RA)_{\alpha\beta}\omega_\beta$. It is natural to introduce a notation where we keep track of the action of group on the entire basis $\Xi = (\psi, \omega)^T$. In this notation

$$R\begin{pmatrix} \psi \\ \omega \end{pmatrix} = \begin{pmatrix} \Delta(R) & 0 \\ 0 & \Delta^*(A^{-1}RA) \end{pmatrix}\begin{pmatrix} \psi \\ \omega \end{pmatrix} \equiv D(R)\Xi. \tag{A.3}$$

The action of antiunitary elements on the basis is off-diagonal and of the form

$$B\begin{pmatrix} \psi \\ \omega \end{pmatrix} = \begin{pmatrix} 0 & \Delta(BA) \\ \Delta^*(A^{-1}B) & 0 \end{pmatrix}\begin{pmatrix} \psi \\ \omega \end{pmatrix} \equiv D(B)\Xi. \tag{A.4}$$

Therefore with knowledge of the group multiplication table and the representations of **G** one may compute co-representations of **M**.

The problem of determining irreducible representations for magnetic groups is the problem of finding a unitary transformation acting on representations that reduce them as far as possible to block diagonal form. For magnetic groups the precise statement is that, to reduce a representation $D$, we look for $U$ acting on $D$ such that

$$U^{-1}D(R)U = \bar{D}(R), \tag{A.5}$$

$$U^{-1}D(B)U^* = \bar{D}(B), \tag{A.6}$$

and where the barred representations are similarly block diagonal.

A key step is to ask whether $\Delta(R)$ and $\Delta^*(A^{-1}RA)$ are unitarily equivalent. As we shall see, if they are then representations of unitary elements may (Case (a)) or may not (Case (b)) be reducible to fully block diagonal form. In case $\Delta(R)$ and $\Delta^*(A^{-1}RA)$ are inequivalent then generally these combine into a single co-representation (Case (c)). These three cases categorize the possible irreducible co-representations starting from unitary irreps. Cases (b) and (c) stand out in leading to irreps of doubled dimensions compared to the dimension of unitary irreps of **G**. This means, in principle, that the dimensions $1, 2, 3$ of unitary point groups may be doubled to $2, 4$ and $6$.

For cases (a) and (b) we assume that $\Delta(R)$ is an irreducible representation (irrep) of **G**. We also suppose that it is unitarily equivalent to $\Delta^*(A^{-1}RA)$ so that there is a matrix $P$ such that

$$\Delta(R) = P\Delta^*(A^{-1}RA)P^{-1}. \tag{A.7}$$

It is straightforward to see that

$$\Delta(R) = P\left(P^*\Delta^{-1}(A^2)\Delta(R)\Delta(A^2)P^{*-1}\right)P^{-1}, \tag{A.8}$$

from which the assumption of irreducibility combined with Schur's lemma gives the condition

$$PP^* = \lambda\Delta(A^2), \tag{A.9}$$

and identity $\Delta(A^2) = P\Delta^*(A^2)P^{-1}$ and unitarity of $P$ and $\Delta(A^2)$ combine to fix $\lambda$ to be real and $|\lambda| = 1$. So

$$PP^* = \pm\Delta(A^2). \tag{A.10}$$

We now see that there are two distinct cases. To see what they imply for reducibility of the co-representation we first use

$$\begin{pmatrix} 1 & 0 \\ 0 & P^{-1} \end{pmatrix} \tag{A.11}$$

to bring $D(R)$ to the form $D'(R) = \Delta(R) \oplus \Delta(R)$ appearing twice. The same transformation leads to

$$D'(A) = \begin{pmatrix} 0 & \Delta(A^2)P^{-1*} \\ P & 0 \end{pmatrix}. \tag{A.12}$$

We now try to reduce with unitary $U$. This should do nothing to the $D'(R)$ which constrains

$$U^{-1} = \begin{pmatrix} \lambda_1 \mathbb{1} & \lambda_2 \mathbb{1} \\ \lambda_3 \mathbb{1} & \lambda_4 \mathbb{1} \end{pmatrix}. \tag{A.13}$$

For $D(A)$ we get

$$\begin{pmatrix} \lambda_1 \mathbb{1} & \lambda_2 \mathbb{1} \\ \lambda_3 \mathbb{1} & \lambda_4 \mathbb{1} \end{pmatrix} \begin{pmatrix} 0 & \Delta(A^2)P^{-1*} \\ P & 0 \end{pmatrix} \begin{pmatrix} \lambda_1 \mathbb{1} & \lambda_3 \mathbb{1} \\ \lambda_2 \mathbb{1} & \lambda_4 \mathbb{1} \end{pmatrix}, \tag{A.14}$$

where we highlight the transpose between the left and right matrices. The constraint that this be block diagonal fixes $\lambda_1 \lambda_4 = \lambda_2 \lambda_3$. Going back to the condition

$$PP^* = \pm \lambda \mathbb{1}, \tag{A.15}$$

we see that reduction to block diagonal form is possible for the choice of positive sign.

**Case (a)** Where the reduction is possible we get

$$D(R) = \Delta(R), \qquad D(B) = \pm \Delta(BA^{-1})P, \tag{A.16}$$

where the sign of $D(B)$ is a matter of convention as these are equivalent.

**Case (b)** Here $PP^* = -\Delta(A^2)$ and, as we have discussed, it is then not possible to reduce the two-by-two magnetic co-representations and so the dimension of the co-representation is twice the dimension of the non-magnetic $\Delta(R)$. The co-representations take the form

$$\begin{pmatrix} \Delta(R) & 0 \\ 0 & \Delta(R) \end{pmatrix}, \qquad \begin{pmatrix} 0 & -\Delta(BA^{-1})P \\ \Delta(BA^{-1})P & 0 \end{pmatrix}. \tag{A.17}$$

**Case (c)** This case corresponds to the situation where two inequivalent unitary irreps $\Delta(R)$ and $\Delta^*(A^{-1}RA)$ are combined into a single co-representation. The co-representations are

$$\begin{pmatrix} \Delta(R) & 0 \\ 0 & \Delta^*(A^{-1}RA) \end{pmatrix}, \qquad \begin{pmatrix} 0 & \Delta(BA) \\ \Delta^*(A^{-1}B) & 0 \end{pmatrix}. \tag{A.18}$$

Inspection of these cases reveals immediately that knowledge of the unitary irreps is sufficient to compute the co-representations in case (c) and for one-dimensional irreps $\Delta(R)$ in all three cases. For cases (a) and (b) for 2D and higher dimensional irreps of the unitary part of the group, we must find a suitable $P$ matrix. If this is not immediately evident one may compute it from

$$P = \frac{1}{|\mathbf{G}|} \sum_{R \in \mathbf{G}} \Delta(R) X \Delta^*(A^{-1}R^{-1}A), \tag{A.19}$$

where $X$ is chosen so that $P$ is unitary.

In summary, we have seen that in order to build irreducible co-representations from irreps of the unitary part of the magnetic group three possible cases may arise and that there is a relatively simple prescription to compute the irreps explicitly. So far, however, it is not immediately evident what determines which case arises. We state a simple criterion known in the literature as the Dimmock test,[10] derived in Ref. [45] to make this determination from the characters $\chi(R)$:

$$\sum_{B \in A\mathbf{G}} \chi(B^2) = \begin{cases} +|\mathbf{G}|, & \text{Case (a)}, \\ -|\mathbf{G}|, & \text{Case (b)}, \\ 0, & \text{Case (c)}. \end{cases} \tag{A.20}$$

---

[10]Notice that when the antiunitary coset representative $A$ is equal to time reversal $\tau$, the Dimmock test corresponds exactly to the Frobenius-Schur indicator for the reality of our representation, with cases (a)-(c) corresponding to real, pseudoreal, and complex representations respectively.

When the group in question has uncountably many elements, as will be the case for collinear groups (containing factors of $\mathbf{SO(2)}$) the sum will be understood to sum over the discrete point group elements and integrate over the rotation angle $\varphi$. In this case, it is necessary to normalize the left-hand side both by $2\pi$ and the size of the discrete factor.

# B  Spin point group tables

Table 7: Table of nontrivial spin groups and the matching magnetic point groups organized by parent group (first column). The enumeration order and symbol for each group coincides with the notation of Litvin [25]. The last column gives the matching magnetic point group in standard notation. The Litvin symbols are color coded according to whether the groups are naturally collinear (blue) with spin-only group $\mathbf{b}^\infty$ or coplanar (orange) with spin-only group $\mathbf{b}^{\mathbb{Z}_2}$. When the Litvin number is bolded, the group exactly corresponds to a regular or black & white point group [25].

| Parent Group | Litvin # | Litvin Symbol | B&W Point Group |
|---|---|---|---|
| **1** | **1** | $^11$ | 1 |
| **$\bar{1}$** | **2** | $^1\bar{1}$ | $-1$ |
| | 3 | $^2\bar{1}$ | $-1$ |
| | 4 | $^m\bar{1}$ | $-1'$ |
| | **5** | $^{\bar{1}}\bar{1}$ | $-1'$ |
| **2** | 6 | $^12$ | 2 |
| | 7 | $^22$ | 2 |
| | **8** | $^m2$ | $2'$ |
| | 9 | $^{\bar{1}}2$ | $2'$ |
| **m** | 10 | $^1m$ | $m$ |
| | **11** | $^2m$ | $m$ |
| | **12** | $^mm$ | $m'$ |
| | 13 | $^{\bar{1}}m$ | $m'$ |
| **2/m** | 14 | $^12/^1m$ | $2/m$ |
| | 15 | $^12/^2m$ | $2/m$ |
| | 16 | $^12/^mm$ | $2/m'$ |
| | 17 | $^12/^{\bar{1}}m$ | $2/m'$ |
| | 18 | $^22/^1m$ | $2/m$ |
| | 19 | $^m2/^1m$ | $2'/m$ |
| | 20 | $^{\bar{1}}2/^1m$ | $2'/m$ |
| | **21** | $^22/^2m$ | $2/m$ |
| | **22** | $^m2/^mm$ | $2'/m'$ |
| | 23 | $^{\bar{1}}2/^{\bar{1}}m$ | $2'/m'$ |
| | 24 | $^{2_z}2/^{2_x}m$ | $2/m$ |
| | 25 | $^{2_z}2/^{m_x}m$ | $2/m'$ |
| | 26 | $^{m_x}2/^{m_y}m$ | $2'/m'$ |
| | 27 | $^{m_x}2/^{2_z}m$ | $2'/m$ |
| | **28** | $^22/^mm$ | $2/m'$ |
| | 29 | $^22/^{\bar{1}}m$ | $2/m'$ |
| | 30 | $^{\bar{1}}2/^mm$ | $2'/m'$ |
| | 31 | $^{\bar{1}}2/^2m$ | $2'/m$ |
| | 32 | $^m2/^{\bar{1}}m$ | $2'/m'$ |
| | **33** | $^m2/^2m$ | $2'/m$ |
| **mm2** | 34 | $^1m^1m^12$ | $mm2$ |
| | 35 | $^2m^2m^12$ | $mm2$ |
| | 36 | $^mm^mm^12$ | $m'm'2$ |
| | 37 | $^{\bar{1}}m^{\bar{1}}m^12$ | $m'm'2$ |
| | 38 | $^1m^2m^22$ | $mm2$ |

| Parent Group | Litvin # | Litvin Symbol | B&W Point Group |
|---|---|---|---|
| | 39 | $^1m^mm^m2$ | $m'm2'$ |
| | 40 | $^1m^{\bar{1}}m^{\bar{1}}2$ | $m'm2'$ |
| | **41** | $^{2_x}m^{2_y}m^{2_z}2$ | $mm2$ |
| | **42** | $^{m_x}m^{m_y}m^{2_z}2$ | $m'm'2$ |
| | **43** | $^{m_x}m^{2_z}m^{m_y}2$ | $m'm2'$ |
| | 44 | $^mm^{\bar{1}}m^22$ | $m'm'2$ |
| | 45 | $^mm^2m^{\bar{1}}2$ | $m'm2'$ |
| | 46 | $^{\bar{1}}m^2m^mm2$ | $m'm2'$ |
| **222** | 47 | $^12^12^12$ | $222$ |
| | 48 | $^12^22^22$ | $222$ |
| | 49 | $^12^m2^m2$ | $2'2'2$ |
| | 50 | $^12^{\bar{1}}2^{\bar{1}}2$ | $2'2'2$ |
| | **51** | $^{2_x}2^{2_y}2^{2_z}2$ | $222$ |
| | **52** | $^{m_x}2^{m_y}2^{2_z}2$ | $2'2'2$ |
| | 53 | $^22^{\bar{1}}2^m2$ | $2'2'2$ |
| **mmm** | 54 | $^1m^1m^1m$ | $mmm$ |
| | 55 | $^1m^2m^2m$ | $mmm$ |
| | 56 | $^1m^mm^mm$ | $m'm'm$ |
| | 57 | $^1m^{\bar{1}}m^{\bar{1}}m$ | $m'm'm$ |
| | 58 | $^1m^1m^2m$ | $mmm$ |
| | 59 | $^1m^1m^mm$ | $m'mm$ |
| | 60 | $^1m^1m^{\bar{1}}m$ | $m'mm$ |
| | 61 | $^2m^2m^2m$ | $mmm$ |
| | 62 | $^mm^mm^mm$ | $m'm'm'$ |
| | 63 | $^{\bar{1}}m^{\bar{1}}m^{\bar{1}}m$ | $m'm'm'$ |
| | 64 | $^{2_z}m^{2_z}m^{2_x}m$ | $mmm$ |
| | 65 | $^{m_y}m^{m_y}m^{m_x}m$ | $m'm'm'$ |
| | 66 | $^{m_y}m^{m_y}m^{2_z}m$ | $m'm'm$ |
| | 67 | $^{2_z}m^{2_z}m^{m_y}m$ | $m'mm$ |
| | 68 | $^mm^mm^{\bar{1}}m$ | $m'm'm'$ |
| | 69 | $^mm^mm^2m$ | $m'm'm$ |
| | 70 | $^2m^2m^{\bar{1}}m$ | $m'mm$ |
| | 71 | $^2m^2m^mm$ | $m'mm$ |
| | 72 | $^{\bar{1}}m^{\bar{1}}m^2m$ | $m'm'm$ |
| | 73 | $^{\bar{1}}m^{\bar{1}}m^mm$ | $m'm'm'$ |
| | 74 | $^{2_x}m^{2_y}m^1m$ | $mmm$ |
| | 75 | $^{m_x}m^{m_y}m^1m$ | $m'm'm$ |
| | 76 | $^{m_y}m^{2_z}m^1m$ | $m'mm$ |

| Parent Group | Litvin # | Litvin Symbol | B&W Point Group | | Parent Group | Litvin # | Litvin Symbol | B&W Point Group |
|---|---|---|---|---|---|---|---|---|
| | 77 | $^m m^{\bar{1}} m^1 m$ | $m'm'm$ | | | 115 | $^{\bar{4}} 4/^1 m$ | $4'/m$ |
| | 78 | $^m m^2 m^1 m$ | $m'mm$ | | | 116 | $^{2_z} 4/^{2_x} m$ | $4/m$ |
| | 79 | $^{\bar{1}} m^2 m^1 m$ | $m'mm$ | | | 117 | $^{2_z} 4/^{m_x} m$ | $4/m'$ |
| | 80 | $^{2_x} m^{2_y} m^{2_z} m$ | $mmm$ | | | 118 | $^{m_x} 4/^{2_z} m$ | $4'/m$ |
| | 81 | $^{m_x} m^{m_y} m^{2_z} m$ | $m'm'm$ | | | 119 | $^{m_x} 4/^{m_y} m$ | $4'/m'$ |
| | 82 | $^2 m^{\bar{1}} m^m m$ | $m'm'm$ | | | 120 | $^2 4/^{\bar{1}} m$ | $4/m'$ |
| | 83 | $^{m_x} m^{m_y} m^{m_z} m$ | $m'm'm'$ | | | 121 | $^2 4/^m m$ | $4/m'$ |
| | 84 | $^{m_x} m^{2_y} m^{2_z} m$ | $m'mm$ | | | 122 | $^{\bar{1}} 4/^2 m$ | $4'/m$ |
| | 85 | $^{m_z} m^{2_x} m^{2_z} m$ | $m'mm$ | | | 123 | $^{\bar{1}} 4/^m m$ | $4'/m'$ |
| | 86 | $^{2_z} m^{m_x} m^{m_z} m$ | $m'm'm$ | | | 124 | $^m 4/^2 m$ | $4'/m$ |
| | 87 | $^{2_z} m^{\bar{1}} m^{m_x} m$ | $m'm'm$ | | | 125 | $^m 4/^{\bar{1}} m$ | $4'/m'$ |
| | 88 | $^{2_z} m^{\bar{1}} m^{2_x} m$ | $m'mm$ | | | 126 | $^4 4/^m m$ | $4/m'$ |
| | 89 | $^{\bar{1}} m^{m_z} m^{m_x} m$ | $m'm'm'$ | | | 127 | $^4 4/^{\bar{1}} m$ | $4/m'$ |
| **4** | 90 | $^1 4$ | $4$ | | | 128 | $^{\bar{4}} 4/^m m$ | $4/m'$ |
| | 91 | $^2 4$ | $4$ | | | 129 | $^{\bar{4}} 4/^{\bar{1}} m$ | $4'/m'$ |
| | 92 | $^m 4$ | $4'$ | | **422** | 130 | $^1 4^1 2^1 2$ | $422$ |
| | 93 | $^{\bar{1}} 4$ | $4'$ | | | 131 | $^1 4^2 2^2 2$ | $422$ |
| | 94 | $^4 4$ | $4$ | | | 132 | $^1 4^m 2^m 2$ | $42'2'$ |
| | 95 | $^{\bar{4}} 4$ | $4'$ | | | 133 | $^1 4^{\bar{1}} 2^1 2$ | $42'2'$ |
| **4̄** | 96 | $^1 \bar{4}$ | $\bar{4}$ | | | 134 | $^2 4^2 2^1 2$ | $422$ |
| | 97 | $^2 \bar{4}$ | $\bar{4}$ | | | 135 | $^m 4^m 2^1 2$ | $4'2'2$ |
| | 98 | $^m \bar{4}$ | $\bar{4}'$ | | | 136 | $^{\bar{1}} 4^{\bar{1}} 2^1 2$ | $4'2'2$ |
| | 99 | $^{\bar{1}} \bar{4}$ | $\bar{4}'$ | | | 137 | $^{2_x} 4^{2_z} 2^{2_y} 2$ | $422$ |
| | 100 | $^4 \bar{4}$ | $\bar{4}$ | | | 138 | $^{2_z} 4^{m_x} 2^{m_y} 2$ | $42'2'$ |
| | 101 | $^{\bar{4}} \bar{4}$ | $\bar{4}'$ | | | 139 | $^{m_x} 4^{2_z} 2^{m_y} 2$ | $4'2'2$ |
| **4/m** | 102 | $^1 4/^1 m$ | $4/m$ | | | 140 | $^2 4^{\bar{1}} 2^m 2$ | $42'2'$ |
| | 103 | $^2 4/^1 m$ | $4/m$ | | | 141 | $^{\bar{1}} 4^2 2^m 2$ | $4'2'2$ |
| | 104 | $^m 4/^1 m$ | $4'/m$ | | | 142 | $^m 4^{\bar{1}} 2^2 2$ | $4'2'2$ |
| | 105 | $^{\bar{1}} 4/^1 m$ | $4'/m$ | | | 143 | $^{4_z} 4^{2_x} 2^{2_{xy}} 2$ | $422$ |
| | 106 | $^2 4/^2 m$ | $4/m$ | | | 144 | $^{4_z} 4^{m_x} 2^{m_{xy}} m$ | $42'2'$ |
| | 107 | $^m 4/^m m$ | $4/m$ | | | 145 | $^{\bar{4z}} 4^{2_x} 2^{m_{xy}} 2$ | $4'2'2$ |
| | 108 | $^{\bar{1}} 4/^{\bar{1}} m$ | $4'/m'$ | | **4mm** | 146 | $^1 4^1 m^1 m$ | $4mm$ |
| | 109 | $^1 4/^2 m$ | $4/m$ | | | 147 | $^1 4^2 m^2 m$ | $4mm$ |
| | 110 | $^1 4/^m m$ | $4/m'$ | | | 148 | $^1 4^m m^m m$ | $4m'm'$ |
| | 111 | $^1 4/^{\bar{1}} m$ | $4/m'$ | | | 149 | $^1 4^{\bar{1}} m^{\bar{1}} m$ | $4m'm'$ |
| | 112 | $^4 4/^2 m$ | $4/m$ | | | 150 | $^2 4^2 m^1 m$ | $4mm$ |
| | 113 | $^{\bar{4}} 4/^2 m$ | $4'/m$ | | | 151 | $^m 4^1 m^m m$ | $4'm'm$ |
| | 114 | $^4 4/^1 m$ | $4/m$ | | | 152 | $^1 4^{\bar{1}} m^1 m$ | $4'm'm$ |

| Parent Group | Litvin # | Litvin Symbol | B&W Point Group |
|---|---|---|---|
| | 153 | $^2_x4^2_y m^2_z m$ | $4mm$ |
| | 154 | $^2_z4^{m}_x m^{m}_y m$ | $4m'm'$ |
| | 155 | $^{m}_x4^2_z m^{m}_y m$ | $4'm'm$ |
| | 156 | $^24\bar1 m^m m$ | $4m'm'$ |
| | 157 | $^{\bar1}4^2 m^m m$ | $4'm'm$ |
| | 158 | $^{m}4\bar1 m^2 m$ | $4'm'm$ |
| | **159** | $^{4}_z4^2_x m^{2_{xy}} m$ | $4mm$ |
| | **160** | $^{4}_z4^{m}_x m^{m_{xy}} m$ | $4m'm'$ |
| | **161** | $^{\bar4}_z4^{m}_x m2^{xy} m$ | $4m'm'$ |
| $\bar42m$ | 162 | $^1\bar4^1 2^1 m$ | $\bar42m$ |
| | 163 | $^1\bar4^2 2^2 m$ | $\bar42m$ |
| | 164 | $^1\bar4^{m}2^{m}m$ | $-\bar42'm'$ |
| | 165 | $^1\bar4^{\bar1}2^{\bar1}m$ | $\bar42'm'$ |
| | 166 | $^2\bar4^2 2^1 m$ | $\bar42m$ |
| | 167 | $^{m}\bar4^{m}2^1 m$ | $\bar4'2'm$ |
| | 168 | $^{\bar1}\bar4^{\bar1}2^1 m$ | $\bar4'2'm$ |
| | 169 | $^2\bar4^1 2^2 m$ | $\bar42m$ |
| | 170 | $^{m}\bar4^1 2^{m}m$ | $\bar4'2m'$ |
| | 171 | $^{\bar1}\bar4^1 2^{\bar1}m$ | $\bar4'2m'$ |
| | 172 | $^2_x\bar4^{2_y}2^{2_z}m$ | $\bar42m$ |
| | 173 | $^2_z\bar4^{m_x}2^{m_y}m$ | $\bar42'm'$ |
| | 174 | $^{m_x}\bar4^{2_z}2^{m_y}m$ | $\bar4'2m'$ |
| | 175 | $^{m_x}\bar4^{m_y}2^{2_z}m$ | $\bar4'2'm$ |
| | 176 | $^2\bar4^{\bar1}2^{m}m$ | $\bar42'm'$ |
| | 177 | $^2\bar4^{m}2^{\bar1}m$ | $\bar42'm'$ |
| | 178 | $^{\bar1}\bar4^2 2^{m}m$ | $\bar4'2m'$ |
| | 179 | $^{\bar1}\bar4^{m}2^2 m$ | $\bar4'2m'$ |
| | 180 | $^{m}\bar4^2 2^{\bar1}m$ | $\bar4'2m'$ |
| | 181 | $^{m}\bar4^{\bar1}2^2 m$ | $\bar4'2m'$ |
| | **182** | $^{4_z}\bar4^{2_x}2^{2_{xy}}m$ | $\bar42m$ |
| | **183** | $^{4_z}\bar4^{m_x}2^{m_{xy}}m$ | $\bar42'm'$ |
| | **184** | $^{\bar4_z}\bar4^{2_x}2^{m_{xy}}m$ | $\bar4'2'm$ |
| | **185** | $^{\bar4_z}\bar4^{m_{xy}}2^{2_{xy}}m$ | $\bar4'2'm$ |
| $4/mmm$ | 186 | $^14/^1m^1m^1m$ | $4/mmm$ |

| Parent Group | Litvin # | Litvin Symbol | B&W Point Group |
|---|---|---|---|
| | 187 | $^24/^2m^2m^1m$ | $4/mmm$ |
| | 188 | $^{m}4/^{m}m^{m}m^1m$ | $4'/m'm'm$ |
| | 189 | $^{\bar1}4/^{\bar1}m^1m^1m$ | $4'/m'm'm$ |
| | 190 | $^14/^2m^1m^1m$ | $4/mmm$ |
| | 191 | $^14/^{m}m^1m^1m$ | $4/m'mm$ |
| | 192 | $^14/^{\bar1}m^1m^1m$ | $4/m'mm$ |
| | 193 | $^24/^1m^1m^2m$ | $4/mmm$ |
| | 194 | $^{m}4/^1m^1m^{m}m$ | $4'/mmm'$ |
| | 195 | $^{\bar1}4/^1m^1m^{\bar1}m$ | $4'/mmm'$ |
| | 196 | $^14/^1m^2m^2m$ | $4/mmm$ |
| | 197 | $^14/^1m^{m}m^{m}m$ | $4/mm'm'$ |
| | 198 | $^14/^1m^{\bar1}m^{\bar1}m$ | $4/mm'm'$ |
| | 199 | $^14/^2m^2m^2m$ | $4/mmm$ |
| | 200 | $^14/^{m}m^{m}m^{m}m$ | $4/m'm'm'$ |
| | 201 | $^14/^{\bar1}m^{\bar1}m^{\bar1}m$ | $4/m'm'm'$ |
| | 202 | $^2_z4/^{2_z}m^{2_x}m^{2_y}m$ | $4/mmm$ |
| | 203 | $^2_z4/^{2_z}m^{m_x}m^{m_y}m$ | $4/mm'm'$ |
| | 204 | $^{m_x}4/^{m_x}m^{2_z}m^{m_y}m$ | $4'/m'm'm$ |
| | 205 | $^24/^2m^{\bar1}m^{m}m$ | $4/mm'm'$ |
| | 206 | $^{\bar1}4/^{\bar1}m^2m^{m}m$ | $4'/m'm'm$ |
| | 207 | $^{m}4/^{m}m^{\bar1}m^2m$ | $4'/m'm'm$ |
| | 208 | $^14/^{2_x}m^{2_y}m^{2_y}m$ | $4/mmm$ |
| | 209 | $^14/^2m^{m_x}m^{m_x}m$ | $4/mm'm'$ |
| | 210 | $^14/^{m_x}m^{2_z}m^{2_z}m$ | $4/m'mm$ |
| | 211 | $^14/^{m_x}m^{m_y}m^{m_y}m$ | $4/m'm'm'$ |
| | 212 | $^14/^2m^{\bar1}m^{\bar1}m$ | $4/mm'm'$ |
| | 213 | $^14/^2m^{m}m^{m}m$ | $4/mm'm'$ |
| | 214 | $^14/^{\bar1}m^2m^2m$ | $4/m'mm$ |
| | 215 | $^14/^{\bar1}m^{m}m^{m}m$ | $4/m'm'm'$ |
| | 216 | $^14/^{m}m^2m^2m$ | $4/m'mm$ |
| | 217 | $^14/^{m}m^{\bar1}m^{\bar1}m$ | $4/m'm'm'$ |
| | 218 | $^2_z4/^1m^{2_x}m^{2_y}m$ | $4/mmm$ |
| | 219 | $^2_z4/^1m^{m_x}m^{m_y}m$ | $4/mm'm'$ |
| | 220 | $^{m_x}4/^1m^{2_z}m^{m_y}m$ | $4'/mmm'$ |
| | 221 | $^24/^1m^{m}m^{\bar1}m$ | $4/mm'm'$ |
| | 222 | $^{\bar1}4/^1m^{m}m^2m$ | $4'/mmm'$ |
| | 223 | $^{m}4/^1m^2m^{\bar1}m$ | $4'/mmm'$ |
| | 224 | $^2_z4/^{2_x}m^1m^{2_z}m$ | $4/mmm$ |

| Parent Group | Litvin # | Litvin Symbol | B&W Point Group |
|---|---|---|---|
|  | 225 | $^{2_z}4/^{m_x}m^1m^{2_z}m$ | $4/m'mm$ |
|  | 226 | $^{m_x}4/^{2_z}m^1m^{m_x}m$ | $4'/mmm'$ |
|  | 227 | $^{m_x}4/^{m_y}m^1m^{m_x}m$ | $4'/m'm'm$ |
|  | 228 | $^24/^{\bar1}m^2m^1m$ | $4/m'mm$ |
|  | 229 | $^24/^mm^1m^2m$ | $4/m'mm$ |
|  | 230 | $^{\bar1}4/^2m^{\bar1}m^1m$ | $4'/mmm'$ |
|  | 231 | $^{\bar1}4/^mm^{\bar1}m^1m$ | $4'/m'm'm$ |
|  | 232 | $^m4/^2m^1m^mm$ | $4'/mmm'$ |
|  | 233 | $^m4/^{\bar1}m^mm^1m$ | $4'/m'm'm$ |
|  | 234 | $^{2_z}4/^{2_x}m^{2_x}m^{2_y}m$ | $4/mmm$ |
|  | 235 | $^{2_z}4/^{m_x}m^{m_x}m^{m_y}m$ | $4/m'm'm'$ |
|  | 236 | $^{m_x}4/^{2_z}m^{2_z}m^{m_y}m$ | $4'/mmm'$ |
|  | 237 | $^{m_x}4/^{m_y}m^{m_y}m^{2_z}m$ | $4'/m'm'm$ |
|  | 238 | $^24/^{\bar1}m^1m^mm$ | $4/m'm'm'$ |
|  | 239 | $^24/^mm^mm^{\bar1}m$ | $4/m'm'm'$ |
|  | 240 | $^{\bar1}4/^2m^2m^mm$ | $4'/mmm'$ |
|  | 241 | $^{\bar1}4/^mm^mm^2m$ | $4'/m'm'm$ |
|  | 242 | $^m4/^2m^2m^{\bar1}m$ | $4'/mmm'$ |
|  | 243 | $^m4/^{\bar1}m^{\bar1}m^2m$ | $4'/m'm'm$ |
|  | 244 | $^{4_z}4/^1m^{2_x}m^{2_{xy}}m$ | $4/mmm$ |
|  | 245 | $^{4_z}4/^1m^{m_x}m^{m_{xy}}m$ | $4/mm'm'$ |
|  | 246 | $^{\bar4_z}4/^1m^{2_x}m^{m_{xy}}m$ | $4'/m'm'm$ |
|  | **247** | $^{4_z}4/^{2_z}m^{2_x}m^{2_{xy}}m$ | $4/mmm$ |
|  | **248** | $^{4_z}4/^{2_z}m^{m_x}m^{m_{xy}}m$ | $4/mm'm'$ |
|  | **249** | $^{\bar4_z}4/^{2_z}m^{m_x}m^{2_{xy}}m$ | $4'/mmm'$ |
|  | 250 | $^{m_z}4/^{2_x}m^{m_x}m^{2_y}m$ | $4'/mmm'$ |
|  | 251 | $^{m_z}4/^{m_x}m^{m_y}m^{2_x}m$ | $4'/m'm'm$ |
|  | 252 | $^{2_z}4/^{m_x}m^{2_x}m^{2_y}m$ | $4/m'mm$ |
|  | 253 | $^{m_z}4/^{2_x}m^{2_x}m^{m_y}m$ | $4'/mmm'$ |
|  | 254 | $^{2_z}4/^{m_z}m^{2_x}m^{2_y}m$ | $4/m'mm$ |
|  | 255 | $^{2_z}4/^{m_z}m^{m_x}m^{m_y}m$ | $4/m'm'm'$ |
|  | 256 | $^{m_z}4/^{2_x}m^{2_z}m^{\bar1}m$ | $4'/mmm'$ |
|  | 257 | $^{m_z}4/^{m_x}m^{\bar1}m^{2_z}m$ | $4'/m'm'm$ |
|  | 258 | $^{m_z}4/^{\bar1}m^{m_x}m^{2_y}m$ | $4'/m'm'm$ |
|  | 259 | $^{2_z}4/^{2_x}m^{m_x}m^{m_y}m$ | $4/mm'm'$ |
|  | 260 | $^{2_z}4/^{2_x}m^{\bar1}m^mm^{m_z}m$ | $4/mm'm'$ |
|  | 261 | $^{2_z}4/^{\bar1}m^{m_x}m^{m_y}m$ | $4/m'm'm'$ |
|  | 262 | $^{2_z}4/^{m_x}m^{\bar1}m^mm^{m_z}m$ | $4/m'm'm'$ |
|  | 263 | $^{2_z}4/^{\bar1}m^{2_y}m^{2_x}m$ | $4/m'mm$ |
|  | 264 | $^{\bar1}4/^{2_x}m^{2_z}m^{m_z}m$ | $4'/mmm'$ |
|  | 265 | $^{\bar1}4/^{m_x}m^{m_z}m^{2_z}m$ | $4'/m'm'm$ |
|  | **266** | $^{4_z}4/^{m_z}m^{m_x}m^{m_{xy}}m$ | $4/m'm'm'$ |
|  | 267 | $^{4_z}4/^{\bar1}m^{m_x}m^{m_{xy}}m$ | $4/m'm'm'$ |
|  | **268** | $^{4_z}4/^{m_z}m^{2_x}m^{2_{xy}}m$ | $4/m'mm$ |
|  | 269 | $^{4_z}4/^{\bar1}m^{2_x}m^{2_{xy}}m$ | $4/m'mm$ |
|  | **270** | $^{\bar4_z}4/^{m_z}m^{2_x}m^{2_{xy}}m$ | $4'/m'm'm$ |
|  | 271 | $^{\bar4_z}4/^{\bar1}m^{m_x}m^{2_{xy}}m$ | $4'/m'm'm$ |
| **3** | 272 | $^13$ | 3 |
|  | **273** | $^33$ | 3 |
| **$\bar3$** | 274 | $^1\bar3$ | $\bar3$ |
|  | 275 | $^2\bar3$ | $\bar3$ |
|  | 276 | $^m\bar3$ | $\bar3'$ |
|  | 277 | $^{\bar1}\bar3$ | $\bar3'$ |
|  | **278** | $^3\bar3$ | $\bar3$ |
|  | 279 | $^6\bar3$ | $\bar3$ |
|  | **280** | $^{\bar3}\bar3$ | $\bar3'$ |
|  | 281 | $^{\bar6}\bar3$ | $\bar3'$ |
| **32** | 282 | $^13^12$ | 32 |
|  | 283 | $^13^22$ | 32 |
|  | 284 | $^13^m2$ | 32' |
|  | 285 | $^13^{\bar1}2$ | 32' |
|  | **286** | $^{3_z}3^{2_x}2$ | 32 |
|  | **287** | $^{3_z}3^{m_x}2$ | 32' |
| **3m** | 288 | $^13^1m$ | 3m |
|  | 289 | $^13^2m$ | 3m |
|  | 290 | $^13^mm$ | 3m' |
|  | 291 | $^13^{\bar1}m$ | 3m' |
|  | **292** | $^{3_z}3^{2_x}m$ | 3m |
|  | **293** | $^{3_z}3^{m_x}m$ | 3m' |
| **$\bar3m$** | 294 | $^1\bar3^1m$ | $\bar3m$ |
|  | 295 | $^1\bar3^2m$ | $\bar3m$ |
|  | 296 | $^1\bar3^mm$ | $\bar3m'$ |
|  | 297 | $^1\bar3^{\bar1}m$ | $\bar3m'$ |
|  | 298 | $^2\bar3^1m$ | $\overline{\bar3m}$ |
|  | 299 | $^m\bar3^{\bar1}m$ | $\bar3'm$ |
|  | 300 | $^{\bar1}\bar3^1m$ | $\bar3'm$ |

| Parent Group | Litvin # | Litvin Symbol | B&W Point Group |
|---|---|---|---|
| | 301 | $^{2}\bar{3}^{2}m$ | $\bar{3}m$ |
| | 302 | $^{m}\bar{3}^{m}m$ | $\bar{3}'m'$ |
| | 303 | $^{\bar{1}}\bar{3}^{\bar{1}}m$ | $\bar{3}'m'$ |
| | 304 | $^{2_x}\bar{3}^{2_z}m$ | $\bar{3}m$ |
| | 305 | $^{2_z}\bar{3}^{m_x}m$ | $\bar{3}m'$ |
| | 306 | $^{m_x}\bar{3}^{m_y}m$ | $\bar{3}'m'$ |
| | 307 | $^{m_x}\bar{3}^{2_z}m$ | $\bar{3}'m$ |
| | 308 | $^{2}\bar{3}^{m}m$ | $\bar{3}m'$ |
| | 309 | $^{2}\bar{3}^{\bar{1}}m$ | $\bar{3}m'$ |
| | 310 | $^{\bar{1}}\bar{3}^{m}m$ | $\bar{3}'m'$ |
| | 311 | $^{\bar{1}}\bar{3}^{2}m$ | $\bar{3}'m$ |
| | 312 | $^{m}\bar{3}^{\bar{1}}m$ | $\bar{3}'m'$ |
| | 313 | $^{m}\bar{3}^{2}m$ | $\bar{3}'m$ |
| | **314** | $^{3_z}\bar{3}^{2_x}m$ | $\bar{3}m$ |
| | **315** | $^{3_z}\bar{3}^{m_x}m$ | $\bar{3}m'$ |
| | 316 | $^{6_z}\bar{3}^{2_y}m$ | $\bar{3}m$ |
| | **317** | $^{\overline{3_z}}\bar{3}^{m_x}m$ | $\bar{3}'m'$ |
| | **318** | $^{\overline{3_z}}\bar{3}^{2_x}m$ | $\bar{3}'m$ |
| | 319 | $^{6_z}\bar{3}^{m_x}m$ | $\bar{3}m'$ |
| | 320 | $^{\overline{6_z}}\bar{3}^{m_x}m$ | $\bar{3}'m'$ |
| | 321 | $^{\overline{6_z}}\bar{3}^{2_x}m$ | $\bar{3}'m$ |
| $\bar{6}$ | 322 | $^{1}\bar{6}$ | $\bar{6}$ |
| | 323 | $^{2}\bar{6}$ | $\bar{6}'$ |
| | 324 | $^{m}\bar{6}$ | $\bar{6}$ |
| | 325 | $^{\bar{1}}\bar{6}$ | $\bar{6}'$ |
| | 326 | $^{3}\bar{6}$ | $\bar{6}$ |
| | **327** | $^{6}\bar{6}$ | $\bar{6}'$ |
| | 328 | $^{\bar{3}}\bar{6}$ | $\bar{6}$ |
| | **329** | $^{\bar{6}}\bar{6}$ | $\bar{6}'$ |
| 6 | 330 | $^{1}6$ | 6 |
| | 331 | $^{2}6$ | 6 |
| | 332 | $^{m}6$ | 6' |
| | 333 | $^{\bar{1}}6$ | 6' |
| | 334 | $^{3}6$ | 6 |
| | **335** | $^{6}6$ | 6 |
| | 336 | $^{\bar{3}}6$ | 6' |
| | **337** | $^{\bar{6}}6$ | 6' |

| Parent Group | Litvin # | Litvin Symbol | B&W Point Group |
|---|---|---|---|
| **622** | 338 | $^{1}6^{1}2^{1}2$ | 622 |
| | 339 | $^{1}6^{2}2^{2}2$ | 622 |
| | 340 | $^{1}6^{m}2^{m}2$ | 62'2' |
| | 341 | $^{1}6^{\bar{1}}2^{\bar{1}}2$ | 62'2' |
| | 342 | $^{2}6^{1}2^{2}2$ | 622 |
| | 343 | $^{m}6^{1}2^{m}2$ | 6'22' |
| | 344 | $^{\bar{1}}6^{1}2^{\bar{1}}2$ | 6'22' |
| | 345 | $^{2_x}6^{2_y}2^{2_z}2$ | 622 |
| | 346 | $^{2_z}6^{m_x}2^{m_y}2$ | 62'2' |
| | 347 | $^{m_x}6^{2_z}2^{m_y}2$ | 6'22' |
| | 348 | $^{2}6^{\bar{1}}2^{m}2$ | 62'2' |
| | 349 | $^{\bar{1}}6^{2}2^{m}2$ | 6'22' |
| | 350 | $^{m}6^{2}2^{\bar{1}}2$ | 6'22' |
| | 351 | $^{3_z}6^{2_x}2^{2_{xy}}2$ | 622 |
| | 352 | $^{3_z}6^{m_x}2^{m_{xy}}2$ | 62'2' |
| | **353** | $^{6_z}6^{2_x}2^{2_l}2$ | 622 |
| | 354 | $^{\overline{3_z}}6^{2_x}2^{m_{xy}}2$ | 6'22' |
| | **355** | $^{6_z}6^{m_x}2^{m_l}2$ | 62'2' |
| | **356** | $^{\overline{6_z}}6^{m_x}2^{2_l}2$ | 6'22' |
| **6/m** | 357 | $^{1}6/^{1}m$ | 6//m |
| | 358 | $^{2}6/^{2}m$ | 6/m |
| | 359 | $^{m}6/^{m}m$ | 6'/m' |
| | 360 | $^{\bar{1}}6/^{\bar{1}}m$ | 6'/m' |
| | 361 | $^{2}6/^{1}m$ | 6/m |
| | 362 | $^{m}6/^{1}m$ | 6'/m |
| | 363 | $^{\bar{1}}6/^{1}m$ | 6'/m |
| | 364 | $^{1}6/^{2}m$ | 6/m |
| | 365 | $^{1}6/^{m}m$ | 6/m' |
| | 366 | $^{1}6/^{\bar{1}}m$ | 6/m' |
| | 367 | $^{3}6/^{1}m$ | 6/m |
| | 368 | $^{2_z}6/^{2_x}m$ | 6/m |
| | 369 | $^{2_z}6/^{m_x}m$ | 6/m' |
| | 370 | $^{m_x}6/^{2_z}m$ | 6'/m |
| | 371 | $^{m_x}6/^{m_y}m$ | 6'/m' |
| | 372 | $^{2}6/^{\bar{1}}m$ | 6/m' |
| | 373 | $^{2}6/^{m}m$ | 6'/m' |
| | 374 | $^{\bar{1}}6/^{2}m$ | 6'/m |
| | 375 | $^{\bar{1}}6/^{m}m$ | 6'/m' |

| Parent Group | Litvin # | Litvin Symbol | B&W Point Group |
|---|---|---|---|
| | 376 | $^m6/^2m$ | $6'/m$ |
| | 377 | $^m6/^{\bar1}m$ | $6'/m'$ |
| | 378 | $^36/^2m$ | $6/m$ |
| | 379 | $^36/^{\bar1}m$ | $6/m'$ |
| | 380 | $^36/^mm$ | $6/m'$ |
| | 381 | $^66/^1m$ | $6/m$ |
| | 382 | $^{\bar3}6/^1m$ | $6'/m$ |
| | 383 | $^{\bar6}6/^1m$ | $6'/m$ |
| | **384** | $^66/^2m$ | $6/m$ |
| | 385 | $^{\bar3}6/^{\bar1}m$ | $6'/m'$ |
| | **386** | $^{\bar6}6/^mm$ | $6'/m'$ |
| | **387** | $^66/^mm$ | $6/m'$ |
| | 388 | $^66/^{\bar1}m$ | $6/m'$ |
| | 389 | $^{\bar3}6/^mm$ | $6'/m'$ |
| | 390 | $^{\bar3}6/^2m$ | $6'/m$ |
| | 391 | $^{\bar6}6/^{\bar1}m$ | $6'/m'$ |
| | **392** | $^{\bar6}6/^2m$ | $6'/m$ |
| **6mm** | 393 | $^16^1m^1m$ | $6mm$ |
| | 394 | $^16^2m^2m$ | $6mm$ |
| | 395 | $^16^mm^mm$ | $6m'm'$ |
| | 396 | $^16^{\bar1}m^{\bar1}m$ | $6m'm'$ |
| | 397 | $^26^1m^2m$ | $6mm$ |
| | 398 | $^m6^1m^mm$ | $6'mm'$ |
| | 399 | $^{\bar1}6^1m^{\bar1}m$ | $6'mm'$ |
| | 400 | $^{2_x}6^{2_y}m^{2_z}m$ | $6mm$ |
| | 401 | $^{2_z}6^{m_x}m^{m_y}m$ | $6m'm'$ |
| | 402 | $^{m_x}6^{2_z}m^{m_y}m$ | $6'mm'$ |
| | 403 | $^26^{\bar1}m^mm$ | $6'm'm'$ |
| | 404 | $^{\bar1}6^2m^mm$ | $6'mm'$ |
| | 405 | $^m6^2m^{\bar1}m$ | $6'mm'$ |
| | 406 | $^{3_z}6^{2_x}m^{2_{xy}}m$ | $6mm$ |
| | 407 | $^{3_z}6^{m_x}m^{m_{xy}}m$ | $6m'm'$ |
| | **408** | $^{6_z}6^{2_x}m^{2_1}m$ | $6mm$ |
| | 409 | $^{\bar3_z}6^{2_x}m^{m_{xy}}m$ | $6'mm'$ |
| | **410** | $^{6_z}6^{m_x}m^{m_1}m$ | $6m'm'$ |
| | **411** | $^{6_z}6^{m_x}m^{2_1}m$ | $6'mm'$ |

| Parent Group | Litvin # | Litvin Symbol | B&W Point Group |
|---|---|---|---|
| $\bar6m2$ | 412 | $^1\bar6^{\bar1}m^12$ | $\bar6m2$ |
| | 413 | $^1\bar6^{\bar2}m^22$ | $\bar6m2$ |
| | 414 | $^1\bar6^mm^m2$ | $\bar6m'2'$ |
| | 415 | $^1\bar6^{\bar1}m^{\bar1}2$ | $\bar6m'2'$ |
| | 416 | $^2\bar6^{\bar1}m^22$ | $\bar6m2$ |
| | 417 | $^m\bar6^{\bar1}m^m2$ | $\bar6'm2'$ |
| | 418 | $^{\bar1}\bar6^{\bar1}m^{\bar1}2$ | $\bar6'm2'$ |
| | 419 | $^2\bar6^{\bar2}m^12$ | $\bar6m2$ |
| | 420 | $^m\bar6^mm^12$ | $\bar6'm2$ |
| | 421 | $^{\bar1}\bar6^{\bar1}m^12$ | $\bar6'm'2$ |
| | 422 | $^{2_z}\bar6^{2_x}m^{2_y}2$ | $\bar6m2$ |
| | 423 | $^{2_z}\bar6^{m_x}m^{m_y}2$ | $\bar6m'2'$ |
| | 424 | $^{m_x}\bar6^{m_y}m^{2_z}2$ | $\bar6'm'2$ |
| | 425 | $^{m_x}\bar6^{2_z}m^{m_y}2$ | $\bar6m'2'$ |
| | 426 | $^2\bar6^mm^{\bar1}2$ | $\bar6m'2'$ |
| | 427 | $^2\bar6^{\bar1}m^m2$ | $\bar6m'2'$ |
| | 428 | $^{\bar1}\bar6^mm^22$ | $\bar6'm'2$ |
| | 429 | $^{\bar1}\bar6^2m^m2$ | $\bar6'm2'$ |
| | 430 | $^m\bar6^{\bar1}m^22$ | $\bar6'm'2$ |
| | 431 | $^m\bar6^2m^{\bar1}2$ | $\bar6'm2'$ |
| | 432 | $^{3_z}\bar6^{2_x}m^{2_{xy}}2$ | $\bar6m2$ |
| | 433 | $^{3_z}\bar6^{m_x}m^{m_{xy}}2$ | $\bar6m'2'$ |
| | **434** | $^{6_z}\bar6^{2_x}m^{2_1}2$ | $\bar6m2$ |
| | 435 | $^{\bar3_z}\bar6^{m_x}m^{2_{xy}}2$ | $\bar6'm'2$ |
| | 436 | $^{\bar3_z}\bar6^{2_x}m^{m_{xy}}2$ | $\bar6'm2'$ |
| | **437** | $^{6_z}\bar6^{m_x}m^{m_1}2$ | $\bar6m'2'$ |
| | **438** | $^{\bar6_z}\bar6^{m_x}m^{2_1}2$ | $\bar6'm'2$ |
| | **439** | $^{\bar6_z}\bar6^{2_x}m^{m_1}2$ | $\bar6'm2'$ |
| 6/mmm | 440 | $^16/^1m^1m^1m$ | $6/mmm$ |
| | 441 | $^26/^2m^1m^2m$ | $6/mmm$ |
| | 442 | $^m6/^mm^1m^mm$ | $6'/m'mm'$ |
| | 443 | $^{\bar1}6/^{\bar1}m^1m^{\bar1}m$ | $6'/m'mm'$ |
| | 444 | $^26/^1m^2m^1m$ | $6/mmm$ |
| | 445 | $^m6/^1m^mm^1m$ | $6'/mmm'$ |
| | 446 | $^{\bar1}6/^1m^{\bar1}m^1m$ | $6'/mmm'$ |
| | 447 | $^16/^1m^2m^2m$ | $6/mmm$ |
| | 448 | $^16/^1m^mm^mm$ | $6/mm'm'$ |
| | 449 | $^16/^1m^{\bar1}m^{\bar1}m$ | $6/mm'm'$ |

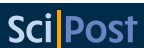

| Parent Group | Litvin # | Litvin Symbol | B&W Point Group |
|---|---|---|---|
| | 450 | $^{1}6/^{2}m^{1}m^{1}m$ | $6/mmm$ |
| | 451 | $^{1}6/^{m}m^{1}m^{1}m$ | $6/m'mm$ |
| | 452 | $^{1}6/^{\bar{1}}m^{1}m^{1}m$ | $6/m'mm$ |
| | 453 | $^{1}6/^{2}m^{2}m^{2}m$ | $6/mmm$ |
| | 454 | $^{1}6/^{m}m^{m}m^{m}m$ | $6/m'm'm'$ |
| | 455 | $^{1}6/^{\bar{1}}m^{1}m^{1}m$ | $6/m'm'm'$ |
| | 456 | $^{2_z}6/^{2_z}m^{2_x}m^{2_y}m$ | $6/mmm$ |
| | 457 | $^{2_z}6/^{2_z}m^{m_x}m^{m_y}m$ | $6/mm'm'$ |
| | 458 | $^{m_x}6/^{m_x}m^{2_z}m^{m_y}m$ | $6'/m'mm'$ |
| | 459 | $^{2}6/^{2}m^{\bar{1}}m^{m}m$ | $6/mm'm'$ |
| | 460 | $^{\bar{1}}6/^{\bar{1}}m^{2}m^{m}m$ | $6'/m'mm'$ |
| | 461 | $^{m}6/^{m}m^{2}m^{\bar{1}}m$ | $6/mm'm'$ |
| | 462 | $^{2_z}6/^{1}m^{2_x}m^{2_y}m$ | $6/mmm$ |
| | 463 | $^{2_z}6/^{1}m^{m_x}m^{m_y}m$ | $6/mm'm'$ |
| | 464 | $^{m_x}6/^{1}m^{2_z}m^{m_y}m$ | $6'/mmm'$ |
| | 465 | $^{2}6/^{1}m^{\bar{1}}m^{m}m$ | $6/mm'm'$ |
| | 466 | $^{\bar{1}}6/^{1}m^{2}m^{m}m$ | $6'/mmm'$ |
| | 467 | $^{m}6/^{1}m^{2}m^{\bar{1}}m$ | $6'/mmm'$ |
| | 468 | $^{1}6/^{2_z}m^{2_x}m^{2_x}m$ | $6/mmm$ |
| | 469 | $^{1}6/^{2_z}m^{m_x}m^{m_x}m$ | $6/mm'm'$ |
| | 470 | $^{1}6/^{m_x}m^{m_y}m^{m_y}m$ | $6/m'm'm'$ |
| | 471 | $^{1}6/^{m_x}m^{2_z}m^{2_z}m$ | $6/m'mm$ |
| | 472 | $^{1}6/^{2}m^{m}m^{m}m$ | $6/mm'm'$ |
| | 473 | $^{1}6/^{2}m^{\bar{1}}m^{\bar{1}}m$ | $6/mm'm'$ |
| | 474 | $^{1}6/^{\bar{1}}m^{m}m^{m}m$ | $6/m'm'm'$ |
| | 475 | $^{1}6/^{\bar{1}}m^{2}m^{2}m$ | $6/m'mm$ |
| | 476 | $^{1}6/^{m}m^{\bar{1}}m^{\bar{1}}m$ | $6/m'm'm'$ |
| | 477 | $^{1}6/^{m}m^{2}m^{2}m$ | $6/m'mm$ |
| | 478 | $^{2_z}6/^{2_x}m^{1}m^{2_z}m$ | $6/mmm$ |
| | 479 | $^{m_x}6/^{m_y}m^{1}m^{m_x}m$ | $6'/m'mm'$ |
| | 480 | $^{m_x}6/^{2_z}m^{1}m^{m_x}m$ | $6'/mmm'$ |
| | 481 | $^{2_z}6/^{m_x}m^{1}m^{2_z}m$ | $6/m'mm$ |
| | 482 | $^{m}6/^{\bar{1}}m^{1}m^{m}m$ | $6'/m'mm'$ |
| | 483 | $^{\bar{1}}6/^{m}m^{1}m^{\bar{1}}m$ | $6'/m'mm'$ |
| | 484 | $^{m}6/^{2}m^{1}m^{m}m$ | $6'/mmm'$ |
| | 485 | $^{2}6/^{m}m^{1}m^{2}m$ | $6/m'mm$ |
| | 486 | $^{\bar{1}}6/^{2}m^{1}m^{\bar{1}}m$ | $6'/mmm'$ |
| | 487 | $^{2}6/^{\bar{1}}m^{1}m^{2}m$ | $6/m'mm$ |
| | 488 | $^{2_z}6/^{2_x}m^{2_y}m^{2_x}m$ | $6/mmm$ |
| | 489 | $^{m_y}6/^{2_z}m^{m_x}m^{2_z}m$ | $6'/mmm'$ |
| | 490 | $^{m_y}6/^{m_x}m^{2_z}m^{m_x}m$ | $6'/m'mm'$ |
| | 491 | $^{2_z}6/^{m_x}m^{m_y}m^{m_x}m$ | $6/m'm'm'$ |
| | 492 | $^{m}6/^{2}m^{\bar{1}}m^{2}m$ | $6'/mmm'$ |
| | 493 | $^{\bar{1}}6/^{2}m^{m}m^{2}m$ | $6'/mmm'$ |
| | 494 | $^{m}6/^{\bar{1}}m^{2}m^{\bar{1}}m$ | $6'/m'mm'$ |
| | 495 | $^{2}6/^{\bar{1}}m^{m}m^{\bar{1}}m$ | $6/m'm'm'$ |
| | 496 | $^{\bar{1}}6/^{m}m^{2}m^{m}m$ | $6'/m'mm'$ |
| | 497 | $^{2}6/^{m}m^{\bar{1}}m^{m}m$ | $6/m'm'm'$ |
| | 498 | $^{3_z}6/^{1}m^{2_x}m^{2_{xy}}m$ | $6/mmm$ |
| | 499 | $^{3_z}6/^{1}m^{m_x}m^{m_{xy}}m$ | $6/mm'm'$ |
| | 500 | $^{m_z}6/^{2_z}m^{m_x}m^{2_y}m$ | $6'/mmm'$ |
| | 501 | $^{2_z}6/^{m_z}m^{m_x}m^{m_y}m$ | $6/m'm'm'$ |
| | 502 | $^{2_z}6/^{m_z}m^{2_x}m^{2_y}m$ | $6/m'mm$ |
| | 503 | $^{m_z}6/^{m_x}m^{2_x}m^{m_y}m$ | $6'/m'mm'$ |
| | 504 | $^{m_z}6/^{2_x}m^{m_x}m^{m_y}m$ | $6'/mmm'$ |
| | 505 | $^{2_z}6/^{2_x}m^{m_y}m^{m_x}m$ | $6/mm'm'$ |
| | 506 | $^{2_z}6/^{m_x}m^{2_y}m^{2_x}m$ | $6/m'mm$ |
| | 507 | $^{m_z}6/^{2_x}m^{\bar{1}}m^{2_z}m$ | $6'/mmm'$ |
| | 508 | $^{m_z}6/^{m_x}m^{2_z}m^{\bar{1}}m$ | $6'/m'mm'$ |
| | 509 | $^{m_z}6/^{\bar{1}}m^{2_x}m^{m_y}m$ | $6'/m'mm'$ |
| | 510 | $^{2_z}6/^{m_x}m^{m_z}m^{\bar{1}}m$ | $6/m'm'm'$ |
| | 511 | $^{2_z}6/^{2_x}m^{m_z}m^{\bar{1}}m$ | $6/mm'm'$ |
| | 512 | $^{2_z}6/^{\bar{1}}m^{m_x}m^{m_y}m$ | $6/m'm'm'$ |
| | 513 | $^{2_z}6/^{\bar{1}}m^{2_x}m^{2_y}m$ | $6/m'mm$ |
| | 514 | $^{\bar{1}}6/^{m_x}m^{2_z}m^{m_z}m$ | $6'/m'mm'$ |
| | 515 | $^{\bar{1}}6/^{2_x}m^{m_z}m^{2_z}m$ | $6'/mmm'$ |
| | **516** | $^{6_z}6/^{2_z}m^{2_x}m^{2_l}m$ | $6/mmm$ |
| | 517 | $^{\bar{3}_z}6/^{\bar{1}}m^{2_x}m^{m_{xy}}m$ | $6'/m'mm'$ |
| | **518** | $^{6_z}6/^{2_z}m^{m_x}m^{m_1}m$ | $6/mm'm'$ |
| | **519** | $^{\bar{6}_z}6/^{m_z}m^{m_x}m^{2_1}m$ | $6'/m'mm'$ |
| | 520 | $^{6_z}6/^{1}m^{2_x}m^{2_l}m$ | $6/mmm$ |
| | 521 | $^{\bar{3}_z}6/^{1}m^{m_x}m^{2_{xy}}m$ | $6'/mmm'$ |
| | 522 | $^{6_z}6/^{1}m^{m_x}m^{m_l}m$ | $6/mm'm'$ |
| | 523 | $^{\bar{6}_z}6/^{1}m^{2_x}m^{m_l}m$ | $6'/mmm'$ |
| | 524 | $^{3_z}6/^{2_z}m^{2_x}m^{2_{xy}}m$ | $6/mmm$ |
| | 525 | $^{3_z}6/^{\bar{1}}m^{m_x}m^{m_{xy}}m$ | $6/m'm'm'$ |

| Parent Group | Litvin # | Litvin Symbol | B&W Point Group |
|---|---|---|---|
| | 526 | $^{3_z}6/^{\bar{1}}m^{2_x}m^{2_{xy}}m$ | $6/m'mm$ |
| | 527 | $^{3_z}6/^{2_z}m^{m_x}m^{m_{xy}}m$ | $6/mm'm'$ |
| | 528 | $^{3_z}6/^{m_z}m^{m_x}m^{m_{xy}}m$ | $6/m'm'm'$ |
| | 529 | $^{3_z}6/^{m_z}m^{2_x}m^{2_{xy}}m$ | $6/m'mm$ |
| | 530 | $^{6_z}6/^{\bar{1}}m^{m_x}m^{m_l}m$ | $6/m'm'm'$ |
| | **531** | $^{6_z}6/^{m_z}m^{m_x}m^{m_l}m$ | $6/m'm'm'$ |
| | 532 | $^{6_z}6/^{\bar{1}}m^{2_x}m^{2_l}m$ | $6/m'mm$ |
| | **533** | $^{6_z}6/^{m_z}m^{2_x}m^{2_l}m$ | $6/m'mm$ |
| | 534 | $^{\bar{3}_z}6/^{m_z}m^{2_x}m^{m_{xy}}m$ | $6'/m'mm'$ |
| | 535 | $^{\bar{3}_z}6/^{2_z}m^{m_x}m^{2_{xy}}m$ | $6'/mmm'$ |
| | 536 | $^{\bar{6}_z}6/^{\bar{1}}m^{2_x}m^{m_l}m$ | $6'/m'mm'$ |
| | **537** | $^{\bar{6}_z}6/^{2_z}m^{m_x}m^{2_l}m$ | $6'/mmm'$ |
| **23** | 538 | $^{1}2^{1}3$ | $23$ |
| | 539 | $^{1}2^{3}3$ | $23$ |
| | **540** | $^{2_z}2^{3_{xyz}}3$ | $23$ |
| **2/m$\bar{3}$** | 541 | $^{1}2/^{1}m^{1}\bar{3}$ | $m\bar{3}$ |
| | 542 | $^{1}2/^{2}m^{2}\bar{3}$ | $m\bar{3}$ |
| | 543 | $^{1}2/^{m}m^{m}\bar{3}$ | $m'\bar{3}'$ |
| | 544 | $^{1}2/^{\bar{1}}m^{1}\bar{3}$ | $m'\bar{3}'$ |
| | 545 | $^{1}2/^{1}m^{3}\bar{3}$ | $m\bar{3}$ |
| | 546 | $^{1}2/^{2}m^{6}\bar{3}$ | $m\bar{3}$ |
| | 547 | $^{1}2/^{\bar{1}}m^{3}\bar{3}$ | $m'\bar{3}'$ |
| | 548 | $^{1}2/^{m}m^{6}\bar{3}$ | $m'\bar{3}'$ |
| | **549** | $^{2_z}2/^{2_z}m^{3_{xyz}}\bar{3}$ | $m\bar{3}$ |
| | **550** | $^{2_z}2/^{m_z}m^{\bar{3}_{xyz}}\bar{3}$ | $m'\bar{3}'$ |
| **$\bar{4}$3m** | 551 | $^{1}\bar{4}^{1}3^{1}m$ | $\bar{4}3m$ |
| | 552 | $^{2}\bar{4}^{1}3^{2}m$ | $\bar{4}3m$ |
| | 553 | $^{m}\bar{4}^{1}3^{m}m$ | $\bar{4}'3m'$ |
| | 554 | $^{\bar{1}}\bar{4}^{1}3^{\bar{1}}m$ | $\bar{4}'3m'$ |
| | 555 | $^{2_x}\bar{4}^{3_z}3^{2_x}m$ | $\bar{4}3m$ |
| | 556 | $^{m_x}\bar{4}^{3_z}3^{m_x}m$ | $\bar{4}'3m'$ |
| | **557** | $^{4_z}\bar{4}^{3_{xyz}}3^{2_{xy}}m$ | $\bar{4}3m$ |
| | **558** | $^{\bar{4}_z}\bar{4}^{3_{xyz}}2^{m_{xy}}m$ | $\bar{4}'3m'$ |
| **432** | 559 | $^{1}4^{1}3^{1}2$ | $432$ |
| | 560 | $^{2}4^{1}3^{2}2$ | $432$ |
| | 561 | $^{m}4^{1}3^{m}2$ | $4'32'$ |
| | 562 | $^{\bar{1}}4^{1}3^{\bar{1}}2$ | $4'32'$ |
| | 563 | $^{2_x}4^{3_z}3^{2_x}2$ | $432$ |

| Parent Group | Litvin # | Litvin Symbol | B&W Point Group |
|---|---|---|---|
| | 564 | $^{m_x}4^{3_z}3^{2_x}2$ | $4'32'$ |
| | **565** | $^{4_z}4^{3_{xyz}}3^{2_{xy}}2$ | $432$ |
| | **566** | $^{\bar{4}_z}4^{3_{xyz}}3^{m_{xy}}2$ | $4'32'$ |
| **m$\bar{3}$m** | 567 | $^{1}4/^{1}m^{1}\bar{3}^{1}2/^{1}m$ | $m\bar{3}m$ |
| | 568 | $^{2}4/^{1}m^{1}\bar{3}^{2}2/^{2}m$ | $m\bar{3}m$ |
| | 569 | $^{m}4/^{1}m^{1}\bar{3}^{m}2/^{m}m$ | $m\bar{3}m'$ |
| | 570 | $^{\bar{1}}4/_{1}m_{1}\bar{3}_{\bar{1}}2/_{\bar{1}}m$ | $m\bar{3}m'$ |
| | 571 | $^{2}4/^{2}m^{2}\bar{3}^{2}2/^{1}m$ | $m\bar{3}m$ |
| | 572 | $^{m}4/^{m}m^{m}\bar{3}^{m}2/^{1}m$ | $m'\bar{3}'m$ |
| | 573 | $^{\bar{1}}4/^{\bar{1}}m^{\bar{1}}\bar{3}^{\bar{1}}2/^{1}m$ | $m'\bar{3}'m$ |
| | 574 | $^{1}4/^{2}m^{2}\bar{3}^{1}2/^{2}m$ | $m\bar{3}m$ |
| | 575 | $^{1}4/^{m}m^{m}\bar{3}^{1}2/^{m}m$ | $m'\bar{3}'m'$ |
| | 576 | $^{1}4/^{\bar{1}}m^{\bar{1}}\bar{3}^{1}2/^{\bar{1}}m$ | $m'\bar{3}'m'$ |
| | 577 | $^{2_x}4/^{2_y}m^{2_y}\bar{3}^{2_x}2/^{2_z}m$ | $m\bar{3}m$ |
| | 578 | $^{2_z}4/^{m_x}m^{m_x}\bar{3}^{2_z}2/^{m_y}m$ | $m'\bar{3}'m'$ |
| | 579 | $^{m_x}4/^{2_z}m^{2_z}\bar{3}^{m_x}2/^{m_y}m$ | $m\bar{3}m'$ |
| | 580 | $^{m_x}4/^{m_y}m^{m_y}\bar{3}^{m_x}2/^{2_z}m$ | $m'\bar{3}'m$ |
| | 581 | $^{2}4/^{\bar{1}}m^{\bar{1}}3^{2}2/^{m}m$ | $m'\bar{3}'m'$ |
| | 582 | $^{2}4/^{m}m^{m}\bar{3}^{2}2/^{\bar{1}}m$ | $m'\bar{3}'m$ |
| | 583 | $^{\bar{1}}4/^{2}m^{2}\bar{3}^{\bar{1}}2/^{m}m$ | $m\bar{3}m'$ |
| | 584 | $^{\bar{1}}4/^{m}m^{m}\bar{3}^{\bar{1}}2/^{2}m$ | $m'\bar{3}'m$ |
| | 585 | $^{m}4/^{2}m^{2}\bar{3}^{m}2/^{\bar{1}}m$ | $m\bar{3}m'$ |
| | 586 | $^{m}4/^{\bar{1}}m^{\bar{1}}\bar{3}^{m}2/^{2}m$ | $m'\bar{3}'m$ |
| | 587 | $^{2_x}4/^{1}m^{3_z}\bar{3}^{2_x}2/^{2_x}m$ | $m\bar{3}m$ |
| | 588 | $^{m_x}4/^{1}m^{3_z}\bar{3}^{m_x}2/^{m_x}m$ | $m\bar{3}m'$ |
| | 589 | $^{2_z}4/^{2_z}m^{6_z}\bar{3}^{2_z}2/^{2_x}m$ | $m\bar{3}m$ |
| | 590 | $^{2_x}4/^{\bar{1}}m^{\bar{3}_z}\bar{3}^{2_x}2/^{m_x}m$ | $m'\bar{3}'m'$ |
| | 591 | $^{m_x}4/^{\bar{1}}m^{3}\bar{3}^{m_x}2/^{2_x}m$ | $m'\bar{3}'m$ |
| | 592 | $^{m_z}4/^{2_z}m^{6_z}\bar{3}^{m_z}2/^{m_x}m$ | $m\bar{3}m'$ |
| | 593 | $^{m_z}4/^{m_z}m^{\bar{6}_z}\bar{3}^{m_z}2/^{2_x}m$ | $m'\bar{3}'m$ |
| | 594 | $^{2_z}4/^{m_z}m^{\bar{6}_z}\bar{3}^{2}2/^{m_x}m$ | $m'\bar{3}'m'$ |
| | **595** | $^{4_z}4/^{2_z}m^{3_{xyz}}\bar{3}^{2_{xy}}2/^{2_{xy}}m$ | $m\bar{3}m$ |
| | **596** | $^{\bar{4}_z}4/^{2_z}m^{3_{xyz}}\bar{3}^{m_{xy}}2/^{m_{xy}}m$ | $m\bar{3}m'$ |
| | **597** | $^{4_z}4/^{m_z}m^{\bar{3}_{xyz}}\bar{3}^{2_{xy}}2/^{m_{xy}}m$ | $m'\bar{3}'m'$ |
| | **598** | $^{\bar{4}_z}4/^{m_z}m^{\bar{3}_{xyz}}\bar{3}^{m_{xy}}2/^{2_{xy}}m$ | $m'\bar{3}'m$ |

## C Isomorphism theorem for constructing spin groups

A general spin group **S** is constructed from a spatial parent group **G** and spin-space parent group **B** where **G**, **B** are the collections (groups) of all the right, left components of the elements $[\cdot\|\cdot]$ of **S** respectively. The spin group **S** is a subgroup of **B**×**G**, i.e. **S** ≤ **B**×**G**. Defining **g** = **G**∩**S** and **b** = **B** ∩ **S**, the isomorphism theorem of Litvin & Opechowski [18] states that **G/g** ≅ **B/b**.

To prove this, first note that $\mathbf{B}, \mathbf{G} \triangleleft \mathbf{B} \times \mathbf{G}$ and use Noether's 2nd isomorphism theorem[11] to conclude that $\mathbf{b}, \mathbf{g} \triangleleft \mathbf{S}$ as well as $\mathbf{S}/\mathbf{b} \cong \mathbf{BS}/\mathbf{B}$ and $\mathbf{S}/\mathbf{g} \cong \mathbf{GS}/\mathbf{G}$. The second set of isomorphisms can be understood in the following way. Given that, for example, $\mathbf{g} \triangleleft \mathbf{S}$, we can express $\mathbf{BS}$ as

$$\mathbf{BS} = \mathbf{B}[E\|\mathbf{g}] + \mathbf{B}[B_2\|G_2\mathbf{g}] + \ldots \mathbf{B}[B_n\|G_n\mathbf{g}] \cong \mathbf{B}[E\|\mathbf{g}] + \mathbf{B}[E\|G_2\mathbf{g}] + \ldots \mathbf{B}[E\|G_n\mathbf{g}],$$

since $\mathbf{B}B_i = \mathbf{B}$. So $\mathbf{BS}/\mathbf{B} \cong [E\|\mathbf{g}] + [E\|G_2\mathbf{g}] + \ldots [E\|G_n\mathbf{g}] \cong \mathbf{G}$. We have shown that

$$\mathbf{S}/\mathbf{b} \cong \mathbf{BS}/\mathbf{B} \cong \mathbf{G}. \tag{C.1}$$

Using an analogous argument,

$$\mathbf{S}/\mathbf{g} \cong \mathbf{GS}/\mathbf{G} \cong \mathbf{B}. \tag{C.2}$$

Next, notice that since $\mathbf{b}, \mathbf{g} \triangleleft \mathbf{S}$ with $\mathbf{b} \cap \mathbf{g} = [E\|E]$, it follows that $\mathbf{b} \times \mathbf{g} \triangleleft \mathbf{S}$. Now we have, for example, $\mathbf{b} \triangleleft \mathbf{b} \times \mathbf{g} \triangleleft \mathbf{S}$. Using Noether's third isomorphism theorem,[12] we have that

$$(\mathbf{S}/\mathbf{b})/(\mathbf{b} \times \mathbf{g}/\mathbf{b}) \cong \mathbf{S}/\mathbf{b} \times \mathbf{g}. \tag{C.3}$$

On the other hand, using equation C.1 and $\mathbf{b} \times \mathbf{g}/\mathbf{b} \cong \mathbf{g}$, we also have

$$(\mathbf{S}/\mathbf{b})/(\mathbf{b} \times \mathbf{g}/\mathbf{b}) \cong \mathbf{G}/\mathbf{g}. \tag{C.4}$$

Using similar arguments and equation C.2 we have

$$(\mathbf{S}/\mathbf{g})/(\mathbf{b} \times \mathbf{g}/\mathbf{g}) \cong \mathbf{S}/\mathbf{b} \times \mathbf{g} \cong \mathbf{B}/\mathbf{b}. \tag{C.5}$$

Comparing equations C.3, C.4 and C.5 we find that

$$\mathbf{G}/\mathbf{g} \cong \mathbf{B}/\mathbf{b}. \tag{C.6}$$

It can be shown that a general spin group $\mathbf{S}$ is of the form $\mathbf{b} \times \mathbf{S}^*$ where $\mathbf{S}^*$ contains no elements of the form $[B\|E]$ (a nontrivial spin group), and $\mathbf{b} = \mathbf{S} \cap \mathbf{B}$ contains *only* elements of the form $[B\|E]$ (a spin-only group). As a result, when searching for the nontrivial spin groups $\mathbf{S}^*$ with spatial parent group $\mathbf{G}$ and spin-space parent group $\mathbf{B}$, it is evident that $\mathbf{b}^* = \mathbf{S}^* \cap \mathbf{B} = \{E\}$, and we have $\mathbf{G}/\mathbf{g} \cong \mathbf{B}$ as was the case in Section 2.

# D   Direct products of antiunitary groups

Assume we have two groups $\mathbf{G}$ and $\mathbf{G}'$ expressable in the forms

$$\mathbf{G} = \mathbf{H} + s\mathbf{H}, \qquad \mathbf{G}' = \mathbf{K} + q\mathbf{K},$$

where $\mathbf{H}$ and $\mathbf{K}$ are halving subgroups of $\mathbf{G}$ and $\mathbf{G}'$ respectively, and $s = \tau u$ and $q = \tau u'$ where $\tau$ will for us be time reversal. We would like to express the product $\mathbf{M} = \mathbf{G} \times \mathbf{G}'$ in terms of $\mathbf{H}$, $\mathbf{K}$, $u$, $u'$, and $\tau$. In particular, we would like to get it in the form of a two-coset decomposition where the first coset is the unitary halving subgroup of the direct product.

For any $m \in \mathbf{M}$, we can express $m$ as $m = (g, g') = (s^\alpha h, q^\beta k)$ where $\alpha, \beta = 0, 1$ and $h \in \mathbf{H}$ and $k \in \mathbf{K}$. Then, define $\tilde{\mathbf{H}} = \{(h, E) | \mathbf{h} \in \mathbf{H}\} \cong \mathbf{H} < \mathbf{G} \triangleleft \mathbf{M}$. In this case, we can express $\mathbf{M}$ via the coset decomposition

$$\mathbf{M} = \tilde{\mathbf{H}} + (s, E)\tilde{\mathbf{H}} + (E, k_1)\tilde{\mathbf{H}} + (s, k_1)\tilde{\mathbf{H}} + \ldots + (E, k_\kappa)\tilde{\mathbf{H}} + (s, k_\kappa)\tilde{\mathbf{H}}$$
$$+ (E, q)\tilde{\mathbf{H}} + (s, q)\tilde{\mathbf{H}} + (E, qk_1)\tilde{\mathbf{H}} + (s, qk_1)\tilde{\mathbf{H}} + \ldots + (E, qk_\kappa)\tilde{\mathbf{H}} + (s, qk_\kappa)\tilde{\mathbf{H}},$$

---

[11] Let $\mathbf{H}$ be a group, with $\mathbf{J} < \mathbf{H}$ and $\mathbf{K} \triangleleft \mathbf{H}$. Then, $\mathbf{K} \cap \mathbf{J} \triangleleft \mathbf{J}$, and $\mathbf{J}/(\mathbf{K} \cap \mathbf{J}) \cong \mathbf{KJ}/\mathbf{K}$, where $\mathbf{JK} \equiv \{jk | j \in \mathbf{J}, k \in \mathbf{K}\}$.

[12] Let $\mathbf{K}, \mathbf{J} \triangleleft \mathbf{H}$ such that $\mathbf{K} < \mathbf{J} < \mathbf{H}$. Then, $(\mathbf{H}/\mathbf{K})/(\mathbf{J}/\mathbf{K}) \cong \mathbf{H}/\mathbf{J}$.

where $|\mathbf{K}| = \kappa$. We can then group this decomposition into four cosets of

$$\tilde{\mathbf{H}} + (E, k_1)\tilde{\mathbf{H}} + \ldots + (E, k_\kappa)\tilde{\mathbf{H}}$$

as follows:

$$\mathbf{M} = \left(\tilde{\mathbf{H}} + (E, k_1)\tilde{\mathbf{H}} + \ldots + (E, k_\kappa)\tilde{\mathbf{H}}\right) + (s, E)\left(\tilde{\mathbf{H}} + (E, k_1)\tilde{\mathbf{H}} + \ldots + (E, k_\kappa)\tilde{\mathbf{H}}\right)$$
$$+ (E, q)\left(\tilde{\mathbf{H}} + (E, k_1)\tilde{\mathbf{H}} + \ldots + (E, k_\kappa)\tilde{\mathbf{H}}\right) + (s, q)\left(\tilde{\mathbf{H}} + (E, k_1)\tilde{\mathbf{H}} + \ldots + (E, k_\kappa)\tilde{\mathbf{H}}\right).$$

In our case, if $\mathbf{G}$ and $\mathbf{G}'$ are spin groups, then $u$ and $u'$ can be expressed in the forms $[u_s\|u_o]$ and $[u'_s\|u'_o]$ respectively, and the element $(s, q)$ is in our case simply

$$sq = \tau[u_s\|u_o]\tau[u'_s\|u'_o] = uu' = (u, u').$$

Further, note that the outer direct product of $\mathbf{H}$ and $\mathbf{K}$ is $\mathbf{H} \times \mathbf{K} = \{(h, k)\|h \in \mathbf{H}, k \in \mathbf{K}\}$, such that $\tilde{\mathbf{H}}, \tilde{\mathbf{K}} \triangleleft \mathbf{H} \times \mathbf{K}$ and $\tilde{\mathbf{H}} \cap \tilde{\mathbf{K}} = (E, E)$, and this outer direct product can be expressed precisely as the coset decomposition

$$\mathbf{H} \times \mathbf{K} = \tilde{\mathbf{H}} + (E, k_1)\tilde{\mathbf{H}} + \ldots + (E, k_\kappa)\tilde{\mathbf{H}}.$$

As a result, we can express the direct product of $\mathbf{G}$ and $\mathbf{G}'$ as

$$\mathbf{M} = \mathbf{H} \times \mathbf{K} + (u, u')\mathbf{H} \times \mathbf{K} + \tau(u, E)\left(\mathbf{H} \times \mathbf{K} + (u, u')\mathbf{H} \times \mathbf{K}\right).$$

In the case that one of the two original groups is unitary, for example if $\mathbf{G}' = \mathbf{K}$, then this result reduces to $\mathbf{M} = \mathbf{H} \times \mathbf{K} + \tau(u, E)\mathbf{H} \times \mathbf{K}$.

# E  Inducing co-irreps of collinear spin point groups

Here we explain how to derive the co-irreps of the collinear spin point groups. In particular, we focus on those collinear groups $\mathbf{X} = \mathbf{b}^\infty \times \mathbf{S}$ whose nontrivial part $\mathbf{S}$ is non-unitary. In this case, we have first to find the irreps of the unitary halving group $\mathbf{X}_{1/2}$ (as introduced in Section 3.2.4). To find these, we use the method of induction from a subgroup of index two. This method which is derived and discussed in references [6, 8, 29, 30] is briefly reviewed in Section E.1. Then, in Section E.2 we present a detailed example of the co-irrep derivation for total collinear spin point group $\mathbf{X} = \mathbf{b}^\infty \times {}^{\overline{1}}4/{}^1m^1m^{\overline{1}}m$, which applies to the zone center in the altermagnetic rutile system introduced in Section 4.1.

## E.1  General technique

As explained in Appendix A, the irreducible co-representations of a non-unitary group can be obtained algorithmically from the irreps of a unitary halving subgroup with the help of the Dimmock test (equation A.20) and equations A.16, A.17 and A.18. In Section 3.2.4 we found that the unitary halving subgroups $\mathbf{X}_{1/2}$ for collinear spin point groups $\mathbf{X} = \mathbf{b}^\infty \times \mathbf{S}$ with non-unitary nontrivial part $\mathbf{S}$ have irreps that are *a priori* unknown. In this section, we describe the method of induction from a subgroup of index two,[13] which we use to induce the irreps of

$$\mathbf{X}_{1/2} = \mathbf{SO(2)} \times \mathbf{S}^* + [2_{\perp \mathbf{n}}\|s_o]\mathbf{SO(2)} \times \mathbf{S}^*,$$

from its index-two subgroup $\mathbf{SO(2)} \times \mathbf{S}^*$, where $\mathbf{S} = \mathbf{S}^* + [\tau\|s_o]\mathbf{S}^*$. The technique we will describe is general, and applies for any group and its subgroup(s) of index two. For derivations and other descriptions of the technique, we refer the reader to references [6, 8, 29, 30].

---

[13]The index of a subgroup $\mathbf{H}$ in $\mathbf{G}$ is the number of left cosets of $\mathbf{H}$ in $\mathbf{G}$.

Let the spatial parent group of $\mathbf{S}^*$ be called $\mathbf{G}_{1/2}$, and note that it is the halving subgroup of the parent spatial group of $\mathbf{S}$, that is, $\mathbf{G} = \mathbf{G}_{1/2} + s_o \mathbf{G}_{1/2}$. Further, let the irreducible representations of our index-two subgroup $\mathbf{SO(2)} \times \mathbf{S}^*$ be denoted by $d^{(\mu,\nu)}$, where integer $\mu$ denotes the irreps of $\mathbf{SO(2)}$ while $\nu$ runs through the irreducible representations of $\mathbf{S}^*$. Since we are considering spin groups with spin rotation symmetry $\mathbf{SO(2)}$, $\mathbf{S}^*$ is a unitary spin point group where each group element has trivial spin part (as explained in Section 2.4), i.e. $s \in \mathbf{S}^*$ are of the form $[E \| g]$ for $g \in \mathbf{G}_{1/2}$. Therefore $\mathbf{S}^*$ is the ordinary point group $\mathbf{G}_{1/2}$, and so $\nu$ runs through the irreps of $\mathbf{G}_{1/2}$ (this also follows directly from the isomorphism theorem introduced by Litvin and Opechowski [18, 25] and revisited in Section 3.1).

We induce the irreducible representations of $\mathbf{X}_{1/2}$ from the irreps $d^{(\mu,\nu)}(\mathbf{SO(2)} \times \mathbf{G}_{1/2})$. The algorithm for induction from an index-two subgroup consists of two main steps:

1. Finding the orbits of the irreducible representations of the index-two subgroup under conjugation by the elements of the total group. One finds that each orbit contains either one irrep or two irreps.

2. To calculate the irreducible representations of the total group, there is an expression corresponding to the orbits of size one (Eq. E.1 below) and another for the orbits of size two (Eq. E.3).

For $x \in \mathbf{X}_{1/2}$, an $x-$conjugated irrep $d_x^{(\mu,\nu)}$ is defined as follows:

$$d_x^{(\mu,\nu)} = \{ d^{(\mu,\nu)}(xyx^{-1}) \,|\, y \in \mathbf{SO(2)} \times \mathbf{G}_{1/2} \}.$$

Because all index-two subgroups are also normal subgroups, it follows that $xyx^{-1} \in \mathbf{SO(2)} \times \mathbf{G}_{1/2}$. The $x-$conjugated irrep can either be equivalent[14] to the original irrep $d_x^{(\mu,\nu)} \sim d^{(\mu,\nu)}$ or a different irrep, $d_x^{(\mu,\nu)} \sim d^{(\rho,\sigma)}$. We define the orbit $O^{(\mu,\nu)}$ of the irrep $d^{(\mu,\nu)}$ to be the collection of distinct irreps that $d_x^{(\mu,\nu)}$ is equivalent to for any $x \in \mathbf{X}_{1/2}$, i.e.

$$O^{(\mu,\nu)} = \{ d^{(\rho,\sigma)} \,|\, \exists\, x \in \mathbf{X}_{1/2} : d_x^{(\mu,\nu)} \sim d^{(\rho,\sigma)} \}.$$

For induction from an index-two subgroup, $O^{(\mu,\nu)}$ contains either one or two irreps, and it turns out that it is sufficient to conjugate only by the coset representative $[\, 2_{\perp\mathbf{n}} \| s_o \,]$ to determine whether the orbit is of size one or two.

The first step of the algorithm is to determine the orbit for every irrep of $\mathbf{SO(2)} \times \mathbf{G}_{1/2}$ under $x-$conjugation. Because we have a partially specified form of $\mathbf{X}_{1/2}$, we can already make some statements about the nature of these orbits. We know that

$$d^{(\mu,\nu)}(\mathbf{SO(2)} \times \mathbf{G}_{1/2}) = \delta^{(\mu)}(\mathbf{SO(2)}) \times \Delta^{(\nu)}(\mathbf{G}_{1/2}),$$

and more specifically $d^{(\mu,\nu)}(\mathbf{SO(2)} \times \mathbf{G}_{1/2})$ are of the form

$$d^{(\mu,\nu)}([\, R_{\mathbf{n}}(\varphi) \| g \,]) = e^{i\mu\varphi} \Delta^{(\nu)}(g),$$

where $R_{\mathbf{n}}(\varphi) \in \mathbf{SO(2)}$ is a rotation about the spin axis and $g \in \mathbf{G}_{1/2}$ is a point group element.

Take $y = [\, R_{\mathbf{n}} \| g \,] \in \mathbf{SO(2)} \times \mathbf{G}_{1/2}$. If we conjugate $y$ by $x = [\, R_{\mathbf{n}}(\psi) \| h \,] \in \mathbf{SO(2)} \times \mathbf{G}_{1/2}$,

$$\begin{aligned} xyx^{-1} &= [\, R_{\mathbf{n}}(\psi) \| h \,][\, R_{\mathbf{n}}(\varphi) \| g \,][\, R_{\mathbf{n}}(-\psi) \| h^{-1} \,] = [\, R_{\mathbf{n}}(\psi)R_{\mathbf{n}}(\varphi)R_{\mathbf{n}}(-\psi) \| hgh^{-1} \,] \\ &= [\, R_{\mathbf{n}}(\varphi) \| hgh^{-1} \,], \end{aligned}$$

---

[14]Two representations of a unitary group are equivalent if they have equal characters.

since rotations about the same axis commute with one another. If we conjugate by $[\,2_{\perp\mathbf{n}}\|s_o\,]x$, we find

$$[\,2_{\perp\mathbf{n}}\|s_o\,]xyx^{-1}[\,2_{\perp\mathbf{n}}\|s_o^{-1}\,] = [\,2_{\perp\mathbf{n}}R_{\mathbf{n}}(\varphi)2_{\perp\mathbf{n}}\|s_o hgh^{-1}s_o^{-1}\,] = [\,R_{\mathbf{n}}(-\varphi)\|s_o hgh^{-1}s_o^{-1}\,],$$

using an identity for conjugating rotations by rotations.[15] Immediately we see that every irrep $d^{(\mu,\nu)}$ will be mapped to the irrep $d^{(-\mu,\nu)}$ when conjugated by any element in the second coset of $\mathbf{X}_{1/2}$. As a result, all irreps with non-zero $\mu$ have orbits of size two, regardless of what happens to the point-group elements. Further, the only irreps with orbits of size one have $\mu = 0$. The orbits are only known once the $\mathbf{G}_{1/2}$ is specified. A detailed example is presented in Section E.2.

The second step of the algorithm is to take each orbit, pick one representative irrep from each orbit, and to them apply one of the following two formulae depending on the size of the orbit. We denote by $y$ the elements of $\mathbf{SO(2)} \times \mathbf{G}_{1/2}$. If the orbit for $d^{(\mu,\nu)}$ is of size one, then we obtain two induced irreducible representations of $\mathbf{X}_{1/2}$ that are given by

$$D^{(\mu,\nu,\pm)}(y) = d^{(\mu,\nu)}(y), \qquad D^{(\mu,\nu,\pm)}([\,2_{\perp\mathbf{n}}\|s_o\,]y) = \pm Z d^{(\mu,\nu)}(y), \qquad \text{(E.1)}$$

where $Z$ is the invertible matrix that imposes the similarity between $d^{(\mu,\nu)}$ and $d^{(\mu,\nu)}_{[2_{\perp\mathbf{n}}\|s_o]}$, that is

$$d^{(\mu,\nu)}_{[2_{\perp\mathbf{n}}\|s_o]}(y) = Z d^{(\mu,\nu)}(y)Z^{-1} \ \forall y \in \mathbf{SO(2)} \times \mathbf{G}_{1/2}, \qquad \text{(E.2)}$$

while also satisfying the constraint that $Z^2 = d^{(\mu,\nu)}([\,2_{\perp\mathbf{n}}\|s_o\,]^2)$.

If the orbit for $d^{(\mu,\nu)}$ is of size two, the induced irreducible representation of $\mathbf{X}_{1/2}$ is given by

$$D^{(\mu,\nu)}(y) = \begin{bmatrix} d^{(\mu,\nu)}(y) & 0 \\ 0 & d^{(\mu,\nu)}_{[2_{\perp\mathbf{n}}\|s_o]}(y) \end{bmatrix}, \quad D^{(\mu,\nu)}([2_{\perp\mathbf{n}}\|s_o]y) = \begin{bmatrix} 0 & \eta\, d^{(\mu,\nu)}_{[2_{\perp\mathbf{n}}\|s_o]}(y) \\ d^{(\mu,\nu)}(y) & 0 \end{bmatrix}, \text{(E.3)}$$

where $\eta = d^{(\mu,\nu)}([\,2_{\perp\mathbf{n}}\|s_o\,]^2)$. It is important to note that for the orbits of size two, the irreps induced from $d^{(\mu,\nu)}$ and $d^{(\mu,\nu)}_{[2_{\perp\mathbf{n}}\|s_o]} \sim d^{(\rho,\sigma)}$ are equivalent, and as a result if we select $d^{(\mu,\nu)}$ as the representative, the new irreps can no longer take on the labels given by $\rho$ and $\sigma$ since they would be redundant.

Choosing different coset representatives or orbit representatives gives rise to equivalent irreducible representations, so these formulae completely determine the irreducible representations of $\mathbf{X}_{1/2}$ in terms of the irreducible representations $d^{(\mu,\nu)}$ of $\mathbf{SO(2)} \times \mathbf{G}_{1/2}$.

Once the irreducible representations of the unitary halving subgroup of our total collinear spin point group are known, we follow the procedure for co-irrep induction as described in Appendix A.

## E.2 Example: $\mathbf{S} = {}^{\bar{1}}4/{}^{1}m^{1}m^{\bar{1}}m$

In Section 4.1 we introduced a model for rutile altermagnetism, where the zone center has symmetry corresponding to the collinear spin point group $\mathbf{X} = \mathbf{b}^{\infty} \times {}^{\bar{1}}4/{}^{1}m^{1}m^{\bar{1}}m$, whose nontrivial spin point group is the non-unitary group $\mathbf{S} = {}^{\bar{1}}4/{}^{1}m^{1}m^{\bar{1}}m$. Here, we derive the irreducible co-representations of this group as a demonstration of the technique used for all collinear spin point groups with non-unitary nontrivial part. First we express $\mathbf{X}$ in the form of a unitary halving subgroup joined with an antiunitary coset (following Appendix D). We can express $\mathbf{S}$ as ${}^{\bar{1}}4/{}^{1}m^{1}m^{\bar{1}}m = {}^{1}m^{1}m^{1}m + [\,\bar{1}\|4_z\,]{}^{1}m^{1}m^{1}m$, and so the unitary halving subgroup

---

[15] $R_{\mathbf{m}}(\psi)R_{\mathbf{n}}(\varphi)R_{\mathbf{m}}(-\psi) = R_{R_{\mathbf{m}}(\psi)\mathbf{n}}(\varphi)$, which can be demonstrated using Rodrigues' rotation formula.

of $\mathbf{S}$ is $\mathbf{S}^* = {}^1m^1m^1m = \mathbf{G}_{1/2}$, which is precisely the ordinary point group $D_{2h}$.[16] Now, the total collinear spin point group is given by

$$\mathbf{X} = \mathbf{X}_{1/2} + [\,\tau 2_{\perp\mathbf{n}} \,\|\, E\,]\mathbf{X}_{1/2}\,,$$

where

$$\mathbf{X}_{1/2} = \mathbf{SO(2)} \times {}^1m^1m^1m + [\,2_{\perp\mathbf{n}} \,\|\, 4_z\,]\,\mathbf{SO(2)} \times {}^1m^1m^1m\,.$$

In order to find the co-irreps of $\mathbf{X}$, we must first find the irreps of $\mathbf{X}_{1/2}$ via induction from an index-two subgroup, $\mathbf{SO(2)} \times {}^1m^1m^1m$. The irreducible representations for $[\,R_{\mathbf{n}}(\varphi)\|g\,] \in \mathbf{SO(2)} \times {}^1m^1m^1m$ are given by

$$d^{(\mu,\alpha)}([\,R_{\mathbf{n}}(\varphi)\|g\,]) = e^{i\mu\varphi}\,\Gamma_\alpha(g)\,,$$

where the irreducible representations $\Gamma_\alpha$ of $D_{2h} = mmm$ are given below.

| d2h | E | $C_{2z}$ | $C_{2x}$ | $C_{2y}$ | $IC_{2z}$ | $I$ | $IC_{2y}$ | $IC_{2x}$ |
|---|---|---|---|---|---|---|---|---|
| $\Gamma_1$ | 1 | 1 | 1 | 1 | 1 | 1 | 1 | 1 |
| $\Gamma_2$ | 1 | 1 | 1 | 1 | -1 | -1 | -1 | -1 |
| $\Gamma_3$ | 1 | 1 | -1 | -1 | 1 | 1 | -1 | -1 |
| $\Gamma_4$ | 1 | 1 | -1 | -1 | -1 | -1 | 1 | 1 |
| $\Gamma_5$ | 1 | -1 | 1 | -1 | 1 | -1 | 1 | -1 |
| $\Gamma_6$ | 1 | -1 | 1 | -1 | -1 | 1 | -1 | 1 |
| $\Gamma_7$ | 1 | -1 | -1 | 1 | 1 | -1 | -1 | 1 |
| $\Gamma_8$ | 1 | -1 | -1 | 1 | -1 | 1 | 1 | -1 |

Following the procedure in Section E.1, we first find the orbits of $d^{(\mu,\alpha)}$ under conjugation by $[\,2_{\perp\mathbf{n}}\|4_z\,]$. For $y = [\,R_{\mathbf{n}}(\varphi)\|g\,] \in \mathbf{SO(2)} \times {}^1m^1m^1m$, we find that

$$[\,2_{\perp\mathbf{n}}\|4_z\,]y[\,2_{\perp\mathbf{n}}\|4_z^{-1}\,] = [\,R_{\mathbf{n}}(-\varphi)\|4_z g 4_z^{-1}\,]\,.$$

If $g = IC_{2x}$ then $4_z IC_{2x} 4_z^{-1} = IC_{2y}$ and vice versa. The remaining elements of $mmm$ remain invariant under conjugation by $4_z$. By inspecting the $IC_{2x}$ and $IC_{2y}$ columns of the $mmm$ irreps, we see that $\Gamma_1$-$\Gamma_4$ remain invariant under $[\,2_{\perp\mathbf{n}}\|4_z\,]$−conjugation (thus having orbits of size one), whereas the remaining four irreps must have orbits of size two. In particular, because characters are functions of the conjugacy classes, we can identify that the $[\,2_{\perp\mathbf{n}}\|4_z\,]$−conjugated $\Gamma_5$ irrep is equivalent to $\Gamma_7$, and $\Gamma_6$ gets mapped to $\Gamma_8$ under this conjugation. From Section E.1, we also know that the $\mu$−th irrep of $\mathbf{SO(2)}$ is mapped to the $-\mu$-th irrep. So, we find that the orbits of the irreps $d^{(\mu,\alpha)}$ (where here $\mu$ takes on all non-zero integer values, and $\alpha$ runs through the irreps of $mmm$) are:

$$O^{(0,1)} = \{d^{(0,1)}\}\,,\qquad O^{(0,2)} = \{d^{(0,2)}\}\,,\qquad O^{(0,3)} = \{d^{(0,3)}\}\,,\qquad O^{(0,4)} = \{d^{(0,4)}\}\,,$$
$$O^{(\mu,1)} = \{d^{(\mu,1)}, d^{(-\mu,1)}\}\,,\qquad O^{(\mu,2)} = \{d^{(\mu,2)}, d^{(-\mu,2)}\}\,,\qquad O^{(\mu,3)} = \{d^{(\mu,3)}, d^{(-\mu,3)}\}\,,$$
$$O^{(\mu,4)} = \{d^{(\mu,4)}, d^{(-\mu,4)}\}\,,\qquad O^{(0,5)} = \{d^{(0,5)}, d^{(0,7)}\}\,,\qquad O^{(0,6)} = \{d^{(0,6)}, d^{(0,8)}\}\,,$$
$$O^{(\mu,5)} = \{d^{(\mu,5)}, d^{(-\mu,7)}\}\,,\qquad O^{(\mu,6)} = \{d^{(\mu,6)}, d^{(-\mu,8)}\}\,.$$

Now, we simply apply equations E.1 and E.3 to the orbits we just found. In total, one should find eight one-dimensional irreps (from the size-one orbits), and eight two-dimensional irreps (from the size-two orbits). We will show one example of the calculation for each type of orbit.

---

[16]Notice how the spatial parent group of $\mathbf{S}$ is $\mathbf{G} = 4/mmm$, and the spatial parent group of $\mathbf{S}^*$, $\mathbf{G}_{1/2} = mmm$, is a halving subgroup of $\mathbf{G}$.

As an example of calculating the irreducible representation for an orbit of size one, let us induce from $d^{(0,2)}$. Since $[\,2_{\perp\mathbf{n}}\,\|\,4_z\,]$—conjugation maps this irrep exactly onto itself, the matrix $Z$ appearing in equation E.1 is equal to one, the $1\times1$ identity matrix,[17] which is consistent with the constraint that $Z^2 = d^{(0,2)}([2_{\perp\mathbf{n}}\|4_z]^2) = d^{(0,2)}([E\|2_z])$. The two irreducible representations induced from $d^{(0,2)}$ are $D^{(0,2,\pm)}$. The elements of $\mathbf{SO(2)}\times{}^1m^1m^1m$ have the same matrix representation in $D^{(0,2,\pm)}$ as in $d^{(0,2)}$. The irrep matrices for the second coset are completely determined by the matrix for the coset representative $D^{(0,2,\pm)}([\,2_{\perp\mathbf{n}}\,\|\,4_z\,])$ by homomorphism, so we only list this matrix here:

$$D^{(0,2,\pm)}([\,2_{\perp\mathbf{n}}\,\|\,4_z\,]) = \pm 1\,,$$

by equation E.1.

As an example of the size two orbits, let us induce from $d^{(\mu,5)}$. Note that inducing from $d^{(\mu,5)}$ and $d^{(-\mu,7)}$ will produce equivalent representations. We list the irrep matrices for each of the generators of $\mathbf{X}_{1/2}$, which are $[\,R_\mathbf{n}(\varphi)\,\|\,E\,]$, $[\,E\,\|\,IC_{2z}\,]$, $[\,E\,\|\,IC_{2x}\,]$, and $[\,2_{\perp\mathbf{n}}\,\|\,4_z\,]$. The remaining group elements' irrep matrices can be found by homomorphism. Using equation E.3 the generators are represented by

$$D^{(\mu,5)}([\,R_\mathbf{n}(\varphi)\,\|\,E\,]) = \begin{bmatrix} e^{i\mu\varphi} & 0 \\ 0 & e^{-i\mu\varphi} \end{bmatrix}, \qquad D^{(\mu,5)}([\,E\,\|\,IC_{2z}\,]) = \begin{bmatrix} 1 & 0 \\ 0 & 1 \end{bmatrix},$$

$$D^{(\mu,5)}([\,E\,\|\,IC_{2x}\,]) = \begin{bmatrix} -1 & 0 \\ 0 & 1 \end{bmatrix}, \qquad D^{(\mu,5)}([\,2_{\perp\mathbf{n}}\,\|\,4_z\,]) = \begin{bmatrix} 0 & -1 \\ 1 & 0 \end{bmatrix}.$$

In this way, we can derive all of the irreducible representations of $\mathbf{X}_{1/2}$, and we are now ready to induce the co-irreps of $\mathbf{X} = \mathbf{X}_{1/2} + [\tau 2_{\perp\mathbf{n}}\|E]\mathbf{X}_{1/2}$, following the procedure laid out in Appendix A. For this example of $\mathbf{S} = {}^{\bar{1}}4/{}^1m^1m^{\bar{1}}m$, it is the case that all irreps of $\mathbf{X}_{1/2}$ belong to case (a).[18] This can be seen by first examining the elements appearing in the (combination integral and) sum of Dimmock's test (A.20), which are the squares of the antiunitary elements. If we let $x = [R_\mathbf{n}(\varphi)\|g] \in \mathbf{SO(2)}\times{}^1m^1m^1m$, then the elements in question are either

$$([\tau 2_{\perp\mathbf{n}}\|E]x)^2 = [2_{\perp\mathbf{n}}R_\mathbf{n}(\varphi)2_{\perp\mathbf{n}}R_\mathbf{n}(\varphi)\|E] = [E\|E]\,,$$

or

$$([\tau 2_{\perp\mathbf{n}}\|E][2_{\perp\mathbf{n}}\|4_z]x)^2 = [R_\mathbf{n}(2\varphi)\|(4_z g)^2] = \begin{cases} [R_\mathbf{n}(2\varphi)\|E], & g \in \{2_x, 2_y, I2_x, I2_y\}\,, \\ [R_\mathbf{n}(2\varphi)\|2_z], & g \in \{E, 2_z, I, I2_z\}\,. \end{cases}$$

In the second case, the integral over $\varphi$ from 0 to $2\pi$ gives zero, meaning only the first case contributes to the Dimmock indicator. Since the identity element is always represented by an identity matrix of the appropriate dimension, this Dimmock indicator will never be zero or negative, and so all irreps of $\mathbf{X}_{1/2}$ belong to case (a) of the Dimmock test. Using Equation A.16, one find the co-irreps for $\mathbf{b}^\infty \times {}^{\bar{1}}4/{}^1m^1m^{\bar{1}}m$ listed on page 70.

# F  Co-irreps for collinear total spin groups

In the tables that follow, $\sigma \in \mathbb{Z}$, $\mu \in \mathbb{Z}\setminus\{0\}$, and $\nu \in \mathbb{N}\setminus\{0\}$. An index of the co-irrep tables is provided for both unitary and non-unitary cases.

---

[17]An irrep in an orbit of size one will *not* generically be mapped exactly onto itself under conjugation by the coset representative. In our own calculations, we have used Mathematica to solve the simultaneous set of linear equations given by equation E.2.

[18]Generically, this is not the case for the total collinear spin point groups.

Table 8: Index of the co-irrep tables for the 32 unitary nontrivial groups.

| Type of $\mathbf{S}$ | Group | Page # |
|---|---|---|
| Unitary | $\mathbf{b}^\infty \times {}^1 1$ | 47 |
| | $\mathbf{b}^\infty \times {}^1 2$ | 47 |
| | $\mathbf{b}^\infty \times {}^1 m$ | 47 |
| | $\mathbf{b}^\infty \times {}^1\overline{1}$ | 47 |
| | $\mathbf{b}^\infty \times {}^1 3$ | 47 |
| | $\mathbf{b}^\infty \times {}^1 4$ | 47 |
| | $\mathbf{b}^\infty \times {}^1 2/{}^1 m$ | 48 |
| | $\mathbf{b}^\infty \times {}^1 m{}^1 m{}^1 2$ | 48 |
| | $\mathbf{b}^\infty \times {}^1 2{}^1 2{}^1 2$ | 48 |
| | $\mathbf{b}^\infty \times {}^1\overline{4}$ | 48 |
| | $\mathbf{b}^\infty \times {}^1 6$ | 49 |
| | $\mathbf{b}^\infty \times {}^1\overline{6}$ | 49 |
| | $\mathbf{b}^\infty \times {}^1\overline{3}$ | 49 |
| | $\mathbf{b}^\infty \times {}^1 3{}^1 m$ | 50 |
| | $\mathbf{b}^\infty \times {}^1 3{}^1 2$ | 50 |
| | $\mathbf{b}^\infty \times {}^1 4/{}^1 m$ | 51 |
| | $\mathbf{b}^\infty \times {}^1 4{}^1 m{}^1 m$ | 51 |
| | $\mathbf{b}^\infty \times {}^1 4{}^1 2{}^1 2$ | 52 |
| | $\mathbf{b}^\infty \times {}^1 m{}^1 m{}^1 m$ | 52 |
| | $\mathbf{b}^\infty \times {}^1\overline{4}{}^1 2{}^1 m$ | 53 |
| | $\mathbf{b}^\infty \times {}^1 6/{}^1 m$ | 53 |
| | $\mathbf{b}^\infty \times {}^1 6{}^1 m{}^1 m$ | 54 |
| | $\mathbf{b}^\infty \times {}^1 6{}^1 2{}^1 2$ | 54 |
| | $\mathbf{b}^\infty \times {}^1\overline{6}{}^1 m{}^1 2$ | 55 |
| | $\mathbf{b}^\infty \times {}^1\overline{3}{}^1 m$ | 55 |
| | $\mathbf{b}^\infty \times {}^1 2{}^1 3$ | 56 |
| | $\mathbf{b}^\infty \times {}^1 4/{}^1 m{}^1 m{}^1 m$ | 56 |
| | $\mathbf{b}^\infty \times {}^1 6/{}^1 m{}^1 m{}^1 m$ | 57 |
| | $\mathbf{b}^\infty \times {}^1 2/{}^1 m{}^1\overline{3}$ | 57 |
| | $\mathbf{b}^\infty \times {}^1\overline{4}{}^1 3{}^1 m$ | 58 |
| | $\mathbf{b}^\infty \times {}^1 4{}^1 3{}^1 2$ | 58 |
| | $\mathbf{b}^\infty \times {}^1 4/{}^1 m{}^1\overline{3}{}^1 2/{}^1 m$ | 59 |

Table 9: Index of the co-irrep tables for the 58 non-unitary nontrivial spin groups.

| Type of **S** | Group | Page # |
|---|---|---|
| Non-unitary | $\mathbf{b}^\infty \times \bar{1}\bar{1}$ | 60 |
| | $\mathbf{b}^\infty \times \bar{1}2$ | 60 |
| | $\mathbf{b}^\infty \times \bar{1}m$ | 60 |
| | $\mathbf{b}^\infty \times {}^1 2/\bar{1}m$ | 60 |
| | $\mathbf{b}^\infty \times \bar{1}2/{}^1m$ | 60 |
| | $\mathbf{b}^\infty \times \bar{1}2/\bar{1}m$ | 61 |
| | $\mathbf{b}^\infty \times \bar{1}m\bar{1}m^1 2$ | 61 |
| | $\mathbf{b}^\infty \times {}^1 m\bar{1}m\bar{1} 2$ | 61 |
| | $\mathbf{b}^\infty \times {}^1 2\bar{1}2\bar{1}2$ | 61 |
| | $\mathbf{b}^\infty \times {}^1 m\bar{1}m\bar{1}m$ | 62 |
| | $\mathbf{b}^\infty \times {}^1 m^1 m\bar{1}m$ | 62 |
| | $\mathbf{b}^\infty \times \bar{1}m\bar{1}m\bar{1}m$ | 63 |
| | $\mathbf{b}^\infty \times \bar{1}4$ | 64 |
| | $\mathbf{b}^\infty \times \bar{1}\bar{4}$ | 64 |
| | $\mathbf{b}^\infty \times \bar{1}4/{}^1m$ | 65 |
| | $\mathbf{b}^\infty \times \bar{1}4/{}^1m$ | 65 |
| | $\mathbf{b}^\infty \times {}^1 4/\bar{1}m$ | 66 |
| | $\mathbf{b}^\infty \times {}^1 4\bar{1}2\bar{1}2$ | 66 |
| | $\mathbf{b}^\infty \times \bar{1}4^1 2\bar{1}2$ | 67 |
| | $\mathbf{b}^\infty \times \bar{1}4^1 m\bar{1}m$ | 67 |
| | $\mathbf{b}^\infty \times \bar{1}\bar{4}^1 2^1 m$ | 67 |
| | $\mathbf{b}^\infty \times \bar{1}\bar{4}^1 2\bar{1}m$ | 67 |
| | $\mathbf{b}^\infty \times {}^1 4\bar{1}m\bar{1}m$ | 68 |
| | $\mathbf{b}^\infty \times {}^1 \bar{4}^{\bar{1}} m\bar{1}m$ | 68 |
| | $\mathbf{b}^\infty \times \bar{1}4/{}^1m\bar{1}m^1 m$ | 69 |
| | $\mathbf{b}^\infty \times {}^1 4/\bar{1}m^1 m^1 m$ | 69 |
| | $\mathbf{b}^\infty \times \bar{1}4/{}^1m^1 m\bar{1}m$ | 70 |
| | $\mathbf{b}^\infty \times {}^1 4/{}^1m\bar{1}m\bar{1}m$ | 71 |
| | $\mathbf{b}^\infty \times {}^1 4/\bar{1}m\bar{1}m\bar{1}m$ | 71 |
| | $\mathbf{b}^\infty \times \bar{1}\bar{3}$ | 72 |
| | $\mathbf{b}^\infty \times {}^1 3\bar{1}2$ | 73 |
| | $\mathbf{b}^\infty \times {}^1 3\bar{1}m$ | 73 |
| | $\mathbf{b}^\infty \times {}^1 3\bar{1}m$ | 74 |
| | $\mathbf{b}^\infty \times {}^1 \bar{3}^{\bar{1}} m$ | 74 |
| | $\mathbf{b}^\infty \times \bar{1}\bar{3}^1 m$ | 75 |
| | $\mathbf{b}^\infty \times \bar{1}\bar{3}^{\bar{1}} m$ | 76 |
| | $\mathbf{b}^\infty \times \bar{1}\bar{6}$ | 76 |
| | $\mathbf{b}^\infty \times \bar{1}6$ | 77 |
| | $\mathbf{b}^\infty \times {}^1 6\bar{1}2\bar{1}2$ | 77 |

Table 10: Index of the co-irrep tables for the 58 non-unitary nontrivial spin groups.

| Type of **S** | Group | Page # |
|---|---|---|
| Non-Unitary | $\mathbf{b}^\infty \times {}^1 6^{\bar{1}} 2^{\bar{1}} 2$ | 78 |
| | $\mathbf{b}^\infty \times {}^{\bar{1}} 6 / {}^{\bar{1}} m$ | 79 |
| | $\mathbf{b}^\infty \times {}^{\bar{1}} 6 / {}^1 m$ | 80 |
| | $\mathbf{b}^\infty \times {}^1 6 / {}^{\bar{1}} m$ | 81 |
| | $\mathbf{b}^\infty \times {}^1 6^{\bar{1}} m^{\bar{1}} m$ | 82 |
| | $\mathbf{b}^\infty \times {}^{\bar{1}} 6^1 m^{\bar{1}} m$ | 82 |
| | $\mathbf{b}^\infty \times {}^1 6^{\bar{1}} m^{\bar{1}} 2$ | 82 |
| | $\mathbf{b}^\infty \times {}^{\bar{1}} 6^1 m^{\bar{1}} 2$ | 83 |
| | $\mathbf{b}^\infty \times {}^{\bar{1}} 6^{\bar{1}} m^1 2$ | 83 |
| | $\mathbf{b}^\infty \times {}^{\bar{1}} 6 / {}^1 m^1 m^{\bar{1}} m$ | 84 |
| | $\mathbf{b}^\infty \times {}^{\bar{1}} 6 / {}^1 m^{\bar{1}} m^{\bar{1}} m$ | 85 |
| | $\mathbf{b}^\infty \times {}^1 6 / {}^1 m^{\bar{1}} m^{\bar{1}} m$ | 86 |
| | $\mathbf{b}^\infty \times {}^1 6 / {}^{\bar{1}} m^1 m^1 m$ | 87 |
| | $\mathbf{b}^\infty \times {}^1 6 / {}^{\bar{1}} m^{\bar{1}} m^{\bar{1}} m$ | 88 |
| | $\mathbf{b}^\infty \times {}^1 2 / {}^{\bar{1}} m \bar{3}$ | 89 |
| | $\mathbf{b}^\infty \times {}^{\bar{1}} \bar{4}^1 3^{\bar{1}} m$ | 90 |
| | $\mathbf{b}^\infty \times {}^{\bar{1}} \bar{4}^1 3^{\bar{1}} 2$ | 90 |
| | $\mathbf{b}^\infty \times {}^{\bar{1}} 4 / {}^1 m^1 \bar{3}^{\bar{1}} 2 / {}^{\bar{1}} m$ | 91 |
| | $\mathbf{b}^\infty \times {}^{\bar{1}} 4 / {}^{\bar{1}} m^{\bar{1}} \bar{3}^{\bar{1}} 2 / {}^1 m$ | 92 |
| | $\mathbf{b}^\infty \times {}^1 4 / {}^{\bar{1}} m^{\bar{1}} \bar{3}^1 2 / {}^{\bar{1}} m$ | 93 |

## F.1 Collinear groups with unitary nontrivial group

| $b^\infty \times {}^1 1$ | $\{R_\varphi, E\}$ | $\{\tau 2_\perp R_\varphi, E\}$ |
|---|---|---|
| $\Gamma_1$ | $e^{i\sigma\varphi}$ | $e^{i\sigma\varphi}$ |

| $b^\infty \times {}^1 2$ | $\{R_\varphi, E\}$ | $\left\{R_\varphi, C_{2z}\right\}$ | $\{\tau 2_\perp R_\varphi, E\}$ | $\left\{\tau 2_\perp R_\varphi, C_{2z}\right\}$ |
|---|---|---|---|---|
| $\Gamma_1$ | $e^{i\sigma\varphi}$ | $e^{i\sigma\varphi}$ | $e^{i\sigma\varphi}$ | $e^{i\sigma\varphi}$ |
| $\Gamma_2$ | $e^{i\sigma\varphi}$ | $-e^{i\sigma\varphi}$ | $e^{i\sigma\varphi}$ | $-e^{i\sigma\varphi}$ |

| $b^\infty \times {}^1 m$ | $\{R_\varphi, E\}$ | $\left\{R_\varphi, IC_{2z}\right\}$ | $\{\tau 2_\perp R_\varphi, E\}$ | $\left\{\tau 2_\perp R_\varphi, IC_{2z}\right\}$ |
|---|---|---|---|---|
| $\Gamma_1$ | $e^{i\sigma\varphi}$ | $e^{i\sigma\varphi}$ | $e^{i\sigma\varphi}$ | $e^{i\sigma\varphi}$ |
| $\Gamma_2$ | $e^{i\sigma\varphi}$ | $-e^{i\sigma\varphi}$ | $e^{i\sigma\varphi}$ | $-e^{i\sigma\varphi}$ |

| $b^\infty \times {}^1 \bar{1}$ | $\{R_\varphi, E\}$ | $\{R_\varphi, I\}$ | $\{\tau 2_\perp R_\varphi, E\}$ | $\{\tau 2_\perp R_\varphi, I\}$ |
|---|---|---|---|---|
| $\Gamma_1$ | $e^{i\sigma\varphi}$ | $e^{i\sigma\varphi}$ | $e^{i\sigma\varphi}$ | $e^{i\sigma\varphi}$ |
| $\Gamma_2$ | $e^{i\sigma\varphi}$ | $-e^{i\sigma\varphi}$ | $e^{i\sigma\varphi}$ | $-e^{i\sigma\varphi}$ |

| $b^\infty \times {}^1 3$ | $\{R_\varphi, E\}$ | $\left\{R_\varphi, C_{3z}\right\}$ | $\{\tau 2_\perp R_\varphi, E\}$ | $\left\{\tau 2_\perp R_\varphi, C_{3z}\right\}$ |
|---|---|---|---|---|
| $\Gamma_1$ | $e^{i\sigma\varphi}$ | $e^{i\sigma\varphi}$ | $e^{i\sigma\varphi}$ | $e^{i\sigma\varphi}$ |
| $\Gamma_2$ | $\begin{pmatrix} e^{i\sigma\varphi} & 0 \\ 0 & e^{i\sigma\varphi} \end{pmatrix}$ | $\begin{pmatrix} e^{\frac{2i\pi}{3}+i\sigma\varphi} & 0 \\ 0 & e^{-\frac{2i\pi}{3}+i\sigma\varphi} \end{pmatrix}$ | $\begin{pmatrix} 0 & e^{i\sigma\varphi} \\ e^{i\sigma\varphi} & 0 \end{pmatrix}$ | $\begin{pmatrix} 0 & e^{\frac{2i\pi}{3}+i\sigma\varphi} \\ e^{-\frac{2i\pi}{3}+i\sigma\varphi} & 0 \end{pmatrix}$ |

| $b^\infty \times {}^1 4$ | $\{R_\varphi, E\}$ | $\left\{R_\varphi, C_{4z}\right\}$ | $\{\tau 2_\perp R_\varphi, E\}$ | $\left\{\tau 2_\perp R_\varphi, C_{4z}\right\}$ |
|---|---|---|---|---|
| $\Gamma_1$ | $e^{i\sigma\varphi}$ | $e^{i\sigma\varphi}$ | $e^{i\sigma\varphi}$ | $e^{i\sigma\varphi}$ |
| $\Gamma_2$ | $e^{i\sigma\varphi}$ | $-e^{i\sigma\varphi}$ | $e^{i\sigma\varphi}$ | $-e^{i\sigma\varphi}$ |
| $\Gamma_3$ | $\begin{pmatrix} e^{i\sigma\varphi} & 0 \\ 0 & e^{i\sigma\varphi} \end{pmatrix}$ | $\begin{pmatrix} -i\, e^{i\sigma\varphi} & 0 \\ 0 & i\, e^{i\sigma\varphi} \end{pmatrix}$ | $\begin{pmatrix} 0 & e^{i\sigma\varphi} \\ e^{i\sigma\varphi} & 0 \end{pmatrix}$ | $\begin{pmatrix} 0 & -i\, e^{i\sigma\varphi} \\ i\, e^{i\sigma\varphi} & 0 \end{pmatrix}$ |

| $b^\infty \times {}^1 2/{}^1 m$ | $\{R_\varphi, E\}$ | $\{R_\varphi, C_{2z}\}$ | $\{R_\varphi, IC_{2z}\}$ | $\{\tau 2_\perp R_\varphi, E\}$ | $\{\tau 2_\perp R_\varphi, C_{2z}\}$ | $\{\tau 2_\perp R_\varphi, IC_{2z}\}$ |
|---|---|---|---|---|---|---|
| $\Gamma_1$ | $e^{i\sigma\varphi}$ | $e^{i\sigma\varphi}$ | $e^{i\sigma\varphi}$ | $e^{i\sigma\varphi}$ | $e^{i\sigma\varphi}$ | $e^{i\sigma\varphi}$ |
| $\Gamma_2$ | $e^{i\sigma\varphi}$ | $e^{i\sigma\varphi}$ | $-e^{i\sigma\varphi}$ | $e^{i\sigma\varphi}$ | $e^{i\sigma\varphi}$ | $-e^{i\sigma\varphi}$ |
| $\Gamma_3$ | $e^{i\sigma\varphi}$ | $-e^{i\sigma\varphi}$ | $e^{i\sigma\varphi}$ | $e^{i\sigma\varphi}$ | $-e^{i\sigma\varphi}$ | $e^{i\sigma\varphi}$ |
| $\Gamma_4$ | $e^{i\sigma\varphi}$ | $-e^{i\sigma\varphi}$ | $-e^{i\sigma\varphi}$ | $e^{i\sigma\varphi}$ | $-e^{i\sigma\varphi}$ | $-e^{i\sigma\varphi}$ |

| $b^\infty \times {}^1 m \, {}^1 m \, {}^1 2$ | $\{R_\varphi, E\}$ | $\{R_\varphi, IC_{2x}\}$ | $\{R_\varphi, IC_{2y}\}$ | $\{\tau 2_\perp R_\varphi, E\}$ | $\{\tau 2_\perp R_\varphi, IC_{2x}\}$ | $\{\tau 2_\perp R_\varphi, IC_{2y}\}$ |
|---|---|---|---|---|---|---|
| $\Gamma_1$ | $e^{i\sigma\varphi}$ | $e^{i\sigma\varphi}$ | $e^{i\sigma\varphi}$ | $e^{i\sigma\varphi}$ | $e^{i\sigma\varphi}$ | $e^{i\sigma\varphi}$ |
| $\Gamma_2$ | $e^{i\sigma\varphi}$ | $-e^{i\sigma\varphi}$ | $-e^{i\sigma\varphi}$ | $e^{i\sigma\varphi}$ | $-e^{i\sigma\varphi}$ | $-e^{i\sigma\varphi}$ |
| $\Gamma_3$ | $e^{i\sigma\varphi}$ | $e^{i\sigma\varphi}$ | $-e^{i\sigma\varphi}$ | $e^{i\sigma\varphi}$ | $e^{i\sigma\varphi}$ | $-e^{i\sigma\varphi}$ |
| $\Gamma_4$ | $e^{i\sigma\varphi}$ | $-e^{i\sigma\varphi}$ | $e^{i\sigma\varphi}$ | $e^{i\sigma\varphi}$ | $-e^{i\sigma\varphi}$ | $e^{i\sigma\varphi}$ |

| $b^\infty \times {}^1 2 \, {}^1 2 \, {}^1 2$ | $\{R_\varphi, E\}$ | $\{R_\varphi, C_{2x}\}$ | $\{R_\varphi, C_{2y}\}$ | $\{\tau 2_\perp R_\varphi, E\}$ | $\{\tau 2_\perp R_\varphi, C_{2x}\}$ | $\{\tau 2_\perp R_\varphi, C_{2y}\}$ |
|---|---|---|---|---|---|---|
| $\Gamma_1$ | $e^{i\sigma\varphi}$ | $e^{i\sigma\varphi}$ | $e^{i\sigma\varphi}$ | $e^{i\sigma\varphi}$ | $e^{i\sigma\varphi}$ | $e^{i\sigma\varphi}$ |
| $\Gamma_2$ | $e^{i\sigma\varphi}$ | $-e^{i\sigma\varphi}$ | $-e^{i\sigma\varphi}$ | $e^{i\sigma\varphi}$ | $-e^{i\sigma\varphi}$ | $-e^{i\sigma\varphi}$ |
| $\Gamma_3$ | $e^{i\sigma\varphi}$ | $e^{i\sigma\varphi}$ | $-e^{i\sigma\varphi}$ | $e^{i\sigma\varphi}$ | $e^{i\sigma\varphi}$ | $-e^{i\sigma\varphi}$ |
| $\Gamma_4$ | $e^{i\sigma\varphi}$ | $-e^{i\sigma\varphi}$ | $e^{i\sigma\varphi}$ | $e^{i\sigma\varphi}$ | $-e^{i\sigma\varphi}$ | $e^{i\sigma\varphi}$ |

| $b^\infty \times {}^1\overline{4}$ | $\{R_\varphi, E\}$ | $\{R_\varphi, IC_{4z}^{-1}\}$ | $\{\tau 2_\perp R_\varphi, E\}$ | $\{\tau 2_\perp R_\varphi, IC_{4z}^{-1}\}$ |
|---|---|---|---|---|
| $\Gamma_1$ | $e^{i\sigma\varphi}$ | $e^{i\sigma\varphi}$ | $e^{i\sigma\varphi}$ | $e^{i\sigma\varphi}$ |
| $\Gamma_2$ | $e^{i\sigma\varphi}$ | $-e^{i\sigma\varphi}$ | $e^{i\sigma\varphi}$ | $-e^{i\sigma\varphi}$ |
| $\Gamma_3$ | $\begin{pmatrix} e^{i\sigma\varphi} & 0 \\ 0 & e^{i\sigma\varphi} \end{pmatrix}$ | $\begin{pmatrix} i\, e^{i\sigma\varphi} & 0 \\ 0 & -i\, e^{i\sigma\varphi} \end{pmatrix}$ | $\begin{pmatrix} 0 & e^{i\sigma\varphi} \\ e^{i\sigma\varphi} & 0 \end{pmatrix}$ | $\begin{pmatrix} 0 & i\, e^{i\sigma\varphi} \\ -i\, e^{i\sigma\varphi} & 0 \end{pmatrix}$ |

| $b^\infty \times {}^1 6$ | $\{R_\varphi,\ E\}$ | $\left\{R_\varphi,\ C_{6z}\right\}$ | $\{\tau 2_\perp R_\varphi,\ E\}$ | $\left\{\tau 2_\perp R_\varphi,\ C_{6z}\right\}$ |
|---|---|---|---|---|
| $\Gamma_1$ | $e^{i\sigma\varphi}$ | $e^{i\sigma\varphi}$ | $e^{i\sigma\varphi}$ | $e^{i\sigma\varphi}$ |
| $\Gamma_2$ | $e^{i\sigma\varphi}$ | $-e^{i\sigma\varphi}$ | $e^{i\sigma\varphi}$ | $-e^{i\sigma\varphi}$ |
| $\Gamma_3$ | $\begin{pmatrix} e^{i\sigma\varphi} & 0 \\ 0 & e^{i\sigma\varphi} \end{pmatrix}$ | $\begin{pmatrix} e^{-\frac{2i\pi}{3}+i\sigma\varphi} & 0 \\ 0 & e^{\frac{2i\pi}{3}+i\sigma\varphi} \end{pmatrix}$ | $\begin{pmatrix} 0 & e^{i\sigma\varphi} \\ e^{i\sigma\varphi} & 0 \end{pmatrix}$ | $\begin{pmatrix} 0 & e^{-\frac{2i\pi}{3}+i\sigma\varphi} \\ e^{\frac{2i\pi}{3}+i\sigma\varphi} & 0 \end{pmatrix}$ |
| $\Gamma_4$ | $\begin{pmatrix} e^{i\sigma\varphi} & 0 \\ 0 & e^{i\sigma\varphi} \end{pmatrix}$ | $\begin{pmatrix} -e^{-\frac{2i\pi}{3}+i\sigma\varphi} & 0 \\ 0 & -e^{\frac{2i\pi}{3}+i\sigma\varphi} \end{pmatrix}$ | $\begin{pmatrix} 0 & e^{i\sigma\varphi} \\ e^{i\sigma\varphi} & 0 \end{pmatrix}$ | $\begin{pmatrix} 0 & -e^{-\frac{2i\pi}{3}+i\sigma\varphi} \\ -e^{\frac{2i\pi}{3}+i\sigma\varphi} & 0 \end{pmatrix}$ |

| $b^\infty \times {}^1\bar{6}$ | $\{R_\varphi,\ E\}$ | $\left\{R_\varphi,\ IC_{6z}^{-1}\right\}$ | $\{\tau 2_\perp R_\varphi,\ E\}$ | $\left\{\tau 2_\perp R_\varphi,\ IC_{6z}^{-1}\right\}$ |
|---|---|---|---|---|
| $\Gamma_1$ | $e^{i\sigma\varphi}$ | $e^{i\sigma\varphi}$ | $e^{i\sigma\varphi}$ | $e^{i\sigma\varphi}$ |
| $\Gamma_2$ | $e^{i\sigma\varphi}$ | $-e^{i\sigma\varphi}$ | $e^{i\sigma\varphi}$ | $-e^{i\sigma\varphi}$ |
| $\Gamma_3$ | $\begin{pmatrix} e^{i\sigma\varphi} & 0 \\ 0 & e^{i\sigma\varphi} \end{pmatrix}$ | $\begin{pmatrix} e^{\frac{2i\pi}{3}+i\sigma\varphi} & 0 \\ 0 & e^{-\frac{2i\pi}{3}+i\sigma\varphi} \end{pmatrix}$ | $\begin{pmatrix} 0 & e^{i\sigma\varphi} \\ e^{i\sigma\varphi} & 0 \end{pmatrix}$ | $\begin{pmatrix} 0 & e^{\frac{2i\pi}{3}+i\sigma\varphi} \\ e^{-\frac{2i\pi}{3}+i\sigma\varphi} & 0 \end{pmatrix}$ |
| $\Gamma_4$ | $\begin{pmatrix} e^{i\sigma\varphi} & 0 \\ 0 & e^{i\sigma\varphi} \end{pmatrix}$ | $\begin{pmatrix} -e^{\frac{2i\pi}{3}+i\sigma\varphi} & 0 \\ 0 & -e^{-\frac{2i\pi}{3}+i\sigma\varphi} \end{pmatrix}$ | $\begin{pmatrix} 0 & e^{i\sigma\varphi} \\ e^{i\sigma\varphi} & 0 \end{pmatrix}$ | $\begin{pmatrix} 0 & -e^{\frac{2i\pi}{3}+i\sigma\varphi} \\ -e^{-\frac{2i\pi}{3}+i\sigma\varphi} & 0 \end{pmatrix}$ |

| $b^\infty \times {}^1\bar{3}$ | $\{R_\varphi,\ E\}$ | $\left\{R_\varphi,\ IC_{3z}^{-1}\right\}$ | $\{\tau 2_\perp R_\varphi,\ E\}$ | $\left\{\tau 2_\perp R_\varphi,\ IC_{3z}^{-1}\right\}$ |
|---|---|---|---|---|
| $\Gamma_1$ | $e^{i\sigma\varphi}$ | $e^{i\sigma\varphi}$ | $e^{i\sigma\varphi}$ | $e^{i\sigma\varphi}$ |
| $\Gamma_2$ | $e^{i\sigma\varphi}$ | $-e^{i\sigma\varphi}$ | $e^{i\sigma\varphi}$ | $-e^{i\sigma\varphi}$ |
| $\Gamma_3$ | $\begin{pmatrix} e^{i\sigma\varphi} & 0 \\ 0 & e^{i\sigma\varphi} \end{pmatrix}$ | $\begin{pmatrix} -e^{-\frac{2i\pi}{3}+i\sigma\varphi} & 0 \\ 0 & -e^{\frac{2i\pi}{3}+i\sigma\varphi} \end{pmatrix}$ | $\begin{pmatrix} 0 & e^{i\sigma\varphi} \\ e^{i\sigma\varphi} & 0 \end{pmatrix}$ | $\begin{pmatrix} 0 & -e^{-\frac{2i\pi}{3}+i\sigma\varphi} \\ -e^{\frac{2i\pi}{3}+i\sigma\varphi} & 0 \end{pmatrix}$ |
| $\Gamma_4$ | $\begin{pmatrix} e^{i\sigma\varphi} & 0 \\ 0 & e^{i\sigma\varphi} \end{pmatrix}$ | $\begin{pmatrix} e^{-\frac{2i\pi}{3}+i\sigma\varphi} & 0 \\ 0 & e^{\frac{2i\pi}{3}+i\sigma\varphi} \end{pmatrix}$ | $\begin{pmatrix} 0 & e^{i\sigma\varphi} \\ e^{i\sigma\varphi} & 0 \end{pmatrix}$ | $\begin{pmatrix} 0 & e^{-\frac{2i\pi}{3}+i\sigma\varphi} \\ e^{\frac{2i\pi}{3}+i\sigma\varphi} & 0 \end{pmatrix}$ |

| $b^\infty \times {}^1 3\, {}^1 m$ | $\{R_\varphi, E\}$ | $\{R_\varphi, C_{3z}\}$ | $\{R_\varphi, IC_{2B}\}$ | $\{\tau 2_\perp R_\varphi, E\}$ | $\{\tau 2_\perp R_\varphi, C_{3z}\}$ | $\{\tau 2_\perp R_\varphi, IC_{2B}\}$ |
|---|---|---|---|---|---|---|
| $\Gamma_1$ | $e^{i\sigma\varphi}$ | $e^{i\sigma\varphi}$ | $e^{i\sigma\varphi}$ | $e^{i\sigma\varphi}$ | $e^{i\sigma\varphi}$ | $e^{i\sigma\varphi}$ |
| $\Gamma_2$ | $e^{i\sigma\varphi}$ | $e^{i\sigma\varphi}$ | $-e^{i\sigma\varphi}$ | $e^{i\sigma\varphi}$ | $e^{i\sigma\varphi}$ | $-e^{i\sigma\varphi}$ |
| $\Gamma_3$ | $\begin{pmatrix} e^{i\sigma\varphi} & 0 \\ 0 & e^{i\sigma\varphi} \end{pmatrix}$ | $\begin{pmatrix} e^{\frac{2i\pi}{3}+i\sigma\varphi} & 0 \\ 0 & e^{-\frac{2i\pi}{3}+i\sigma\varphi} \end{pmatrix}$ | $\begin{pmatrix} 0 & e^{\frac{2i\pi}{3}+i\sigma\varphi} \\ e^{-\frac{2i\pi}{3}+i\sigma\varphi} & 0 \end{pmatrix}$ | $\begin{pmatrix} 0 & e^{i\sigma\varphi} \\ e^{i\sigma\varphi} & 0 \end{pmatrix}$ | $\begin{pmatrix} 0 & e^{\frac{2i\pi}{3}+i\sigma\varphi} \\ e^{-\frac{2i\pi}{3}+i\sigma\varphi} & 0 \end{pmatrix}$ | $\begin{pmatrix} e^{\frac{2i\pi}{3}+i\sigma\varphi} & 0 \\ 0 & e^{-\frac{2i\pi}{3}+i\sigma\varphi} \end{pmatrix}$ |

| $b^\infty \times {}^1 3\, {}^1 2$ | $\{R_\varphi, E\}$ | $\{R_\varphi, C_{3z}\}$ | $\{R_\varphi, C_{2x}\}$ | $\{\tau 2_\perp R_\varphi, E\}$ | $\{\tau 2_\perp R_\varphi, C_{3z}\}$ | $\{\tau 2_\perp R_\varphi, C_{2x}\}$ |
|---|---|---|---|---|---|---|
| $\Gamma_1$ | $e^{i\sigma\varphi}$ | $e^{i\sigma\varphi}$ | $e^{i\sigma\varphi}$ | $e^{i\sigma\varphi}$ | $e^{i\sigma\varphi}$ | $e^{i\sigma\varphi}$ |
| $\Gamma_2$ | $e^{i\sigma\varphi}$ | $e^{i\sigma\varphi}$ | $-e^{i\sigma\varphi}$ | $e^{i\sigma\varphi}$ | $e^{i\sigma\varphi}$ | $-e^{i\sigma\varphi}$ |
| $\Gamma_3$ | $\begin{pmatrix} e^{i\sigma\varphi} & 0 \\ 0 & e^{i\sigma\varphi} \end{pmatrix}$ | $\begin{pmatrix} e^{\frac{2i\pi}{3}+i\sigma\varphi} & 0 \\ 0 & e^{-\frac{2i\pi}{3}+i\sigma\varphi} \end{pmatrix}$ | $\begin{pmatrix} 0 & e^{i\sigma\varphi} \\ e^{i\sigma\varphi} & 0 \end{pmatrix}$ | $\begin{pmatrix} 0 & e^{i\sigma\varphi} \\ e^{i\sigma\varphi} & 0 \end{pmatrix}$ | $\begin{pmatrix} 0 & e^{\frac{2i\pi}{3}+i\sigma\varphi} \\ e^{-\frac{2i\pi}{3}+i\sigma\varphi} & 0 \end{pmatrix}$ | $\begin{pmatrix} e^{i\sigma\varphi} & 0 \\ 0 & e^{i\sigma\varphi} \end{pmatrix}$ |

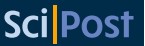

| $b^\infty \times {}^14/_{1_m}$ | $\{R_\varphi,\, E\}$ | $\{R_\varphi,\, C_{4z}\}$ | $\{R_\varphi,\, IC_{2z}\}$ | $\{\tau 2_\perp R_\varphi,\, E\}$ | $\{\tau 2_\perp R_\varphi,\, C_{4z}\}$ | $\{\tau 2_\perp R_\varphi,\, IC_{2z}\}$ |
|---|---|---|---|---|---|---|
| $\Gamma_1$ | $e^{i\sigma\varphi}$ | $e^{i\sigma\varphi}$ | $e^{i\sigma\varphi}$ | $e^{i\sigma\varphi}$ | $e^{i\sigma\varphi}$ | $e^{i\sigma\varphi}$ |
| $\Gamma_2$ | $e^{i\sigma\varphi}$ | $e^{i\sigma\varphi}$ | $-e^{i\sigma\varphi}$ | $e^{i\sigma\varphi}$ | $e^{i\sigma\varphi}$ | $-e^{i\sigma\varphi}$ |
| $\Gamma_3$ | $e^{i\sigma\varphi}$ | $-e^{i\sigma\varphi}$ | $e^{i\sigma\varphi}$ | $e^{i\sigma\varphi}$ | $-e^{i\sigma\varphi}$ | $e^{i\sigma\varphi}$ |
| $\Gamma_4$ | $e^{i\sigma\varphi}$ | $-e^{i\sigma\varphi}$ | $-e^{i\sigma\varphi}$ | $e^{i\sigma\varphi}$ | $-e^{i\sigma\varphi}$ | $-e^{i\sigma\varphi}$ |
| $\Gamma_5$ | $\begin{pmatrix} e^{i\sigma\varphi} & 0 \\ 0 & e^{i\sigma\varphi}\end{pmatrix}$ | $\begin{pmatrix} -i\,e^{i\sigma\varphi} & 0 \\ 0 & i\,e^{i\sigma\varphi}\end{pmatrix}$ | $\begin{pmatrix} e^{i\sigma\varphi} & 0 \\ 0 & e^{i\sigma\varphi}\end{pmatrix}$ | $\begin{pmatrix} 0 & e^{i\sigma\varphi} \\ e^{i\sigma\varphi} & 0\end{pmatrix}$ | $\begin{pmatrix} 0 & -i\,e^{i\sigma\varphi} \\ i\,e^{i\sigma\varphi} & 0\end{pmatrix}$ | $\begin{pmatrix} 0 & e^{i\sigma\varphi} \\ e^{i\sigma\varphi} & 0\end{pmatrix}$ |
| $\Gamma_6$ | $\begin{pmatrix} e^{i\sigma\varphi} & 0 \\ 0 & e^{i\sigma\varphi}\end{pmatrix}$ | $\begin{pmatrix} -i\,e^{i\sigma\varphi} & 0 \\ 0 & i\,e^{i\sigma\varphi}\end{pmatrix}$ | $\begin{pmatrix} -e^{i\sigma\varphi} & 0 \\ 0 & -e^{i\sigma\varphi}\end{pmatrix}$ | $\begin{pmatrix} 0 & e^{i\sigma\varphi} \\ e^{i\sigma\varphi} & 0\end{pmatrix}$ | $\begin{pmatrix} 0 & -i\,e^{i\sigma\varphi} \\ i\,e^{i\sigma\varphi} & 0\end{pmatrix}$ | $\begin{pmatrix} 0 & -e^{i\sigma\varphi} \\ -e^{i\sigma\varphi} & 0\end{pmatrix}$ |

| $b^\infty \times {}^14\,{}^1m\,{}^1m$ | $\{R_\varphi,\, E\}$ | $\{R_\varphi,\, C_{4z}\}$ | $\{R_\varphi,\, IC_{2y}\}$ | $\{\tau 2_\perp R_\varphi,\, E\}$ | $\{\tau 2_\perp R_\varphi,\, C_{4z}\}$ | $\{\tau 2_\perp R_\varphi,\, IC_{2y}\}$ |
|---|---|---|---|---|---|---|
| $\Gamma_1$ | $e^{i\sigma\varphi}$ | $e^{i\sigma\varphi}$ | $e^{i\sigma\varphi}$ | $e^{i\sigma\varphi}$ | $e^{i\sigma\varphi}$ | $e^{i\sigma\varphi}$ |
| $\Gamma_2$ | $e^{i\sigma\varphi}$ | $e^{i\sigma\varphi}$ | $-e^{i\sigma\varphi}$ | $e^{i\sigma\varphi}$ | $e^{i\sigma\varphi}$ | $-e^{i\sigma\varphi}$ |
| $\Gamma_3$ | $e^{i\sigma\varphi}$ | $-e^{i\sigma\varphi}$ | $e^{i\sigma\varphi}$ | $e^{i\sigma\varphi}$ | $-e^{i\sigma\varphi}$ | $e^{i\sigma\varphi}$ |
| $\Gamma_4$ | $e^{i\sigma\varphi}$ | $-e^{i\sigma\varphi}$ | $-e^{i\sigma\varphi}$ | $e^{i\sigma\varphi}$ | $-e^{i\sigma\varphi}$ | $-e^{i\sigma\varphi}$ |
| $\Gamma_5$ | $\begin{pmatrix} e^{i\sigma\varphi} & 0 \\ 0 & e^{i\sigma\varphi}\end{pmatrix}$ | $\begin{pmatrix} -i\,e^{i\sigma\varphi} & 0 \\ 0 & i\,e^{i\sigma\varphi}\end{pmatrix}$ | $\begin{pmatrix} 0 & -e^{i\sigma\varphi} \\ -e^{i\sigma\varphi} & 0\end{pmatrix}$ | $\begin{pmatrix} e^{i\sigma\varphi} & 0 \\ 0 & e^{i\sigma\varphi}\end{pmatrix}$ | $\begin{pmatrix} 0 & -i\,e^{i\sigma\varphi} \\ i\,e^{i\sigma\varphi} & 0\end{pmatrix}$ | $\begin{pmatrix} 0 & -e^{i\sigma\varphi} \\ -e^{i\sigma\varphi} & 0\end{pmatrix}$ |



| $b^\infty \times {}^14\,{}^12\,{}^12$ | $\{R_\varphi, E\}$ | $\{R_\varphi, C_{4z}\}$ | $\{R_\varphi, C_{2x}\}$ | $\{\tau2_\perp R_\varphi, E\}$ | $\{\tau2_\perp R_\varphi, C_{4z}\}$ | $\{\tau2_\perp R_\varphi, C_{2x}\}$ |
|---|---|---|---|---|---|---|
| $\Gamma_1$ | $e^{i\sigma\varphi}$ | $e^{i\sigma\varphi}$ | $e^{i\sigma\varphi}$ | $e^{i\sigma\varphi}$ | $e^{i\sigma\varphi}$ | $e^{i\sigma\varphi}$ |
| $\Gamma_2$ | $e^{i\sigma\varphi}$ | $e^{i\sigma\varphi}$ | $-e^{i\sigma\varphi}$ | $e^{i\sigma\varphi}$ | $e^{i\sigma\varphi}$ | $-e^{i\sigma\varphi}$ |
| $\Gamma_3$ | $e^{i\sigma\varphi}$ | $-e^{i\sigma\varphi}$ | $e^{i\sigma\varphi}$ | $e^{i\sigma\varphi}$ | $-e^{i\sigma\varphi}$ | $e^{i\sigma\varphi}$ |
| $\Gamma_4$ | $e^{i\sigma\varphi}$ | $-e^{i\sigma\varphi}$ | $-e^{i\sigma\varphi}$ | $e^{i\sigma\varphi}$ | $-e^{i\sigma\varphi}$ | $-e^{i\sigma\varphi}$ |
| $\Gamma_5$ | $\begin{pmatrix} e^{i\sigma\varphi} & 0 \\ 0 & e^{i\sigma\varphi}\end{pmatrix}$ | $\begin{pmatrix} -ie^{i\sigma\varphi} & 0 \\ 0 & ie^{i\sigma\varphi}\end{pmatrix}$ | $\begin{pmatrix} 0 & e^{i\sigma\varphi} \\ e^{i\sigma\varphi} & 0\end{pmatrix}$ | $\begin{pmatrix} 0 & e^{i\sigma\varphi} \\ e^{i\sigma\varphi} & 0\end{pmatrix}$ | $\begin{pmatrix} 0 & -ie^{i\sigma\varphi} \\ ie^{i\sigma\varphi} & 0\end{pmatrix}$ | $\begin{pmatrix} e^{i\sigma\varphi} & 0 \\ 0 & e^{i\sigma\varphi}\end{pmatrix}$ |

| $b^\infty \times {}^1m\,{}^1m\,{}^1m$ | $\{R_\varphi, E\}$ | $\{R_\varphi, IC_{2z}\}$ | $\{R_\varphi, IC_{2y}\}$ | $\{R_\varphi, IC_{2x}\}$ | $\{\tau2_\perp R_\varphi, E\}$ | $\{\tau2_\perp R_\varphi, IC_{2z}\}$ | $\{\tau2_\perp R_\varphi, IC_{2y}\}$ | $\{\tau2_\perp R_\varphi, IC_{2x}\}$ |
|---|---|---|---|---|---|---|---|---|
| $\Gamma_1$ | $e^{i\sigma\varphi}$ | $e^{i\sigma\varphi}$ | $e^{i\sigma\varphi}$ | $e^{i\sigma\varphi}$ | $e^{i\sigma\varphi}$ | $e^{i\sigma\varphi}$ | $e^{i\sigma\varphi}$ | $e^{i\sigma\varphi}$ |
| $\Gamma_2$ | $e^{i\sigma\varphi}$ | $-e^{i\sigma\varphi}$ | $-e^{i\sigma\varphi}$ | $-e^{i\sigma\varphi}$ | $e^{i\sigma\varphi}$ | $-e^{i\sigma\varphi}$ | $-e^{i\sigma\varphi}$ | $-e^{i\sigma\varphi}$ |
| $\Gamma_3$ | $e^{i\sigma\varphi}$ | $e^{i\sigma\varphi}$ | $-e^{i\sigma\varphi}$ | $-e^{i\sigma\varphi}$ | $e^{i\sigma\varphi}$ | $e^{i\sigma\varphi}$ | $-e^{i\sigma\varphi}$ | $-e^{i\sigma\varphi}$ |
| $\Gamma_4$ | $e^{i\sigma\varphi}$ | $-e^{i\sigma\varphi}$ | $e^{i\sigma\varphi}$ | $e^{i\sigma\varphi}$ | $e^{i\sigma\varphi}$ | $-e^{i\sigma\varphi}$ | $e^{i\sigma\varphi}$ | $e^{i\sigma\varphi}$ |
| $\Gamma_5$ | $e^{i\sigma\varphi}$ | $e^{i\sigma\varphi}$ | $e^{i\sigma\varphi}$ | $-e^{i\sigma\varphi}$ | $e^{i\sigma\varphi}$ | $e^{i\sigma\varphi}$ | $e^{i\sigma\varphi}$ | $-e^{i\sigma\varphi}$ |
| $\Gamma_6$ | $e^{i\sigma\varphi}$ | $-e^{i\sigma\varphi}$ | $-e^{i\sigma\varphi}$ | $e^{i\sigma\varphi}$ | $e^{i\sigma\varphi}$ | $-e^{i\sigma\varphi}$ | $-e^{i\sigma\varphi}$ | $e^{i\sigma\varphi}$ |
| $\Gamma_7$ | $e^{i\sigma\varphi}$ | $e^{i\sigma\varphi}$ | $-e^{i\sigma\varphi}$ | $e^{i\sigma\varphi}$ | $e^{i\sigma\varphi}$ | $e^{i\sigma\varphi}$ | $-e^{i\sigma\varphi}$ | $-e^{i\sigma\varphi}$ |
| $\Gamma_8$ | $e^{i\sigma\varphi}$ | $-e^{i\sigma\varphi}$ | $e^{i\sigma\varphi}$ | $-e^{i\sigma\varphi}$ | $e^{i\sigma\varphi}$ | $-e^{i\sigma\varphi}$ | $e^{i\sigma\varphi}$ | $-e^{i\sigma\varphi}$ |

| $b^\infty \times {}^1\bar{4}\,{}^1 2\,{}^1 m$ | $\{R_\varphi, E\}$ | $\{R_\varphi, C_{2x}\}$ | $\{R_\varphi, IC_{4z}^{-1}\}$ | $\{\tau 2_\perp R_\varphi, E\}$ | $\{\tau 2_\perp R_\varphi, C_{2x}\}$ | $\{\tau 2_\perp R_\varphi, IC_{4z}^{-1}\}$ |
|---|---|---|---|---|---|---|
| $\Gamma_1$ | $e^{i\sigma\varphi}$ | $e^{i\sigma\varphi}$ | $e^{i\sigma\varphi}$ | $e^{i\sigma\varphi}$ | $e^{i\sigma\varphi}$ | $e^{i\sigma\varphi}$ |
| $\Gamma_2$ | $e^{i\sigma\varphi}$ | $e^{i\sigma\varphi}$ | $-e^{i\sigma\varphi}$ | $e^{i\sigma\varphi}$ | $e^{i\sigma\varphi}$ | $-e^{i\sigma\varphi}$ |
| $\Gamma_3$ | $e^{i\sigma\varphi}$ | $-e^{i\sigma\varphi}$ | $-e^{i\sigma\varphi}$ | $e^{i\sigma\varphi}$ | $-e^{i\sigma\varphi}$ | $-e^{i\sigma\varphi}$ |
| $\Gamma_4$ | $e^{i\sigma\varphi}$ | $-e^{i\sigma\varphi}$ | $e^{i\sigma\varphi}$ | $e^{i\sigma\varphi}$ | $-e^{i\sigma\varphi}$ | $e^{i\sigma\varphi}$ |
| $\Gamma_5$ | $\begin{pmatrix} e^{i\sigma\varphi} & 0 \\ 0 & e^{i\sigma\varphi} \end{pmatrix}$ | $\begin{pmatrix} e^{i\sigma\varphi} & 0 \\ 0 & -e^{i\sigma\varphi} \end{pmatrix}$ | $\begin{pmatrix} 0 & -e^{i\sigma\varphi} \\ e^{i\sigma\varphi} & 0 \end{pmatrix}$ | $\begin{pmatrix} e^{i\sigma\varphi} & 0 \\ 0 & e^{i\sigma\varphi} \end{pmatrix}$ | $\begin{pmatrix} e^{i\sigma\varphi} & 0 \\ 0 & -e^{i\sigma\varphi} \end{pmatrix}$ | $\begin{pmatrix} 0 & -e^{i\sigma\varphi} \\ e^{i\sigma\varphi} & 0 \end{pmatrix}$ |

| $b^\infty \times {}^1 6/{}^1 m$ | $\{R_\varphi, E\}$ | $\{R_\varphi, C_{6z}\}$ | $\{R_\varphi, IC_{2z}\}$ | $\{\tau 2_\perp R_\varphi, E\}$ | $\{\tau 2_\perp R_\varphi, C_{6z}\}$ | $\{\tau 2_\perp R_\varphi, IC_{2z}\}$ |
|---|---|---|---|---|---|---|
| $\Gamma_1$ | $e^{i\sigma\varphi}$ | $e^{i\sigma\varphi}$ | $e^{i\sigma\varphi}$ | $e^{i\sigma\varphi}$ | $e^{i\sigma\varphi}$ | $e^{i\sigma\varphi}$ |
| $\Gamma_2$ | $e^{i\sigma\varphi}$ | $e^{i\sigma\varphi}$ | $-e^{i\sigma\varphi}$ | $e^{i\sigma\varphi}$ | $e^{i\sigma\varphi}$ | $-e^{i\sigma\varphi}$ |
| $\Gamma_3$ | $e^{i\sigma\varphi}$ | $-e^{i\sigma\varphi}$ | $e^{i\sigma\varphi}$ | $e^{i\sigma\varphi}$ | $-e^{i\sigma\varphi}$ | $e^{i\sigma\varphi}$ |
| $\Gamma_4$ | $e^{i\sigma\varphi}$ | $-e^{i\sigma\varphi}$ | $-e^{i\sigma\varphi}$ | $e^{i\sigma\varphi}$ | $-e^{i\sigma\varphi}$ | $-e^{i\sigma\varphi}$ |
| $\Gamma_5$ | $\begin{pmatrix} e^{-\frac{2i\pi}{3}+i\sigma\varphi} & 0 \\ 0 & e^{\frac{2i\pi}{3}+i\sigma\varphi} \end{pmatrix}$ | $\begin{pmatrix} e^{-\frac{2i\pi}{3}+i\sigma\varphi} & 0 \\ 0 & e^{\frac{2i\pi}{3}+i\sigma\varphi} \end{pmatrix}$ | $\begin{pmatrix} e^{i\sigma\varphi} & 0 \\ 0 & e^{i\sigma\varphi} \end{pmatrix}$ | $\begin{pmatrix} 0 & e^{i\sigma\varphi} \\ e^{i\sigma\varphi} & 0 \end{pmatrix}$ | $\begin{pmatrix} 0 & e^{-\frac{2i\pi}{3}+i\sigma\varphi} \\ e^{\frac{2i\pi}{3}+i\sigma\varphi} & 0 \end{pmatrix}$ | $\begin{pmatrix} 0 & e^{i\sigma\varphi} \\ e^{i\sigma\varphi} & 0 \end{pmatrix}$ |
| $\Gamma_6$ | $\begin{pmatrix} e^{-\frac{2i\pi}{3}+i\sigma\varphi} & 0 \\ 0 & e^{\frac{2i\pi}{3}+i\sigma\varphi} \end{pmatrix}$ | $\begin{pmatrix} e^{-\frac{2i\pi}{3}+i\sigma\varphi} & 0 \\ 0 & e^{\frac{2i\pi}{3}+i\sigma\varphi} \end{pmatrix}$ | $\begin{pmatrix} -e^{i\sigma\varphi} & 0 \\ 0 & -e^{i\sigma\varphi} \end{pmatrix}$ | $\begin{pmatrix} 0 & e^{i\sigma\varphi} \\ e^{i\sigma\varphi} & 0 \end{pmatrix}$ | $\begin{pmatrix} 0 & -e^{-\frac{2i\pi}{3}+i\sigma\varphi} \\ -e^{\frac{2i\pi}{3}+i\sigma\varphi} & 0 \end{pmatrix}$ | $\begin{pmatrix} 0 & -e^{i\sigma\varphi} \\ -e^{i\sigma\varphi} & 0 \end{pmatrix}$ |
| $\Gamma_7$ | $\begin{pmatrix} e^{-\frac{2i\pi}{3}+i\sigma\varphi} & 0 \\ 0 & e^{\frac{2i\pi}{3}+i\sigma\varphi} \end{pmatrix}$ | $\begin{pmatrix} -e^{-\frac{2i\pi}{3}+i\sigma\varphi} & 0 \\ 0 & -e^{\frac{2i\pi}{3}+i\sigma\varphi} \end{pmatrix}$ | $\begin{pmatrix} e^{i\sigma\varphi} & 0 \\ 0 & e^{i\sigma\varphi} \end{pmatrix}$ | $\begin{pmatrix} 0 & e^{i\sigma\varphi} \\ e^{i\sigma\varphi} & 0 \end{pmatrix}$ | $\begin{pmatrix} 0 & -e^{-\frac{2i\pi}{3}+i\sigma\varphi} \\ -e^{\frac{2i\pi}{3}+i\sigma\varphi} & 0 \end{pmatrix}$ | $\begin{pmatrix} 0 & e^{i\sigma\varphi} \\ e^{i\sigma\varphi} & 0 \end{pmatrix}$ |
| $\Gamma_8$ | $\begin{pmatrix} e^{-\frac{2i\pi}{3}+i\sigma\varphi} & 0 \\ 0 & e^{\frac{2i\pi}{3}+i\sigma\varphi} \end{pmatrix}$ | $\begin{pmatrix} -e^{-\frac{2i\pi}{3}+i\sigma\varphi} & 0 \\ 0 & -e^{\frac{2i\pi}{3}+i\sigma\varphi} \end{pmatrix}$ | $\begin{pmatrix} -e^{i\sigma\varphi} & 0 \\ 0 & -e^{i\sigma\varphi} \end{pmatrix}$ | $\begin{pmatrix} 0 & e^{i\sigma\varphi} \\ e^{i\sigma\varphi} & 0 \end{pmatrix}$ | $\begin{pmatrix} 0 & -e^{-\frac{2i\pi}{3}+i\sigma\varphi} \\ -e^{\frac{2i\pi}{3}+i\sigma\varphi} & 0 \end{pmatrix}$ | $\begin{pmatrix} 0 & -e^{i\sigma\varphi} \\ -e^{i\sigma\varphi} & 0 \end{pmatrix}$ |

| $b^\infty \times {}^16\,{}^1m\,{}^1m$ | $\{R_\varphi,\,E\}$ | $\{R_\varphi,\,C_{6z}\}$ | $\{R_\varphi,\,IC_{2y}\}$ | $\{\tau2_\perp R_\varphi,\,E\}$ | $\{\tau2_\perp R_\varphi,\,C_{6z}\}$ | $\{\tau2_\perp R_\varphi,\,IC_{2y}\}$ |
|---|---|---|---|---|---|---|
| $\Gamma_1$ | $e^{i\sigma\varphi}$ | $e^{i\sigma\varphi}$ | $e^{i\sigma\varphi}$ | $e^{i\sigma\varphi}$ | $e^{i\sigma\varphi}$ | $e^{i\sigma\varphi}$ |
| $\Gamma_2$ | $e^{i\sigma\varphi}$ | $e^{i\sigma\varphi}$ | $-e^{i\sigma\varphi}$ | $e^{i\sigma\varphi}$ | $e^{i\sigma\varphi}$ | $-e^{i\sigma\varphi}$ |
| $\Gamma_3$ | $e^{i\sigma\varphi}$ | $-e^{i\sigma\varphi}$ | $-e^{i\sigma\varphi}$ | $e^{i\sigma\varphi}$ | $-e^{i\sigma\varphi}$ | $-e^{i\sigma\varphi}$ |
| $\Gamma_4$ | $e^{i\sigma\varphi}$ | $-e^{i\sigma\varphi}$ | $e^{i\sigma\varphi}$ | $e^{i\sigma\varphi}$ | $-e^{i\sigma\varphi}$ | $e^{i\sigma\varphi}$ |
| $\Gamma_5$ | $\begin{pmatrix} e^{i\sigma\varphi} & 0 \\ 0 & e^{i\sigma\varphi} \end{pmatrix}$ | $\begin{pmatrix} e^{-\frac{2i\pi}{3}+i\sigma\varphi} & 0 \\ 0 & e^{\frac{2i\pi}{3}+i\sigma\varphi} \end{pmatrix}$ | $\begin{pmatrix} 0 & e^{i\sigma\varphi} \\ e^{i\sigma\varphi} & 0 \end{pmatrix}$ | $\begin{pmatrix} 0 & e^{i\sigma\varphi} \\ e^{i\sigma\varphi} & 0 \end{pmatrix}$ | $\begin{pmatrix} 0 & e^{-\frac{2i\pi}{3}+i\sigma\varphi} \\ e^{\frac{2i\pi}{3}+i\sigma\varphi} & 0 \end{pmatrix}$ | $\begin{pmatrix} e^{i\sigma\varphi} & 0 \\ 0 & e^{i\sigma\varphi} \end{pmatrix}$ |
| $\Gamma_6$ | $\begin{pmatrix} e^{i\sigma\varphi} & 0 \\ 0 & e^{i\sigma\varphi} \end{pmatrix}$ | $\begin{pmatrix} -e^{-\frac{2i\pi}{3}+i\sigma\varphi} & 0 \\ 0 & -e^{\frac{2i\pi}{3}+i\sigma\varphi} \end{pmatrix}$ | $\begin{pmatrix} 0 & -e^{i\sigma\varphi} \\ -e^{i\sigma\varphi} & 0 \end{pmatrix}$ | $\begin{pmatrix} 0 & e^{i\sigma\varphi} \\ e^{i\sigma\varphi} & 0 \end{pmatrix}$ | $\begin{pmatrix} 0 & -e^{-\frac{2i\pi}{3}+i\sigma\varphi} \\ -e^{\frac{2i\pi}{3}+i\sigma\varphi} & 0 \end{pmatrix}$ | $\begin{pmatrix} -e^{i\sigma\varphi} & 0 \\ 0 & -e^{i\sigma\varphi} \end{pmatrix}$ |

| $b^\infty \times {}^16\,{}^12\,{}^12$ | $\{R_\varphi,\,E\}$ | $\{R_\varphi,\,C_{6z}\}$ | $\{R_\varphi,\,C_{2x}\}$ | $\{\tau2_\perp R_\varphi,\,E\}$ | $\{\tau2_\perp R_\varphi,\,C_{6z}\}$ | $\{\tau2_\perp R_\varphi,\,C_{2x}\}$ |
|---|---|---|---|---|---|---|
| $\Gamma_1$ | $e^{i\sigma\varphi}$ | $e^{i\sigma\varphi}$ | $e^{i\sigma\varphi}$ | $e^{i\sigma\varphi}$ | $e^{i\sigma\varphi}$ | $e^{i\sigma\varphi}$ |
| $\Gamma_2$ | $e^{i\sigma\varphi}$ | $e^{i\sigma\varphi}$ | $-e^{i\sigma\varphi}$ | $e^{i\sigma\varphi}$ | $e^{i\sigma\varphi}$ | $-e^{i\sigma\varphi}$ |
| $\Gamma_3$ | $e^{i\sigma\varphi}$ | $-e^{i\sigma\varphi}$ | $e^{i\sigma\varphi}$ | $e^{i\sigma\varphi}$ | $-e^{i\sigma\varphi}$ | $e^{i\sigma\varphi}$ |
| $\Gamma_4$ | $e^{i\sigma\varphi}$ | $-e^{i\sigma\varphi}$ | $-e^{i\sigma\varphi}$ | $e^{i\sigma\varphi}$ | $-e^{i\sigma\varphi}$ | $-e^{i\sigma\varphi}$ |
| $\Gamma_5$ | $\begin{pmatrix} e^{i\sigma\varphi} & 0 \\ 0 & e^{i\sigma\varphi} \end{pmatrix}$ | $\begin{pmatrix} e^{-\frac{2i\pi}{3}+i\sigma\varphi} & 0 \\ 0 & e^{\frac{2i\pi}{3}+i\sigma\varphi} \end{pmatrix}$ | $\begin{pmatrix} 0 & e^{i\sigma\varphi} \\ e^{i\sigma\varphi} & 0 \end{pmatrix}$ | $\begin{pmatrix} 0 & e^{i\sigma\varphi} \\ e^{i\sigma\varphi} & 0 \end{pmatrix}$ | $\begin{pmatrix} 0 & e^{-\frac{2i\pi}{3}+i\sigma\varphi} \\ e^{\frac{2i\pi}{3}+i\sigma\varphi} & 0 \end{pmatrix}$ | $\begin{pmatrix} e^{i\sigma\varphi} & 0 \\ 0 & e^{i\sigma\varphi} \end{pmatrix}$ |
| $\Gamma_6$ | $\begin{pmatrix} e^{i\sigma\varphi} & 0 \\ 0 & e^{i\sigma\varphi} \end{pmatrix}$ | $\begin{pmatrix} -e^{-\frac{2i\pi}{3}+i\sigma\varphi} & 0 \\ 0 & -e^{\frac{2i\pi}{3}+i\sigma\varphi} \end{pmatrix}$ | $\begin{pmatrix} 0 & e^{i\sigma\varphi} \\ e^{i\sigma\varphi} & 0 \end{pmatrix}$ | $\begin{pmatrix} 0 & e^{i\sigma\varphi} \\ e^{i\sigma\varphi} & 0 \end{pmatrix}$ | $\begin{pmatrix} 0 & -e^{-\frac{2i\pi}{3}+i\sigma\varphi} \\ -e^{\frac{2i\pi}{3}+i\sigma\varphi} & 0 \end{pmatrix}$ | $\begin{pmatrix} e^{i\sigma\varphi} & 0 \\ 0 & e^{i\sigma\varphi} \end{pmatrix}$ |

**Table:** $b^\infty \times {}^1\bar{6}\,{}^1m\,{}^12$

| | $\{R_\varphi, E\}$ | $\{R_\varphi, IC_{6z}^{-1}\}$ | $\{R_\varphi, IC_{2y}\}$ | $\{\tau 2_\perp R_\varphi, E\}$ | $\{\tau 2_\perp R_\varphi, IC_{6z}^{-1}\}$ | $\{\tau 2_\perp R_\varphi, IC_{2y}\}$ |
|---|---|---|---|---|---|---|
| $\Gamma_1$ | $e^{i\sigma\varphi}$ | $e^{i\sigma\varphi}$ | $e^{i\sigma\varphi}$ | $e^{i\sigma\varphi}$ | $e^{i\sigma\varphi}$ | $e^{i\sigma\varphi}$ |
| $\Gamma_2$ | $e^{i\sigma\varphi}$ | $-e^{i\sigma\varphi}$ | $-e^{i\sigma\varphi}$ | $e^{i\sigma\varphi}$ | $-e^{i\sigma\varphi}$ | $-e^{i\sigma\varphi}$ |
| $\Gamma_3$ | $e^{i\sigma\varphi}$ | $e^{i\sigma\varphi}$ | $-e^{i\sigma\varphi}$ | $e^{i\sigma\varphi}$ | $e^{i\sigma\varphi}$ | $-e^{i\sigma\varphi}$ |
| $\Gamma_4$ | $e^{i\sigma\varphi}$ | $-e^{i\sigma\varphi}$ | $e^{i\sigma\varphi}$ | $e^{i\sigma\varphi}$ | $-e^{i\sigma\varphi}$ | $e^{i\sigma\varphi}$ |
| $\Gamma_5$ | $\begin{pmatrix} e^{i\sigma\varphi} & 0 \\ 0 & e^{i\sigma\varphi} \end{pmatrix}$ | $\begin{pmatrix} e^{\frac{2i\pi}{3}+i\sigma\varphi} & 0 \\ 0 & e^{-\frac{2i\pi}{3}+i\sigma\varphi} \end{pmatrix}$ | $\begin{pmatrix} 0 & e^{i\sigma\varphi} \\ e^{i\sigma\varphi} & 0 \end{pmatrix}$ | $\begin{pmatrix} 0 & e^{i\sigma\varphi} \\ e^{i\sigma\varphi} & 0 \end{pmatrix}$ | $\begin{pmatrix} 0 & e^{\frac{2i\pi}{3}+i\sigma\varphi} \\ e^{-\frac{2i\pi}{3}+i\sigma\varphi} & 0 \end{pmatrix}$ | $\begin{pmatrix} e^{i\sigma\varphi} & 0 \\ 0 & e^{i\sigma\varphi} \end{pmatrix}$ |
| $\Gamma_6$ | $\begin{pmatrix} e^{i\sigma\varphi} & 0 \\ 0 & e^{i\sigma\varphi} \end{pmatrix}$ | $\begin{pmatrix} -e^{\frac{2i\pi}{3}+i\sigma\varphi} & 0 \\ 0 & -e^{-\frac{2i\pi}{3}+i\sigma\varphi} \end{pmatrix}$ | $\begin{pmatrix} 0 & -e^{i\sigma\varphi} \\ -e^{i\sigma\varphi} & 0 \end{pmatrix}$ | $\begin{pmatrix} 0 & e^{i\sigma\varphi} \\ e^{i\sigma\varphi} & 0 \end{pmatrix}$ | $\begin{pmatrix} 0 & -e^{\frac{2i\pi}{3}+i\sigma\varphi} \\ -e^{-\frac{2i\pi}{3}+i\sigma\varphi} & 0 \end{pmatrix}$ | $\begin{pmatrix} -e^{i\sigma\varphi} & 0 \\ 0 & -e^{i\sigma\varphi} \end{pmatrix}$ |

**Table:** $b^\infty \times {}^1\bar{3}\,{}^1m$

| | $\{R_\varphi, E\}$ | $\{R_\varphi, C_{2x}\}$ | $\{R_\varphi, IC_{3z}^{-1}\}$ | $\{\tau 2_\perp R_\varphi, E\}$ | $\{\tau 2_\perp R_\varphi, C_{2x}\}$ | $\{\tau 2_\perp R_\varphi, IC_{3z}^{-1}\}$ |
|---|---|---|---|---|---|---|
| $\Gamma_1$ | $e^{i\sigma\varphi}$ | $e^{i\sigma\varphi}$ | $e^{i\sigma\varphi}$ | $e^{i\sigma\varphi}$ | $e^{i\sigma\varphi}$ | $e^{i\sigma\varphi}$ |
| $\Gamma_2$ | $e^{i\sigma\varphi}$ | $e^{i\sigma\varphi}$ | $-e^{i\sigma\varphi}$ | $e^{i\sigma\varphi}$ | $e^{i\sigma\varphi}$ | $-e^{i\sigma\varphi}$ |
| $\Gamma_3$ | $e^{i\sigma\varphi}$ | $-e^{i\sigma\varphi}$ | $e^{i\sigma\varphi}$ | $e^{i\sigma\varphi}$ | $-e^{i\sigma\varphi}$ | $e^{i\sigma\varphi}$ |
| $\Gamma_4$ | $e^{i\sigma\varphi}$ | $-e^{i\sigma\varphi}$ | $-e^{i\sigma\varphi}$ | $e^{i\sigma\varphi}$ | $-e^{i\sigma\varphi}$ | $-e^{i\sigma\varphi}$ |
| $\Gamma_5$ | $\begin{pmatrix} e^{i\sigma\varphi} & 0 \\ 0 & e^{i\sigma\varphi} \end{pmatrix}$ | $\begin{pmatrix} 0 & e^{i\sigma\varphi} \\ e^{i\sigma\varphi} & 0 \end{pmatrix}$ | $\begin{pmatrix} e^{-\frac{2i\pi}{3}+i\sigma\varphi} & 0 \\ 0 & e^{\frac{2i\pi}{3}+i\sigma\varphi} \end{pmatrix}$ | $\begin{pmatrix} e^{i\sigma\varphi} & 0 \\ 0 & e^{i\sigma\varphi} \end{pmatrix}$ | $\begin{pmatrix} 0 & e^{i\sigma\varphi} \\ e^{i\sigma\varphi} & 0 \end{pmatrix}$ | $\begin{pmatrix} e^{-\frac{2i\pi}{3}+i\sigma\varphi} & 0 \\ 0 & e^{\frac{2i\pi}{3}+i\sigma\varphi} \end{pmatrix}$ |
| $\Gamma_6$ | $\begin{pmatrix} e^{i\sigma\varphi} & 0 \\ 0 & e^{i\sigma\varphi} \end{pmatrix}$ | $\begin{pmatrix} 0 & e^{i\sigma\varphi} \\ e^{i\sigma\varphi} & 0 \end{pmatrix}$ | $\begin{pmatrix} -e^{-\frac{2i\pi}{3}+i\sigma\varphi} & 0 \\ 0 & -e^{\frac{2i\pi}{3}+i\sigma\varphi} \end{pmatrix}$ | $\begin{pmatrix} e^{i\sigma\varphi} & 0 \\ 0 & e^{i\sigma\varphi} \end{pmatrix}$ | $\begin{pmatrix} 0 & e^{i\sigma\varphi} \\ e^{i\sigma\varphi} & 0 \end{pmatrix}$ | $\begin{pmatrix} -e^{-\frac{2i\pi}{3}+i\sigma\varphi} & 0 \\ 0 & -e^{\frac{2i\pi}{3}+i\sigma\varphi} \end{pmatrix}$ |

**Table 1**

| $b^\infty \times {}^{1}2\,{}^{1}3$ | $\{R_\varphi, E\}$ | $\{R_\varphi, C_{2x}\}$ | $\{R_\varphi, C_{3\delta}\}$ | $\{\tau 2_\perp R_\varphi, E\}$ | $\{\tau 2_\perp R_\varphi, C_{2x}\}$ | $\{\tau 2_\perp R_\varphi, C_{3\delta}\}$ |
|---|---|---|---|---|---|---|
| $\Gamma_1$ | $e^{i\sigma\varphi}$ | $e^{i\sigma\varphi}$ | $e^{i\sigma\varphi}$ | $e^{i\sigma\varphi}$ | $e^{i\sigma\varphi}$ | $e^{i\sigma\varphi}$ |
| $\Gamma_2$ | $\begin{pmatrix} e^{i\sigma\varphi} & 0 & 0 \\ 0 & e^{i\sigma\varphi} & 0 \\ 0 & 0 & e^{i\sigma\varphi} \end{pmatrix}$ | $\begin{pmatrix} -e^{i\sigma\varphi} & 0 & 0 \\ 0 & -e^{i\sigma\varphi} & 0 \\ 0 & 0 & e^{i\sigma\varphi} \end{pmatrix}$ | $\begin{pmatrix} 0 & e^{i\sigma\varphi} & 0 \\ 0 & 0 & e^{i\sigma\varphi} \\ e^{i\sigma\varphi} & 0 & 0 \end{pmatrix}$ | $\begin{pmatrix} e^{i\sigma\varphi} & 0 & 0 \\ 0 & e^{i\sigma\varphi} & 0 \\ 0 & 0 & e^{i\sigma\varphi} \end{pmatrix}$ | $\begin{pmatrix} -e^{i\sigma\varphi} & 0 & 0 \\ 0 & -e^{i\sigma\varphi} & 0 \\ 0 & 0 & e^{i\sigma\varphi} \end{pmatrix}$ | $\begin{pmatrix} 0 & e^{i\sigma\varphi} & 0 \\ 0 & 0 & e^{i\sigma\varphi} \\ e^{i\sigma\varphi} & 0 & 0 \end{pmatrix}$ |
| $\Gamma_3$ | $\begin{pmatrix} e^{i\sigma\varphi} & 0 \\ 0 & e^{i\sigma\varphi} \end{pmatrix}$ | $\begin{pmatrix} e^{i\sigma\varphi} & 0 \\ 0 & e^{i\sigma\varphi} \end{pmatrix}$ | $\begin{pmatrix} e^{-\frac{2i\pi}{3}+i\sigma\varphi} & 0 \\ 0 & e^{\frac{2i\pi}{3}+i\sigma\varphi} \end{pmatrix}$ | $\begin{pmatrix} 0 & e^{i\sigma\varphi} \\ e^{i\sigma\varphi} & 0 \end{pmatrix}$ | $\begin{pmatrix} 0 & e^{i\sigma\varphi} \\ e^{i\sigma\varphi} & 0 \end{pmatrix}$ | $\begin{pmatrix} 0 & e^{-\frac{2i\pi}{3}+i\sigma\varphi} \\ e^{\frac{2i\pi}{3}+i\sigma\varphi} & 0 \end{pmatrix}$ |

**Table 2**

| $b^\infty \times {}^{1}4/{}_{1m}\,{}^{1m}\,{}^{1m}$ | $\{R_\varphi, E\}$ | $\{R_\varphi, C_{4z}\}$ | $\{R_\varphi, IC_{2z}\}$ | $\{R_\varphi, IC_{2y}\}$ | $\{\tau 2_\perp R_\varphi, E\}$ | $\{\tau 2_\perp R_\varphi, C_{4z}\}$ | $\{\tau 2_\perp R_\varphi, IC_{2z}\}$ | $\{\tau 2_\perp R_\varphi, IC_{2y}\}$ |
|---|---|---|---|---|---|---|---|---|
| $\Gamma_1$ | $e^{i\sigma\varphi}$ | $e^{i\sigma\varphi}$ | $e^{i\sigma\varphi}$ | $e^{i\sigma\varphi}$ | $e^{i\sigma\varphi}$ | $e^{i\sigma\varphi}$ | $e^{i\sigma\varphi}$ | $e^{i\sigma\varphi}$ |
| $\Gamma_2$ | $e^{i\sigma\varphi}$ | $e^{i\sigma\varphi}$ | $-e^{i\sigma\varphi}$ | $-e^{i\sigma\varphi}$ | $e^{i\sigma\varphi}$ | $e^{i\sigma\varphi}$ | $-e^{i\sigma\varphi}$ | $-e^{i\sigma\varphi}$ |
| $\Gamma_3$ | $e^{i\sigma\varphi}$ | $e^{i\sigma\varphi}$ | $e^{i\sigma\varphi}$ | $-e^{i\sigma\varphi}$ | $e^{i\sigma\varphi}$ | $e^{i\sigma\varphi}$ | $e^{i\sigma\varphi}$ | $-e^{i\sigma\varphi}$ |
| $\Gamma_4$ | $e^{i\sigma\varphi}$ | $e^{i\sigma\varphi}$ | $-e^{i\sigma\varphi}$ | $e^{i\sigma\varphi}$ | $e^{i\sigma\varphi}$ | $e^{i\sigma\varphi}$ | $-e^{i\sigma\varphi}$ | $e^{i\sigma\varphi}$ |
| $\Gamma_5$ | $e^{i\sigma\varphi}$ | $-e^{i\sigma\varphi}$ | $e^{i\sigma\varphi}$ | $e^{i\sigma\varphi}$ | $e^{i\sigma\varphi}$ | $-e^{i\sigma\varphi}$ | $e^{i\sigma\varphi}$ | $e^{i\sigma\varphi}$ |
| $\Gamma_6$ | $e^{i\sigma\varphi}$ | $-e^{i\sigma\varphi}$ | $-e^{i\sigma\varphi}$ | $-e^{i\sigma\varphi}$ | $e^{i\sigma\varphi}$ | $-e^{i\sigma\varphi}$ | $-e^{i\sigma\varphi}$ | $-e^{i\sigma\varphi}$ |
| $\Gamma_7$ | $e^{i\sigma\varphi}$ | $-e^{i\sigma\varphi}$ | $e^{i\sigma\varphi}$ | $-e^{i\sigma\varphi}$ | $e^{i\sigma\varphi}$ | $-e^{i\sigma\varphi}$ | $e^{i\sigma\varphi}$ | $-e^{i\sigma\varphi}$ |
| $\Gamma_8$ | $e^{i\sigma\varphi}$ | $-e^{i\sigma\varphi}$ | $-e^{i\sigma\varphi}$ | $e^{i\sigma\varphi}$ | $e^{i\sigma\varphi}$ | $-e^{i\sigma\varphi}$ | $-e^{i\sigma\varphi}$ | $e^{i\sigma\varphi}$ |
| $\Gamma_9$ | $\begin{pmatrix} e^{i\sigma\varphi} & 0 \\ 0 & e^{i\sigma\varphi} \end{pmatrix}$ | $\begin{pmatrix} -ie^{i\sigma\varphi} & 0 \\ 0 & ie^{i\sigma\varphi} \end{pmatrix}$ | $\begin{pmatrix} e^{i\sigma\varphi} & 0 \\ 0 & e^{i\sigma\varphi} \end{pmatrix}$ | $\begin{pmatrix} 0 & e^{i\sigma\varphi} \\ e^{i\sigma\varphi} & 0 \end{pmatrix}$ | $\begin{pmatrix} e^{i\sigma\varphi} & 0 \\ 0 & e^{i\sigma\varphi} \end{pmatrix}$ | $\begin{pmatrix} -ie^{i\sigma\varphi} & 0 \\ 0 & ie^{i\sigma\varphi} \end{pmatrix}$ | $\begin{pmatrix} e^{i\sigma\varphi} & 0 \\ 0 & e^{i\sigma\varphi} \end{pmatrix}$ | $\begin{pmatrix} 0 & e^{i\sigma\varphi} \\ e^{i\sigma\varphi} & 0 \end{pmatrix}$ |
| $\Gamma_{10}$ | $\begin{pmatrix} e^{i\sigma\varphi} & 0 \\ 0 & e^{i\sigma\varphi} \end{pmatrix}$ | $\begin{pmatrix} -ie^{i\sigma\varphi} & 0 \\ 0 & ie^{i\sigma\varphi} \end{pmatrix}$ | $\begin{pmatrix} -e^{i\sigma\varphi} & 0 \\ 0 & -e^{i\sigma\varphi} \end{pmatrix}$ | $\begin{pmatrix} 0 & -e^{i\sigma\varphi} \\ -e^{i\sigma\varphi} & 0 \end{pmatrix}$ | $\begin{pmatrix} e^{i\sigma\varphi} & 0 \\ 0 & e^{i\sigma\varphi} \end{pmatrix}$ | $\begin{pmatrix} -ie^{i\sigma\varphi} & 0 \\ 0 & ie^{i\sigma\varphi} \end{pmatrix}$ | $\begin{pmatrix} -e^{i\sigma\varphi} & 0 \\ 0 & -e^{i\sigma\varphi} \end{pmatrix}$ | $\begin{pmatrix} 0 & -e^{i\sigma\varphi} \\ -e^{i\sigma\varphi} & 0 \end{pmatrix}$ |

| $b^\infty \times {}^1\bar{4}\,{}^13\,{}^1m$ | $\{R_\varphi, E\}$ | $\{R_\varphi, IC_{4z}^{-1}\}$ | $\{R_\varphi, IC_{2f}\}$ | $\{\tau 2_\perp R_\varphi, E\}$ | $\{\tau 2_\perp R_\varphi, IC_{4z}^{-1}\}$ | $\{\tau 2_\perp R_\varphi, IC_{2f}\}$ |
|---|---|---|---|---|---|---|
| $\Gamma_1$ | $e^{i\sigma\varphi}$ | $e^{i\sigma\varphi}$ | $e^{i\sigma\varphi}$ | $e^{i\sigma\varphi}$ | $e^{i\sigma\varphi}$ | $e^{i\sigma\varphi}$ |
| $\Gamma_2$ | $e^{i\sigma\varphi}$ | $-e^{i\sigma\varphi}$ | $-e^{i\sigma\varphi}$ | $e^{i\sigma\varphi}$ | $-e^{i\sigma\varphi}$ | $-e^{i\sigma\varphi}$ |
| $\Gamma_3$ | $\begin{pmatrix} e^{i\sigma\varphi} & 0 \\ 0 & e^{i\sigma\varphi} \end{pmatrix}$ | $\begin{pmatrix} e^{i\sigma\varphi} & 0 \\ 0 & -e^{i\sigma\varphi} \end{pmatrix}$ | $\begin{pmatrix} 0 & -e^{i\sigma\varphi} \\ -e^{i\sigma\varphi} & 0 \end{pmatrix}$ | $\begin{pmatrix} e^{i\sigma\varphi} & 0 \\ 0 & e^{i\sigma\varphi} \end{pmatrix}$ | $\begin{pmatrix} e^{i\sigma\varphi} & 0 \\ 0 & -e^{i\sigma\varphi} \end{pmatrix}$ | $\begin{pmatrix} 0 & -e^{i\sigma\varphi} \\ -e^{i\sigma\varphi} & 0 \end{pmatrix}$ |
| $\Gamma_4$ | $\begin{pmatrix} e^{i\sigma\varphi} & 0 \\ 0 & e^{i\sigma\varphi} \end{pmatrix}$ | $\begin{pmatrix} -e^{i\sigma\varphi} & 0 \\ 0 & e^{i\sigma\varphi} \end{pmatrix}$ | $\begin{pmatrix} 0 & e^{i\sigma\varphi} \\ e^{i\sigma\varphi} & 0 \end{pmatrix}$ | $\begin{pmatrix} e^{i\sigma\varphi} & 0 \\ 0 & e^{i\sigma\varphi} \end{pmatrix}$ | $\begin{pmatrix} -e^{i\sigma\varphi} & 0 \\ 0 & e^{i\sigma\varphi} \end{pmatrix}$ | $\begin{pmatrix} 0 & e^{i\sigma\varphi} \\ e^{i\sigma\varphi} & 0 \end{pmatrix}$ |
| $\Gamma_5$ | $\begin{pmatrix} e^{i\sigma\varphi} & 0 \\ 0 & e^{i\sigma\varphi} \end{pmatrix}$ | $\begin{pmatrix} 0 & e^{i\sigma\varphi} \\ e^{i\sigma\varphi} & 0 \end{pmatrix}$ | $\begin{pmatrix} e^{\frac{2i\pi}{3}+i\sigma\varphi} & 0 \\ 0 & e^{-\frac{2i\pi}{3}+i\sigma\varphi} \end{pmatrix}$ | $\begin{pmatrix} 0 & e^{i\sigma\varphi} \\ e^{i\sigma\varphi} & 0 \end{pmatrix}$ | $\begin{pmatrix} e^{i\sigma\varphi} & 0 \\ 0 & e^{i\sigma\varphi} \end{pmatrix}$ | $\begin{pmatrix} e^{-\frac{2i\pi}{3}+i\sigma\varphi} & 0 \\ 0 & e^{\frac{2i\pi}{3}+i\sigma\varphi} \end{pmatrix}$ |

| $b^\infty \times {}^14\,{}^13\,{}^12$ | $\{R_\varphi, E\}$ | $\{R_\varphi, C_{2a}\}$ | $\{R_\varphi, C_{4x}\}$ | $\{\tau 2_\perp R_\varphi, E\}$ | $\{\tau 2_\perp R_\varphi, C_{2a}\}$ | $\{\tau 2_\perp R_\varphi, C_{4x}\}$ |
|---|---|---|---|---|---|---|
| $\Gamma_1$ | $e^{i\sigma\varphi}$ | $e^{i\sigma\varphi}$ | $e^{i\sigma\varphi}$ | $e^{i\sigma\varphi}$ | $e^{i\sigma\varphi}$ | $e^{i\sigma\varphi}$ |
| $\Gamma_2$ | $e^{i\sigma\varphi}$ | $-e^{i\sigma\varphi}$ | $-e^{i\sigma\varphi}$ | $e^{i\sigma\varphi}$ | $-e^{i\sigma\varphi}$ | $-e^{i\sigma\varphi}$ |
| $\Gamma_3$ | $\begin{pmatrix} e^{i\sigma\varphi} & 0 \\ 0 & e^{i\sigma\varphi} \end{pmatrix}$ | $\begin{pmatrix} -e^{i\sigma\varphi} & 0 \\ 0 & e^{i\sigma\varphi} \end{pmatrix}$ | $\begin{pmatrix} 0 & e^{i\sigma\varphi} \\ e^{i\sigma\varphi} & 0 \end{pmatrix}$ | $\begin{pmatrix} e^{i\sigma\varphi} & 0 \\ 0 & e^{i\sigma\varphi} \end{pmatrix}$ | $\begin{pmatrix} -e^{i\sigma\varphi} & 0 \\ 0 & e^{i\sigma\varphi} \end{pmatrix}$ | $\begin{pmatrix} 0 & e^{i\sigma\varphi} \\ e^{i\sigma\varphi} & 0 \end{pmatrix}$ |
| $\Gamma_4$ | $\begin{pmatrix} e^{i\sigma\varphi} & 0 \\ 0 & e^{i\sigma\varphi} \end{pmatrix}$ | $\begin{pmatrix} 0 & -e^{i\sigma\varphi} \\ -e^{i\sigma\varphi} & 0 \end{pmatrix}$ | $\begin{pmatrix} 0 & -e^{i\sigma\varphi} \\ e^{i\sigma\varphi} & 0 \end{pmatrix}$ | $\begin{pmatrix} e^{i\sigma\varphi} & 0 \\ 0 & e^{i\sigma\varphi} \end{pmatrix}$ | $\begin{pmatrix} 0 & -e^{i\sigma\varphi} \\ -e^{i\sigma\varphi} & 0 \end{pmatrix}$ | $\begin{pmatrix} 0 & -e^{i\sigma\varphi} \\ e^{i\sigma\varphi} & 0 \end{pmatrix}$ |
| $\Gamma_5$ | $\begin{pmatrix} e^{i\sigma\varphi} & 0 \\ 0 & e^{i\sigma\varphi} \end{pmatrix}$ | $\begin{pmatrix} 0 & e^{i\sigma\varphi} \\ e^{i\sigma\varphi} & 0 \end{pmatrix}$ | $\begin{pmatrix} e^{\frac{2i\pi}{3}+i\sigma\varphi} & 0 \\ 0 & e^{-\frac{2i\pi}{3}+i\sigma\varphi} \end{pmatrix}$ | $\begin{pmatrix} 0 & e^{i\sigma\varphi} \\ e^{i\sigma\varphi} & 0 \end{pmatrix}$ | $\begin{pmatrix} e^{i\sigma\varphi} & 0 \\ 0 & e^{i\sigma\varphi} \end{pmatrix}$ | $\begin{pmatrix} e^{-\frac{2i\pi}{3}+i\sigma\varphi} & 0 \\ 0 & e^{\frac{2i\pi}{3}+i\sigma\varphi} \end{pmatrix}$ |

| $b^\infty \times \frac{1_4/1_m}{1_{\bar 3}}$ $1_2/1_m$ | $\{R_\varphi, E\}$ | $\{R_\varphi, IC_{3\beta}\}$ | $\{R_\varphi, IC_{2f}\}$ | $\{\tau 2_\perp R_\varphi, E\}$ | $\{\tau 2_\perp R_\varphi, IC_{3\beta}\}$ | $\{\tau 2_\perp R_\varphi, IC_{2f}\}$ |
|---|---|---|---|---|---|---|
| $\Gamma_1$ | $e^{i\sigma\varphi}$ | $e^{i\sigma\varphi}$ | $e^{i\sigma\varphi}$ | $e^{i\sigma\varphi}$ | $e^{i\sigma\varphi}$ | $e^{i\sigma\varphi}$ |
| $\Gamma_2$ | $e^{i\sigma\varphi}$ | $-e^{i\sigma\varphi}$ | $-e^{i\sigma\varphi}$ | $e^{i\sigma\varphi}$ | $-e^{i\sigma\varphi}$ | $-e^{i\sigma\varphi}$ |
| $\Gamma_3$ | $e^{i\sigma\varphi}$ | $e^{i\sigma\varphi}$ | $-e^{i\sigma\varphi}$ | $e^{i\sigma\varphi}$ | $e^{i\sigma\varphi}$ | $-e^{i\sigma\varphi}$ |
| $\Gamma_4$ | $e^{i\sigma\varphi}$ | $-e^{i\sigma\varphi}$ | $e^{i\sigma\varphi}$ | $e^{i\sigma\varphi}$ | $-e^{i\sigma\varphi}$ | $e^{i\sigma\varphi}$ |
| $\Gamma_5$ | $\begin{pmatrix} e^{i\sigma\varphi} & 0 & 0 \\ 0 & e^{i\sigma\varphi} & 0 \\ 0 & 0 & e^{i\sigma\varphi} \end{pmatrix}$ | $\begin{pmatrix} 0 & 0 & e^{i\sigma\varphi} \\ 0 & -e^{i\sigma\varphi} & 0 \\ -e^{i\sigma\varphi} & 0 & 0 \end{pmatrix}$ | $\begin{pmatrix} 0 & -e^{i\sigma\varphi} & 0 \\ -e^{i\sigma\varphi} & 0 & 0 \\ 0 & 0 & -e^{i\sigma\varphi} \end{pmatrix}$ | $\begin{pmatrix} e^{i\sigma\varphi} & 0 & 0 \\ 0 & e^{i\sigma\varphi} & 0 \\ 0 & 0 & e^{i\sigma\varphi} \end{pmatrix}$ | $\begin{pmatrix} 0 & 0 & e^{i\sigma\varphi} \\ 0 & -e^{i\sigma\varphi} & 0 \\ -e^{i\sigma\varphi} & 0 & 0 \end{pmatrix}$ | $\begin{pmatrix} 0 & -e^{i\sigma\varphi} & 0 \\ -e^{i\sigma\varphi} & 0 & 0 \\ 0 & 0 & -e^{i\sigma\varphi} \end{pmatrix}$ |
| $\Gamma_6$ | $\begin{pmatrix} e^{i\sigma\varphi} & 0 & 0 \\ 0 & e^{i\sigma\varphi} & 0 \\ 0 & 0 & e^{i\sigma\varphi} \end{pmatrix}$ | $\begin{pmatrix} 0 & 0 & -e^{i\sigma\varphi} \\ 0 & e^{i\sigma\varphi} & 0 \\ e^{i\sigma\varphi} & 0 & 0 \end{pmatrix}$ | $\begin{pmatrix} 0 & e^{i\sigma\varphi} & 0 \\ e^{i\sigma\varphi} & 0 & 0 \\ 0 & 0 & e^{i\sigma\varphi} \end{pmatrix}$ | $\begin{pmatrix} e^{i\sigma\varphi} & 0 & 0 \\ 0 & e^{i\sigma\varphi} & 0 \\ 0 & 0 & e^{i\sigma\varphi} \end{pmatrix}$ | $\begin{pmatrix} 0 & 0 & -e^{i\sigma\varphi} \\ 0 & e^{i\sigma\varphi} & 0 \\ e^{i\sigma\varphi} & 0 & 0 \end{pmatrix}$ | $\begin{pmatrix} 0 & e^{i\sigma\varphi} & 0 \\ e^{i\sigma\varphi} & 0 & 0 \\ 0 & 0 & e^{i\sigma\varphi} \end{pmatrix}$ |
| $\Gamma_7$ | $\begin{pmatrix} e^{i\sigma\varphi} & 0 & 0 \\ 0 & e^{i\sigma\varphi} & 0 \\ 0 & 0 & e^{i\sigma\varphi} \end{pmatrix}$ | $\begin{pmatrix} 0 & 0 & e^{i\sigma\varphi} \\ 0 & -e^{i\sigma\varphi} & 0 \\ -e^{i\sigma\varphi} & 0 & 0 \end{pmatrix}$ | $\begin{pmatrix} 0 & e^{i\sigma\varphi} & 0 \\ e^{i\sigma\varphi} & 0 & 0 \\ 0 & 0 & e^{i\sigma\varphi} \end{pmatrix}$ | $\begin{pmatrix} e^{i\sigma\varphi} & 0 & 0 \\ 0 & e^{i\sigma\varphi} & 0 \\ 0 & 0 & e^{i\sigma\varphi} \end{pmatrix}$ | $\begin{pmatrix} 0 & 0 & e^{i\sigma\varphi} \\ 0 & -e^{i\sigma\varphi} & 0 \\ -e^{i\sigma\varphi} & 0 & 0 \end{pmatrix}$ | $\begin{pmatrix} 0 & e^{i\sigma\varphi} & 0 \\ e^{i\sigma\varphi} & 0 & 0 \\ 0 & 0 & e^{i\sigma\varphi} \end{pmatrix}$ |
| $\Gamma_8$ | $\begin{pmatrix} e^{i\sigma\varphi} & 0 & 0 \\ 0 & e^{i\sigma\varphi} & 0 \\ 0 & 0 & e^{i\sigma\varphi} \end{pmatrix}$ | $\begin{pmatrix} 0 & 0 & -e^{i\sigma\varphi} \\ 0 & e^{i\sigma\varphi} & 0 \\ e^{i\sigma\varphi} & 0 & 0 \end{pmatrix}$ | $\begin{pmatrix} 0 & -e^{i\sigma\varphi} & 0 \\ -e^{i\sigma\varphi} & 0 & 0 \\ 0 & 0 & -e^{i\sigma\varphi} \end{pmatrix}$ | $\begin{pmatrix} e^{i\sigma\varphi} & 0 & 0 \\ 0 & e^{i\sigma\varphi} & 0 \\ 0 & 0 & e^{i\sigma\varphi} \end{pmatrix}$ | $\begin{pmatrix} 0 & 0 & -e^{i\sigma\varphi} \\ 0 & e^{i\sigma\varphi} & 0 \\ e^{i\sigma\varphi} & 0 & 0 \end{pmatrix}$ | $\begin{pmatrix} 0 & -e^{i\sigma\varphi} & 0 \\ -e^{i\sigma\varphi} & 0 & 0 \\ 0 & 0 & -e^{i\sigma\varphi} \end{pmatrix}$ |
| $\Gamma_9$ | $\begin{pmatrix} e^{i\sigma\varphi} & 0 \\ 0 & e^{i\sigma\varphi} \end{pmatrix}$ | $\begin{pmatrix} e^{-\frac{2i\pi}{3}+i\sigma\varphi} & 0 \\ 0 & e^{\frac{2i\pi}{3}+i\sigma\varphi} \end{pmatrix}$ | $\begin{pmatrix} 0 & e^{-\frac{2i\pi}{3}+i\sigma\varphi} \\ e^{\frac{2i\pi}{3}+i\sigma\varphi} & 0 \end{pmatrix}$ | $\begin{pmatrix} 0 & e^{i\sigma\varphi} \\ e^{i\sigma\varphi} & 0 \end{pmatrix}$ | $\begin{pmatrix} 0 & e^{-\frac{2i\pi}{3}+i\sigma\varphi} \\ e^{\frac{2i\pi}{3}+i\sigma\varphi} & 0 \end{pmatrix}$ | $\begin{pmatrix} e^{-\frac{2i\pi}{3}+i\sigma\varphi} & 0 \\ 0 & e^{\frac{2i\pi}{3}+i\sigma\varphi} \end{pmatrix}$ |
| $\Gamma_{10}$ | $\begin{pmatrix} e^{i\sigma\varphi} & 0 \\ 0 & e^{i\sigma\varphi} \end{pmatrix}$ | $\begin{pmatrix} -e^{-\frac{2i\pi}{3}+i\sigma\varphi} & 0 \\ 0 & -e^{\frac{2i\pi}{3}+i\sigma\varphi} \end{pmatrix}$ | $\begin{pmatrix} 0 & -e^{-\frac{2i\pi}{3}+i\sigma\varphi} \\ -e^{\frac{2i\pi}{3}+i\sigma\varphi} & 0 \end{pmatrix}$ | $\begin{pmatrix} 0 & e^{i\sigma\varphi} \\ e^{i\sigma\varphi} & 0 \end{pmatrix}$ | $\begin{pmatrix} 0 & -e^{-\frac{2i\pi}{3}+i\sigma\varphi} \\ -e^{\frac{2i\pi}{3}+i\sigma\varphi} & 0 \end{pmatrix}$ | $\begin{pmatrix} -e^{-\frac{2i\pi}{3}+i\sigma\varphi} & 0 \\ 0 & -e^{\frac{2i\pi}{3}+i\sigma\varphi} \end{pmatrix}$ |

## F.2 Collinear groups with non-unitary nontrivial group

| $b^\infty \times \bar{1}\bar{1}$ | $\{R_\varphi, E\}$ | $\{2_\perp R_\varphi, I\}$ | $\{\tau 2_\perp R_\varphi, E\}$ | $\{\tau R_\varphi, I\}$ |
|---|---|---|---|---|
| $\Gamma_1$ | 1 | 1 | 1 | 1 |
| $\Gamma_2$ | 1 | $-1$ | 1 | $-1$ |
| $\Gamma_3$ | $\begin{pmatrix} e^{i\nu\varphi} & 0 \\ 0 & e^{-i\nu\varphi} \end{pmatrix}$ | $\begin{pmatrix} 0 & e^{-i\nu\varphi} \\ e^{i\nu\varphi} & 0 \end{pmatrix}$ | $\begin{pmatrix} e^{i\nu\varphi} & 0 \\ 0 & e^{-i\nu\varphi} \end{pmatrix}$ | $\begin{pmatrix} 0 & e^{-i\nu\varphi} \\ e^{i\nu\varphi} & 0 \end{pmatrix}$ |

| $b^\infty \times \bar{1}2$ | $\{R_\varphi, E\}$ | $\{2_\perp R_\varphi, C_{2z}\}$ | $\{\tau 2_\perp R_\varphi, E\}$ | $\{\tau R_\varphi, C_{2z}\}$ |
|---|---|---|---|---|
| $\Gamma_1$ | 1 | 1 | 1 | 1 |
| $\Gamma_2$ | 1 | $-1$ | 1 | $-1$ |
| $\Gamma_3$ | $\begin{pmatrix} e^{i\nu\varphi} & 0 \\ 0 & e^{-i\nu\varphi} \end{pmatrix}$ | $\begin{pmatrix} 0 & e^{-i\nu\varphi} \\ e^{i\nu\varphi} & 0 \end{pmatrix}$ | $\begin{pmatrix} e^{i\nu\varphi} & 0 \\ 0 & e^{-i\nu\varphi} \end{pmatrix}$ | $\begin{pmatrix} 0 & e^{-i\nu\varphi} \\ e^{i\nu\varphi} & 0 \end{pmatrix}$ |

| $b^\infty \times \bar{1}m$ | $\{R_\varphi, E\}$ | $\{2_\perp R_\varphi, IC_{2z}\}$ | $\{\tau 2_\perp R_\varphi, E\}$ | $\{\tau R_\varphi, IC_{2z}\}$ |
|---|---|---|---|---|
| $\Gamma_1$ | 1 | 1 | 1 | 1 |
| $\Gamma_2$ | 1 | $-1$ | 1 | $-1$ |
| $\Gamma_3$ | $\begin{pmatrix} e^{i\nu\varphi} & 0 \\ 0 & e^{-i\nu\varphi} \end{pmatrix}$ | $\begin{pmatrix} 0 & e^{-i\nu\varphi} \\ e^{i\nu\varphi} & 0 \end{pmatrix}$ | $\begin{pmatrix} e^{i\nu\varphi} & 0 \\ 0 & e^{-i\nu\varphi} \end{pmatrix}$ | $\begin{pmatrix} 0 & e^{-i\nu\varphi} \\ e^{i\nu\varphi} & 0 \end{pmatrix}$ |

| $b^\infty \times {}^1 2/\bar{1}_m$ | $\{R_\varphi, E\}$ | $\{R_\varphi, C_{2z}\}$ | $\{2_\perp R_\varphi, IC_{2z}\}$ | $\{\tau 2_\perp R_\varphi, E\}$ | $\{\tau 2_\perp R_\varphi, C_{2z}\}$ | $\{\tau R_\varphi, IC_{2z}\}$ |
|---|---|---|---|---|---|---|
| $\Gamma_1$ | 1 | 1 | 1 | 1 | 1 | 1 |
| $\Gamma_2$ | 1 | 1 | $-1$ | 1 | 1 | $-1$ |
| $\Gamma_3$ | 1 | $-1$ | 1 | 1 | $-1$ | 1 |
| $\Gamma_4$ | 1 | $-1$ | $-1$ | 1 | $-1$ | $-1$ |
| $\Gamma_5$ | $\begin{pmatrix} e^{i\nu\varphi} & 0 \\ 0 & e^{-i\nu\varphi} \end{pmatrix}$ | $\begin{pmatrix} e^{i\nu\varphi} & 0 \\ 0 & e^{-i\nu\varphi} \end{pmatrix}$ | $\begin{pmatrix} 0 & e^{-i\nu\varphi} \\ e^{i\nu\varphi} & 0 \end{pmatrix}$ | $\begin{pmatrix} e^{i\nu\varphi} & 0 \\ 0 & e^{-i\nu\varphi} \end{pmatrix}$ | $\begin{pmatrix} e^{i\nu\varphi} & 0 \\ 0 & e^{-i\nu\varphi} \end{pmatrix}$ | $\begin{pmatrix} 0 & e^{-i\nu\varphi} \\ e^{i\nu\varphi} & 0 \end{pmatrix}$ |
| $\Gamma_6$ | $\begin{pmatrix} e^{i\nu\varphi} & 0 \\ 0 & e^{-i\nu\varphi} \end{pmatrix}$ | $\begin{pmatrix} -e^{i\nu\varphi} & 0 \\ 0 & -e^{-i\nu\varphi} \end{pmatrix}$ | $\begin{pmatrix} 0 & e^{-i\nu\varphi} \\ e^{i\nu\varphi} & 0 \end{pmatrix}$ | $\begin{pmatrix} e^{i\nu\varphi} & 0 \\ 0 & e^{-i\nu\varphi} \end{pmatrix}$ | $\begin{pmatrix} -e^{i\nu\varphi} & 0 \\ 0 & -e^{-i\nu\varphi} \end{pmatrix}$ | $\begin{pmatrix} 0 & e^{-i\nu\varphi} \\ e^{i\nu\varphi} & 0 \end{pmatrix}$ |

| $b^\infty \times {}^1 2/{}^1_m$ | $\{R_\varphi, E\}$ | $\{R_\varphi, IC_{2z}\}$ | $\{2_\perp R_\varphi, C_{2z}\}$ | $\{\tau 2_\perp R_\varphi, E\}$ | $\{\tau 2_\perp R_\varphi, IC_{2z}\}$ | $\{\tau R_\varphi, C_{2z}\}$ |
|---|---|---|---|---|---|---|
| $\Gamma_1$ | 1 | 1 | 1 | 1 | 1 | 1 |
| $\Gamma_2$ | 1 | 1 | $-1$ | 1 | 1 | $-1$ |
| $\Gamma_3$ | 1 | $-1$ | 1 | 1 | $-1$ | 1 |
| $\Gamma_4$ | 1 | $-1$ | $-1$ | 1 | $-1$ | $-1$ |
| $\Gamma_5$ | $\begin{pmatrix} e^{i\nu\varphi} & 0 \\ 0 & e^{-i\nu\varphi} \end{pmatrix}$ | $\begin{pmatrix} e^{i\nu\varphi} & 0 \\ 0 & e^{-i\nu\varphi} \end{pmatrix}$ | $\begin{pmatrix} 0 & e^{-i\nu\varphi} \\ e^{i\nu\varphi} & 0 \end{pmatrix}$ | $\begin{pmatrix} e^{i\nu\varphi} & 0 \\ 0 & e^{-i\nu\varphi} \end{pmatrix}$ | $\begin{pmatrix} e^{i\nu\varphi} & 0 \\ 0 & e^{-i\nu\varphi} \end{pmatrix}$ | $\begin{pmatrix} 0 & e^{-i\nu\varphi} \\ e^{i\nu\varphi} & 0 \end{pmatrix}$ |
| $\Gamma_6$ | $\begin{pmatrix} e^{i\nu\varphi} & 0 \\ 0 & e^{-i\nu\varphi} \end{pmatrix}$ | $\begin{pmatrix} -e^{i\nu\varphi} & 0 \\ 0 & -e^{-i\nu\varphi} \end{pmatrix}$ | $\begin{pmatrix} 0 & e^{-i\nu\varphi} \\ e^{i\nu\varphi} & 0 \end{pmatrix}$ | $\begin{pmatrix} e^{i\nu\varphi} & 0 \\ 0 & e^{-i\nu\varphi} \end{pmatrix}$ | $\begin{pmatrix} -e^{i\nu\varphi} & 0 \\ 0 & -e^{-i\nu\varphi} \end{pmatrix}$ | $\begin{pmatrix} 0 & e^{-i\nu\varphi} \\ e^{i\nu\varphi} & 0 \end{pmatrix}$ |

| $b^\infty \times \bar{1}2/\bar{1}_m$ | $\{R_\varphi, E\}$ | $\{2_\perp R_\varphi, C_{2z}\}$ | $\{2_\perp R_\varphi, IC_{2z}\}$ | $\{\tau 2_\perp R_\varphi, E\}$ | $\{\tau R_\varphi, C_{2z}\}$ | $\{\tau R_\varphi, IC_{2z}\}$ |
|---|---|---|---|---|---|---|
| $\Gamma_1$ | $1$ | $1$ | $1$ | $1$ | $1$ | $1$ |
| $\Gamma_2$ | $1$ | $-1$ | $-1$ | $1$ | $-1$ | $-1$ |
| $\Gamma_3$ | $1$ | $1$ | $-1$ | $1$ | $1$ | $-1$ |
| $\Gamma_4$ | $1$ | $-1$ | $1$ | $1$ | $-1$ | $1$ |
| $\Gamma_5$ | $\begin{pmatrix} e^{i\nu\varphi} & 0 \\ 0 & e^{-i\nu\varphi} \end{pmatrix}$ | $\begin{pmatrix} 0 & e^{-i\nu\varphi} \\ e^{i\nu\varphi} & 0 \end{pmatrix}$ | $\begin{pmatrix} 0 & e^{-i\nu\varphi} \\ e^{i\nu\varphi} & 0 \end{pmatrix}$ | $\begin{pmatrix} e^{i\nu\varphi} & 0 \\ 0 & e^{-i\nu\varphi} \end{pmatrix}$ | $\begin{pmatrix} 0 & e^{-i\nu\varphi} \\ e^{i\nu\varphi} & 0 \end{pmatrix}$ | $\begin{pmatrix} 0 & e^{-i\nu\varphi} \\ e^{i\nu\varphi} & 0 \end{pmatrix}$ |
| $\Gamma_6$ | $\begin{pmatrix} e^{i\nu\varphi} & 0 \\ 0 & e^{-i\nu\varphi} \end{pmatrix}$ | $\begin{pmatrix} 0 & e^{-i\nu\varphi} \\ e^{i\nu\varphi} & 0 \end{pmatrix}$ | $\begin{pmatrix} 0 & -e^{-i\nu\varphi} \\ -e^{i\nu\varphi} & 0 \end{pmatrix}$ | $\begin{pmatrix} e^{i\nu\varphi} & 0 \\ 0 & e^{-i\nu\varphi} \end{pmatrix}$ | $\begin{pmatrix} 0 & e^{-i\nu\varphi} \\ e^{i\nu\varphi} & 0 \end{pmatrix}$ | $\begin{pmatrix} 0 & -e^{-i\nu\varphi} \\ -e^{i\nu\varphi} & 0 \end{pmatrix}$ |

| $b^\infty \times \bar{1}m\,\bar{1}m\,^{1}2$ | $\{R_\varphi, E\}$ | $\{2_\perp R_\varphi, IC_{2x}\}$ | $\{2_\perp R_\varphi, IC_{2y}\}$ | $\{\tau 2_\perp R_\varphi, E\}$ | $\{\tau R_\varphi, IC_{2x}\}$ | $\{\tau R_\varphi, IC_{2y}\}$ |
|---|---|---|---|---|---|---|
| $\Gamma_1$ | $1$ | $1$ | $1$ | $1$ | $1$ | $1$ |
| $\Gamma_2$ | $1$ | $-1$ | $-1$ | $1$ | $-1$ | $-1$ |
| $\Gamma_3$ | $1$ | $1$ | $-1$ | $1$ | $1$ | $-1$ |
| $\Gamma_4$ | $1$ | $-1$ | $1$ | $1$ | $-1$ | $1$ |
| $\Gamma_5$ | $\begin{pmatrix} e^{i\nu\varphi} & 0 \\ 0 & e^{-i\nu\varphi} \end{pmatrix}$ | $\begin{pmatrix} 0 & e^{-i\nu\varphi} \\ e^{i\nu\varphi} & 0 \end{pmatrix}$ | $\begin{pmatrix} 0 & e^{-i\nu\varphi} \\ e^{i\nu\varphi} & 0 \end{pmatrix}$ | $\begin{pmatrix} e^{i\nu\varphi} & 0 \\ 0 & e^{-i\nu\varphi} \end{pmatrix}$ | $\begin{pmatrix} 0 & e^{-i\nu\varphi} \\ e^{i\nu\varphi} & 0 \end{pmatrix}$ | $\begin{pmatrix} 0 & e^{-i\nu\varphi} \\ e^{i\nu\varphi} & 0 \end{pmatrix}$ |
| $\Gamma_6$ | $\begin{pmatrix} e^{i\nu\varphi} & 0 \\ 0 & e^{-i\nu\varphi} \end{pmatrix}$ | $\begin{pmatrix} 0 & e^{-i\nu\varphi} \\ e^{i\nu\varphi} & 0 \end{pmatrix}$ | $\begin{pmatrix} 0 & -e^{-i\nu\varphi} \\ -e^{i\nu\varphi} & 0 \end{pmatrix}$ | $\begin{pmatrix} e^{i\nu\varphi} & 0 \\ 0 & e^{-i\nu\varphi} \end{pmatrix}$ | $\begin{pmatrix} 0 & e^{-i\nu\varphi} \\ e^{i\nu\varphi} & 0 \end{pmatrix}$ | $\begin{pmatrix} 0 & -e^{-i\nu\varphi} \\ -e^{i\nu\varphi} & 0 \end{pmatrix}$ |

| $b^\infty \times {}^{1}m\,\bar{1}m\,\bar{1}2$ | $\{R_\varphi, E\}$ | $\{R_\varphi, IC_{2z}\}$ | $\{2_\perp R_\varphi, C_{2z}\}$ | $\{\tau 2_\perp R_\varphi, E\}$ | $\{\tau 2_\perp R_\varphi, IC_{2z}\}$ | $\{\tau R_\varphi, C_{2z}\}$ |
|---|---|---|---|---|---|---|
| $\Gamma_1$ | $1$ | $1$ | $1$ | $1$ | $1$ | $1$ |
| $\Gamma_2$ | $1$ | $1$ | $-1$ | $1$ | $1$ | $-1$ |
| $\Gamma_3$ | $1$ | $-1$ | $1$ | $1$ | $-1$ | $1$ |
| $\Gamma_4$ | $1$ | $-1$ | $-1$ | $1$ | $-1$ | $-1$ |
| $\Gamma_5$ | $\begin{pmatrix} e^{i\nu\varphi} & 0 \\ 0 & e^{-i\nu\varphi} \end{pmatrix}$ | $\begin{pmatrix} e^{i\nu\varphi} & 0 \\ 0 & e^{-i\nu\varphi} \end{pmatrix}$ | $\begin{pmatrix} 0 & e^{-i\nu\varphi} \\ e^{i\nu\varphi} & 0 \end{pmatrix}$ | $\begin{pmatrix} e^{i\nu\varphi} & 0 \\ 0 & e^{-i\nu\varphi} \end{pmatrix}$ | $\begin{pmatrix} e^{i\nu\varphi} & 0 \\ 0 & e^{-i\nu\varphi} \end{pmatrix}$ | $\begin{pmatrix} 0 & e^{-i\nu\varphi} \\ e^{i\nu\varphi} & 0 \end{pmatrix}$ |
| $\Gamma_6$ | $\begin{pmatrix} e^{i\nu\varphi} & 0 \\ 0 & e^{-i\nu\varphi} \end{pmatrix}$ | $\begin{pmatrix} -e^{i\nu\varphi} & 0 \\ 0 & -e^{-i\nu\varphi} \end{pmatrix}$ | $\begin{pmatrix} 0 & e^{-i\nu\varphi} \\ e^{i\nu\varphi} & 0 \end{pmatrix}$ | $\begin{pmatrix} e^{i\nu\varphi} & 0 \\ 0 & e^{-i\nu\varphi} \end{pmatrix}$ | $\begin{pmatrix} -e^{i\nu\varphi} & 0 \\ 0 & -e^{-i\nu\varphi} \end{pmatrix}$ | $\begin{pmatrix} 0 & e^{-i\nu\varphi} \\ e^{i\nu\varphi} & 0 \end{pmatrix}$ |

| $b^\infty \times {}^{1}2\,\bar{1}2\,\bar{1}2$ | $\{R_\varphi, E\}$ | $\{2_\perp R_\varphi, C_{2x}\}$ | $\{2_\perp R_\varphi, C_{2y}\}$ | $\{\tau 2_\perp R_\varphi, E\}$ | $\{\tau R_\varphi, C_{2x}\}$ | $\{\tau R_\varphi, C_{2y}\}$ |
|---|---|---|---|---|---|---|
| $\Gamma_1$ | $1$ | $1$ | $1$ | $1$ | $1$ | $1$ |
| $\Gamma_2$ | $1$ | $-1$ | $-1$ | $1$ | $-1$ | $-1$ |
| $\Gamma_3$ | $1$ | $1$ | $-1$ | $1$ | $1$ | $-1$ |
| $\Gamma_4$ | $1$ | $-1$ | $1$ | $1$ | $-1$ | $1$ |
| $\Gamma_5$ | $\begin{pmatrix} e^{i\nu\varphi} & 0 \\ 0 & e^{-i\nu\varphi} \end{pmatrix}$ | $\begin{pmatrix} 0 & e^{-i\nu\varphi} \\ e^{i\nu\varphi} & 0 \end{pmatrix}$ | $\begin{pmatrix} 0 & e^{-i\nu\varphi} \\ e^{i\nu\varphi} & 0 \end{pmatrix}$ | $\begin{pmatrix} e^{i\nu\varphi} & 0 \\ 0 & e^{-i\nu\varphi} \end{pmatrix}$ | $\begin{pmatrix} 0 & e^{-i\nu\varphi} \\ e^{i\nu\varphi} & 0 \end{pmatrix}$ | $\begin{pmatrix} 0 & e^{-i\nu\varphi} \\ e^{i\nu\varphi} & 0 \end{pmatrix}$ |
| $\Gamma_6$ | $\begin{pmatrix} e^{i\nu\varphi} & 0 \\ 0 & e^{-i\nu\varphi} \end{pmatrix}$ | $\begin{pmatrix} 0 & e^{-i\nu\varphi} \\ e^{i\nu\varphi} & 0 \end{pmatrix}$ | $\begin{pmatrix} 0 & -e^{-i\nu\varphi} \\ -e^{i\nu\varphi} & 0 \end{pmatrix}$ | $\begin{pmatrix} e^{i\nu\varphi} & 0 \\ 0 & e^{-i\nu\varphi} \end{pmatrix}$ | $\begin{pmatrix} 0 & e^{-i\nu\varphi} \\ e^{i\nu\varphi} & 0 \end{pmatrix}$ | $\begin{pmatrix} 0 & -e^{-i\nu\varphi} \\ -e^{i\nu\varphi} & 0 \end{pmatrix}$ |

**Table 1**

| $b^\infty \times {}^1m \,/\, {}^{\bar1}m\,{}^{\bar1}m$ | $\{R_\varphi, E\}$ | $\{R_\varphi, IC_{2z}\}$ | $\{2_\perp R_\varphi, IC_{2y}\}$ | $\{2_\perp R_\varphi, IC_{2x}\}$ | $\{\tau 2_\perp R_\varphi, E\}$ | $\{\tau 2_\perp R_\varphi, IC_{2z}\}$ | $\{\tau R_\varphi, IC_{2y}\}$ | $\{\tau R_\varphi, IC_{2x}\}$ |
|---|---|---|---|---|---|---|---|---|
| $\Gamma_1$ | $1$ | $1$ | $1$ | $1$ | $1$ | $1$ | $1$ | $1$ |
| $\Gamma_2$ | $1$ | $1$ | $-1$ | $-1$ | $1$ | $1$ | $-1$ | $-1$ |
| $\Gamma_3$ | $1$ | $-1$ | $-1$ | $-1$ | $1$ | $-1$ | $-1$ | $-1$ |
| $\Gamma_4$ | $1$ | $-1$ | $1$ | $1$ | $1$ | $-1$ | $1$ | $1$ |
| $\Gamma_5$ | $1$ | $1$ | $1$ | $-1$ | $1$ | $1$ | $1$ | $-1$ |
| $\Gamma_6$ | $1$ | $1$ | $-1$ | $1$ | $1$ | $1$ | $-1$ | $1$ |
| $\Gamma_7$ | $1$ | $-1$ | $-1$ | $1$ | $1$ | $-1$ | $-1$ | $1$ |
| $\Gamma_8$ | $1$ | $-1$ | $1$ | $-1$ | $1$ | $-1$ | $1$ | $-1$ |
| $\Gamma_9$ | $\begin{pmatrix} e^{i\varphi} & 0 \\ 0 & e^{-i\varphi} \end{pmatrix}$ | $\begin{pmatrix} e^{i\varphi} & 0 \\ 0 & e^{-i\varphi} \end{pmatrix}$ | $\begin{pmatrix} 0 & e^{-i\varphi} \\ e^{i\varphi} & 0 \end{pmatrix}$ | $\begin{pmatrix} 0 & e^{-i\varphi} \\ e^{i\varphi} & 0 \end{pmatrix}$ | $\begin{pmatrix} e^{i\varphi} & 0 \\ 0 & e^{-i\varphi} \end{pmatrix}$ | $\begin{pmatrix} e^{i\varphi} & 0 \\ 0 & e^{-i\varphi} \end{pmatrix}$ | $\begin{pmatrix} 0 & e^{-i\varphi} \\ e^{i\varphi} & 0 \end{pmatrix}$ | $\begin{pmatrix} 0 & e^{-i\varphi} \\ e^{i\varphi} & 0 \end{pmatrix}$ |
| $\Gamma_{10}$ | $\begin{pmatrix} e^{i\varphi} & 0 \\ 0 & e^{-i\varphi} \end{pmatrix}$ | $\begin{pmatrix} -e^{i\varphi} & 0 \\ 0 & -e^{-i\varphi} \end{pmatrix}$ | $\begin{pmatrix} 0 & -e^{-i\varphi} \\ -e^{i\varphi} & 0 \end{pmatrix}$ | $\begin{pmatrix} 0 & -e^{-i\varphi} \\ -e^{i\varphi} & 0 \end{pmatrix}$ | $\begin{pmatrix} e^{i\varphi} & 0 \\ 0 & e^{-i\varphi} \end{pmatrix}$ | $\begin{pmatrix} -e^{i\varphi} & 0 \\ 0 & -e^{-i\varphi} \end{pmatrix}$ | $\begin{pmatrix} 0 & -e^{-i\varphi} \\ -e^{i\varphi} & 0 \end{pmatrix}$ | $\begin{pmatrix} 0 & -e^{-i\varphi} \\ -e^{i\varphi} & 0 \end{pmatrix}$ |
| $\Gamma_{11}$ | $\begin{pmatrix} e^{i\varphi} & 0 \\ 0 & e^{-i\varphi} \end{pmatrix}$ | $\begin{pmatrix} e^{i\varphi} & 0 \\ 0 & e^{-i\varphi} \end{pmatrix}$ | $\begin{pmatrix} 0 & e^{-i\varphi} \\ e^{i\varphi} & 0 \end{pmatrix}$ | $\begin{pmatrix} 0 & -e^{-i\varphi} \\ -e^{i\varphi} & 0 \end{pmatrix}$ | $\begin{pmatrix} e^{i\varphi} & 0 \\ 0 & e^{-i\varphi} \end{pmatrix}$ | $\begin{pmatrix} e^{i\varphi} & 0 \\ 0 & e^{-i\varphi} \end{pmatrix}$ | $\begin{pmatrix} 0 & e^{-i\varphi} \\ e^{i\varphi} & 0 \end{pmatrix}$ | $\begin{pmatrix} 0 & -e^{-i\varphi} \\ -e^{i\varphi} & 0 \end{pmatrix}$ |
| $\Gamma_{12}$ | $\begin{pmatrix} e^{i\varphi} & 0 \\ 0 & e^{-i\varphi} \end{pmatrix}$ | $\begin{pmatrix} -e^{i\varphi} & 0 \\ 0 & -e^{-i\varphi} \end{pmatrix}$ | $\begin{pmatrix} 0 & -e^{-i\varphi} \\ -e^{i\varphi} & 0 \end{pmatrix}$ | $\begin{pmatrix} 0 & e^{-i\varphi} \\ e^{i\varphi} & 0 \end{pmatrix}$ | $\begin{pmatrix} e^{i\varphi} & 0 \\ 0 & e^{-i\varphi} \end{pmatrix}$ | $\begin{pmatrix} -e^{i\varphi} & 0 \\ 0 & -e^{-i\varphi} \end{pmatrix}$ | $\begin{pmatrix} 0 & -e^{-i\varphi} \\ -e^{i\varphi} & 0 \end{pmatrix}$ | $\begin{pmatrix} 0 & e^{-i\varphi} \\ e^{i\varphi} & 0 \end{pmatrix}$ |

**Table 2**

| $b^\infty \times {}^1m \,/\, {}^{\bar1}m\,{}^{\bar1}m$ | $\{R_\varphi, E\}$ | $\{R_\varphi, IC_{2x}\}$ | $\{R_\varphi, IC_{2y}\}$ | $\{2_\perp R_\varphi, IC_{2z}\}$ | $\{\tau 2_\perp R_\varphi, E\}$ | $\{\tau 2_\perp R_\varphi, IC_{2x}\}$ | $\{\tau 2_\perp R_\varphi, IC_{2y}\}$ | $\{\tau R_\varphi, IC_{2z}\}$ |
|---|---|---|---|---|---|---|---|---|
| $\Gamma_1$ | $1$ | $1$ | $1$ | $1$ | $1$ | $1$ | $1$ | $1$ |
| $\Gamma_2$ | $1$ | $1$ | $1$ | $-1$ | $1$ | $1$ | $1$ | $-1$ |
| $\Gamma_3$ | $1$ | $-1$ | $-1$ | $-1$ | $1$ | $-1$ | $-1$ | $-1$ |
| $\Gamma_4$ | $1$ | $-1$ | $-1$ | $1$ | $1$ | $-1$ | $-1$ | $1$ |
| $\Gamma_5$ | $1$ | $1$ | $-1$ | $-1$ | $1$ | $1$ | $-1$ | $-1$ |
| $\Gamma_6$ | $1$ | $1$ | $-1$ | $1$ | $1$ | $1$ | $-1$ | $1$ |
| $\Gamma_7$ | $1$ | $-1$ | $1$ | $1$ | $1$ | $-1$ | $1$ | $1$ |
| $\Gamma_8$ | $1$ | $-1$ | $1$ | $-1$ | $1$ | $-1$ | $1$ | $-1$ |
| $\Gamma_9$ | $\begin{pmatrix} e^{i\varphi} & 0 \\ 0 & e^{-i\varphi} \end{pmatrix}$ | $\begin{pmatrix} 0 & e^{-i\varphi} \\ e^{i\varphi} & 0 \end{pmatrix}$ | $\begin{pmatrix} 0 & e^{-i\varphi} \\ e^{i\varphi} & 0 \end{pmatrix}$ | $\begin{pmatrix} e^{i\varphi} & 0 \\ 0 & e^{-i\varphi} \end{pmatrix}$ | $\begin{pmatrix} e^{i\varphi} & 0 \\ 0 & e^{-i\varphi} \end{pmatrix}$ | $\begin{pmatrix} 0 & e^{-i\varphi} \\ e^{i\varphi} & 0 \end{pmatrix}$ | $\begin{pmatrix} 0 & e^{-i\varphi} \\ e^{i\varphi} & 0 \end{pmatrix}$ | $\begin{pmatrix} e^{i\varphi} & 0 \\ 0 & e^{-i\varphi} \end{pmatrix}$ |
| $\Gamma_{10}$ | $\begin{pmatrix} e^{i\varphi} & 0 \\ 0 & e^{-i\varphi} \end{pmatrix}$ | $\begin{pmatrix} 0 & -e^{-i\varphi} \\ -e^{i\varphi} & 0 \end{pmatrix}$ | $\begin{pmatrix} 0 & -e^{-i\varphi} \\ -e^{i\varphi} & 0 \end{pmatrix}$ | $\begin{pmatrix} -e^{i\varphi} & 0 \\ 0 & -e^{-i\varphi} \end{pmatrix}$ | $\begin{pmatrix} e^{i\varphi} & 0 \\ 0 & e^{-i\varphi} \end{pmatrix}$ | $\begin{pmatrix} 0 & -e^{-i\varphi} \\ -e^{i\varphi} & 0 \end{pmatrix}$ | $\begin{pmatrix} 0 & -e^{-i\varphi} \\ -e^{i\varphi} & 0 \end{pmatrix}$ | $\begin{pmatrix} -e^{i\varphi} & 0 \\ 0 & -e^{-i\varphi} \end{pmatrix}$ |
| $\Gamma_{11}$ | $\begin{pmatrix} e^{i\varphi} & 0 \\ 0 & e^{-i\varphi} \end{pmatrix}$ | $\begin{pmatrix} 0 & e^{-i\varphi} \\ e^{i\varphi} & 0 \end{pmatrix}$ | $\begin{pmatrix} 0 & -e^{-i\varphi} \\ -e^{i\varphi} & 0 \end{pmatrix}$ | $\begin{pmatrix} e^{i\varphi} & 0 \\ 0 & e^{-i\varphi} \end{pmatrix}$ | $\begin{pmatrix} e^{i\varphi} & 0 \\ 0 & e^{-i\varphi} \end{pmatrix}$ | $\begin{pmatrix} 0 & e^{-i\varphi} \\ e^{i\varphi} & 0 \end{pmatrix}$ | $\begin{pmatrix} 0 & -e^{-i\varphi} \\ -e^{i\varphi} & 0 \end{pmatrix}$ | $\begin{pmatrix} e^{i\varphi} & 0 \\ 0 & e^{-i\varphi} \end{pmatrix}$ |
| $\Gamma_{12}$ | $\begin{pmatrix} e^{i\varphi} & 0 \\ 0 & e^{-i\varphi} \end{pmatrix}$ | $\begin{pmatrix} 0 & -e^{-i\varphi} \\ -e^{i\varphi} & 0 \end{pmatrix}$ | $\begin{pmatrix} 0 & e^{-i\varphi} \\ e^{i\varphi} & 0 \end{pmatrix}$ | $\begin{pmatrix} -e^{i\varphi} & 0 \\ 0 & -e^{-i\varphi} \end{pmatrix}$ | $\begin{pmatrix} e^{i\varphi} & 0 \\ 0 & e^{-i\varphi} \end{pmatrix}$ | $\begin{pmatrix} 0 & -e^{-i\varphi} \\ -e^{i\varphi} & 0 \end{pmatrix}$ | $\begin{pmatrix} 0 & e^{-i\varphi} \\ e^{i\varphi} & 0 \end{pmatrix}$ | $\begin{pmatrix} -e^{i\varphi} & 0 \\ 0 & -e^{-i\varphi} \end{pmatrix}$ |



| $b^\infty \times \bar{1}m$ $\bar{1}m\,\bar{1}m$ | $\{R_\varphi,\,E\}$ | $\{2_\perp R_\varphi,\,IC_{2z}\}$ | $\{2_\perp R_\varphi,\,IC_{2y}\}$ | $\{2_\perp R_\varphi,\,IC_{2x}\}$ | $\{\tau 2_\perp R_\varphi,\,E\}$ | $\{\tau R_\varphi,\,IC_{2z}\}$ | $\{\tau R_\varphi,\,IC_{2y}\}$ | $\{\tau R_\varphi,\,IC_{2x}\}$ |
|---|---|---|---|---|---|---|---|---|
| $\Gamma_1$ | $1$ | $1$ | $1$ | $1$ | $1$ | $1$ | $1$ | $1$ |
| $\Gamma_2$ | $1$ | $-1$ | $-1$ | $-1$ | $1$ | $-1$ | $-1$ | $-1$ |
| $\Gamma_3$ | $1$ | $1$ | $-1$ | $-1$ | $1$ | $1$ | $-1$ | $-1$ |
| $\Gamma_4$ | $1$ | $-1$ | $1$ | $1$ | $1$ | $-1$ | $1$ | $1$ |
| $\Gamma_5$ | $1$ | $1$ | $1$ | $-1$ | $1$ | $1$ | $1$ | $-1$ |
| $\Gamma_6$ | $1$ | $-1$ | $-1$ | $1$ | $1$ | $-1$ | $-1$ | $1$ |
| $\Gamma_7$ | $1$ | $1$ | $-1$ | $1$ | $1$ | $1$ | $-1$ | $1$ |
| $\Gamma_8$ | $1$ | $-1$ | $1$ | $-1$ | $1$ | $-1$ | $1$ | $-1$ |
| $\Gamma_9$ | $\begin{pmatrix} e^{i\nu\varphi} & 0 \\ 0 & e^{-i\nu\varphi} \end{pmatrix}$ | $\begin{pmatrix} 0 & e^{-i\nu\varphi} \\ e^{i\nu\varphi} & 0 \end{pmatrix}$ | $\begin{pmatrix} 0 & e^{-i\nu\varphi} \\ e^{i\nu\varphi} & 0 \end{pmatrix}$ | $\begin{pmatrix} 0 & e^{-i\nu\varphi} \\ e^{i\nu\varphi} & 0 \end{pmatrix}$ | $\begin{pmatrix} e^{i\nu\varphi} & 0 \\ 0 & e^{-i\nu\varphi} \end{pmatrix}$ | $\begin{pmatrix} e^{-i\nu\varphi} & 0 \\ 0 & e^{i\nu\varphi} \end{pmatrix}$ | $\begin{pmatrix} e^{-i\nu\varphi} & 0 \\ 0 & e^{i\nu\varphi} \end{pmatrix}$ | $\begin{pmatrix} e^{-i\nu\varphi} & 0 \\ 0 & e^{i\nu\varphi} \end{pmatrix}$ |
| $\Gamma_{10}$ | $\begin{pmatrix} e^{i\nu\varphi} & 0 \\ 0 & e^{-i\nu\varphi} \end{pmatrix}$ | $\begin{pmatrix} 0 & e^{-i\nu\varphi} \\ e^{i\nu\varphi} & 0 \end{pmatrix}$ | $\begin{pmatrix} 0 & -e^{-i\nu\varphi} \\ -e^{i\nu\varphi} & 0 \end{pmatrix}$ | $\begin{pmatrix} 0 & -e^{-i\nu\varphi} \\ -e^{i\nu\varphi} & 0 \end{pmatrix}$ | $\begin{pmatrix} e^{i\nu\varphi} & 0 \\ 0 & e^{-i\nu\varphi} \end{pmatrix}$ | $\begin{pmatrix} e^{-i\nu\varphi} & 0 \\ 0 & e^{i\nu\varphi} \end{pmatrix}$ | $\begin{pmatrix} -e^{-i\nu\varphi} & 0 \\ 0 & -e^{i\nu\varphi} \end{pmatrix}$ | $\begin{pmatrix} -e^{-i\nu\varphi} & 0 \\ 0 & -e^{i\nu\varphi} \end{pmatrix}$ |
| $\Gamma_{11}$ | $\begin{pmatrix} e^{i\nu\varphi} & 0 \\ 0 & e^{-i\nu\varphi} \end{pmatrix}$ | $\begin{pmatrix} 0 & e^{-i\nu\varphi} \\ e^{i\nu\varphi} & 0 \end{pmatrix}$ | $\begin{pmatrix} 0 & e^{-i\nu\varphi} \\ e^{i\nu\varphi} & 0 \end{pmatrix}$ | $\begin{pmatrix} 0 & -e^{-i\nu\varphi} \\ -e^{i\nu\varphi} & 0 \end{pmatrix}$ | $\begin{pmatrix} e^{i\nu\varphi} & 0 \\ 0 & e^{-i\nu\varphi} \end{pmatrix}$ | $\begin{pmatrix} e^{-i\nu\varphi} & 0 \\ 0 & e^{i\nu\varphi} \end{pmatrix}$ | $\begin{pmatrix} e^{-i\nu\varphi} & 0 \\ 0 & e^{i\nu\varphi} \end{pmatrix}$ | $\begin{pmatrix} -e^{-i\nu\varphi} & 0 \\ 0 & -e^{i\nu\varphi} \end{pmatrix}$ |
| $\Gamma_{12}$ | $\begin{pmatrix} e^{i\nu\varphi} & 0 \\ 0 & e^{-i\nu\varphi} \end{pmatrix}$ | $\begin{pmatrix} 0 & e^{-i\nu\varphi} \\ e^{i\nu\varphi} & 0 \end{pmatrix}$ | $\begin{pmatrix} 0 & -e^{-i\nu\varphi} \\ -e^{i\nu\varphi} & 0 \end{pmatrix}$ | $\begin{pmatrix} 0 & e^{-i\nu\varphi} \\ e^{i\nu\varphi} & 0 \end{pmatrix}$ | $\begin{pmatrix} e^{i\nu\varphi} & 0 \\ 0 & e^{-i\nu\varphi} \end{pmatrix}$ | $\begin{pmatrix} e^{-i\nu\varphi} & 0 \\ 0 & e^{i\nu\varphi} \end{pmatrix}$ | $\begin{pmatrix} -e^{-i\nu\varphi} & 0 \\ 0 & -e^{i\nu\varphi} \end{pmatrix}$ | $\begin{pmatrix} e^{-i\nu\varphi} & 0 \\ 0 & e^{i\nu\varphi} \end{pmatrix}$ |

| $b^\infty \times \bar{1}4$ | $\{R_\varphi, E\}$ | $\{2_\perp R_\varphi, C_{4z}\}$ | $\{\tau 2_\perp R_\varphi, E\}$ | $\{\tau R_\varphi, C_{4z}\}$ |
|---|---|---|---|---|
| $\Gamma_1$ | 1 | 1 | 1 | 1 |
| $\Gamma_2$ | 1 | $-1$ | 1 | $-1$ |
| $\Gamma_3$ | $\begin{pmatrix} e^{i\nu\varphi} & 0 \\ 0 & e^{-i\nu\varphi} \end{pmatrix}$ | $\begin{pmatrix} 0 & e^{-i\nu\varphi} \\ e^{i\nu\varphi} & 0 \end{pmatrix}$ | $\begin{pmatrix} e^{i\nu\varphi} & 0 \\ 0 & e^{-i\nu\varphi} \end{pmatrix}$ | $\begin{pmatrix} 0 & e^{-i\nu\varphi} \\ e^{i\nu\varphi} & 0 \end{pmatrix}$ |
| $\Gamma_4$ | $\begin{pmatrix} e^{i\nu\varphi} & 0 \\ 0 & e^{-i\nu\varphi} \end{pmatrix}$ | $\begin{pmatrix} 0 & -e^{-i\nu\varphi} \\ e^{i\nu\varphi} & 0 \end{pmatrix}$ | $\begin{pmatrix} e^{i\nu\varphi} & 0 \\ 0 & e^{-i\nu\varphi} \end{pmatrix}$ | $\begin{pmatrix} 0 & -e^{-i\nu\varphi} \\ e^{i\nu\varphi} & 0 \end{pmatrix}$ |
| $\Gamma_5$ | $\begin{pmatrix} 1 & 0 \\ 0 & 1 \end{pmatrix}$ | $\begin{pmatrix} -i & 0 \\ 0 & i \end{pmatrix}$ | $\begin{pmatrix} 0 & 1 \\ 1 & 0 \end{pmatrix}$ | $\begin{pmatrix} 0 & -i \\ i & 0 \end{pmatrix}$ |

| $b^\infty \times \bar{1}\bar{4}$ | $\{R_\varphi, E\}$ | $\{2_\perp R_\varphi, IC_{4z}^{-1}\}$ | $\{\tau 2_\perp R_\varphi, E\}$ | $\{\tau R_\varphi, IC_{4z}^{-1}\}$ |
|---|---|---|---|---|
| $\Gamma_1$ | 1 | 1 | 1 | 1 |
| $\Gamma_2$ | 1 | $-1$ | 1 | $-1$ |
| $\Gamma_3$ | $\begin{pmatrix} e^{i\nu\varphi} & 0 \\ 0 & e^{-i\nu\varphi} \end{pmatrix}$ | $\begin{pmatrix} 0 & e^{-i\nu\varphi} \\ e^{i\nu\varphi} & 0 \end{pmatrix}$ | $\begin{pmatrix} e^{i\nu\varphi} & 0 \\ 0 & e^{-i\nu\varphi} \end{pmatrix}$ | $\begin{pmatrix} 0 & e^{-i\nu\varphi} \\ e^{i\nu\varphi} & 0 \end{pmatrix}$ |
| $\Gamma_4$ | $\begin{pmatrix} e^{i\nu\varphi} & 0 \\ 0 & e^{-i\nu\varphi} \end{pmatrix}$ | $\begin{pmatrix} 0 & e^{-i\nu\varphi} \\ -e^{i\nu\varphi} & 0 \end{pmatrix}$ | $\begin{pmatrix} e^{i\nu\varphi} & 0 \\ 0 & e^{-i\nu\varphi} \end{pmatrix}$ | $\begin{pmatrix} 0 & e^{-i\nu\varphi} \\ -e^{i\nu\varphi} & 0 \end{pmatrix}$ |
| $\Gamma_5$ | $\begin{pmatrix} 1 & 0 \\ 0 & 1 \end{pmatrix}$ | $\begin{pmatrix} i & 0 \\ 0 & -i \end{pmatrix}$ | $\begin{pmatrix} 0 & 1 \\ 1 & 0 \end{pmatrix}$ | $\begin{pmatrix} 0 & i \\ -i & 0 \end{pmatrix}$ |

**Table 1.** $b^\infty \times \overline{1}_4/1_m$

| $b^\infty \times \overline{1}_4/1_m$ | $\{R_\psi, E\}$ | $\{R_\psi, IC_{2z}\}$ | $\{2_\perp R_\psi, C_{4z}\}$ | $\{\tau 2_\perp R_\psi, E\}$ | $\{\tau 2_\perp R_\psi, IC_{2z}\}$ | $\{\tau R_\psi, C_{4z}\}$ |
|---|---|---|---|---|---|---|
| $\Gamma_1$ | $1$ | $1$ | $1$ | $1$ | $1$ | $1$ |
| $\Gamma_2$ | $1$ | $1$ | $-1$ | $1$ | $1$ | $-1$ |
| $\Gamma_3$ | $1$ | $-1$ | $-1$ | $1$ | $-1$ | $1$ |
| $\Gamma_4$ | $1$ | $-1$ | $-1$ | $1$ | $-1$ | $-1$ |
| $\Gamma_5$ | $\begin{pmatrix} e^{i\nu\varphi} & 0 \\ 0 & e^{-i\nu\varphi} \end{pmatrix}$ | $\begin{pmatrix} e^{i\nu\varphi} & 0 \\ 0 & e^{-i\nu\varphi} \end{pmatrix}$ | $\begin{pmatrix} 0 & e^{-i\nu\varphi} \\ e^{i\nu\varphi} & 0 \end{pmatrix}$ | $\begin{pmatrix} e^{i\nu\varphi} & 0 \\ 0 & e^{-i\nu\varphi} \end{pmatrix}$ | $\begin{pmatrix} e^{i\nu\varphi} & 0 \\ 0 & e^{-i\nu\varphi} \end{pmatrix}$ | $\begin{pmatrix} 0 & e^{-i\nu\varphi} \\ e^{i\nu\varphi} & 0 \end{pmatrix}$ |
| $\Gamma_6$ | $\begin{pmatrix} e^{i\nu\varphi} & 0 \\ 0 & e^{-i\nu\varphi} \end{pmatrix}$ | $\begin{pmatrix} -e^{i\nu\varphi} & 0 \\ 0 & -e^{-i\nu\varphi} \end{pmatrix}$ | $\begin{pmatrix} 0 & e^{-i\nu\varphi} \\ e^{i\nu\varphi} & 0 \end{pmatrix}$ | $\begin{pmatrix} e^{i\nu\varphi} & 0 \\ 0 & e^{-i\nu\varphi} \end{pmatrix}$ | $\begin{pmatrix} -e^{i\nu\varphi} & 0 \\ 0 & -e^{-i\nu\varphi} \end{pmatrix}$ | $\begin{pmatrix} 0 & e^{-i\nu\varphi} \\ e^{i\nu\varphi} & 0 \end{pmatrix}$ |
| $\Gamma_7$ | $\begin{pmatrix} e^{i\nu\varphi} & 0 \\ 0 & e^{-i\nu\varphi} \end{pmatrix}$ | $\begin{pmatrix} e^{i\nu\varphi} & 0 \\ 0 & e^{-i\nu\varphi} \end{pmatrix}$ | $\begin{pmatrix} 0 & -e^{-i\nu\varphi} \\ e^{i\nu\varphi} & 0 \end{pmatrix}$ | $\begin{pmatrix} e^{i\nu\varphi} & 0 \\ 0 & e^{-i\nu\varphi} \end{pmatrix}$ | $\begin{pmatrix} -e^{i\nu\varphi} & 0 \\ 0 & e^{-i\nu\varphi} \end{pmatrix}$ | $\begin{pmatrix} 0 & -e^{-i\nu\varphi} \\ e^{i\nu\varphi} & 0 \end{pmatrix}$ |
| $\Gamma_8$ | $\begin{pmatrix} 1 & 0 \\ 0 & 1 \end{pmatrix}$ | $\begin{pmatrix} 1 & 0 \\ 0 & 1 \end{pmatrix}$ | $\begin{pmatrix} -i & 0 \\ 0 & i \end{pmatrix}$ | $\begin{pmatrix} 0 & 1 \\ 1 & 0 \end{pmatrix}$ | $\begin{pmatrix} 0 & 1 \\ 1 & 0 \end{pmatrix}$ | $\begin{pmatrix} 0 & -i \\ i & 0 \end{pmatrix}$ |
| $\Gamma_9$ | $\begin{pmatrix} 1 & 0 \\ 0 & 1 \end{pmatrix}$ | $\begin{pmatrix} 1 & 0 \\ 0 & 1 \end{pmatrix}$ | $\begin{pmatrix} -i & 0 \\ 0 & i \end{pmatrix}$ | $\begin{pmatrix} 0 & 1 \\ 1 & 0 \end{pmatrix}$ | $\begin{pmatrix} 0 & -1 \\ -1 & 0 \end{pmatrix}$ | $\begin{pmatrix} 0 & -i \\ i & 0 \end{pmatrix}$ |
| $\Gamma_{10}$ | | | | | | |

**Table 2.** $b^\infty \times \overline{1}_4/\overline{1}_m$

| $b^\infty \times \overline{1}_4/\overline{1}_m$ | $\{R_\psi, E\}$ | $\{2_\perp R_\psi, C_{4z}\}$ | $\{2_\perp R_\psi, IC_{2z}\}$ | $\{\tau 2_\perp R_\psi, E\}$ | $\{\tau R_\psi, C_{4z}\}$ | $\{\tau R_\psi, IC_{2z}\}$ |
|---|---|---|---|---|---|---|
| $\Gamma_1$ | $1$ | $1$ | $1$ | $1$ | $1$ | $1$ |
| $\Gamma_2$ | $1$ | $-1$ | $-1$ | $1$ | $-1$ | $-1$ |
| $\Gamma_3$ | $1$ | $1$ | $-1$ | $1$ | $1$ | $-1$ |
| $\Gamma_4$ | $1$ | $-1$ | $1$ | $1$ | $-1$ | $1$ |
| $\Gamma_5$ | $\begin{pmatrix} e^{i\nu\varphi} & 0 \\ 0 & e^{-i\nu\varphi} \end{pmatrix}$ | $\begin{pmatrix} 0 & e^{-i\nu\varphi} \\ e^{i\nu\varphi} & 0 \end{pmatrix}$ | $\begin{pmatrix} 0 & e^{-i\nu\varphi} \\ e^{i\nu\varphi} & 0 \end{pmatrix}$ | $\begin{pmatrix} e^{i\nu\varphi} & 0 \\ 0 & e^{-i\nu\varphi} \end{pmatrix}$ | $\begin{pmatrix} 0 & e^{-i\nu\varphi} \\ e^{i\nu\varphi} & 0 \end{pmatrix}$ | $\begin{pmatrix} 0 & e^{-i\nu\varphi} \\ e^{i\nu\varphi} & 0 \end{pmatrix}$ |
| $\Gamma_6$ | $\begin{pmatrix} e^{i\nu\varphi} & 0 \\ 0 & e^{-i\nu\varphi} \end{pmatrix}$ | $\begin{pmatrix} 0 & -e^{-i\nu\varphi} \\ e^{i\nu\varphi} & 0 \end{pmatrix}$ | $\begin{pmatrix} 0 & -e^{-i\nu\varphi} \\ -e^{i\nu\varphi} & 0 \end{pmatrix}$ | $\begin{pmatrix} e^{i\nu\varphi} & 0 \\ 0 & e^{-i\nu\varphi} \end{pmatrix}$ | $\begin{pmatrix} 0 & -e^{-i\nu\varphi} \\ -e^{i\nu\varphi} & 0 \end{pmatrix}$ | $\begin{pmatrix} 0 & -e^{-i\nu\varphi} \\ -e^{i\nu\varphi} & 0 \end{pmatrix}$ |
| $\Gamma_7$ | $\begin{pmatrix} 1 & 0 \\ 0 & 1 \end{pmatrix}$ | $\begin{pmatrix} -i & 0 \\ 0 & i \end{pmatrix}$ | $\begin{pmatrix} -1 & 0 \\ 0 & -1 \end{pmatrix}$ | $\begin{pmatrix} 0 & 1 \\ 1 & 0 \end{pmatrix}$ | $\begin{pmatrix} 0 & -i \\ i & 0 \end{pmatrix}$ | $\begin{pmatrix} 0 & -1 \\ -1 & 0 \end{pmatrix}$ |
| $\Gamma_8$ | $\begin{pmatrix} 1 & 0 \\ 0 & 1 \end{pmatrix}$ | $\begin{pmatrix} i & 0 \\ 0 & -i \end{pmatrix}$ | $\begin{pmatrix} 1 & 0 \\ 0 & 1 \end{pmatrix}$ | $\begin{pmatrix} 0 & 1 \\ 1 & 0 \end{pmatrix}$ | $\begin{pmatrix} 0 & i \\ -i & 0 \end{pmatrix}$ | $\begin{pmatrix} 1 & 0 \\ 0 & 1 \end{pmatrix}$ |
| $\Gamma_9$ | | | | | | |

**Table (left):**

| $\mathfrak{b}^{\infty}\times{}^{1}4_{2}/{}^{1}\bar{1}_{m}$ | $\{R_{\psi},E\}$ | $\{R_{\psi},C_{4z}\}$ | $\{2_{\perp}R_{\psi},IC_{2z}\}$ | $\{\tau2_{\perp}R_{\psi},E\}$ | $\{\tau2_{\perp}R_{\psi},C_{4z}\}$ | $\{\tau R_{\psi},IC_{2z}\}$ |
|---|---|---|---|---|---|---|
| $\Gamma_1$ | $1$ | $1$ | $1$ | $1$ | $1$ | $1$ |
| $\Gamma_2$ | $1$ | $1$ | $-1$ | $1$ | $1$ | $-1$ |
| $\Gamma_3$ | $1$ | $-1$ | $1$ | $1$ | $-1$ | $1$ |
| $\Gamma_4$ | $1$ | $-1$ | $-1$ | $1$ | $-1$ | $-1$ |
| $\Gamma_5$ | $\begin{pmatrix} e^{i\nu\varphi} & 0 \\ 0 & e^{-i\nu\varphi}\end{pmatrix}$ | $\begin{pmatrix} e^{i\nu\varphi} & 0 \\ 0 & e^{-i\nu\varphi}\end{pmatrix}$ | $\begin{pmatrix} 0 & e^{-i\nu\varphi} \\ e^{i\nu\varphi} & 0\end{pmatrix}$ | $\begin{pmatrix} e^{i\nu\varphi} & 0 \\ 0 & e^{-i\nu\varphi}\end{pmatrix}$ | $\begin{pmatrix} e^{i\nu\varphi} & 0 \\ 0 & e^{-i\nu\varphi}\end{pmatrix}$ | $\begin{pmatrix} 0 & e^{-i\nu\varphi} \\ e^{i\nu\varphi} & 0\end{pmatrix}$ |
| $\Gamma_6$ | $\begin{pmatrix} e^{i\nu\varphi} & 0 \\ 0 & e^{-i\nu\varphi}\end{pmatrix}$ | $\begin{pmatrix} -e^{i\nu\varphi} & 0 \\ 0 & -e^{-i\nu\varphi}\end{pmatrix}$ | $\begin{pmatrix} 0 & e^{-i\nu\varphi} \\ e^{i\nu\varphi} & 0\end{pmatrix}$ | $\begin{pmatrix} e^{i\nu\varphi} & 0 \\ 0 & e^{-i\nu\varphi}\end{pmatrix}$ | $\begin{pmatrix} -e^{i\nu\varphi} & 0 \\ 0 & -e^{-i\nu\varphi}\end{pmatrix}$ | $\begin{pmatrix} 0 & e^{-i\nu\varphi} \\ e^{i\nu\varphi} & 0\end{pmatrix}$ |
| $\Gamma_7$ | $\begin{pmatrix} 1 & 0 \\ 0 & 1\end{pmatrix}$ | $\begin{pmatrix} -i & 0 \\ 0 & i\end{pmatrix}$ | $\begin{pmatrix} 1 & 0 \\ 0 & 1\end{pmatrix}$ | $\begin{pmatrix} 0 & 1 \\ 1 & 0\end{pmatrix}$ | $\begin{pmatrix} 0 & -i \\ i & 0\end{pmatrix}$ | $\begin{pmatrix} 0 & 1 \\ 1 & 0\end{pmatrix}$ |
| $\Gamma_8$ | $\begin{pmatrix} 1 & 0 \\ 0 & 1\end{pmatrix}$ | $\begin{pmatrix} -i & 0 \\ 0 & i\end{pmatrix}$ | $\begin{pmatrix} -1 & 0 \\ 0 & -1\end{pmatrix}$ | $\begin{pmatrix} 0 & 1 \\ 1 & 0\end{pmatrix}$ | $\begin{pmatrix} 0 & -i \\ i & 0\end{pmatrix}$ | $\begin{pmatrix} 0 & -1 \\ -1 & 0\end{pmatrix}$ |
| $\Gamma_9$ | $\begin{pmatrix} e^{i\nu\varphi} & 0 & 0 & 0 \\ 0 & e^{-i\nu\varphi} & 0 & 0 \\ 0 & 0 & e^{i\nu\varphi} & 0 \\ 0 & 0 & 0 & e^{-i\nu\varphi}\end{pmatrix}$ | $\begin{pmatrix} -ie^{i\nu\varphi} & 0 & 0 & 0 \\ 0 & -ie^{-i\nu\varphi} & 0 & 0 \\ 0 & 0 & ie^{i\nu\varphi} & 0 \\ 0 & 0 & 0 & ie^{-i\nu\varphi}\end{pmatrix}$ | $\begin{pmatrix} 0 & e^{-i\nu\varphi} & 0 & 0 \\ e^{i\nu\varphi} & 0 & 0 & 0 \\ 0 & 0 & 0 & e^{-i\nu\varphi} \\ 0 & 0 & e^{i\nu\varphi} & 0\end{pmatrix}$ | $\begin{pmatrix} 0 & 0 & e^{i\nu\varphi} & 0 \\ 0 & 0 & 0 & e^{-i\nu\varphi} \\ e^{i\nu\varphi} & 0 & 0 & 0 \\ 0 & e^{-i\nu\varphi} & 0 & 0\end{pmatrix}$ | $\begin{pmatrix} 0 & 0 & -ie^{i\nu\varphi} & 0 \\ 0 & 0 & 0 & -ie^{-i\nu\varphi} \\ ie^{i\nu\varphi} & 0 & 0 & 0 \\ 0 & ie^{-i\nu\varphi} & 0 & 0\end{pmatrix}$ | $\begin{pmatrix} 0 & 0 & 0 & e^{-i\nu\varphi} \\ 0 & 0 & e^{i\nu\varphi} & 0 \\ 0 & e^{-i\nu\varphi} & 0 & 0 \\ e^{i\nu\varphi} & 0 & 0 & 0\end{pmatrix}$ |

**Table (right):**

| $\mathfrak{b}^{\infty}\times{}^{1}4\,{}^{1}2\,{}^{1}2$ | $\{R_{\psi},E\}$ | $\{R_{\psi},C_{4z}\}$ | $\{2_{\perp}R_{\psi},C_{2x}\}$ | $\{\tau2_{\perp}R_{\psi},E\}$ | $\{\tau2_{\perp}R_{\psi},C_{4z}\}$ | $\{\tau R_{\psi},C_{2x}\}$ |
|---|---|---|---|---|---|---|
| $\Gamma_1$ | $1$ | $1$ | $1$ | $1$ | $1$ | $1$ |
| $\Gamma_2$ | $1$ | $1$ | $-1$ | $1$ | $1$ | $-1$ |
| $\Gamma_3$ | $1$ | $-1$ | $1$ | $1$ | $-1$ | $1$ |
| $\Gamma_4$ | $1$ | $-1$ | $-1$ | $1$ | $-1$ | $-1$ |
| $\Gamma_5$ | $\begin{pmatrix} 1 & 0 \\ 0 & 1\end{pmatrix}$ | $\begin{pmatrix} -i & 0 \\ 0 & i\end{pmatrix}$ | $\begin{pmatrix} 0 & 1 \\ 1 & 0\end{pmatrix}$ | $\begin{pmatrix} 0 & 1 \\ 1 & 0\end{pmatrix}$ | $\begin{pmatrix} 0 & -i \\ i & 0\end{pmatrix}$ | $\begin{pmatrix} 1 & 0 \\ 0 & 1\end{pmatrix}$ |
| $\Gamma_6$ | $\begin{pmatrix} e^{i\nu\varphi} & 0 \\ 0 & e^{-i\nu\varphi}\end{pmatrix}$ | $\begin{pmatrix} e^{i\nu\varphi} & 0 \\ 0 & e^{-i\nu\varphi}\end{pmatrix}$ | $\begin{pmatrix} 0 & e^{-i\nu\varphi} \\ e^{i\nu\varphi} & 0\end{pmatrix}$ | $\begin{pmatrix} e^{i\nu\varphi} & 0 \\ 0 & e^{-i\nu\varphi}\end{pmatrix}$ | $\begin{pmatrix} e^{i\nu\varphi} & 0 \\ 0 & e^{-i\nu\varphi}\end{pmatrix}$ | $\begin{pmatrix} 0 & e^{-i\nu\varphi} \\ e^{i\nu\varphi} & 0\end{pmatrix}$ |
| $\Gamma_7$ | $\begin{pmatrix} e^{i\nu\varphi} & 0 \\ 0 & e^{-i\nu\varphi}\end{pmatrix}$ | $\begin{pmatrix} -e^{i\nu\varphi} & 0 \\ 0 & -e^{-i\nu\varphi}\end{pmatrix}$ | $\begin{pmatrix} 0 & e^{-i\nu\varphi} \\ e^{i\nu\varphi} & 0\end{pmatrix}$ | $\begin{pmatrix} e^{i\nu\varphi} & 0 \\ 0 & e^{-i\nu\varphi}\end{pmatrix}$ | $\begin{pmatrix} -e^{i\nu\varphi} & 0 \\ 0 & -e^{-i\nu\varphi}\end{pmatrix}$ | $\begin{pmatrix} 0 & e^{-i\nu\varphi} \\ e^{i\nu\varphi} & 0\end{pmatrix}$ |
| $\Gamma_8$ | $\begin{pmatrix} e^{i\nu\varphi} & 0 & 0 & 0 \\ 0 & e^{-i\nu\varphi} & 0 & 0 \\ 0 & 0 & e^{i\nu\varphi} & 0 \\ 0 & 0 & 0 & e^{-i\nu\varphi}\end{pmatrix}$ | $\begin{pmatrix} -ie^{i\nu\varphi} & 0 & 0 & 0 \\ 0 & ie^{-i\nu\varphi} & 0 & 0 \\ 0 & 0 & ie^{i\nu\varphi} & 0 \\ 0 & 0 & 0 & -ie^{-i\nu\varphi}\end{pmatrix}$ | $\begin{pmatrix} 0 & e^{-i\nu\varphi} & 0 & 0 \\ e^{i\nu\varphi} & 0 & 0 & 0 \\ 0 & 0 & 0 & e^{-i\nu\varphi} \\ 0 & 0 & e^{i\nu\varphi} & 0\end{pmatrix}$ | $\begin{pmatrix} 0 & 0 & e^{i\nu\varphi} & 0 \\ 0 & 0 & 0 & e^{-i\nu\varphi} \\ e^{i\nu\varphi} & 0 & 0 & 0 \\ 0 & e^{-i\nu\varphi} & 0 & 0\end{pmatrix}$ | $\begin{pmatrix} 0 & 0 & -ie^{i\nu\varphi} & 0 \\ 0 & 0 & 0 & ie^{-i\nu\varphi} \\ ie^{i\nu\varphi} & 0 & 0 & 0 \\ 0 & -ie^{-i\nu\varphi} & 0 & 0\end{pmatrix}$ | $\begin{pmatrix} 0 & 0 & 0 & e^{-i\nu\varphi} \\ 0 & 0 & e^{i\nu\varphi} & 0 \\ 0 & e^{-i\nu\varphi} & 0 & 0 \\ e^{i\nu\varphi} & 0 & 0 & 0\end{pmatrix}$ |

| $b^{\infty} \times {}^{\bar{1}}4\,{}^{1}2\,{}^{\bar{1}}2$ | $\{R_{\varphi}, E\}$ | $\{R_{\varphi}, C_{2x}\}$ | $\{2_{\perp}R_{\varphi}, C_{4z}\}$ | $\{\tau 2_{\perp}R_{\varphi}, E\}$ | $\{\tau 2_{\perp}R_{\varphi}, C_{2x}\}$ | $\{\tau R_{\varphi}, C_{4z}\}$ |
|---|---|---|---|---|---|---|
| $\Gamma_1$ | $1$ | $1$ | $1$ | $1$ | $1$ | $1$ |
| $\Gamma_2$ | $1$ | $1$ | $-1$ | $1$ | $1$ | $-1$ |
| $\Gamma_3$ | $1$ | $-1$ | $1$ | $1$ | $-1$ | $1$ |
| $\Gamma_4$ | $1$ | $-1$ | $-1$ | $1$ | $-1$ | $-1$ |
| $\Gamma_5$ | $\begin{pmatrix}1&0\\0&1\end{pmatrix}$ | $\begin{pmatrix}1&0\\0&-1\end{pmatrix}$ | $\begin{pmatrix}0&-1\\1&0\end{pmatrix}$ | $\begin{pmatrix}1&0\\0&1\end{pmatrix}$ | $\begin{pmatrix}1&0\\0&-1\end{pmatrix}$ | $\begin{pmatrix}0&-1\\1&0\end{pmatrix}$ |
| $\Gamma_6$ | $\begin{pmatrix}e^{i\nu\varphi}&0\\0&e^{-i\nu\varphi}\end{pmatrix}$ | $\begin{pmatrix}e^{i\nu\varphi}&0\\0&e^{-i\nu\varphi}\end{pmatrix}$ | $\begin{pmatrix}0&e^{-i\nu\varphi}\\e^{i\nu\varphi}&0\end{pmatrix}$ | $\begin{pmatrix}e^{i\nu\varphi}&0\\0&e^{-i\nu\varphi}\end{pmatrix}$ | $\begin{pmatrix}e^{i\nu\varphi}&0\\0&e^{-i\nu\varphi}\end{pmatrix}$ | $\begin{pmatrix}0&e^{-i\nu\varphi}\\e^{i\nu\varphi}&0\end{pmatrix}$ |
| $\Gamma_7$ | $\begin{pmatrix}e^{i\nu\varphi}&0\\0&e^{-i\nu\varphi}\end{pmatrix}$ | $\begin{pmatrix}-e^{i\nu\varphi}&0\\0&-e^{-i\nu\varphi}\end{pmatrix}$ | $\begin{pmatrix}0&e^{-i\nu\varphi}\\e^{i\nu\varphi}&0\end{pmatrix}$ | $\begin{pmatrix}e^{i\nu\varphi}&0\\0&e^{-i\nu\varphi}\end{pmatrix}$ | $\begin{pmatrix}-e^{i\nu\varphi}&0\\0&-e^{-i\nu\varphi}\end{pmatrix}$ | $\begin{pmatrix}0&e^{-i\nu\varphi}\\e^{i\nu\varphi}&0\end{pmatrix}$ |
| $\Gamma_8$ | $\begin{pmatrix}e^{i\mu\varphi}&0\\0&e^{-i\mu\varphi}\end{pmatrix}$ | $\begin{pmatrix}e^{i\mu\varphi}&0\\0&-e^{-i\mu\varphi}\end{pmatrix}$ | $\begin{pmatrix}0&-e^{-i\mu\varphi}\\e^{i\mu\varphi}&0\end{pmatrix}$ | $\begin{pmatrix}e^{i\mu\varphi}&0\\0&e^{-i\mu\varphi}\end{pmatrix}$ | $\begin{pmatrix}e^{i\mu\varphi}&0\\0&-e^{-i\mu\varphi}\end{pmatrix}$ | $\begin{pmatrix}0&-e^{-i\mu\varphi}\\e^{i\mu\varphi}&0\end{pmatrix}$ |

| $b^{\infty} \times {}^{\bar{1}}4\,{}^{1}m\,{}^{\bar{1}}m$ | $\{R_{\varphi}, E\}$ | $\{R_{\varphi}, IC_{2y}\}$ | $\{2_{\perp}R_{\varphi}, C_{4z}\}$ | $\{\tau 2_{\perp}R_{\varphi}, E\}$ | $\{\tau 2_{\perp}R_{\varphi}, IC_{2y}\}$ | $\{\tau R_{\varphi}, C_{4z}\}$ |
|---|---|---|---|---|---|---|
| $\Gamma_1$ | $1$ | $1$ | $1$ | $1$ | $1$ | $1$ |
| $\Gamma_2$ | $1$ | $1$ | $-1$ | $1$ | $1$ | $-1$ |
| $\Gamma_3$ | $1$ | $-1$ | $1$ | $1$ | $-1$ | $1$ |
| $\Gamma_4$ | $1$ | $-1$ | $-1$ | $1$ | $-1$ | $-1$ |
| $\Gamma_5$ | $\begin{pmatrix}1&0\\0&1\end{pmatrix}$ | $\begin{pmatrix}-1&0\\0&1\end{pmatrix}$ | $\begin{pmatrix}0&-1\\1&0\end{pmatrix}$ | $\begin{pmatrix}1&0\\0&1\end{pmatrix}$ | $\begin{pmatrix}-1&0\\0&1\end{pmatrix}$ | $\begin{pmatrix}0&-1\\1&0\end{pmatrix}$ |
| $\Gamma_6$ | $\begin{pmatrix}e^{i\nu\varphi}&0\\0&e^{-i\nu\varphi}\end{pmatrix}$ | $\begin{pmatrix}e^{i\nu\varphi}&0\\0&e^{-i\nu\varphi}\end{pmatrix}$ | $\begin{pmatrix}0&e^{-i\nu\varphi}\\e^{i\nu\varphi}&0\end{pmatrix}$ | $\begin{pmatrix}e^{i\nu\varphi}&0\\0&e^{-i\nu\varphi}\end{pmatrix}$ | $\begin{pmatrix}e^{i\nu\varphi}&0\\0&e^{-i\nu\varphi}\end{pmatrix}$ | $\begin{pmatrix}0&e^{-i\nu\varphi}\\e^{i\nu\varphi}&0\end{pmatrix}$ |
| $\Gamma_7$ | $\begin{pmatrix}e^{i\nu\varphi}&0\\0&e^{-i\nu\varphi}\end{pmatrix}$ | $\begin{pmatrix}-e^{i\nu\varphi}&0\\0&-e^{-i\nu\varphi}\end{pmatrix}$ | $\begin{pmatrix}0&e^{-i\nu\varphi}\\e^{i\nu\varphi}&0\end{pmatrix}$ | $\begin{pmatrix}e^{i\nu\varphi}&0\\0&e^{-i\nu\varphi}\end{pmatrix}$ | $\begin{pmatrix}-e^{i\nu\varphi}&0\\0&-e^{-i\nu\varphi}\end{pmatrix}$ | $\begin{pmatrix}0&e^{-i\nu\varphi}\\e^{i\nu\varphi}&0\end{pmatrix}$ |
| $\Gamma_8$ | $\begin{pmatrix}e^{i\mu\varphi}&0\\0&e^{-i\mu\varphi}\end{pmatrix}$ | $\begin{pmatrix}-e^{i\mu\varphi}&0\\0&e^{-i\mu\varphi}\end{pmatrix}$ | $\begin{pmatrix}0&-e^{-i\mu\varphi}\\e^{i\mu\varphi}&0\end{pmatrix}$ | $\begin{pmatrix}e^{i\mu\varphi}&0\\0&e^{-i\mu\varphi}\end{pmatrix}$ | $\begin{pmatrix}-e^{i\mu\varphi}&0\\0&e^{-i\mu\varphi}\end{pmatrix}$ | $\begin{pmatrix}0&-e^{-i\mu\varphi}\\e^{i\mu\varphi}&0\end{pmatrix}$ |

| $b^{\infty} \times {}^{\bar{1}}\bar{4}\,{}^{\bar{1}}2\,{}^{1}m$ | $\{R_{\varphi}, E\}$ | $\{R_{\varphi}, IC_{2y}\}$ | $\{2_{\perp}R_{\varphi}, IC_{4z}^{-1}\}$ | $\{\tau 2_{\perp}R_{\varphi}, E\}$ | $\{\tau 2_{\perp}R_{\varphi}, IC_{2y}\}$ | $\{\tau R_{\varphi}, IC_{4z}^{-1}\}$ |
|---|---|---|---|---|---|---|
| $\Gamma_1$ | $1$ | $1$ | $1$ | $1$ | $1$ | $1$ |
| $\Gamma_2$ | $1$ | $1$ | $-1$ | $1$ | $1$ | $-1$ |
| $\Gamma_3$ | $1$ | $-1$ | $1$ | $1$ | $-1$ | $1$ |
| $\Gamma_4$ | $1$ | $-1$ | $-1$ | $1$ | $-1$ | $-1$ |
| $\Gamma_5$ | $\begin{pmatrix}1&0\\0&1\end{pmatrix}$ | $\begin{pmatrix}-1&0\\0&1\end{pmatrix}$ | $\begin{pmatrix}0&1\\-1&0\end{pmatrix}$ | $\begin{pmatrix}1&0\\0&1\end{pmatrix}$ | $\begin{pmatrix}-1&0\\0&1\end{pmatrix}$ | $\begin{pmatrix}0&1\\-1&0\end{pmatrix}$ |
| $\Gamma_6$ | $\begin{pmatrix}e^{i\nu\varphi}&0\\0&e^{-i\nu\varphi}\end{pmatrix}$ | $\begin{pmatrix}e^{i\nu\varphi}&0\\0&e^{-i\nu\varphi}\end{pmatrix}$ | $\begin{pmatrix}0&e^{-i\nu\varphi}\\e^{i\nu\varphi}&0\end{pmatrix}$ | $\begin{pmatrix}e^{i\nu\varphi}&0\\0&e^{-i\nu\varphi}\end{pmatrix}$ | $\begin{pmatrix}e^{i\nu\varphi}&0\\0&e^{-i\nu\varphi}\end{pmatrix}$ | $\begin{pmatrix}0&e^{-i\nu\varphi}\\e^{i\nu\varphi}&0\end{pmatrix}$ |
| $\Gamma_7$ | $\begin{pmatrix}e^{i\nu\varphi}&0\\0&e^{-i\nu\varphi}\end{pmatrix}$ | $\begin{pmatrix}-e^{i\nu\varphi}&0\\0&-e^{-i\nu\varphi}\end{pmatrix}$ | $\begin{pmatrix}0&e^{-i\nu\varphi}\\e^{i\nu\varphi}&0\end{pmatrix}$ | $\begin{pmatrix}e^{i\nu\varphi}&0\\0&e^{-i\nu\varphi}\end{pmatrix}$ | $\begin{pmatrix}-e^{i\nu\varphi}&0\\0&-e^{-i\nu\varphi}\end{pmatrix}$ | $\begin{pmatrix}0&e^{-i\nu\varphi}\\e^{i\nu\varphi}&0\end{pmatrix}$ |
| $\Gamma_8$ | $\begin{pmatrix}e^{i\mu\varphi}&0\\0&e^{-i\mu\varphi}\end{pmatrix}$ | $\begin{pmatrix}-e^{i\mu\varphi}&0\\0&e^{-i\mu\varphi}\end{pmatrix}$ | $\begin{pmatrix}0&e^{-i\mu\varphi}\\-e^{i\mu\varphi}&0\end{pmatrix}$ | $\begin{pmatrix}e^{i\mu\varphi}&0\\0&e^{-i\mu\varphi}\end{pmatrix}$ | $\begin{pmatrix}-e^{i\mu\varphi}&0\\0&e^{-i\mu\varphi}\end{pmatrix}$ | $\begin{pmatrix}0&e^{-i\mu\varphi}\\-e^{i\mu\varphi}&0\end{pmatrix}$ |

| $b^{\infty} \times {}^{\bar{1}}\bar{4}\,{}^{1}2\,{}^{\bar{1}}m$ | $\{R_{\varphi}, E\}$ | $\{R_{\varphi}, C_{2x}\}$ | $\{2_{\perp}R_{\varphi}, IC_{4z}^{-1}\}$ | $\{\tau 2_{\perp}R_{\varphi}, E\}$ | $\{\tau 2_{\perp}R_{\varphi}, C_{2x}\}$ | $\{\tau R_{\varphi}, IC_{4z}^{-1}\}$ |
|---|---|---|---|---|---|---|
| $\Gamma_1$ | $1$ | $1$ | $1$ | $1$ | $1$ | $1$ |
| $\Gamma_2$ | $1$ | $1$ | $-1$ | $1$ | $1$ | $-1$ |
| $\Gamma_3$ | $1$ | $-1$ | $1$ | $1$ | $-1$ | $1$ |
| $\Gamma_4$ | $1$ | $-1$ | $-1$ | $1$ | $-1$ | $-1$ |
| $\Gamma_5$ | $\begin{pmatrix}1&0\\0&1\end{pmatrix}$ | $\begin{pmatrix}1&0\\0&-1\end{pmatrix}$ | $\begin{pmatrix}0&1\\-1&0\end{pmatrix}$ | $\begin{pmatrix}1&0\\0&1\end{pmatrix}$ | $\begin{pmatrix}1&0\\0&-1\end{pmatrix}$ | $\begin{pmatrix}0&1\\-1&0\end{pmatrix}$ |
| $\Gamma_6$ | $\begin{pmatrix}e^{i\nu\varphi}&0\\0&e^{-i\nu\varphi}\end{pmatrix}$ | $\begin{pmatrix}e^{i\nu\varphi}&0\\0&e^{-i\nu\varphi}\end{pmatrix}$ | $\begin{pmatrix}0&e^{-i\nu\varphi}\\e^{i\nu\varphi}&0\end{pmatrix}$ | $\begin{pmatrix}e^{i\nu\varphi}&0\\0&e^{-i\nu\varphi}\end{pmatrix}$ | $\begin{pmatrix}e^{i\nu\varphi}&0\\0&e^{-i\nu\varphi}\end{pmatrix}$ | $\begin{pmatrix}0&e^{-i\nu\varphi}\\e^{i\nu\varphi}&0\end{pmatrix}$ |
| $\Gamma_7$ | $\begin{pmatrix}e^{i\nu\varphi}&0\\0&e^{-i\nu\varphi}\end{pmatrix}$ | $\begin{pmatrix}-e^{i\nu\varphi}&0\\0&-e^{-i\nu\varphi}\end{pmatrix}$ | $\begin{pmatrix}0&e^{-i\nu\varphi}\\e^{i\nu\varphi}&0\end{pmatrix}$ | $\begin{pmatrix}e^{i\nu\varphi}&0\\0&e^{-i\nu\varphi}\end{pmatrix}$ | $\begin{pmatrix}-e^{i\nu\varphi}&0\\0&-e^{-i\nu\varphi}\end{pmatrix}$ | $\begin{pmatrix}0&e^{-i\nu\varphi}\\e^{i\nu\varphi}&0\end{pmatrix}$ |
| $\Gamma_8$ | $\begin{pmatrix}e^{i\mu\varphi}&0\\0&e^{-i\mu\varphi}\end{pmatrix}$ | $\begin{pmatrix}e^{i\mu\varphi}&0\\0&-e^{-i\mu\varphi}\end{pmatrix}$ | $\begin{pmatrix}0&e^{-i\mu\varphi}\\-e^{i\mu\varphi}&0\end{pmatrix}$ | $\begin{pmatrix}e^{i\mu\varphi}&0\\0&e^{-i\mu\varphi}\end{pmatrix}$ | $\begin{pmatrix}e^{i\mu\varphi}&0\\0&-e^{-i\mu\varphi}\end{pmatrix}$ | $\begin{pmatrix}0&e^{-i\mu\varphi}\\-e^{i\mu\varphi}&0\end{pmatrix}$ |

**Table 1**

| $b^\infty \times {}^1_4 \bar{1}_m \bar{1}_m$ | $\{R_\psi, E\}$ | $\{R_\psi, C_{4z}\}$ | $\{2_\perp R_\psi, IC_{2y}\}$ | $\{\tau 2_\perp R_\psi, E\}$ | $\{\tau 2_\perp R_\psi, C_{4z}\}$ | $\{\tau R_\psi, IC_{2y}\}$ |
|---|---|---|---|---|---|---|
| $\Gamma_1$ | $1$ | $1$ | $1$ | $1$ | $1$ | $1$ |
| $\Gamma_2$ | $1$ | $1$ | $-1$ | $1$ | $1$ | $-1$ |
| $\Gamma_3$ | $1$ | $-1$ | $1$ | $1$ | $-1$ | $1$ |
| $\Gamma_4$ | $1$ | $-1$ | $-1$ | $1$ | $-1$ | $-1$ |
| $\Gamma_5$ | $\begin{pmatrix}1&0\\0&1\end{pmatrix}$ | $\begin{pmatrix}-i&0\\0&i\end{pmatrix}$ | $\begin{pmatrix}0&-1\\-1&0\end{pmatrix}$ | $\begin{pmatrix}0&1\\1&0\end{pmatrix}$ | $\begin{pmatrix}0&-i\\i&0\end{pmatrix}$ | $\begin{pmatrix}-1&0\\0&-1\end{pmatrix}$ |
| $\Gamma_6$ | $\begin{pmatrix}e^{i\mu\varphi}&0\\0&e^{-i\mu\varphi}\end{pmatrix}$ | $\begin{pmatrix}e^{i\mu\varphi}&0\\0&e^{-i\mu\varphi}\end{pmatrix}$ | $\begin{pmatrix}0&e^{-i\mu\varphi}\\e^{i\mu\varphi}&0\end{pmatrix}$ | $\begin{pmatrix}e^{i\mu\varphi}&0\\0&e^{-i\mu\varphi}\end{pmatrix}$ | $\begin{pmatrix}e^{i\mu\varphi}&0\\0&e^{-i\mu\varphi}\end{pmatrix}$ | $\begin{pmatrix}0&e^{-i\mu\varphi}\\e^{i\mu\varphi}&0\end{pmatrix}$ |
| $\Gamma_7$ | $\begin{pmatrix}e^{i\mu\varphi}&0\\0&e^{-i\mu\varphi}\end{pmatrix}$ | $\begin{pmatrix}-e^{i\mu\varphi}&0\\0&-e^{-i\mu\varphi}\end{pmatrix}$ | $\begin{pmatrix}0&e^{-i\mu\varphi}\\e^{i\mu\varphi}&0\end{pmatrix}$ | $\begin{pmatrix}e^{i\mu\varphi}&0\\0&e^{-i\mu\varphi}\end{pmatrix}$ | $\begin{pmatrix}-e^{i\mu\varphi}&0\\0&-e^{-i\mu\varphi}\end{pmatrix}$ | $\begin{pmatrix}0&e^{-i\mu\varphi}\\e^{i\mu\varphi}&0\end{pmatrix}$ |
| $\Gamma_8$ | $\begin{pmatrix}e^{i\mu\varphi}&0&0&0\\0&e^{-i\mu\varphi}&0&0\\0&0&e^{i\mu\varphi}&0\\0&0&0&e^{-i\mu\varphi}\end{pmatrix}$ | $\begin{pmatrix}-ie^{i\mu\varphi}&0&0&0\\0&ie^{-i\mu\varphi}&0&0\\0&0&ie^{i\mu\varphi}&0\\0&0&0&-ie^{-i\mu\varphi}\end{pmatrix}$ | $\begin{pmatrix}0&-e^{-i\mu\varphi}&0&0\\-e^{i\mu\varphi}&0&0&0\\0&0&0&-e^{-i\mu\varphi}\\0&0&-e^{i\mu\varphi}&0\end{pmatrix}$ | $\begin{pmatrix}e^{i\mu\varphi}&0&0&0\\0&e^{-i\mu\varphi}&0&0\\0&0&e^{i\mu\varphi}&0\\0&0&0&e^{-i\mu\varphi}\end{pmatrix}$ | $\begin{pmatrix}-ie^{i\mu\varphi}&0&0&0\\0&ie^{-i\mu\varphi}&0&0\\0&0&ie^{i\mu\varphi}&0\\0&0&0&-ie^{-i\mu\varphi}\end{pmatrix}$ | $\begin{pmatrix}0&-e^{-i\mu\varphi}&0&0\\-e^{i\mu\varphi}&0&0&0\\0&0&0&-e^{-i\mu\varphi}\\0&0&-e^{i\mu\varphi}&0\end{pmatrix}$ |

**Table 2**

| $b^\infty \times {}^1_4 \bar{1}_2 \bar{1}_m$ | $\{R_\psi, E\}$ | $\{R_\psi, IC_{4z}^1\}$ | $\{2_\perp R_\psi, C_{2x}\}$ | $\{\tau 2_\perp R_\psi, E\}$ | $\{\tau 2_\perp R_\psi, IC_{4z}^1\}$ | $\{\tau R_\psi, C_{2x}\}$ |
|---|---|---|---|---|---|---|
| $\Gamma_1$ | $1$ | $1$ | $1$ | $1$ | $1$ | $1$ |
| $\Gamma_2$ | $1$ | $1$ | $-1$ | $1$ | $1$ | $-1$ |
| $\Gamma_3$ | $1$ | $-1$ | $1$ | $1$ | $-1$ | $1$ |
| $\Gamma_4$ | $1$ | $-1$ | $-1$ | $1$ | $-1$ | $-1$ |
| $\Gamma_5$ | $\begin{pmatrix}1&0\\0&1\end{pmatrix}$ | $\begin{pmatrix}i&0\\0&-i\end{pmatrix}$ | $\begin{pmatrix}0&1\\1&0\end{pmatrix}$ | $\begin{pmatrix}0&1\\1&0\end{pmatrix}$ | $\begin{pmatrix}0&i\\-i&0\end{pmatrix}$ | $\begin{pmatrix}1&0\\0&1\end{pmatrix}$ |
| $\Gamma_6$ | $\begin{pmatrix}e^{i\nu\varphi}&0\\0&e^{-i\nu\varphi}\end{pmatrix}$ | $\begin{pmatrix}e^{i\nu\varphi}&0\\0&e^{-i\nu\varphi}\end{pmatrix}$ | $\begin{pmatrix}0&e^{-i\nu\varphi}\\e^{i\nu\varphi}&0\end{pmatrix}$ | $\begin{pmatrix}e^{i\nu\varphi}&0\\0&e^{-i\nu\varphi}\end{pmatrix}$ | $\begin{pmatrix}e^{i\nu\varphi}&0\\0&e^{-i\nu\varphi}\end{pmatrix}$ | $\begin{pmatrix}0&e^{-i\nu\varphi}\\e^{i\nu\varphi}&0\end{pmatrix}$ |
| $\Gamma_7$ | $\begin{pmatrix}e^{i\nu\varphi}&0\\0&e^{-i\nu\varphi}\end{pmatrix}$ | $\begin{pmatrix}-e^{i\nu\varphi}&0\\0&-e^{-i\nu\varphi}\end{pmatrix}$ | $\begin{pmatrix}0&e^{-i\nu\varphi}\\e^{i\nu\varphi}&0\end{pmatrix}$ | $\begin{pmatrix}e^{i\nu\varphi}&0\\0&e^{-i\nu\varphi}\end{pmatrix}$ | $\begin{pmatrix}-e^{i\nu\varphi}&0\\0&-e^{-i\nu\varphi}\end{pmatrix}$ | $\begin{pmatrix}0&e^{-i\nu\varphi}\\e^{i\nu\varphi}&0\end{pmatrix}$ |
| $\Gamma_8$ | $\begin{pmatrix}e^{i\nu\varphi}&0&0&0\\0&e^{-i\nu\varphi}&0&0\\0&0&e^{i\nu\varphi}&0\\0&0&0&e^{-i\nu\varphi}\end{pmatrix}$ | $\begin{pmatrix}ie^{i\nu\varphi}&0&0&0\\0&-ie^{-i\nu\varphi}&0&0\\0&0&-ie^{i\nu\varphi}&0\\0&0&0&ie^{-i\nu\varphi}\end{pmatrix}$ | $\begin{pmatrix}0&e^{-i\nu\varphi}&0&0\\e^{i\nu\varphi}&0&0&0\\0&0&0&e^{-i\nu\varphi}\\0&0&e^{i\nu\varphi}&0\end{pmatrix}$ | $\begin{pmatrix}e^{i\nu\varphi}&0&0&0\\0&e^{-i\nu\varphi}&0&0\\0&0&e^{i\nu\varphi}&0\\0&0&0&e^{-i\nu\varphi}\end{pmatrix}$ | $\begin{pmatrix}ie^{i\nu\varphi}&0&0&0\\0&-ie^{-i\nu\varphi}&0&0\\0&0&-ie^{i\nu\varphi}&0\\0&0&0&ie^{-i\nu\varphi}\end{pmatrix}$ | $\begin{pmatrix}0&e^{-i\nu\varphi}&0&0\\e^{i\nu\varphi}&0&0&0\\0&0&0&e^{-i\nu\varphi}\\0&0&e^{i\nu\varphi}&0\end{pmatrix}$ |

| $b^\infty_\perp \times \overline{1}_{4/\overline{1}_m}$ / $1_m \overline{1}_m$ | $\{R_\varphi, E\}$ | $\{R_\varphi, IC_{2z}\}$ | $\{R_\varphi, IC_{2y}\}$ | $\{2_\perp R_\varphi, C_{4z}\}$ | $\{\tau 2_\perp R_\varphi, E\}$ | $\{\tau 2_\perp R_\varphi, IC_{2z}\}$ | $\{\tau 2_\perp R_\varphi, IC_{2y}\}$ | $\{\tau R_\varphi, C_{4z}\}$ |
|---|---|---|---|---|---|---|---|---|
| $\Gamma_1$ | $1$ | $1$ | $1$ | $1$ | $1$ | $1$ | $1$ | $1$ |
| $\Gamma_2$ | $1$ | $1$ | $1$ | $-1$ | $1$ | $1$ | $1$ | $-1$ |
| $\Gamma_3$ | $1$ | $-1$ | $-1$ | $1$ | $1$ | $-1$ | $-1$ | $1$ |
| $\Gamma_4$ | $1$ | $-1$ | $-1$ | $-1$ | $1$ | $-1$ | $-1$ | $-1$ |
| $\Gamma_5$ | $1$ | $1$ | $-1$ | $1$ | $1$ | $1$ | $-1$ | $1$ |
| $\Gamma_6$ | $1$ | $1$ | $-1$ | $-1$ | $1$ | $1$ | $-1$ | $-1$ |
| $\Gamma_7$ | $1$ | $-1$ | $1$ | $1$ | $1$ | $-1$ | $1$ | $1$ |
| $\Gamma_8$ | $1$ | $-1$ | $1$ | $-1$ | $1$ | $-1$ | $1$ | $-1$ |
| $\Gamma_9$ | $\begin{pmatrix}1&0\\0&1\end{pmatrix}$ | $\begin{pmatrix}1&0\\0&1\end{pmatrix}$ | $\begin{pmatrix}1&0\\0&-1\end{pmatrix}$ | $\begin{pmatrix}0&-1\\1&0\end{pmatrix}$ | $\begin{pmatrix}1&0\\0&1\end{pmatrix}$ | $\begin{pmatrix}1&0\\0&1\end{pmatrix}$ | $\begin{pmatrix}1&0\\0&-1\end{pmatrix}$ | $\begin{pmatrix}0&-1\\1&0\end{pmatrix}$ |
| $\Gamma_{10}$ | $\begin{pmatrix}1&0\\0&1\end{pmatrix}$ | $\begin{pmatrix}-1&0\\0&-1\end{pmatrix}$ | $\begin{pmatrix}-1&0\\0&1\end{pmatrix}$ | $\begin{pmatrix}0&-1\\1&0\end{pmatrix}$ | $\begin{pmatrix}1&0\\0&1\end{pmatrix}$ | $\begin{pmatrix}-1&0\\0&-1\end{pmatrix}$ | $\begin{pmatrix}-1&0\\0&1\end{pmatrix}$ | $\begin{pmatrix}0&-1\\1&0\end{pmatrix}$ |
| $\Gamma_{11}$ | $\begin{pmatrix}e^{i\nu\varphi}&0\\0&e^{-i\nu\varphi}\end{pmatrix}$ | $\begin{pmatrix}e^{i\nu\varphi}&0\\0&e^{-i\nu\varphi}\end{pmatrix}$ | $\begin{pmatrix}0&e^{-i\nu\varphi}\\e^{i\nu\varphi}&0\end{pmatrix}$ | $\begin{pmatrix}0&e^{-i\nu\varphi}\\e^{i\nu\varphi}&0\end{pmatrix}$ | $\begin{pmatrix}e^{i\nu\varphi}&0\\0&e^{-i\nu\varphi}\end{pmatrix}$ | $\begin{pmatrix}e^{i\nu\varphi}&0\\0&e^{-i\nu\varphi}\end{pmatrix}$ | $\begin{pmatrix}0&e^{-i\nu\varphi}\\e^{i\nu\varphi}&0\end{pmatrix}$ | $\begin{pmatrix}0&e^{-i\nu\varphi}\\e^{i\nu\varphi}&0\end{pmatrix}$ |
| $\Gamma_{12}$ | $\begin{pmatrix}e^{i\nu\varphi}&0\\0&e^{-i\nu\varphi}\end{pmatrix}$ | $\begin{pmatrix}-e^{i\nu\varphi}&0\\0&-e^{-i\nu\varphi}\end{pmatrix}$ | $\begin{pmatrix}0&-e^{-i\nu\varphi}\\-e^{i\nu\varphi}&0\end{pmatrix}$ | $\begin{pmatrix}0&e^{-i\nu\varphi}\\e^{i\nu\varphi}&0\end{pmatrix}$ | $\begin{pmatrix}e^{i\nu\varphi}&0\\0&e^{-i\nu\varphi}\end{pmatrix}$ | $\begin{pmatrix}-e^{i\nu\varphi}&0\\0&-e^{-i\nu\varphi}\end{pmatrix}$ | $\begin{pmatrix}0&-e^{-i\nu\varphi}\\-e^{i\nu\varphi}&0\end{pmatrix}$ | $\begin{pmatrix}0&e^{-i\nu\varphi}\\e^{i\nu\varphi}&0\end{pmatrix}$ |
| $\Gamma_{13}$ | $\begin{pmatrix}e^{i\nu\varphi}&0\\0&e^{-i\nu\varphi}\end{pmatrix}$ | $\begin{pmatrix}e^{i\nu\varphi}&0\\0&e^{-i\nu\varphi}\end{pmatrix}$ | $\begin{pmatrix}0&-e^{-i\nu\varphi}\\-e^{i\nu\varphi}&0\end{pmatrix}$ | $\begin{pmatrix}0&e^{-i\nu\varphi}\\e^{i\nu\varphi}&0\end{pmatrix}$ | $\begin{pmatrix}e^{i\nu\varphi}&0\\0&e^{-i\nu\varphi}\end{pmatrix}$ | $\begin{pmatrix}e^{i\nu\varphi}&0\\0&e^{-i\nu\varphi}\end{pmatrix}$ | $\begin{pmatrix}0&-e^{-i\nu\varphi}\\-e^{i\nu\varphi}&0\end{pmatrix}$ | $\begin{pmatrix}0&e^{-i\nu\varphi}\\e^{i\nu\varphi}&0\end{pmatrix}$ |
| $\Gamma_{14}$ | $\begin{pmatrix}e^{i\nu\varphi}&0\\0&e^{-i\nu\varphi}\end{pmatrix}$ | $\begin{pmatrix}-e^{i\nu\varphi}&0\\0&-e^{-i\nu\varphi}\end{pmatrix}$ | $\begin{pmatrix}0&e^{-i\nu\varphi}\\e^{i\nu\varphi}&0\end{pmatrix}$ | $\begin{pmatrix}0&-e^{-i\nu\varphi}\\e^{i\nu\varphi}&0\end{pmatrix}$ | $\begin{pmatrix}e^{i\nu\varphi}&0\\0&e^{-i\nu\varphi}\end{pmatrix}$ | $\begin{pmatrix}-e^{i\nu\varphi}&0\\0&-e^{-i\nu\varphi}\end{pmatrix}$ | $\begin{pmatrix}0&e^{-i\nu\varphi}\\e^{i\nu\varphi}&0\end{pmatrix}$ | $\begin{pmatrix}0&-e^{-i\nu\varphi}\\e^{i\nu\varphi}&0\end{pmatrix}$ |
| $\Gamma_{15}$ | $\begin{pmatrix}e^{i\mu\nu\varphi}&0\\0&e^{-i\mu\nu\varphi}\end{pmatrix}$ | $\begin{pmatrix}e^{i\mu\nu\varphi}&0\\0&e^{-i\mu\nu\varphi}\end{pmatrix}$ | $\begin{pmatrix}0&e^{-i\mu\nu\varphi}\\e^{i\mu\nu\varphi}&0\end{pmatrix}$ | $\begin{pmatrix}0&-e^{-i\mu\nu\varphi}\\e^{i\mu\nu\varphi}&0\end{pmatrix}$ | $\begin{pmatrix}e^{i\mu\nu\varphi}&0\\0&e^{-i\mu\nu\varphi}\end{pmatrix}$ | $\begin{pmatrix}e^{i\mu\nu\varphi}&0\\0&e^{-i\mu\nu\varphi}\end{pmatrix}$ | $\begin{pmatrix}0&e^{-i\mu\nu\varphi}\\e^{i\mu\nu\varphi}&0\end{pmatrix}$ | $\begin{pmatrix}0&-e^{-i\mu\nu\varphi}\\e^{i\mu\nu\varphi}&0\end{pmatrix}$ |
| $\Gamma_{16}$ | $\begin{pmatrix}e^{i\mu\nu\varphi}&0\\0&e^{-i\mu\nu\varphi}\end{pmatrix}$ | $\begin{pmatrix}-e^{i\mu\nu\varphi}&0\\0&-e^{-i\mu\nu\varphi}\end{pmatrix}$ | $\begin{pmatrix}0&e^{-i\mu\nu\varphi}\\e^{i\mu\nu\varphi}&0\end{pmatrix}$ | $\begin{pmatrix}0&-e^{-i\mu\nu\varphi}\\e^{i\mu\nu\varphi}&0\end{pmatrix}$ | $\begin{pmatrix}e^{i\mu\nu\varphi}&0\\0&e^{-i\mu\nu\varphi}\end{pmatrix}$ | $\begin{pmatrix}-e^{i\mu\nu\varphi}&0\\0&-e^{-i\mu\nu\varphi}\end{pmatrix}$ | $\begin{pmatrix}0&e^{-i\mu\nu\varphi}\\e^{i\mu\nu\varphi}&0\end{pmatrix}$ | $\begin{pmatrix}0&-e^{-i\mu\nu\varphi}\\e^{i\mu\nu\varphi}&0\end{pmatrix}$ |

| $b^\infty \times \bar{1}3$ | $\{R_\varphi, E\}$ | $\left\{2_\perp R_\varphi, IC_{3z}^{-1}\right\}$ | $\{\tau 2_\perp R_\varphi, E\}$ | $\left\{\tau R_\varphi, IC_{3z}^{-1}\right\}$ |
|---|---|---|---|---|
| $\Gamma_1$ | $1$ | $1$ | $1$ | $1$ |
| $\Gamma_2$ | $1$ | $-1$ | $1$ | $-1$ |
| $\Gamma_3$ | $\begin{pmatrix} e^{i\vee\varphi} & 0 \\ 0 & e^{-i\vee\varphi} \end{pmatrix}$ | $\begin{pmatrix} 0 & e^{-i\vee\varphi} \\ e^{i\vee\varphi} & 0 \end{pmatrix}$ | $\begin{pmatrix} e^{i\vee\varphi} & 0 \\ 0 & e^{-i\vee\varphi} \end{pmatrix}$ | $\begin{pmatrix} 0 & e^{-i\vee\varphi} \\ e^{i\vee\varphi} & 0 \end{pmatrix}$ |
| $\Gamma_4$ | $\begin{pmatrix} 1 & 0 \\ 0 & 1 \end{pmatrix}$ | $\begin{pmatrix} -e^{-\frac{2i\pi}{3}} & 0 \\ 0 & -e^{\frac{2i\pi}{3}} \end{pmatrix}$ | $\begin{pmatrix} 0 & 1 \\ 1 & 0 \end{pmatrix}$ | $\begin{pmatrix} 0 & -e^{\frac{2i\pi}{3}} \\ -e^{\frac{2i\pi}{3}} & 0 \end{pmatrix}$ |
| $\Gamma_5$ | $\begin{pmatrix} 1 & 0 \\ 0 & 1 \end{pmatrix}$ | $\begin{pmatrix} e^{-\frac{2i\pi}{3}} & 0 \\ 0 & e^{\frac{2i\pi}{3}} \end{pmatrix}$ | $\begin{pmatrix} 0 & 1 \\ 1 & 0 \end{pmatrix}$ | $\begin{pmatrix} 0 & e^{-\frac{2i\pi}{3}} \\ e^{\frac{2i\pi}{3}} & 0 \end{pmatrix}$ |
| $\Gamma_6$ | $\begin{pmatrix} e^{i\vee\varphi} & 0 & 0 & 0 \\ 0 & e^{-i\vee\varphi} & 0 & 0 \\ 0 & 0 & e^{i\vee\varphi} & 0 \\ 0 & 0 & 0 & e^{-i\vee\varphi} \end{pmatrix}$ | $\begin{pmatrix} 0 & e^{-i\vee\varphi} & 0 & 0 \\ e^{\frac{2i\pi}{3}+i\vee\varphi} & 0 & 0 & 0 \\ 0 & 0 & 0 & e^{-i\vee\varphi} \\ 0 & 0 & e^{-\frac{2i\pi}{3}+i\vee\varphi} & 0 \end{pmatrix}$ | $\begin{pmatrix} 0 & 0 & e^{i\vee\varphi} & 0 \\ 0 & 0 & 0 & e^{-i\vee\varphi} \\ e^{i\vee\varphi} & 0 & 0 & 0 \\ 0 & e^{-i\vee\varphi} & 0 & 0 \end{pmatrix}$ | $\begin{pmatrix} 0 & 0 & 0 & e^{-i\vee\varphi} \\ 0 & 0 & e^{\frac{2i\pi}{3}+i\vee\varphi} & 0 \\ 0 & e^{-i\vee\varphi} & 0 & 0 \\ e^{-\frac{2i\pi}{3}+i\vee\varphi} & 0 & 0 & 0 \end{pmatrix}$ |

| $b^\infty_x\, 13\, \bar{1}_2$ | $\{R_\varphi, E\}$ | $\{R_\varphi, C_{3z}\}$ | $\{2_\perp R_\varphi, C_{2x}\}$ | $\{\tau 2_\perp R_\varphi, E\}$ | $\{\tau 2_\perp R_\varphi, C_{3z}\}$ | $\{\tau R_\varphi, C_{2x}\}$ |
|---|---|---|---|---|---|---|
| $\Gamma_1$ | 1 | 1 | 1 | 1 | 1 | 1 |
| $\Gamma_2$ | 1 | 1 | $-1$ | 1 | 1 | $-1$ |
| $\Gamma_3$ | $\begin{pmatrix}1 & 0\\0 & 1\end{pmatrix}$ | $\begin{pmatrix}e^{\frac{2i\pi}{3}} & 0\\0 & e^{-\frac{2i\pi}{3}}\end{pmatrix}$ | $\begin{pmatrix}0 & 1\\1 & 0\end{pmatrix}$ | $\begin{pmatrix}0 & 1\\1 & 0\end{pmatrix}$ | $\begin{pmatrix}0 & e^{\frac{2i\pi}{3}}\\e^{-\frac{2i\pi}{3}} & 0\end{pmatrix}$ | $\begin{pmatrix}1 & 0\\0 & 1\end{pmatrix}$ |
| $\Gamma_4$ | $\begin{pmatrix}e^{i\varphi} & 0\\0 & e^{-i\varphi}\end{pmatrix}$ | $\begin{pmatrix}e^{i\varphi} & 0\\0 & e^{-i\varphi}\end{pmatrix}$ | $\begin{pmatrix}0 & e^{-i\varphi}\\e^{i\varphi} & 0\end{pmatrix}$ | $\begin{pmatrix}e^{i\varphi} & 0\\0 & e^{-i\varphi}\end{pmatrix}$ | $\begin{pmatrix}e^{i\varphi} & 0\\0 & e^{-i\varphi}\end{pmatrix}$ | $\begin{pmatrix}0 & e^{-i\varphi}\\e^{i\varphi} & 0\end{pmatrix}$ |
| $\Gamma_5$ | (4×4 matrix) | (4×4 matrix) | (4×4 matrix) | (4×4 matrix) | (4×4 matrix) | (4×4 matrix) |

| $b^\infty_x\, 13\, \bar{1}_m$ | $\{R_\varphi, E\}$ | $\{R_\varphi, C_{3z}\}$ | $\{2_\perp R_\varphi, IC_{2B}\}$ | $\{\tau 2_\perp R_\varphi, E\}$ | $\{\tau 2_\perp R_\varphi, C_{3z}\}$ | $\{\tau R_\varphi, IC_{2B}\}$ |
|---|---|---|---|---|---|---|
| $\Gamma_1$ | 1 | 1 | 1 | 1 | 1 | 1 |
| $\Gamma_2$ | 1 | 1 | $-1$ | 1 | 1 | $-1$ |
| $\Gamma_3$ | $\begin{pmatrix}1 & 0\\0 & 1\end{pmatrix}$ | $\begin{pmatrix}e^{\frac{2i\pi}{3}} & 0\\0 & e^{-\frac{2i\pi}{3}}\end{pmatrix}$ | $\begin{pmatrix}0 & e^{\frac{2i\pi}{3}}\\e^{-\frac{2i\pi}{3}} & 0\end{pmatrix}$ | $\begin{pmatrix}0 & 1\\1 & 0\end{pmatrix}$ | $\begin{pmatrix}0 & e^{\frac{2i\pi}{3}}\\e^{-\frac{2i\pi}{3}} & 0\end{pmatrix}$ | $\begin{pmatrix}0 & e^{\frac{2i\pi}{3}}\\e^{-\frac{2i\pi}{3}} & 0\end{pmatrix}$ |
| $\Gamma_4$ | $\begin{pmatrix}e^{i\varphi} & 0\\0 & e^{-i\varphi}\end{pmatrix}$ | $\begin{pmatrix}e^{i\varphi} & 0\\0 & e^{-i\varphi}\end{pmatrix}$ | $\begin{pmatrix}0 & e^{-i\varphi}\\e^{i\varphi} & 0\end{pmatrix}$ | $\begin{pmatrix}e^{i\varphi} & 0\\0 & e^{-i\varphi}\end{pmatrix}$ | $\begin{pmatrix}e^{i\varphi} & 0\\0 & e^{-i\varphi}\end{pmatrix}$ | $\begin{pmatrix}0 & e^{-i\varphi}\\e^{i\varphi} & 0\end{pmatrix}$ |
| $\Gamma_5$ | (4×4 matrix) | (4×4 matrix) | (4×4 matrix) | (4×4 matrix) | (4×4 matrix) | (4×4 matrix) |

| $b^\infty \times \bar{I}\frac{1}{3}\,T_m$ | $\{R_\psi, E\}$ | $\{R_\psi^2, C_{2x}\}$ | $\{2_\perp R_\psi, IC_{3z}^1\}$ | $\{\tau 2_\perp R_\psi, E\}$ | $\{\tau 2_\perp R_\psi, C_{2x}\}$ | $\{\tau R_\psi, IC_{3z}^1\}$ |
|---|---|---|---|---|---|---|
| $\Gamma_1$ | $1$ | $1$ | $1$ | $1$ | $1$ | $1$ |
| $\Gamma_2$ | $1$ | $1$ | $-1$ | $1$ | $1$ | $-1$ |
| $\Gamma_3$ | $1$ | $-1$ | $1$ | $1$ | $-1$ | $1$ |
| $\Gamma_4$ | $1$ | $-1$ | $-1$ | $1$ | $-1$ | $-1$ |
| $\Gamma_5$ | $\begin{pmatrix}1&0\\0&1\end{pmatrix}$ | $\begin{pmatrix}0&1\\1&0\end{pmatrix}$ | $\begin{pmatrix}-e^{-\frac{2i\pi}{3}}&0\\0&-e^{\frac{2i\pi}{3}}\end{pmatrix}$ | $\begin{pmatrix}0&1\\1&0\end{pmatrix}$ | $\begin{pmatrix}1&0\\0&1\end{pmatrix}$ | $\begin{pmatrix}0&-e^{-\frac{2i\pi}{3}}\\-e^{\frac{2i\pi}{3}}&0\end{pmatrix}$ |
| $\Gamma_6$ | $\begin{pmatrix}1&0\\0&1\end{pmatrix}$ | $\begin{pmatrix}0&1\\1&0\end{pmatrix}$ | $\begin{pmatrix}e^{-\frac{2i\pi}{3}}&0\\0&e^{\frac{2i\pi}{3}}\end{pmatrix}$ | $\begin{pmatrix}0&1\\1&0\end{pmatrix}$ | $\begin{pmatrix}1&0\\0&1\end{pmatrix}$ | $\begin{pmatrix}0&e^{-\frac{2i\pi}{3}}\\e^{\frac{2i\pi}{3}}&0\end{pmatrix}$ |
| $\Gamma_7$ | $\begin{pmatrix}e^{i\nu\varphi}&0\\0&e^{-i\nu\varphi}\end{pmatrix}$ | $\begin{pmatrix}e^{i\nu\varphi}&0\\0&e^{-i\nu\varphi}\end{pmatrix}$ | $\begin{pmatrix}e^{i\nu\varphi}&0\\0&e^{-i\nu\varphi}\end{pmatrix}$ | $\begin{pmatrix}e^{i\nu\varphi}&0\\0&e^{-i\nu\varphi}\end{pmatrix}$ | $\begin{pmatrix}e^{i\nu\varphi}&0\\0&e^{-i\nu\varphi}\end{pmatrix}$ | $\begin{pmatrix}0&e^{-i\nu\varphi}\\e^{i\nu\varphi}&0\end{pmatrix}$ |
| $\Gamma_8$ | $\begin{pmatrix}e^{i\nu\varphi}&0\\0&e^{-i\nu\varphi}\end{pmatrix}$ | $\begin{pmatrix}-e^{i\nu\varphi}&0\\0&-e^{-i\nu\varphi}\end{pmatrix}$ | $\begin{pmatrix}e^{i\nu\varphi}&0\\0&e^{-i\nu\varphi}\end{pmatrix}$ | $\begin{pmatrix}e^{i\nu\varphi}&0\\0&e^{-i\nu\varphi}\end{pmatrix}$ | $\begin{pmatrix}-e^{i\nu\varphi}&0\\0&-e^{-i\nu\varphi}\end{pmatrix}$ | $\begin{pmatrix}0&e^{-i\nu\varphi}\\e^{i\nu\varphi}&0\end{pmatrix}$ |
| $\Gamma_9$ | $\begin{pmatrix}e^{i\nu\varphi}&0&0\\0&e^{i\nu\varphi}&0\\0&0&e^{-i\nu\varphi}\end{pmatrix}$ | $\begin{pmatrix}0&e^{i\nu\varphi}&0\\e^{i\nu\varphi}&0&0\\0&0&e^{-\frac{2i\pi}{3}-i\nu\varphi}\end{pmatrix}$ | $\begin{pmatrix}e^{i\nu\varphi}&0&0\\0&e^{i\nu\varphi}&0\\0&0&e^{-\frac{2i\pi}{3}-i\nu\varphi}\end{pmatrix}$ | $\begin{pmatrix}0&e^{i\nu\varphi}&0\\e^{i\nu\varphi}&0&0\\0&0&e^{-i\nu\varphi}\end{pmatrix}$ | $\begin{pmatrix}e^{i\nu\varphi}&0&0\\0&e^{i\nu\varphi}&0\\0&0&e^{\frac{2i\pi}{3}-i\nu\varphi}\end{pmatrix}$ | $\begin{pmatrix}0&e^{-i\nu\varphi}&0\\0&e^{-i\nu\varphi}&0\\e^{\frac{2i\pi}{3}-i\nu\varphi}&0&0\end{pmatrix}$ |

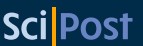

**Left table** (header: $b^\infty \times \bar{1}6\,\bar{1}2\,\bar{1}2$)

| | $\{R_\psi, E\}$ | $\{R_\psi, C_{6z}\}$ | $\{R_\psi, C_{2x}\}$ | $\{2_\perp R_\psi, C_{2x}\}$ | $\{\tau 2_\perp R_\psi, E\}$ | $\{\tau 2_\perp R_\psi, C_{6z}\}$ | $\{\tau R_\psi, C_{2x}\}$ |
|---|---|---|---|---|---|---|---|
| $\Gamma_1$ | $1$ | $1$ | $1$ | $1$ | $1$ | $1$ | $1$ |
| $\Gamma_2$ | $1$ | $1$ | $-1$ | $-1$ | $1$ | $1$ | $-1$ |
| $\Gamma_3$ | $1$ | $-1$ | $1$ | $-1$ | $1$ | $-1$ | $1$ |
| $\Gamma_4$ | $1$ | $-1$ | $-1$ | $-1$ | $1$ | $-1$ | $-1$ |
| $\Gamma_5$ | $\left(\begin{smallmatrix}1&0\\0&1\end{smallmatrix}\right)$ | $\left(\begin{smallmatrix}e^{-\frac{2i\pi}{3}}&0\\0&e^{\frac{2i\pi}{3}}\end{smallmatrix}\right)$ | $\left(\begin{smallmatrix}0&1\\1&0\end{smallmatrix}\right)$ | $\left(\begin{smallmatrix}0&1\\1&0\end{smallmatrix}\right)$ | $\left(\begin{smallmatrix}0&1\\1&0\end{smallmatrix}\right)$ | $\left(\begin{smallmatrix}0&e^{-\frac{2i\pi}{3}}\\e^{\frac{2i\pi}{3}}&0\end{smallmatrix}\right)$ | $\left(\begin{smallmatrix}1&0\\0&1\end{smallmatrix}\right)$ |

*(remaining representation matrices $\Gamma_6$–$\Gamma_{10}$ and the right-hand table for $b^\infty \times \bar{1}6\,\bar{1}2\,\bar{1}2$ with columns $\{R_\psi,E\}$, $\{R_\psi,C_{2x}\}$, $\{2_\perp R_\psi,C_{6z}\}$, $\{\tau 2_\perp R_\psi,E\}$, $\{\tau 2_\perp R_\psi,C_{2x}\}$, $\{\tau R_\psi,C_{6z}\}$ and rows $\Gamma_1$–$\Gamma_9$ consist of higher-dimensional representation matrices)*

| $b^\infty \times {}^{I_6}/I_m$ | $(R_\varphi, E)$ | $\{2_\perp R_\varphi, C_{6z}\}$ | $\{2_\perp R_\varphi, IC_{2z}\}$ | $(\tau 2_\perp R_\varphi, E)$ | $\{\tau R_\varphi, C_{6z}\}$ | $\{\tau R_\varphi, IC_{2z}\}$ |
|---|---|---|---|---|---|---|
| $\Gamma_1$ | $1$ | $1$ | $1$ | $1$ | $1$ | $1$ |
| $\Gamma_2$ | $1$ | $-1$ | $-1$ | $1$ | $-1$ | $-1$ |
| $\Gamma_3$ | $1$ | $-1$ | $-1$ | $1$ | $-1$ | $-1$ |
| $\Gamma_4$ | $1$ | $-1$ | $-1$ | $1$ | $-1$ | $-1$ |
| $\Gamma_5$ | $\begin{pmatrix} e^{i\nu\varphi} & 0 \\ 0 & e^{-i\nu\varphi} \end{pmatrix}$ | $\begin{pmatrix} 0 & e^{-i\nu\varphi} \\ e^{i\nu\varphi} & 0 \end{pmatrix}$ | $\begin{pmatrix} 0 & e^{-i\nu\varphi} \\ e^{i\nu\varphi} & 0 \end{pmatrix}$ | $\begin{pmatrix} e^{i\nu\varphi} & 0 \\ 0 & e^{-i\nu\varphi} \end{pmatrix}$ | $\begin{pmatrix} 0 & e^{-i\nu\varphi} \\ e^{i\nu\varphi} & 0 \end{pmatrix}$ | $\begin{pmatrix} 0 & e^{-i\nu\varphi} \\ e^{i\nu\varphi} & 0 \end{pmatrix}$ |
| $\Gamma_6$ | $\begin{pmatrix} e^{i\nu\varphi} & 0 \\ 0 & e^{-i\nu\varphi} \end{pmatrix}$ | $\begin{pmatrix} 0 & e^{-i\nu\varphi} \\ e^{i\nu\varphi} & 0 \end{pmatrix}$ | $\begin{pmatrix} 0 & -e^{-i\nu\varphi} \\ -e^{i\nu\varphi} & 0 \end{pmatrix}$ | $\begin{pmatrix} 0 & e^{-i\nu\varphi} \\ e^{i\nu\varphi} & 0 \end{pmatrix}$ | $\begin{pmatrix} 0 & e^{-i\nu\varphi} \\ e^{i\nu\varphi} & 0 \end{pmatrix}$ | $\begin{pmatrix} 0 & -e^{-i\nu\varphi} \\ -e^{i\nu\varphi} & 0 \end{pmatrix}$ |
| $\Gamma_7$ | $\begin{pmatrix} 1 & 0 \\ 0 & 1 \end{pmatrix}$ | $\begin{pmatrix} -e^{-\frac{2i\pi}{3}} & 0 \\ 0 & e^{\frac{2i\pi}{3}} \end{pmatrix}$ | $\begin{pmatrix} -1 & 0 \\ 0 & -1 \end{pmatrix}$ | $\begin{pmatrix} 0 & 1 \\ 1 & 0 \end{pmatrix}$ | $\begin{pmatrix} 0 & e^{\frac{2i\pi}{3}} \\ e^{\frac{2i\pi}{3}} & 0 \end{pmatrix}$ | $\begin{pmatrix} 0 & -1 \\ -1 & 0 \end{pmatrix}$ |
| $\Gamma_8$ | $\begin{pmatrix} 1 & 0 \\ 0 & 1 \end{pmatrix}$ | $\begin{pmatrix} -e^{-\frac{2i\pi}{3}} & 0 \\ 0 & -e^{\frac{2i\pi}{3}} \end{pmatrix}$ | $\begin{pmatrix} 1 & 0 \\ 0 & 1 \end{pmatrix}$ | $\begin{pmatrix} 0 & 1 \\ 1 & 0 \end{pmatrix}$ | $\begin{pmatrix} 0 & -e^{-\frac{2i\pi}{3}} \\ -e^{\frac{2i\pi}{3}} & 0 \end{pmatrix}$ | $\begin{pmatrix} 0 & 1 \\ 1 & 0 \end{pmatrix}$ |
| $\Gamma_9$ | $\begin{pmatrix} 1 & 0 \\ 0 & 1 \end{pmatrix}$ | $\begin{pmatrix} e^{-\frac{2i\pi}{3}} & 0 \\ 0 & e^{\frac{2i\pi}{3}} \end{pmatrix}$ | $\begin{pmatrix} 1 & 0 \\ 0 & 1 \end{pmatrix}$ | $\begin{pmatrix} 0 & 1 \\ 1 & 0 \end{pmatrix}$ | $\begin{pmatrix} 0 & e^{-\frac{2i\pi}{3}} \\ e^{\frac{2i\pi}{3}} & 0 \end{pmatrix}$ | $\begin{pmatrix} 0 & 1 \\ 1 & 0 \end{pmatrix}$ |
| $\Gamma_{10}$ | $\begin{pmatrix} 1 & 0 \\ 0 & 1 \end{pmatrix}$ | $\begin{pmatrix} -e^{-\frac{2i\pi}{3}} & 0 \\ 0 & -e^{\frac{2i\pi}{3}} \end{pmatrix}$ | $\begin{pmatrix} -1 & 0 \\ 0 & -1 \end{pmatrix}$ | $\begin{pmatrix} 0 & 1 \\ 1 & 0 \end{pmatrix}$ | $\begin{pmatrix} 0 & -e^{-\frac{2i\pi}{3}} \\ -e^{\frac{2i\pi}{3}} & 0 \end{pmatrix}$ | $\begin{pmatrix} 0 & -1 \\ -1 & 0 \end{pmatrix}$ |
| $\Gamma_{11}$ | $\begin{pmatrix} e^{i\nu\varphi} & 0 & 0 \\ 0 & e^{-i\nu\varphi} & 0 \\ 0 & 0 & e^{-i\nu\varphi} \end{pmatrix}$ | $\begin{pmatrix} 0 & 0 & 0 \\ e^{i\nu\varphi} & 0 & 0 \\ 0 & e^{\frac{2i\pi}{3}-i\nu\varphi} & 0 \end{pmatrix}$ | $\begin{pmatrix} 0 & 0 & 0 \\ e^{\frac{2i\pi}{3}+i\nu\varphi} & 0 & 0 \\ -e^{-\frac{2i\pi}{3}} & 0 & 0 \end{pmatrix}$ | $\begin{pmatrix} 0 & 0 & 0 \\ e^{i\nu\varphi} & 0 & 0 \\ 0 & e^{-i\nu\varphi} & 0 \end{pmatrix}$ | $\begin{pmatrix} 0 & 0 & e^{\frac{2i\pi}{3}-i\nu\varphi} \\ e^{i\nu\varphi} & 0 & 0 \\ 0 & -e^{-\frac{2i\pi}{3}} & 0 \end{pmatrix}$ | $\begin{pmatrix} 0 & 0 & -e^{\frac{2i\pi}{3}-i\nu\varphi} \\ 0 & e^{-\frac{2i\pi}{3}+i\nu\varphi} & 0 \\ -e^{\frac{2i\pi}{3}} & 0 & 0 \end{pmatrix}$ |
| $\Gamma_{12}$ | $\begin{pmatrix} e^{i\nu\varphi} & 0 & 0 \\ 0 & e^{-i\nu\varphi} & 0 \\ 0 & 0 & e^{-i\nu\varphi} \end{pmatrix}$ | $\begin{pmatrix} 0 & 0 & 0 \\ e^{i\nu\varphi} & 0 & 0 \\ 0 & e^{\frac{2i\pi}{3}-i\nu\varphi} & 0 \end{pmatrix}$ | $\begin{pmatrix} 0 & 0 & 0 \\ e^{\frac{2i\pi}{3}+i\nu\varphi} & 0 & 0 \\ e^{\frac{2i\pi}{3}} & 0 & 0 \end{pmatrix}$ | $\begin{pmatrix} 0 & 0 & 0 \\ e^{i\nu\varphi} & 0 & 0 \\ 0 & e^{-i\nu\varphi} & 0 \end{pmatrix}$ | $\begin{pmatrix} 0 & 0 & e^{\frac{2i\pi}{3}-i\nu\varphi} \\ e^{i\nu\varphi} & 0 & 0 \\ 0 & e^{\frac{2i\pi}{3}+i\nu\varphi} & 0 \end{pmatrix}$ | $\begin{pmatrix} 0 & 0 & e^{\frac{2i\pi}{3}-i\nu\varphi} \\ 0 & e^{\frac{2i\pi}{3}+i\nu\varphi} & 0 \\ e^{\frac{2i\pi}{3}} & 0 & 0 \end{pmatrix}$ |

| $b^\infty_\infty \times {}^{\bar{1}}6/1_m$ | $\{R_\psi, E\}$ | $\{R_\psi, IC_{2z}\}$ | $\{2_\perp R_\psi, C_{6z}\}$ | $\{\tau 2_\perp R_\psi, E\}$ | $\{\tau 2_\perp R_\psi, IC_{2z}\}$ | $\{\tau R_\psi, C_{6z}\}$ |
|---|---|---|---|---|---|---|
| $\Gamma_1$ | $1$ | $1$ | $1$ | $1$ | $1$ | $1$ |
| $\Gamma_2$ | $1$ | $1$ | $-1$ | $1$ | $1$ | $-1$ |
| $\Gamma_3$ | $1$ | $-1$ | $1$ | $1$ | $-1$ | $1$ |
| $\Gamma_4$ | $1$ | $-1$ | $-1$ | $1$ | $-1$ | $-1$ |
| $\Gamma_5$ | $\begin{pmatrix} e^{i\nu\varphi} & 0 \\ 0 & e^{-i\nu\varphi} \end{pmatrix}$ | $\begin{pmatrix} e^{i\nu\varphi} & 0 \\ 0 & e^{-i\nu\varphi} \end{pmatrix}$ | $\begin{pmatrix} 0 & e^{-i\nu\varphi} \\ e^{i\nu\varphi} & 0 \end{pmatrix}$ | $\begin{pmatrix} e^{i\nu\varphi} & 0 \\ 0 & e^{-i\nu\varphi} \end{pmatrix}$ | $\begin{pmatrix} e^{i\nu\varphi} & 0 \\ 0 & e^{-i\nu\varphi} \end{pmatrix}$ | $\begin{pmatrix} 0 & e^{-i\nu\varphi} \\ e^{i\nu\varphi} & 0 \end{pmatrix}$ |
| $\Gamma_6$ | $\begin{pmatrix} e^{i\nu\varphi} & 0 \\ 0 & e^{-i\nu\varphi} \end{pmatrix}$ | $\begin{pmatrix} -e^{i\nu\varphi} & 0 \\ 0 & -e^{-i\nu\varphi} \end{pmatrix}$ | $\begin{pmatrix} 0 & e^{-i\nu\varphi} \\ e^{i\nu\varphi} & 0 \end{pmatrix}$ | $\begin{pmatrix} e^{i\nu\varphi} & 0 \\ 0 & e^{-i\nu\varphi} \end{pmatrix}$ | $\begin{pmatrix} -e^{i\nu\varphi} & 0 \\ 0 & -e^{-i\nu\varphi} \end{pmatrix}$ | $\begin{pmatrix} 0 & e^{-i\nu\varphi} \\ e^{i\nu\varphi} & 0 \end{pmatrix}$ |
| $\Gamma_7$ | $\begin{pmatrix} 1 & 0 \\ 0 & 1 \end{pmatrix}$ | $\begin{pmatrix} 1 & 0 \\ 0 & 1 \end{pmatrix}$ | $\begin{pmatrix} e^{-\frac{2i\pi}{3}} & 0 \\ 0 & e^{\frac{2i\pi}{3}} \end{pmatrix}$ | $\begin{pmatrix} 0 & 1 \\ 1 & 0 \end{pmatrix}$ | $\begin{pmatrix} 0 & 1 \\ 1 & 0 \end{pmatrix}$ | $\begin{pmatrix} 0 & e^{\frac{2i\pi}{3}} \\ e^{-\frac{2i\pi}{3}} & 0 \end{pmatrix}$ |
| $\Gamma_8$ | $\begin{pmatrix} 1 & 0 \\ 0 & 1 \end{pmatrix}$ | $\begin{pmatrix} 1 & 0 \\ 0 & 1 \end{pmatrix}$ | $\begin{pmatrix} -e^{-\frac{2i\pi}{3}} & 0 \\ 0 & -e^{\frac{2i\pi}{3}} \end{pmatrix}$ | $\begin{pmatrix} 0 & 1 \\ 1 & 0 \end{pmatrix}$ | $\begin{pmatrix} 0 & 1 \\ 1 & 0 \end{pmatrix}$ | $\begin{pmatrix} 0 & -e^{\frac{2i\pi}{3}} \\ -e^{-\frac{2i\pi}{3}} & 0 \end{pmatrix}$ |
| $\Gamma_9$ | $\begin{pmatrix} 1 & 0 \\ 0 & 1 \end{pmatrix}$ | $\begin{pmatrix} -1 & 0 \\ 0 & -1 \end{pmatrix}$ | $\begin{pmatrix} e^{-\frac{2i\pi}{3}} & 0 \\ 0 & e^{\frac{2i\pi}{3}} \end{pmatrix}$ | $\begin{pmatrix} 0 & 1 \\ 1 & 0 \end{pmatrix}$ | $\begin{pmatrix} 0 & -1 \\ -1 & 0 \end{pmatrix}$ | $\begin{pmatrix} 0 & e^{\frac{2i\pi}{3}} \\ e^{-\frac{2i\pi}{3}} & 0 \end{pmatrix}$ |
| $\Gamma_{10}$ | $\begin{pmatrix} 1 & 0 \\ 0 & 1 \end{pmatrix}$ | $\begin{pmatrix} -1 & 0 \\ 0 & -1 \end{pmatrix}$ | $\begin{pmatrix} -e^{-\frac{2i\pi}{3}} & 0 \\ 0 & -e^{\frac{2i\pi}{3}} \end{pmatrix}$ | $\begin{pmatrix} 0 & 1 \\ 1 & 0 \end{pmatrix}$ | $\begin{pmatrix} 0 & -1 \\ -1 & 0 \end{pmatrix}$ | $\begin{pmatrix} 0 & -e^{\frac{2i\pi}{3}} \\ -e^{-\frac{2i\pi}{3}} & 0 \end{pmatrix}$ |
| $\Gamma_{11}$ | $\begin{pmatrix} e^{i\nu\varphi} & 0 & 0 & 0 \\ 0 & e^{-i\nu\varphi} & 0 & 0 \\ 0 & 0 & e^{i\nu\varphi} & 0 \\ 0 & 0 & 0 & e^{-i\nu\varphi} \end{pmatrix}$ | $\begin{pmatrix} e^{i\nu\varphi} & 0 & 0 & 0 \\ 0 & e^{-i\nu\varphi} & 0 & 0 \\ 0 & 0 & 0 & 0 \\ 0 & 0 & 0 & 0 \end{pmatrix}$ | $\begin{pmatrix} e^{\frac{2i\pi}{3}-i\nu\varphi} & 0 & & \\ 0 & e^{i\nu\varphi} & & \\ & & & \\ & & & \end{pmatrix}$ | $\begin{pmatrix} e^{i\nu\varphi} & 0 & 0 & 0 \\ 0 & e^{-i\nu\varphi} & 0 & 0 \\ 0 & 0 & & \\ 0 & 0 & & \end{pmatrix}$ | $\begin{pmatrix} e^{i\nu\varphi} & 0 & & \\ 0 & e^{-i\nu\varphi} & & \\ & & & \\ & & & \end{pmatrix}$ | $\begin{pmatrix} e^{\frac{2i\pi}{3}-i\nu\varphi} & 0 & 0 & 0 \\ 0 & e^{i\nu\varphi} & 0 & 0 \\ 0 & 0 & & \\ 0 & 0 & e^{i\nu\varphi} \end{pmatrix}$ |
| $\Gamma_{12}$ | $\begin{pmatrix} e^{i\nu\varphi} & 0 & 0 & 0 \\ 0 & e^{-i\nu\varphi} & 0 & 0 \\ 0 & 0 & e^{i\nu\varphi} & 0 \\ 0 & 0 & 0 & e^{-i\nu\varphi} \end{pmatrix}$ | $\begin{pmatrix} -e^{i\nu\varphi} & 0 & 0 & 0 \\ 0 & -e^{-i\nu\varphi} & 0 & 0 \\ 0 & 0 & 0 & 0 \\ 0 & 0 & 0 & 0 \end{pmatrix}$ | $\begin{pmatrix} e^{\frac{2i\pi}{3}-i\nu\varphi} & 0 \\ 0 & e^{i\nu\varphi} \\ \end{pmatrix}$ | $\begin{pmatrix} e^{i\nu\varphi} & 0 & 0 & 0 \\ 0 & e^{-i\nu\varphi} & 0 & 0 \\ \end{pmatrix}$ | $\begin{pmatrix} -e^{i\nu\varphi} & 0 & 0 & 0 \\ 0 & -e^{-i\nu\varphi} & 0 & 0 \\ \end{pmatrix}$ | $\begin{pmatrix} e^{\frac{2i\pi}{3}-i\nu\varphi} & 0 & 0 & 0 \\ 0 & e^{i\nu\varphi} & 0 & 0 \\ 0 & 0 & & \\ 0 & 0 & e^{i\nu\varphi} \end{pmatrix}$ |

| $b^\infty \times {}^{16}_7 I_m$ | $\{R_\psi, E\}$ | $\{R_\psi, C_{6z}\}$ | $\{2_\perp R_\psi, IC_{2z}\}$ | $\{\tau 2_\perp R_\psi, E\}$ | $\{\tau 2_\perp R_\psi, C_{6z}\}$ | $\{\tau R_\psi, IC_{2z}\}$ |
|---|---|---|---|---|---|---|
| $\Gamma_1$ | | | | | | |
| $\Gamma_2$ | | | | | | |
| $\Gamma_3$ | | | | | | |
| $\Gamma_4$ | | | | | | |
| $\Gamma_5$ | | | | | | |
| $\Gamma_6$ | | | | | | |
| $\Gamma_7$ | | | | | | |
| $\Gamma_8$ | | | | | | |
| $\Gamma_9$ | | | | | | |
| $\Gamma_{10}$ | | | | | | |
| $\Gamma_{11}$ | | | | | | |
| $\Gamma_{12}$ | | | | | | |

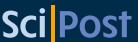

| $b^\infty \times {}^16\,\bar{1}_m\,\bar{1}_m$ | $\{R_\psi, E\}$ | $\{R_\psi, C_{6z}\}$ | $\{2_\perp R_\psi, IC_{2y}\}$ | $\{\tau 2_\perp R_\psi, E\}$ | $\{\tau 2_\perp R_\psi, C_{6z}\}$ | $\{\tau R_\psi, IC_{2y}\}$ |
|---|---|---|---|---|---|---|
| $\Gamma_1$ | $1$ | $1$ | $1$ | $1$ | $1$ | $1$ |
| $\Gamma_2$ | $1$ | $1$ | $-1$ | $1$ | $1$ | $-1$ |
| $\Gamma_3$ | $1$ | $-1$ | $-1$ | $1$ | $-1$ | $-1$ |
| $\Gamma_4$ | $1$ | $-1$ | $1$ | $1$ | $-1$ | $1$ |
| $\Gamma_5$ | $\begin{pmatrix}1&0\\0&1\end{pmatrix}$ | $\begin{pmatrix}e^{-\frac{2i\pi}{3}}&0\\0&e^{\frac{2i\pi}{3}}\end{pmatrix}$ | $\begin{pmatrix}0&1\\1&0\end{pmatrix}$ | $\begin{pmatrix}0&1\\1&0\end{pmatrix}$ | $\begin{pmatrix}0&e^{-\frac{2i\pi}{3}}\\e^{\frac{2i\pi}{3}}&0\end{pmatrix}$ | $\begin{pmatrix}1&0\\0&1\end{pmatrix}$ |
| $\Gamma_6$ | $\begin{pmatrix}1&0\\0&1\end{pmatrix}$ | $\begin{pmatrix}-e^{-\frac{2i\pi}{3}}&0\\0&-e^{\frac{2i\pi}{3}}\end{pmatrix}$ | $\begin{pmatrix}0&-1\\-1&0\end{pmatrix}$ | $\begin{pmatrix}0&1\\1&0\end{pmatrix}$ | $\begin{pmatrix}0&-e^{-\frac{2i\pi}{3}}\\-e^{\frac{2i\pi}{3}}&0\end{pmatrix}$ | $\begin{pmatrix}-1&0\\0&-1\end{pmatrix}$ |
| $\Gamma_7$ | $\begin{pmatrix}e^{i\nu\varphi}&0\\0&e^{-i\nu\varphi}\end{pmatrix}$ | $\begin{pmatrix}-e^{i\nu\varphi}&0\\0&e^{-i\nu\varphi}\end{pmatrix}$ | $\begin{pmatrix}0&e^{-i\nu\varphi}\\e^{i\nu\varphi}&0\end{pmatrix}$ | $\begin{pmatrix}e^{i\nu\varphi}&0\\0&e^{-i\nu\varphi}\end{pmatrix}$ | $\begin{pmatrix}e^{i\nu\varphi}&0\\0&-e^{-i\nu\varphi}\end{pmatrix}$ | $\begin{pmatrix}0&e^{-i\nu\varphi}\\e^{i\nu\varphi}&0\end{pmatrix}$ |
| $\Gamma_8$ | $\begin{pmatrix}e^{i\nu\varphi}&0\\0&e^{-i\nu\varphi}\end{pmatrix}$ | $\begin{pmatrix}-e^{i\nu\varphi}&0\\0&-e^{-i\nu\varphi}\end{pmatrix}$ | $\begin{pmatrix}0&-e^{-i\nu\varphi}\\-e^{i\nu\varphi}&0\end{pmatrix}$ | $\begin{pmatrix}e^{i\nu\varphi}&0\\0&e^{-i\nu\varphi}\end{pmatrix}$ | $\begin{pmatrix}-e^{i\nu\varphi}&0\\0&-e^{-i\nu\varphi}\end{pmatrix}$ | $\begin{pmatrix}0&-e^{-i\nu\varphi}\\-e^{i\nu\varphi}&0\end{pmatrix}$ |
| $\Gamma_9$ | $\begin{pmatrix}e^{i\nu\varphi}&0&0\\0&e^{-i\nu\varphi}&0\\0&0&e^{i\nu\varphi}\end{pmatrix}$ | $\begin{pmatrix}e^{-\frac{2i\pi}{3}+i\nu\varphi}&0&0\\0&e^{\frac{2i\pi}{3}-i\nu\varphi}&0\\0&0&e^{\frac{2i\pi}{3}-i\nu\varphi}\end{pmatrix}$ | $\begin{pmatrix}0&e^{-i\nu\varphi}&0\\e^{i\nu\varphi}&0&0\\0&0&e^{i\nu\varphi}\end{pmatrix}$ | $\begin{pmatrix}e^{i\nu\varphi}&0&0\\0&e^{-i\nu\varphi}&0\\0&0&e^{-i\nu\varphi}\end{pmatrix}$ | $\begin{pmatrix}0&e^{-\frac{2i\pi}{3}-i\nu\varphi}&0\\e^{\frac{2i\pi}{3}+i\nu\varphi}&0&0\\0&0&e^{\frac{2i\pi}{3}}\end{pmatrix}$ | $\begin{pmatrix}0&e^{-i\nu\varphi}&0&0\\e^{i\nu\varphi}&0&0\\0&0&e^{i\nu\varphi}\end{pmatrix}$ |
| $\Gamma_{10}$ | $\begin{pmatrix}e^{i\nu\varphi}&0&0\\0&e^{-i\nu\varphi}&0\\0&0&e^{i\nu\varphi}\end{pmatrix}$ | $\begin{pmatrix}-e^{-\frac{2i\pi}{3}+i\nu\varphi}&0&0\\0&-e^{\frac{2i\pi}{3}-i\nu\varphi}&0\\0&0&-e^{-\frac{2i\pi}{3}-i\nu\varphi}\end{pmatrix}$ | $\begin{pmatrix}0&-e^{-i\nu\varphi}&0\\-e^{i\nu\varphi}&0&0\\0&0&e^{i\nu\varphi}\end{pmatrix}$ | $\begin{pmatrix}e^{i\nu\varphi}&0&0\\0&e^{-i\nu\varphi}&0\\0&0&e^{-i\nu\varphi}\end{pmatrix}$ | $\begin{pmatrix}-e^{\frac{2i\pi}{3}+i\nu\varphi}&0&0\\0&-e^{-\frac{2i\pi}{3}-i\nu\varphi}&0\\0&0&-e^{-\frac{2i\pi}{3}-i\nu\varphi}\end{pmatrix}$ | $\begin{pmatrix}-e^{-i\nu\varphi}&0&0\\0&-e^{i\nu\varphi}&0\\0&0&-e^{-i\nu\varphi}\end{pmatrix}$ |

| $\mathbf{b}^{\nu}\times{}^{\tilde{g}}\rho_{i_n}$ $\quad i_m\;i_m$ | $(R_9,\,E)$ | $\{2_\perp R_9,\,C_{6z}\}$ | $\{2_\perp R_9,\,IC_{2z}\}$ | $\{2_\perp R_9,\,IC_{2y}\}$ | $(\tau2_\perp R_9,\,E)$ | $\{\tau R_9,\,C_{6z}\}$ | $\{\tau R_9,\,IC_{2z}\}$ | $\{\tau R_9,\,IC_{2y}\}$ |
|---|---|---|---|---|---|---|---|---|
| $\Gamma_1$ | 1 | 1 | 1 | 1 | 1 | 1 | 1 | 1 |
| $\Gamma_2$ | 1 | 1 | $-1$ | $-1$ | 1 | $-1$ | $-1$ | $-1$ |
| $\Gamma_3$ | 1 | 1 | $-1$ | $-1$ | 1 | $-1$ | $-1$ | $-1$ |
| $\Gamma_4$ | 1 | $-1$ | 1 | 1 | 1 | $-1$ | 1 | 1 |
| $\Gamma_5$ | 1 | 1 | 1 | 1 | 1 | 1 | 1 | 1 |
| $\Gamma_6$ | 1 | 1 | $-1$ | $-1$ | 1 | $-1$ | $-1$ | $-1$ |
| $\Gamma_7$ | 1 | $-1$ | 1 | $-1$ | 1 | $-1$ | 1 | $-1$ |
| $\Gamma_8$ | 1 | $-1$ | $-1$ | 1 | 1 | $-1$ | $-1$ | 1 |

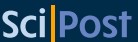

| $b^{\infty}_{\perp}C_{\infty z}\times {}^1_6{}^6 T_{-m}$ $T_{-m}$ | $(R_\varphi, E)$ | $\{R_\varphi, c_{6z}\}$ | $\{R_\varphi, IC_{2y}\}$ | $\{2_\perp R_\varphi, IC_{2z}\}$ | $(2_\perp R_\varphi, E)$ | $\{2_\perp R_\varphi, c_{6z}\}$ | $(2_\perp R_\varphi, E)$ | $\{2_\perp R_\varphi, IC_{2y}\}$ | $\{2_\perp R_\varphi, IC_{2z}\}$ |
|---|---|---|---|---|---|---|---|---|---|
| $\Gamma_1$ | 1 | 1 | 1 | 1 | 1 | 1 | 1 | 1 | 1 |
| $\Gamma_2$ | 1 | 1 | 1 | -1 | 1 | 1 | 1 | 1 | -1 |
| $\Gamma_3$ | 1 | 1 | -1 | -1 | 1 | 1 | 1 | -1 | 1 |
| $\Gamma_4$ | 1 | 1 | -1 | 1 | 1 | 1 | 1 | -1 | -1 |
| $\Gamma_5$ | 1 | -1 | -1 | 1 | 1 | -1 | 1 | -1 | 1 |
| $\Gamma_6$ | 1 | -1 | -1 | -1 | 1 | -1 | 1 | -1 | -1 |
| $\Gamma_7$ | 1 | -1 | 1 | -1 | 1 | -1 | 1 | 1 | -1 |
| $\Gamma_8$ | 1 | -1 | 1 | 1 | 1 | -1 | 1 | 1 | 1 |
| $\Gamma_9$ | | | | | | | | | |
| $\Gamma_{10}$ | | | | | | | | | |
| $\Gamma_{11}$ | | | | | | | | | |
| $\Gamma_{12}$ | | | | | | | | | |
| $\Gamma_{13}$ | | | | | | | | | |
| $\Gamma_{14}$ | | | | | | | | | |
| $\Gamma_{15}$ | | | | | | | | | |
| $\Gamma_{16}$ | | | | | | | | | |
| $\Gamma_{17}$ | | | | | | | | | |
| $\Gamma_{18}$ | | | | | | | | | |

| $b^\infty \times \frac{\bar{1}_4/1_{i_m}}{\bar{1}_3+1_{i_m}}$ | $\{R_0, E\}$ | $\{R_0, IC_{2r}\}$ | $\{2_\perp R_0, IC_{3\delta}\}$ | $\{T_{2\perp}R_0, E\}$ | $\{2_\perp R_0, IC_{2r}\}$ | $\{T_{2\perp}R_0, IC_{3\delta}\}$ |
|---|---|---|---|---|---|---|
| $\Gamma_1$ | | | | | | |
| $\Gamma_2$ | | | | | | |
| $\Gamma_3$ | | | | | | |
| $\Gamma_4$ | | | | | | |
| $\Gamma_5$ | | | | | | |
| $\Gamma_6$ | | | | | | |
| $\Gamma_7$ | | | | | | |
| $\Gamma_8$ | | | | | | |
| $\Gamma_9$ | | | | | | |
| $\Gamma_{10}$ | | | | | | |
| $\Gamma_{11}$ | | | | | | |
| $\Gamma_{12}$ | | | | | | |
| $\Gamma_{13}$ | | | | | | |
| $\Gamma_{14}$ | | | | | | |
| $\Gamma_{15}$ | | | | | | |

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
