# Peer review of "The Crystallographic Spin Point Groups and their Representations"

_SciPost Physics, doi:SciPost Phys. 18, 109 (2025)_

## Round 2 · Referee Report · Anonymous (Referee 1) · 2024-4-6

Strengths

  1. The data produced is expected to be useful for researchers working in the field.

Weaknesses

  1. The article should provide more specificity regarding the novelty of the presented data, especially concerning previous works on the subject.

Report

Presented here is a comprehensive review of spin point groups and their representations, featuring an exhaustive list of data. This resource is expected to be utilized by researchers in the field of condensed matter physics, especially those investigating systems with spin ordering that is either decoupled or weakly coupled to orbital degrees of freedom.
While the authors provide valuable insights, they could enhance clarity regarding the novelty of the presented data, particularly in comparison to prior works such as Litvin's study [22] on spin point groups (Acta Crystallographica Section A 33(2), 279, 1977) and Damnjanovic and Vujicic research [26] on subgroups of weak-direct products and magnetic axial point groups (Journal of Physics A General Physics 14, 1055, 1981).
Additionally, reference [28] by Damnjanovic, "Symmetry in Quantum Nonrelativistic Physics", Faculty of Physics Belgrade 2014, http://www.ff.bg.ac.rs/Katedre/QMF/SiteQMF/pdf/sqnp2e.pdf, should be fully specified and cited within the text of the article, i.e. within section 2 ("Introduction to the Spin Point Groups") and section 3 ("Review of Magnetic Representation Theory").

Requested changes

  1. Within the Introduction enhance the clarity regarding the novelty of the presented data in comparison to prior works such as Litvin's study [22] on spin point groups (Acta Crystallographica Section A 33(2), 279, 1977) and Damnjanovic and Vujicic research [26] on subgroups of weak-direct products and magnetic axial point groups (Journal of Physics A General Physics 14, 1055, 1981).
  2. The reference [28] Damnjanovic, "Symmetry in Quantum Nonrelativistic Physics" Faculty of Physics Belgrade 2014, http://www.ff.bg.ac.rs/Katedre/QMF/SiteQMF/pdf/sqnp2e.pdf, should be fully specified and cited within the text of the article, i.e. within section 2 ("Introduction to the Spin Point Groups") and section 3 ("Review of Magnetic Representation Theory").

---

## Round 2 · Referee Report · Anonymous (Referee 2) · 2024-4-10

Report

The paper addresses considerably interesting subject of spin groups. Generally, this is one of the nowadays frequent articles with exhausting group theoretical data. The main general remark is that the style is more appropriate for textbook: most of the methodological (mathematical) considerations are well known, making redundant many rederivations, proofs, constructions and long explanations. Instead, only basic information with references (which are correctly given) are expected. So, the body of the paper is to be essentially reduced.

Particular comments 1.Title "Spin Point Groups..." is missleading, as only 32 crystallographic point groups are considered, while infinitely many other ones are omitted. In the introduction there are some remarks on the great role that symmetry has in physics, solid state in particular. Maybe reference to Poincare groups is a sort of show-off, basically out of context; also, group theory enters in physics through representation theory, and formulations like “The group theory and representation theory of crystals are the foundations…” is pleonastic.

  1. Section 2 can be reduced, in particular 2.4. In fact, the standard definition is correctly referred to, but then also fully elaborated (without new details) in 2.1, while in the 2.2, 2.3 and 2.4 the first paragraph gives relevant references, and there is no need for long repetitions of the contained results. Also, a footnote 1 is amusing: Pin and SPin groups are hardly within the scope of the researchers reading this paper.

  2. Section 3 is very detailed explanation of the theory of induction of (co-) representations from the index-two subgroups. It seems that the authors are not aware that this theory is well developed, as no reference is offered besides Bradley and Cracknell (e.g. Wigner's classical book, or Jansen and Boon, Theory of Finite Groups: Applications in Physics). So, this Section is to be drastically compressed (if not omitted). Also, characterization of the types of the representations is given in terms of Dimock's criterion, with comparison to the Frobenius-Schur (footnote), but Wigner's is not even mentioned (later on Frobenius-Schur test is used).

  3. As far as formalism is considered, Section 4 contains some new results. Although it is correct, I am also here puzzled by the chosen mathematical terminology. Instead of only giving result for SO(3) (and SO(2)) integrals, here the Haar measure is mentioned in 4.2, but neither corresponding parameterization (Euler's angles?) nor measure itself is explicated; btw, in such “higher mathematical” framework, methodological hierarchy suggests to start with the compactness of the group, and two-sided measure. Simply, in this Section ordinary language is more adequate, in particular taking into account possible audience.

  4. Section 5 is nice, and correctly written. I read it with interest.

  5. In the last section, the summary is given. Besides emphasizing some results which are well known in the literature (e.g. doubling of co-representation for particular values of Dimock's indicator), it is correctly written.

  6. Appendices B and C, and pure theoretical part (which is known) of D are not necessary, being mostly rephrasing of the known results. On the other hand, the remaining parts of D, with examples and concrete derivation are important, and instructive.

To conclude, in this form manuscript is not publicable. However, major revision can help, and I will be ready to reconsider it.

Recommendation

Ask for major revision

---

## Round 5 · Referee Report · Anonymous (Referee 2) · 2024-12-6

Strengths

Interesting point, possibly important.

Weaknesses

Elaborated in too many details

Report

The paper is partly amended according to the suggested lines.
I still find it is too large, with too many repetitions of the well known theory. In the answer to referees the authors try to make some general, almost ontological discussion on the aim of the published papers. I cannot
enter this type of polemic. I only state again that the paper is written rather in the textbook style, with too many "review" aspects. As for the contents alone, it is OK. So, the rest is on the policy of the journal, and I do not want to make additional barriers.

Recommendation

Publish (meets expectations and criteria for this Journal)

---

## Round 5 · Referee Report · Anonymous (Referee 1) · 2024-12-21

Report

The revised version of the paper aligns well with the reviewers' feedback, effectively addressing their concerns. It meets the necessary standards and is suitable for publication in its current form.

Recommendation

Publish (meets expectations and criteria for this Journal)

---

## Round 5 · Author Response

Reply to Referees: Crystallographic Spin Point Groups and their Representations

Dear Editor,

We would like to resubmit our paper Crystallographic Spin Point Groups for publication in SciPost Physics. We thank the two anonymous reviewers for their thoughtful recommendations; we found their feedback valuable, and have largely incorporated the suggestions. Below we describe the changes made, and our response to the comments of the referees.

Sincerely, Hana Schiff (on behalf of all coauthors)

Reply to Report 1 Report: "Strengths. The data produced is expected to be useful for researchers working in the field. Weaknesses. The article should provide more specificity regarding the novelty of the presented data, especially concerning previous works on the subject. Report. Presented here is a comprehensive review of spin point groups and their representations, featuring an exhaustive list of data. This resource is expected to be utilized by researchers in the field of condensed matter physics, especially those investigating systems with spin ordering that is either decoupled or weakly coupled to orbital degrees of freedom. While the authors provide valuable insights, they could enhance clarity regarding the novelty of the presented data, particularly in comparison to prior works such as Litvin's study [22] on spin point groups (Acta Crystallographica Section A 33(2), 279, 1977) and Damnjanovic and Vujicic research [26] on subgroups of weak-direct products and magnetic axial point groups (Journal of Physics A General Physics 14, 1055, 1981). Additionally, reference [28] by Damnjanovic, "Symmetry in Quantum Nonrelativistic Physics", Faculty of Physics Belgrade 2014, http://www.ff.bg.ac.rs/Katedre/QMF/SiteQMF/pdf/sqnp2e.pdf, should be fully specified and cited within the text of the article, i.e. within section 2 ("Introduction to the Spin Point Groups") and section 3 ("Review of Magnetic Representation Theory").

Requested changes. 1. Within the Introduction enhance the clarity regarding the novelty of the presented data in comparison to prior works such as Litvin's study [22] on spin point groups (Acta Crystallographica Section A 33(2), 279, 1977) and Damnjanovic and Vujicic research [26] on subgroups of weak-direct products and magnetic axial point groups (Journal of Physics A General Physics 14, 1055, 1981). 2. The reference [28] Damnjanovic, "Symmetry in Quantum Nonrelativistic Physics" Faculty of Physics Belgrade 2014, http://www.ff.bg.ac.rs/Katedre/QMF/SiteQMF/pdf/sqnp2e.pdf, should be fully specified and cited within the text of the article, i.e. within section 2 ("Introduction to the Spin Point Groups") and section 3 ("Review of Magnetic Representation Theory")."

Author response: We thank the referee for taking the time to read and comment on our manuscript. We are glad that the results in the paper have a broadly positive response.

Of the perceived weakness of the paper, we appreciate the remarks of the referee. It appears that the second referee has similar views. Our manuscript aspires to give a reasonably self-contained and comprehensive discussion of the theory of spin point groups relevant to condensed matter systems. This has meant that we have devoted some sections to a review of results that can be found elsewhere in the literature. In the revised version we have attempted to improve the presentation so that the reader can more easily discern which results are new to this work and which are reviewed: in particular, to highlight that the novelty lies in the derivation of irreducible representation tables which were unknown to the field prior to this manuscript, and a new extension to the spin group classification extending beyond the work of Litvin and Opechowski.

Relatedly we note that our paper was not optimally internally referenced. To address both issues we have enlarged the final paragraph of the introduction where the contents of the paper are outlined referencing the different sections. There we have explicitly pointed out what is new to this work and which parts are review sections making contact with the work of Litvin. The new text is as follows:
The organization of the paper is as follows. In an effort to be as self-contained as is practicable, we give a complete discussion of what the spin point groups are, introducing notions of nontrivial spin point groups and spin-only groups and how to enumerate both (described in the first three parts of Section 2). Once we have both nontrivial spin point groups and the spin-only groups we may classify the total spin point groups (Section 2.4). A complete table of the nontrivial spin point groups is given in Appendix B. This table distinguishes the collinear and coplanar spin groups. These sections review material that can be found elsewhere in the literature [6,8,15–18,25–28]. In particular, the nontrivial spin point groups were enumerated by Litvin [25], the spin line groups by Lazic, Milivojevic and Damnjanovic [27] and the pairing of spin groups with the nontrivial spin groups was investigated by Liu et al. [26]. We then turn to the representation theory of these groups which had not previously been worked out. Making use of the main isomorphism theorem of Litvin and Opechowski [18], we demonstrate that the 598 nontrivial spin point groups have co-irreps corresponding to the regular or black and white point groups (Section 3.1). In Sections 3.2, we describe the effect on the representations of including the spin-only group to form the total spin group. We show that the coplanar spin groups are isomorphic to paramagnetic spin groups. Of particular interest are the spin groups corresponding to collinear magnetic structures with continuous spin rotation symmetry. These have new co-irreps that we compute and tabulate. The computation method is described in Section 3.2.4 with various technical results relegated to appendices. Complete tables of the co-irreps of the collinear spin groups are listed in Appendix F. In Section 4, we give some examples of how to put information about the representation theory to use in applications from band theory. In doing so, we remark on physically relevant extensions to the Litvin- Opechowski spin-only groups. Finally, we conclude with a broader perspective on the spin group representation theory including general results that may be inferred from the co- irrep tables. As a guide to the reader who may not be familiar with the (co-)representation theory of magnetic groups we review the relevant material in Appendix A.
We have added citations to the book “Symmetry in quantum nonrelativistic physics” as a general reference on group theory in our review on magnetic representation theory and in the section introducing spin groups.

Reply to Report 2 Report: "The paper addresses considerably interesting subject of spin groups. Generally, this is one of the nowadays frequent articles with exhausting group theoretical data. The main general remark is that the style is more appropriate for textbook: most of the methodological (mathematical) considerations are well known, making redundant many rederivations, proofs, constructions and long explanations. Instead, only basic information with references (which are correctly given) are expected. So, the body of the paper is to be essentially reduced.

Particular comments 1.Title "Spin Point Groups..." is missleading, as only 32 crystallographic point groups are considered, while infinitely many other ones are omitted. In the introduction there are some remarks on the great role that symmetry has in physics, solid state in particular. Maybe reference to Poincare groups is a sort of show-off, basically out of context; also, group theory enters in physics through representation theory, and formulations like “The group theory and representation theory of crystals are the foundations…” is pleonastic.

  1. Section 2 can be reduced, in particular 2.4. In fact, the standard definition is correctly referred to, but then also fully elaborated (without new details) in 2.1, while in the 2.2, 2.3 and 2.4 the first paragraph gives relevant references, and there is no need for long repetitions of the contained results. Also, a footnote 1 is amusing: Pin and SPin groups are hardly within the scope of the researchers reading this paper.

  2. Section 3 is very detailed explanation of the theory of induction of (co-) representations from the index-two subgroups. It seems that the authors are not aware that this theory is well developed, as no reference is offered besides Bradley and Cracknell (e.g. Wigner's classical book, or Jansen and Boon, Theory of Finite Groups: Applications in Physics). So, this Section is to be drastically compressed (if not omitted). Also, characterization of the types of the representations is given in terms of Dimock's criterion, with comparison to the Frobenius-Schur (footnote), but Wigner's is not even mentioned (later on Frobenius-Schur test is used).

  3. As far as formalism is considered, Section 4 contains some new results. Although it is correct, I am also here puzzled by the chosen mathematical terminology. Instead of only giving result for SO(3) (and SO(2)) integrals, here the Haar measure is mentioned in 4.2, but neither corresponding parameterization (Euler's angles?) nor measure itself is explicated; btw, in such “higher mathematical” framework, methodological hierarchy suggests to start with the compactness of the group, and two-sided measure. Simply, in this Section ordinary language is more adequate, in particular taking into account possible audience.

  4. Section 5 is nice, and correctly written. I read it with interest.

  5. In the last section, the summary is given. Besides emphasizing some results which are well known in the literature (e.g. doubling of co-representation for particular values of Dimock's indicator), it is correctly written.

  6. Appendices B and C, and pure theoretical part (which is known) of D are not necessary, being mostly rephrasing of the known results. On the other hand, the remaining parts of D, with examples and concrete derivation are important, and instructive.

To conclude, in this form manuscript is not publicable. However, major revision can help, and I will be ready to reconsider it."

Author response: We thank the referee for reviewing our manuscript and for the kind words about our new results.

The principal point made in the review is that the existing manuscript contains discussion beyond the new material and that this should be removed. For example, the manuscript discusses essentially standard results on magnetic co-representation theory in Section 3 and (largely) reviews existing results on spin point groups in section 2. The main new results are in Sections 4 and 5 and, of course, the exhaustive list of spin group character tables as well as appendices spelling out details of the calculations that, we felt, are not necessary to appreciate the main new results laid out in the main text.

There are several possible reasons why one might wish to streamline the presentation by removing review sections. These include the following. (1) If readers are already familiar with the background material already and will look at papers only for the new results then review sections are obviously redundant and can be removed. (2) As a matter of style one might argue that papers should minimize material that can be found elsewhere. (3) One might be concerned that including both old and new material runs the risk of confusing the reader about the truly new advances in the paper. (4) Finally, one could be keen to see papers shortened to improve their accessibility to readers.

To the first point, we would ideally make few assumptions about the background knowledge of any readers. That said, the authors are condensed matter physicists writing for a broad-based physics journal not a specialized group theory or mathematical physics journal. The material reflects what we believe is important for physicists wanting to understand magnetic symmetries in condensed matter. Including the review sections has the advantages of unifying notation and sparing the reader the trouble of studying textbooks for co-representation theory or the long papers on spin groups to which we refer in Section 2. We also believe the section on spin groups has pedagogical value as this section is not merely a regurgitation of the literature and it may therefore help some readers better appreciate the material. Including these sections has the disadvantages of delaying the introduction of new results and of increasing the length. But the journal has no length limit and, at 30 pages in the main text, does not seem excessively long. As new applications of the spin groups arise, the authors believe many readers would find value in the pedagogical and self-contained approach of this work.

To the second point, again, our view is that telegraphic papers run the risk of requiring too much background knowledge of their readers and the advantage of the long-form of Scipost is that there is ample space to describe the necessary (reasonable) background.

To the third point, Report 1 contains a similar concern and we sympathize with this view. Although we had clearly cited previous work and had tended to split older work into separate sections, the style of the paper did level the material in keeping with a more pedagogical style. We have now added a paragraph at the end of the introduction to make more explicit which results originate with us (the representation theory of the crystallographic spin point groups resting on the representation theory of the non-magnetic and magnetic point groups and including all of the tables at the end of the paper) and which do not (everything else). This paragraph also references the various sections of the paper again spelling out which contain new results.

For the fourth point, the comments from the referee have caused us to reflect on the presentation. We appreciate that some people may be put off by the length of the main part of the paper and we see no particular harm in having the section on magnetic representation theory as an appendix where it now appears in the updated manuscript.

On the point about references, we have, in addition to Bradley and Cracknell, now cited Wigner's book and Jansen and Boon.

We are in agreement with the referee regarding the title of our paper. It is more correct to be clear from the outset that we restrict our attention to the crystallographic spin point groups. Therefore we have changed the title and, in addition, we point out that the same methods in our paper can be used to obtain the co-representations of non-crystallographic spin point groups. We have also made changes to the wording of the introduction to address the other concerns raised in point number 1. We have retained the footnote clarifying the ambiguity in the term “spin groups” as the alternative usage is more common and it may therefore be useful to people stumbling on this paper as we fully agree that there is no likely overlap in readership.

In Section 4, the referee commented on the Haar measure. In the updated manuscript we have spelt out the form of the measure by parametrizing the elements of SO(3) with Euler angles.

List of Changes

  • Section 4 has been revised and the representation theory review has been moved into an appendix.
  • Title changed to “The crystallographic spin point groups and their representations” and “crystallographic” added to abstract
  • Comments added on non-crystallographic point groups
  • At the end of the introduction there is now a detailed outline of the contents.
  • Wordings in introduction altered to address point 1 e.g. no reference now made to Poincaré group
  • Changed c to ć in last names at the end of the introduction
  • We have added some new references
  • The Applications section (section 4) has been edited for clarity and expanded. In particular we have added a new subsection on MnTe - a second altermagnetic system - to show how the spin point groups are relevant to that case. We have also spelt out the group elements at different momenta in the rutile and MnTe examples with the hope that the section is now more readable. Comment is made about SPGs extending the Litvin classification both here, and briefly in the discussion section.
  • Details added about Haar measure – after “...R(t, f, p) in SO(3)”: Here we have chosen the axis-angle parameterization, where $\theta$ and $\varphi$ define the axis of the rotation $R$, and $\psi$ defines the angle of rotation about the axis. Within this convention, the measure is given by $2(1-\mathrm{cos}\psi)\marthm{sin}\theta \, d\theta\, d\varphi \, d\psi$ and the normalization factor will be $\frac{1}{8\pi^{2}}.$ Updated figures 2 and 3.

---

## Round 5 · List of Changes

List of Changes

  • Section 4 has been revised and the representation theory review has been moved into an appendix.
  • Title changed to “The crystallographic spin point groups and their representations” and “crystallographic” added to abstract
  • Comments added on non-crystallographic point groups
  • At the end of the introduction there is now a detailed outline of the contents.
  • Wordings in introduction altered to address point 1 e.g. no reference now made to Poincaré group
  • Changed c to ć in last names at the end of the introduction
  • We have added some new references
  • The Applications section (section 4) has been edited for clarity and expanded. In particular we have added a new subsection on MnTe - a second altermagnetic system - to show how the spin point groups are relevant to that case. We have also spelt out the group elements at different momenta in the rutile and MnTe examples with the hope that the section is now more readable. Comment is made about SPGs extending the Litvin classification both here, and briefly in the discussion section.
  • Details added about Haar measure – after “...R(t, f, p) in SO(3)”: Here we have chosen the axis-angle parameterization, where $\theta$ and $\varphi$ define the axis of the rotation $R$, and $\psi$ defines the angle of rotation about the axis. Within this convention, the measure is given by $2(1-\mathrm{cos}\psi)\marthm{sin}\theta \, d\theta\, d\varphi \, d\psi$ and the normalization factor will be $\frac{1}{8\pi^{2}}.$ Updated figures 2 and 3.

---

## Editorial Decision

published